behaviour/ecology/evolution

Uganda, chimpanzee, hominin evolution, cluster analysis, landscape use

**Author for correspondence:**
Marian I. Hamilton
e-mail: marian.hamilton@unco.edu

# Using strontium isotopes to determine philopatry and dispersal in primates: a case study from Kibale National Park

Marian I. Hamilton[1,3], Diego P. Fernandez[2]
and Sherry V. Nelson[3]

[1]Department of Anthropology, University of Northern Colorado, Greeley, CO, 80639-6900, USA
[2]Department of Geology and Geochemistry, University of Utah, Salt Lake City, UT, USA
[3]Department of Anthropology, University of New Mexico, Albuquerque, NM, 87111, USA

MIH, 0000-0001-6321-4768

Strontium isotope ratios ($^{87}Sr/^{86}Sr$) allow researchers to track changes in mobility throughout an animal's life and could theoretically be used to reconstruct sex-biases in philopatry and dispersal patterns in primates. Dispersal patterns are a life-history variable that correlate with numerous aspects of behaviour and socio-ecology that are elusive in the fossil record. The present study demonstrates that the standard archaeological method used to differentiate between 'local' and 'non-local' individuals, which involves comparing faunal isotopic ratios with environmental isotopic minima and maxima, is not always reliable; aspects of primate behaviour, local environments, geologic heterogeneity and the availability of detailed geologic maps may compromise its utility in certain situations. This study instead introduces a different methodological approach: calculating offset values to compare $^{87}Sr/^{86}Sr$ of teeth with that of bone or local environments. We demonstrate this method's effectiveness using data from five species of primates, including chimpanzees, from Kibale National Park, Uganda. Tooth-to-bone offsets reliably indicate sex-biases in dispersal for primates with small home ranges while tooth-to-environment offset comparisons are more reliable for primates with larger home ranges. Overall, tooth-to-environment offsets yield the most reliable predictions of species' sex-biases in dispersal.

# 1. Introduction and background

Patterns of mobility and landscape use, such as sex-biased philopatry (remaining in one's natal group through adulthood) and dispersal (leaving one's natal group around the age of sexual maturity), have direct consequences for primate social behaviour. Ancestral behaviour and socioecological conditions for the hominin clade are currently unresolved [1–3]. Reconstructing dispersal patterns could provide a proxy for reconstructing behavioural variables that are not directly observable in the fossil record, including patterns of aggression and affiliation [4], social bonding, coalition formation [5] and the evolutionary trajectory of traits which differentiate humans from our fellow extant great apes (e.g. extended life histories, pair-bonding, cooperative hunting, multi-family social structures, female–female bonding). Previous work relies heavily on modern primate behavioural models and assumptions based on patterns of sexual dimorphism [6,7]. Recently, researchers have employed more direct proxies to reconstruct behaviour, including hormonal measurements [8] and geochemical data, such as strontium isotope ratios [9].

Strontium isotope ratios differ across varying surface geologies and have the potential to act as a mobility tracer, providing empirical insights into the way extinct species used their landscapes [10]. In geologic formations, the ratio of a heavier isotope of strontium ($^{87}Sr$) to a lighter isotope ($^{86}Sr$) varies in accordance with bedrock age and type. The strontium isotope ratio ($^{87}Sr/^{86}Sr$) of underlying bedrock is incorporated into soil and plants and then into calcium-bearing tissues of animals eating those plants with minimal fractionation [10–12]. Therefore, the $^{87}Sr/^{86}Sr$ within faunal tissues, including bones and tooth enamel, reflects the isotopic fingerprint of the area where the animal lived during that tissue's formation. Different tissues capture different life stages. Tooth enamel forms early in life and then is metabolically inert, although different enamel locations within the same tooth do calcify at different times [13]. Enamel that mineralizes at and after weaning captures a juvenile (pre-dispersal) isotopic fingerprint. Enamel is also robust against diagenetic effects due to its high density and low porosity [14]. Bone turns over throughout life and therefore reflects the time period closer to the end of an individual's life (often the adult, post-dispersal period). Bone is more susceptible to diagenetic alterations than tooth enamel due to increased porosity, making them less useful for palaeontological reconstructions [14]. Botanical samples, soil and contemporaneous micromammal remains from the area of a fossil's deposition can also approximate an 'adult' isotopic signal, if the fossilized individuals lived at or near the depositional site. Comparisons of juvenile (e.g. tooth enamel) to adult (e.g. bone or local environment) isotopic signatures can, therefore, trace an individual's movement across geologically heterogeneous landscapes.

Researchers have used strontium isotope ratios to assess residence patterns [15] and track mobility across a wide array of environments and species [16–22]. However, research to date has limited applications in the hominin fossil record [9,13,23,24] and none in modern primate communities. Chimpanzees are frequently used as models for the last common ancestor (LCA) of chimpanzees and humans [25–28], although this model is not without criticisms [2,29,30]. In this male-philopatric species, dispersal patterns have downstream impacts including male coalition formation and violence, decreased female gregariousness in comparison with males and complex social bond formation. Evidence that hominin ancestors also followed a male-philopatric pattern would provide support for continuing to use chimpanzees as models for the LCA and ancestral hominin behaviour. It is critical to assess if strontium isotope ratios can accurately indicate such mobility patterns, especially when home ranges are large and social structures are complex. To this end, the purposes of this investigation are to (i) assess the reliability of strontium isotope ratios for predicting dispersal patterns in a modern primate community and (ii) identify the most accurate methods to reconstruct these dispersal patterns.

The most common method in archaeological strontium isotope literature [9,14,22,24,31–36] to differentiate local (presumably philopatric) and non-local (dispersing) individuals is to first, use botanical, soil or micromammal samples to define minimum and maximum strontium isotope ratios for a predefined local site, such as discrete geologic formations, site associations or community boundaries, and second, compare an individual's enamel $^{87}Sr/^{86}Sr$ with this range. For this standard archaeological approach, individuals with $^{87}Sr/^{86}Sr_{enamel}$ falling outside of the local isotopic minimum and maximum are considered non-local and individuals with $^{87}Sr/^{86}Sr_{enamel}$ falling within the local range are considered local [10–12,14]. In this paper, we first assess the reliability of this method for categorizing philopatric versus dispersing primates by applying it to a modern primate community at Kibale National Park, Uganda in which the expectations for dispersing and philopatric sex are known for each species. Next, we determine the reliability of a different approach: comparing the offset between adult and juvenile isotopic proxies to differentiate between dispersing and philopatric individuals, rather than using direct isotopic ratios for binary categorization. We compare the efficacy

of two such offset proxies: tooth enamel-to-bone ($^{87}Sr/^{86}Sr_{enamel-bone}$) and tooth enamel-to-local environment ($^{87}Sr/^{86}Sr_{enamel-environment}$). We assume the local environment as equivalent to the local bioavailable $^{87}Sr/^{86}Sr$ and define it by statistical clustering of botanical isotopic values. We expect all offsets will be greater in the dispersing sex than the philopatric sex. This offset approach builds on previous work which compares the strontium isotope ratios from different tissues [37–40] or tissues and the environment by using the quantified differences, or offsets, to not only identify individual mobility patterns but also to elucidate broader patterns of sex-biases in dispersal. By comparing each individual's quantified offset values with those from other members of its species from the same location, broader species-level mobility patterns can be uncovered.

## 1.1. Strontium sources and the influence of local geology on bioavailable $^{87}Sr/^{86}Sr$

Although for many areas the local bedrock is the main source of bioavailable soil strontium, important contributions from atmospheric or hydrological origin exist in some ecosystems [41], and bioavailable $^{87}Sr/^{86}Sr$ does not always correspond with underlying geology [42]. In general, a number of processes contribute to the strontium contained in the biosphere: for example, weathering of bedrock from groundwater [43], plant root uptake from soil [44,45] and bedrock [46], precipitation containing marine aerosols [47], dust transport [10,48] and ecosystem recycling [46]. For instance, when allochthonous sediments (e.g. glacial or aeolian deposits) are the main soil parent material, the $^{87}Sr/^{86}Sr$ can be very different from the bedrock [49]. The local $^{87}Sr/^{86}Sr$ is best known from bioavailable Sr contained in plants or small-range mammal teeth [44], although when these data do not exist, the bedrock geology can be a starting place for hypothesizing values for local water and vegetation in some localities [41,50]. No matter the location, there is no substitute for empirical isoscape mapping [42].

For old oxisols in tropical savannah or forests like those in East Africa, the parent material (bedrock, till, alluvium, etc.) and its associated strontium isotope ratio would be almost completely consumed by weathering. Then, bioavailable strontium may be primarily derived from atmospheric sources [51]. In this way, the local geology may play a subordinate role to the strontium found in the soil. In addition, bioavailable strontium in soil within the riparian zone typically reflects the isotopic ratio of the strontium dissolved in the river water, which in turn is related to the weathering of distal upstream rocks at higher elevation [52]. The local geology in the riparian zone soil plays again a secondary role as a source of the strontium in the plant and animal tissue. Bioavailable $^{87}Sr/^{86}Sr$ signature is thus best determined in these cases from plant and small-range animal tissue.

Kibale National Park is located within the Albertine Rift system, the farthest north section of the western branch of the East African Rift System [53]. It is underlain by the Buganda-Toro belt and basement locally migmatized gneiss. The pre-rift geology of the region is the result of a mid-Proterozoic collision zone leading to mixing of Proterozoic schists and Archean gneisses on the scale of kilometres [54]. This geology lends itself to high isotopic variability for numerous reasons [55]. First, the geologic formations here are quite old, dating to the early and middle Palaeoproterozoic era (approx. 2–2.5 billion year ago). The Toro formation comprises undifferentiated acid and basic gneisses while the neighbouring Buganda formation comprises primarily quartzites [56]. These felsic geologies, in addition to being very ancient, are high in their Rb/Sr ratios [50]. Taken together, these factors predict highly radiogenic strontium isotope ratios. Second, we can expect a high degree of isotopic heterogeneity even within a single geologic formation due to differential mineral weathering. Previous research within East Africa illustrates high $^{87}Sr/^{86}Sr$ variability within the region due to the age, lithology, mineral composition and fluvial systems within the region [42,57,58]. We, therefore, cannot expect that bioavailable $^{87}Sr/^{86}Sr$ in Kibale will necessarily track with underlying geology. We anticipate that there will be meaningful variability within geologic formations and that neighbouring geologic formations may have a high degree of isotopic overlap, making them difficult to distinguish geochemically even when they are well defined geographically (see Discussion for more details). Capturing this variability will be crucial to reconstructing primate mobility in this location.

# 2. Material and methods

## 2.1. Sample collection

This study samples faunal and botanical strontium isotope ratios in Kibale National Park, a rainforest in southwestern Uganda. We collected bone and tooth enamel samples from 57 primates housed in existing

**Table 1.** Kibale primate ecological parameters and sample sizes.

| species (scientific name) | species (common name) | home range | dispersing sex | sample size |
|---|---|---|---|---|
| Colobus guereza | black and white colobus monkey | 0.12–0.28 km$^2$ [59] | males | 6 |
| Papio anubis | olive baboon | ∼5 km$^2$ [60] | males | 9 |
| Cercopithecus ascanius | redtail monkey (guenon) | 0.28–0.68 km$^2$ [61] | males | 5 |
| Cercopithcus mitis | blue monkey (guenon) | 0.5–3.3 km$^2$ [62] | males | 2 |
| Pan troglodytes schweinfurthii | common chimpanzee | 28–41 km$^2$ [63,64] | females | 17 |
| Procolobus badius | red colobus | ∼0.65 km$^2$ [61] | females | 18 |

opportunistically gathered skeletal collections at the Makerere Biological Research Station in Kibale National Park, Uganda, with assistance from the Kibale Chimpanzee Project and the Ngogo Chimpanzee Project, in summer 2016. These samples included four female-philopatric species—black and white colobus monkeys (Colobus guereza), olive baboons (Papio anubis), and two species of guenons, blue monkeys (Cercopithecus mitis) and redtail monkeys (C. ascanius); and two male-philopatric species—chimpanzees (Pan troglodytes schweinfurthii) and red colobus monkeys (Procolobus badius) (table 1). Teeth and bones were first cleaned of surface dirt and debris using a Dremel drill. We then removed approximately 15 mg of enamel or bone using a diamond-tipped Dremel drill bit. Areas of cancellous bone (such as the ends of long bones, ribs or areas on the skull) which have a more rapid turnover rate than areas of cortical bone [65,66] were preferentially sampled when available to increase the likelihood of a post-dispersal isotopic signature in bones. First molars (M1s), the first permanent molar teeth to erupt in primates [67], were sampled when available to ensure pre-dispersal isotopic signatures in teeth. Sillen & Balter [13] demonstrate that enamel closer to the dentine-enamel junction (DEJ), which mineralizes earlier in life than the outer enamel, can still reflect maternal residence patterns due to mobilization of maternal skeletal elements during lactation [13]. First molars erupt coincident with weaning in most primates [68]. Even for species who continue to suckle after the emergence of M1 (such as chimpanzees), they are consuming adult levels of solid foods with very little energy transfer from nursing by the time of its emergence [69–71]. Our samples comprise primarily later-forming enamel from the outer enamel layer of M1. This minimizes maternal effects while still ensuring a pre-dispersal isotopic signal, although it is not possible to entirely eliminate the contribution of maternal strontium within first molar samples. Six individuals did not have first molars available to sample and instead first premolars, first incisors or second molars were sampled (see Discussion for more details on age of tooth eruption for sampled species). Details for each individual are provided in electronic supplementary material, table S1. Only fully mature adult individuals were included in the sample. Individuals were assigned provenance based on collection location (electronic supplementary material, table S1).

Botanical samples were GPS-referenced plant leaves collected opportunistically along existing trails in approximately 1 km grids during the summers of 2014, 2015 and 2016. These samples came from four study areas of Kibale National Park: Kanyawara, Sebitole, Kanyanchu and Ngogo (figure 1, electronic supplementary material, table S2). These areas are underlain by two geologic formations: Sebitole is primarily on the Toro Formation (an undifferentiated early Palaeoproterozoic gneiss), Kanyanchu is primarily on the Buganda formation (a middle Palaeoproterozoic quartzite) and Kanyawara falls across both the Toro and Buganda formations. Ngogo falls on an area designated an 'unidentified radiometric anomaly' by the Ugandan Society for Geology and Mines [56]. Available geologic maps for this area lack high-resolution detail making it difficult to assess with confidence which geologic area plant samples are from. For the 'unidentified radiometric anomaly' area, no additional geologic data are available.

Given the lack of detail in available geologic maps, the ancient nature of Kibale's geology and anticipated high variability of bioavailable $^{87}Sr/^{86}Sr$, using boundaries of geologic formations as default definitions for local areas may not be appropriate for this site. Relying on them could lead to faunal $^{87}Sr/^{86}Sr_{enamel}$ equally likely to be from multiple areas, if the geologic areas have extensive isotopic overlap. This would make it impossible to draw further conclusions. Site-specific associations

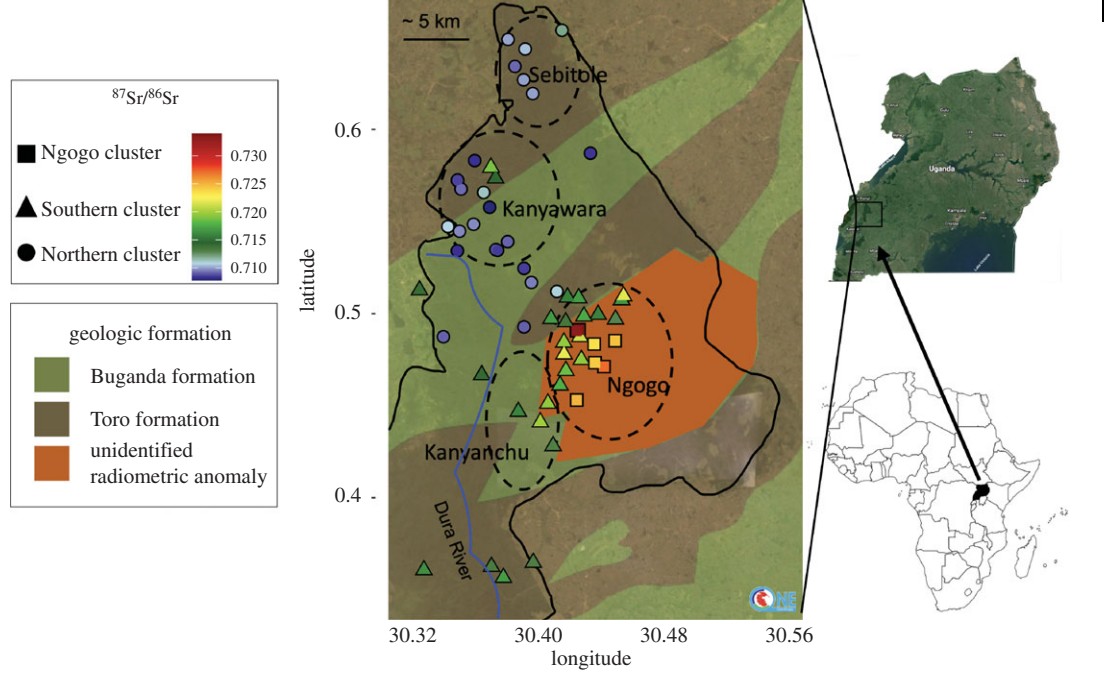

**Figure 1.** Location and geologic map of Kibale National Park, with locations and strontium isotope ratios of collected plant leaf samples. The boundaries of the park are outlined with a solid black line while approximate park area boundaries are shown with dotted black lines; green shaded areas represent the Buganda Formation (middle Palaeoproterozoic quartzite); brown shaded areas represent the Toro Formation (early Palaeoproterozoic gneiss); orange shaded areas represent an 'unidentified radiometric anomaly' [52]. Circles represent samples assigned to the Northern cluster through cluster analysis; triangles represent those assigned to the Southern cluster; squares represent those assigned to the Ngogo cluster. Clusters were generated using a dendrogram with average linkage distance to group similar data points together and mapped using the 'ggmap' package in R Studio (v. 0.99.902). Map reproduced with the permission of the OneGeology. All rights reserved.

are also commonly used to define local areas, although most frequently in studies where the geographical distance between local sites of interest is greater than a few kilometres (as is the case at Kibale). Here, we used instead a hierarchical cluster analysis to statistically identify isotopic clusters using plant samples, which we then used to distinguish possible local areas (see electronic supplementary material, table S2). This approach is beneficial because it can be applied in areas lacking detailed geologic maps or in areas without clear community boundary delineations. It provides an alternative approach for defining geochemically meaningful local areas when neighbouring geologies or neighbouring sites do in fact have overlapping isotopic profiles.

Riparian plants did not differ significantly from non-riparian plant samples in Kibale National Park, as has been observed elsewhere [24,52]. To define isotopic clusters on the landscape, we included all non-riparian leaf samples ($N = 64$) in a hierarchical cluster analysis using an average linkage distance and then used a dendrogram [72] to assign each plant sample to a cluster group. We compared the isotopic profiles of the resulting clusters using Kruskal–Wallace rank sum tests and qualitatively assessed their geographical boundaries, including their relationship to park study areas and geological formations, based on maps generated using the 'ggmap' package in R Studio (v. 0.99.902) (figure 1).

## 2.2. Laboratory analysis

Immediately following collection in Kibale National Park, we removed any surface dust and dirt from plant samples before drying them in a food dehydrator between 50 and 60°C. We manually homogenized the sample for storage in an airtight bag. We performed all additional sample preparation and $^{87}Sr/^{86}Sr$ analysis in a clean laboratory following standard protocols at the University of Utah, Salt Lake City, UT, USA in the Department of Geology and Geophysics. We weighed 50 mg of plant sample and 5 mg of powdered tooth or bone for digestion. Plants were digested using a microwave system (Ethos EZ, Milestone, Inc., Shelton, CT, USA) at 200°C for 20 min in PTFE micro-vessels with 2 ml of $HNO_3$ (70%, trace metal grade). Powdered tooth and bone were digested in 2 ml

of $HNO_3$ (70%, trace metal grade) at room temperature. Strontium concentration in all digests was determined using inductively coupled plasma mass spectrometry (ICP-MS, Agilent 7500ce, Santa Clara, CA, USA) with an external calibration method with $10 \, ng \, l^{-1}$ indium as internal standard. An aliquot containing 200 ng Sr from each digest was purified by chromatography using an automatic system (PrepFASTMC, ESI, Omaha, NE, USA). Strontium fractions were dried on a hot plate, rehydrated in 2 ml of 2.4% $HNO_3$ and run in a multi-collector ICP-MS (Neptune Plus, Thermo Finnigan, Bremen, Germany). Certified reference material sample (NIST SRM 987, National Institute of Standards and Technology, Gaithersburg, MD, USA, certified value $0.71034 \pm 0.00026$) was analysed every three samples, with blanks measured before each sample or SRM 987. Our mean and standard deviation for SRM 987 was $0.710287 \pm 0.000011$ ($N = 30$).

## 2.3. Data analysis

To compare the efficacy of the standard archaeological approach with our offset methods (tooth enamel-to-bone offsets and tooth enamel-to-environment offsets), we categorized each primate in our sample as expected 'local/philopatric' or expected 'non-local/disperser', based on sex and species-level sex-biases in dispersal patterns. We then compared this expected attribution with the assigned attribution based on the isotopic data. This categorization (expected to be local/philopatric or expected to be non-local/disperser) necessitates an idealized expectation that all adult primates of the dispersing sex for each species were in fact individuals who locationally dispersed (non-locals), and all those of the philopatric sex were individuals who did not disperse (locals). The authors recognize that this is an overly simplified version of a dispersal model; in reality, rates of dispersal from natal groups can vary widely and are dependent on resource availability, social factors, habitat fragmentation and other factors [73,74]. For example, in male-philopatric chimpanzees, male dispersals are virtually unheard of due to lethal intergroup aggression among males [75], but female dispersal rates from natal groups vary from 50% to nearly 100% [76–79]. In fact, we know that within our own dataset KCP 6 was an adult female chimpanzee born in Kanyawara who did not ever disperse (M. Muller 2015, personal communication). Even in situations where dispersal by members of a species' dispersing sex is not ubiquitous, it is reasonable that a greater proportion of members of this sex will disperse compared with members of the species' philopatric sex [80–83]. We can, therefore, use the frequency of matches versus mismatches between the ecologically expected attribution (based on species-level sex-biases) and the assigned attribution (from the isotopic model) to determine the degree of reliability for a given modelling approach, even knowing that some expected attributions may be inaccurate for some individuals.

To test the efficacy of the standard archaeological approach, we defined minimum and maximum 'local' $^{87}Sr/^{86}Sr$ thresholds based on plant leaves collected on the Toro Formation for Kanyawara primates and plant leaves collected on the 'unidentified radiometric anomaly' area for Ngogo primates. We compared the $^{87}Sr/^{86}Sr_{enamel}$ of primates collected from each area with these ranges. Individuals with $^{87}Sr/^{86}Sr_{enamel}$ falling outside of the local area were assigned a 'non-local/ disperser' attribution while individuals with $^{87}Sr/^{86}Sr_{enamel}$ falling within the local area were assigned a 'local/philopatric' attribution. We then calculated the number of mismatches between the expected and assigned attributions. We assessed efficacy at the individual level (how many individuals had assigned attributions which matched their expected attributions) and at the species level (if a greater proportion of dispersing sex individuals were assigned 'non-local/disperser' attributions based on the isotopic data compared with the proportion of philopatric individuals). We repeated this procedure using two additional 'local' definitions instead of geologic formation association: one based on the park area within which the plant sample was collected, and one based on statistically identified clusters (see Results for more information on the formation of these cluster definition). The plants collected within the Kanyawara park area ranged from 0.7078 to 0.7189 and those collected within the Ngogo park area ranged from 0.7129 to 0.7339. When using the statistically identified clusters, we compared the Kanyawara primates with the minimum and maximum $^{87}Sr/^{86}Sr$ from the Northern cluster (0.7078–0.712) and Ngogo primates with the minimum and maximum $^{87}Sr/^{86}Sr$ from the combined Southern and Ngogo cluster (0.7129–0.7339), as the park area falls equally on both cluster regions.

Next, we tested the efficacy of our new offset method. To determine offsets between juvenile and adult strontium signals for each individual, we first calculated the difference between the strontium isotope ratio of an individual's tooth enamel and bone ($^{87}Sr/^{86}Sr_{enamel-bone}$). Then, we calculated the offset between an individual's tooth enamel and the mean of the local environment, as defined by the average $^{87}Sr/^{86}Sr$ of the plants within the isotopic cluster on which the remains were recovered ($^{87}Sr/^{86}Sr_{enamel-environment}$). The local environmental mean with which each individual's tooth enamel $^{87}Sr/^{86}Sr$ was compared changed

based on the cluster(s) underlying the park area on which the individual was collected (see Results for detailed information on the cluster(s) underlying each park area). Finally, we tested the efficacy of our novel offset method when defining the local environment as the average $^{87}Sr/^{86}Sr$ of plants gathered within the boundaries of each park area's underlying geologic formation, as assessed using a geologic map from the Ugandan Society of Geology and Mines [56], instead of the statistically identified isotopic clusters. As with the standard archaeological method, we assessed efficacy at the individual and species level for each offset. For both offset methods, we transformed each offset into its absolute value for analysis because we are interested in the *magnitude* of the difference between the juvenile and adult signatures, rather than the directionality.

Due to small sample sizes, which is also a common problem in palaeontological studies, we could not use standard frequentist statistics to investigate differences in offset values. Instead, we used parametric bootstrapping, which uses the mean and standard deviation of offset values collected for each sex in each primate group to draw normal distributions. We randomly sampled 100 data points from these distributions and compared the offsets between the sexes for each species using Student's *t*-tests. We repeated this simulation 10 000 times and determined the percentage of times that the *p*-value was significant ($p < 0.05$) (see the electronic supplementary material). Parametric bootstrapping guards against under-estimations of true population variance and spurious fine structures that non-parametric methods can propagate. By iterating the bootstrap 10 000 times, we could go beyond a frequentist interpretation to assess the probability of retrieving a significant *p*-value. This method was considered effective if such a probability was greater than 90%.

Finally, we categorically classified each individual as either 'large offset' or 'small offset' for both $^{87}Sr/^{86}Sr_{enamel-bone}$ and $^{87}Sr/^{86}Sr_{enamel-environment}$. Large offset individuals were those with offsets above the species' mean offset value; small offset individuals were those with offsets below the species' mean. In an idealized dataset, large offset individuals would be non-local/dispersers, while small offset individuals would be local/philopatric. We calculated the proportion of individuals with an assigned attribution (local/non-local) that matched their expected attribution based on sex and species dispersal patterns. Finally, we compared the proportions of large and small offset individuals between sexes to determine if a higher proportion of large offsets came from the dispersing sex. This allowed us to assess if the method could accurately predict species-level sex-biases in dispersal patterns.

# 3. Results

## 3.1. Environmental

Leaf samples show extensive and patterned isotopic variability across Kibale National Park (figures 1 and 2a). The Toro Formation (mean $^{87}Sr/^{86}Sr = 0.7097$) and Buganda Formation (mean $^{87}Sr/^{86}Sr = 0.71135$) did not differ isotopically (Wilcox Mann–Whitney *U*-test, $U = 157$, $p > 0.1$), and samples collected on the radiometric anomaly (mean $^{87}Sr/^{86}Sr = 0.72005$) included many values considerably more radiogenic than on either formation. When samples are sorted by nearest park area association, there is considerable overlap between the isotopic profiles (figure 2b). Kanyawara and Sebitole are indistinguishable (Wilcox Mann–Whitney *U*-test, $U = 38$, $p > 0.1$) as are Ngogo and Kanyanchu (Wilcox Mann–Whitney *U*-test, $U = 66$, $p > 0.1$). Distinctions between more radiogenic values in the southeast of the park and less radiogenic values in the northwest of the park are more clearly delineated using park area associations than when using geologic formation associations. However, the wide variability captured by the Ngogo and Kanyanchu park areas, in particular, and the isotopic overlap between the park areas makes distinguishing residence within versus migration between them difficult.

Hierarchical cluster analysis identified three isotopically unique clusters in Kibale National Park: a Northern cluster including the Kanyawara and Sebitole park areas (mean $^{87}Sr/^{86}Sr = 0.7094 \pm 0.001$, circles in figure 1), a Southern cluster including the Kanyanchu park area as well as a portion of the Ngogo park area (mean $^{87}Sr/^{86}Sr = 0.7169 \pm 0.0026$, triangles in figure 1), and the Ngogo cluster comprising an extremely geographically constrained underlying the remainder of the Ngogo park area (mean $^{87}Sr/^{86}Sr = 0.7273 \pm 0.004$, squares in figure 1) (Kruskal–Wallace rank sum test, $\chi^2 = 51.712$, d.f. = 2, $p < 0.001$, figure 2c). These clusters do not neatly correspond to underlying geologic formations when compared with existing geologic maps. The Northern cluster includes portions of the Buganda and Toro Formation. The Southern cluster includes parts of the Buganda and Toro formations as well as a portion of the unidentified radiometric anomaly [56]. The small Ngogo cluster falls exclusively on the radiometric anomaly area but does not cover it in its entirety.

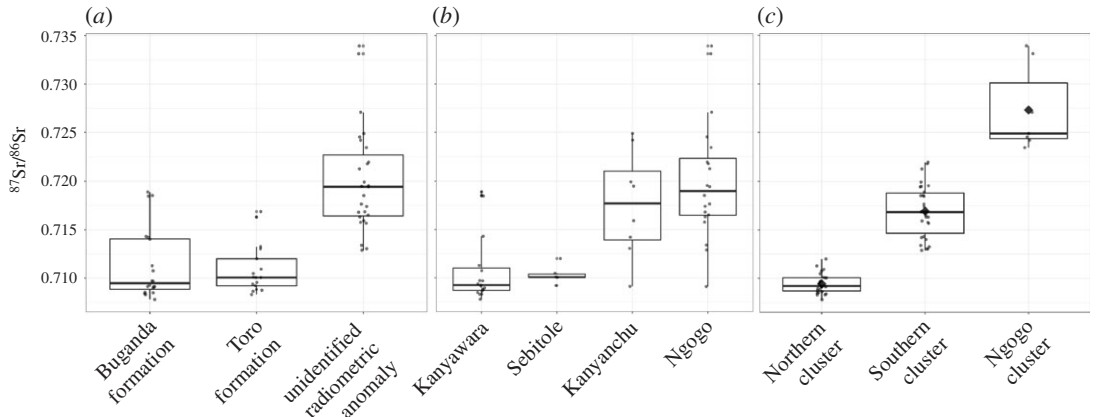

**Figure 2.** Boxplots of strontium isotope ratios from plant samples grouped by (*a*) the geologic formation on which they were collected, (*b*) the nearest park area with which they can be associated (see electronic supplementary material, table S2) and (*c*) the statistical cluster to which they were assigned through cluster analysis. Strontium isotope ratios of plant samples by isotopic clusters—'Northern': 0.7078–0.712, mean = 0.7094 ± 0.0010; 'Southern': 0.7129–0.7219, mean = 0.7169 ± 0.0026; 'Ngogo' cluster: 0.7234–0.7339, mean = 0.7273 ± 0.0043.

## 3.2. Local versus non-local (standard archaeological method)

The $^{87}Sr/^{86}Sr_{enamel}$ of Kanyawara primates were first compared with the minimum and maximum values of plant samples collected on the northern Toro Formation (0.7083–0.712), the geologic formation on which Kanyawara sits according to available geological maps [35]. The $^{87}Sr/^{86}Sr_{enamel}$ of Ngogo primates were compared with the minimum and maximum values of plant samples collected on the unidentified radiometric anomaly area (0.7129–0.7339), on which Ngogo sits according to the same maps. Fifty-six per cent of individual Kibale primates were assigned their expected attribution (local/ philopatric or non-local/dispersing) using this method (table 2 and figure 3), including 54% of individuals from Kanyawara 56% of individuals from Ngogo. There is a strong bias towards matching expected/assigned attributions for locals/philopatric individuals and mismatched expected/assigned attributions for non-locals/dispersers: 76% of expected local/philopatric individuals and only 36% of expected non-local/dispersing individuals were assigned their expected attribution. At the species level, three out of five species (female-philopatric chimpanzees, female-philopatric red colobus monkeys and male-philopatric olive baboons) had a higher proportion of individuals of the dispersing sex fall outside of the local minima and maxima, which would lead researchers to infer the correct sex-bias in dispersal patterns if this were an unknown assemblage. These results were identical when statistically identified clusters were used instead of geologic boundaries to define local environmental areas, with the exception of a single female chimpanzee from Kanyawara (KFB 150) who was correctly identified as a non-local/disperser using the geologic boundaries but not when using the statistical clusters. When park area associations were used to define the local area instead of geologic formations, the results from Ngogo remained unchanged. At Kanyawara, every individual was classified as 'local' with the exception of one male baboon (KFB 181(b)), the only correctly identified non-local/disperser from the site, and three male chimpanzees (KCP 1, KFB 150 and KFB 154), who were misclassified as non-locals/dispersers. This change was driven primarily by three anomalously radiogenic plant samples collected within the northeast corner of Kanyawara (figure 1); if these are omitted, then the results remain unchanged from those seen when using geologic formation associations.

## 3.3. Enamel–bone offsets (bootstrapping)

Nearly all species had larger $^{87}Sr/^{86}Sr_{enamel-bone}$ offset for the expected dispersing sex (figure 4*a*). Olive baboons and guenons had significantly larger $^{87}Sr/^{84}Sr_{enamel-bone}$ offsets in the expected dispersing sex compared with the expected philopatric sex ($p < 0.05$) in 100% of bootstrapped simulations, black and white colobus monkeys in 92% of simulations, and red colobus monkeys in 99% of simulations. Chimpanzees, however, had significantly larger offsets in the expected dispersing sex in only 5% of simulations (figure 5*a* and table 3).

**Table 2.** Comparison of expected and assigned 'local/philopatric' and 'non-local/dispersing' attributions of Kibale Primates. Expected attributions are based on biological sex and species dispersal patterns. Assigned attributions are based on $^{87}Sr/^{86}Sr_{enamel}$ in comparison with local isotopic minima and maxima as determined by plant $^{87}Sr/^{86}Sr$ collected on the geologic formation underlying each park area (standard archaeological method).

| species | no. of samples | expected no. of locals/philopatric sex[a] | % of expected locals assigned 'local' attribution[b] | expected no. of non-locals/dispersing sex[b] | % of non-locals assigned 'non-local' attribution[c] | % of total samples assigned to their expected attribution (local/non-local) | dispersing sex correctly identified for this species?[d] |
|---|---|---|---|---|---|---|---|
| chimpanzees | 17 | 11 | 7 (64%) | 6 | 4 (67%) | 11 (65%) | yes |
| red colobus monkey | 18 | 8 | 8 (100%) | 10 | 2 (20%) | 10 (56%) | yes |
| black and white colobus monkeys | 6 | 2 | 0 | 4 | 2 (50%) | 2 (33%) | no |
| olive baboons | 9 | 4 | 4 (100%) | 5 | 2 (40%) | 6 (67%) | yes |
| guenons | 7 | 4 | 3 (75%) | 3 | 0 | 3 (43%) | no |
| total | 57 | 29 | 22 (76%) | 28 | 10 (36%) | 32 (56%) | 60% of species |

[a]The expected number of philopatric individuals and dispersing individuals for each species is based on the sex of each individual and the overall pattern for sex-biased dispersal within that species.

[b]Any individual falling within the local environment isotopic minima/maxima was assigned a local/philopatric attribution.

[c]Any individual falling outside of local environment isotopic minima/maxima was assigned a non-local/dispersing attribution.

[d]The dispersing sex was considered 'correctly identified' if a greater proportion of the dispersing sex fell outside of the local isotopic minima and maxima compared with the proportion of the philopatric sex falling outside the local isotopic minima and maxima.

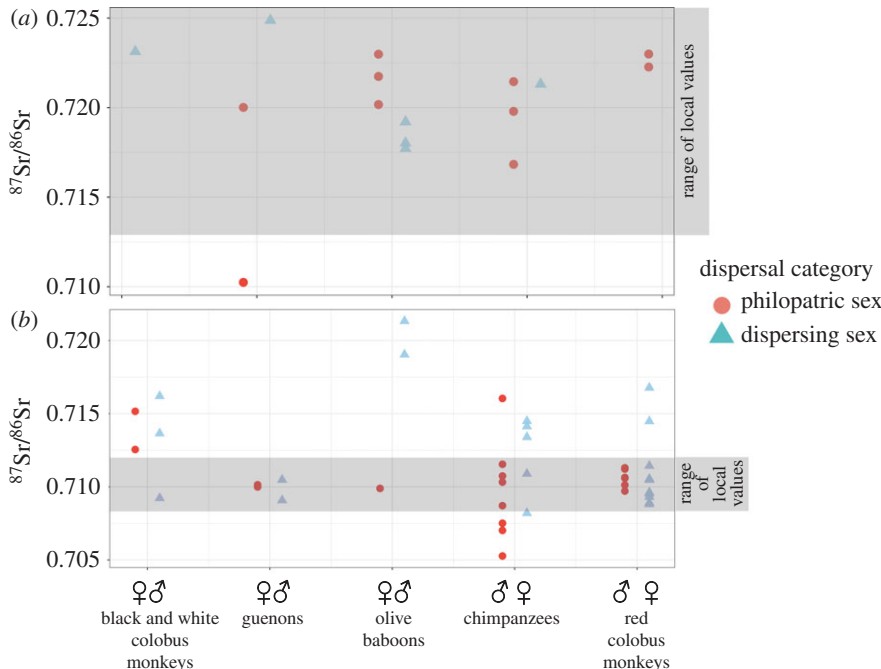

**Figure 3.** $^{87}Sr/^{86}Sr$ of tooth enamel: (*a*) Ngogo fauna; (*b*) Kanyawara fauna. Grey bands show minimum and maximum values of local plants: for Kanyawara, the minimum and maximum values of plant samples collected on the northern Toro Formation (0.70832–0.712); for Ngogo, the minimum and maximum values of plant samples collected on the unidentified radiometric anomaly area (0.71289–0.73392). Other study areas (Kanyanchu, Sebitole) not included due to small sample sizes.

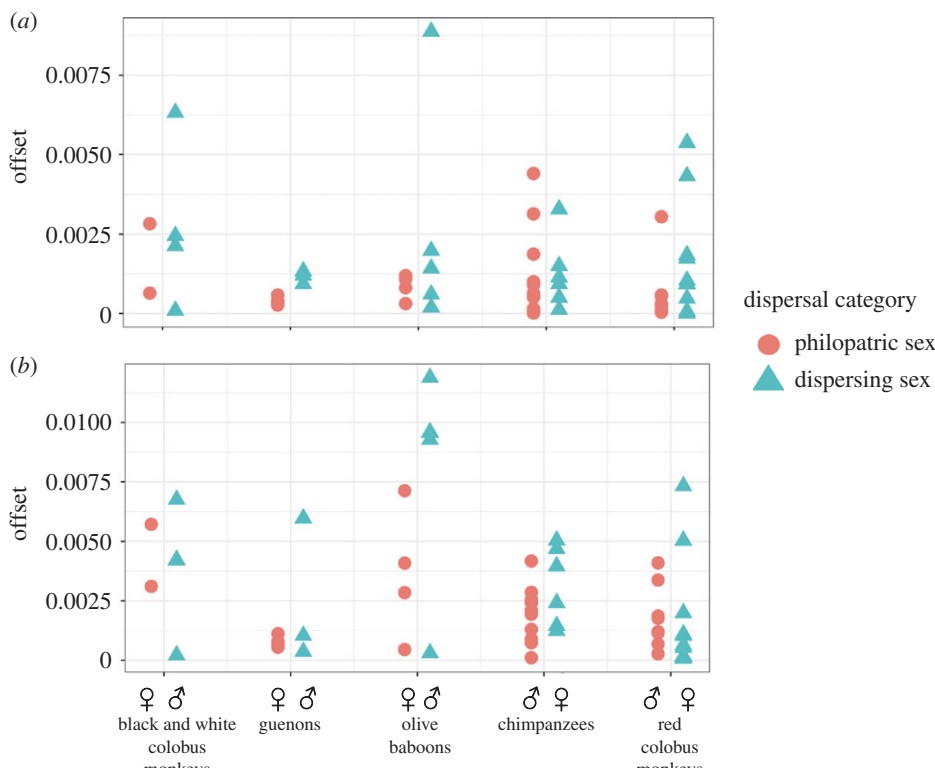

**Figure 4.** Offsets between tooth and bone strontium isotope ratios (*a*) and tooth and local environment strontium isotope ratios (*b*) for individuals of each species of primate by sex. Offset values are shown as absolute values in order to compare the magnitude of each offset.

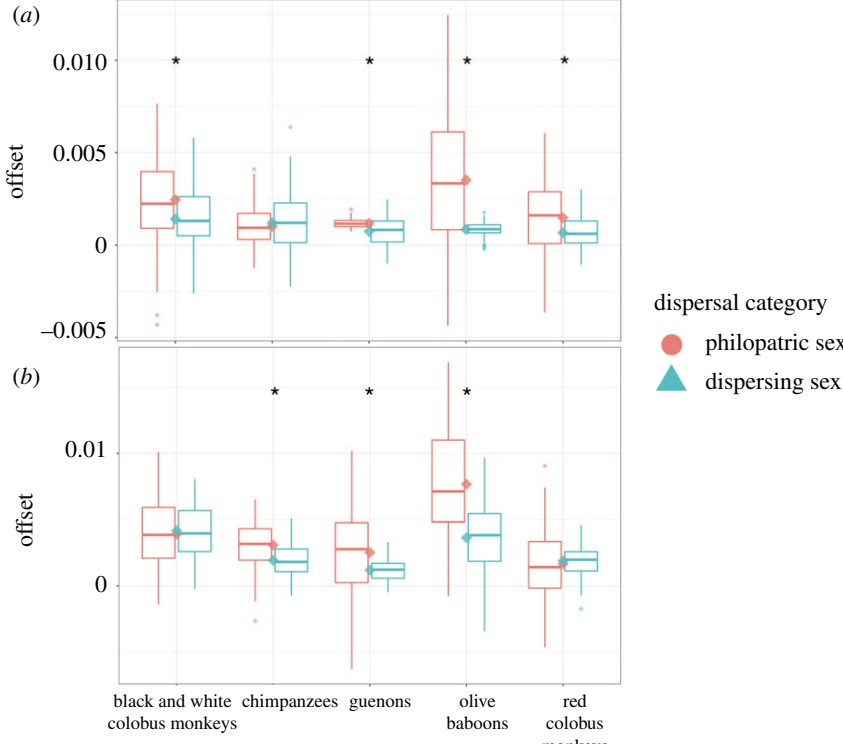

**Figure 5.** Boxplots illustrating bootstrapped comparison of $^{87}$Sr/$^{86}$Sr offsets in enamel and bone (*a*), and enamel and environment (*b*) between the expected dispersing and philopatric sex for each species of primate. Bars show median offset for each sex; diamonds show mean offset. * indicates a significant difference ($p < 0.05$) between the sexes in greater than 90% of bootstrap simulations.

**Table 3.** Proportion of bootstrap simulations in which there was a significant difference ($p < 0.05$, Student's *t*-test) in offset values ($^{87}$Sr/$^{86}$Sr$_{enamel-bone}$ and $^{87}$Sr/$^{86}$Sr$_{enamel-environment}$) between males and females.

| species | $^{87}$Sr/$^{86}$Sr$_{enamel-bone}$ (%) | $^{87}$Sr/$^{86}$Sr$_{enamel-environment}$ (%) |
|---|---|---|
| chimpanzees | 5 | 98 |
| red colobus monkey | 99 | 100 |
| black and white colobus monkeys | 92 | 39 |
| olive baboons | 100 | 100 |
| guenons | 100 | 96 |

## 3.4. Enamel–bone offsets (categorical classification)

Individuals with large offsets (greater than the species' mean offset) were assigned a 'non-local/disperser' attribution; individuals with small offsets (a lower offset than the species' mean) were assigned a 'local/philopatric' attribution. Sixty-seven per cent of individual primates were assigned an attribution which matched their expected attribution based on sex and species (67% of Kanyawara individuals and 67% of Ngogo individuals). As with the standard archaeological method, there was a bias in favour of matching expected/assigned attributions for local/philopatric sex individuals (86% matching) and mismatched expected/assigned attributions for non-local/dispersing sex individuals (only 46% matching) (table 4). This method was nonetheless more accurate than the standard archaeological method. At the species level, red colobus monkeys, olive baboons and guenons all had a greater proportion of dispersing sex individuals with large offsets compared with the proportion of philopatric sex individuals. Chimpanzees and black and white colobus monkeys had approximately equal proportions of each sex fall above the group mean.

**Table 4.** Comparison of expected and assigned 'local/philopatric' versus 'non-local/dispersing' attribution of Kibale primates. Expected attributions are based on biological sex and species dispersal patterns. Assigned attributions are determined by $^{87}Sr/^{86}Sr_{enamel-bone}$ offsets, categorized as 'small'[b] or 'large'[c].

| species | no. of samples | expected no. of locals/philopatric sex[a] | % of expected locals assigned 'local' attribution[b] (%) | expected no. of non-locals/dispersing sex[a] | % of non-locals assigned 'non-local' attribution[c] (%) | % of total samples assigned to their expected attribution (local/non-local) (%) | dispersing sex correctly identified for this species?[d] |
|---|---|---|---|---|---|---|---|
| chimpanzees | 17 | 11 | 72 | 6 | 33 | 59 | inconclusive |
| red colobus monkey | 18 | 8 | 88 | 10 | 40 | 61 | yes |
| black and white colobus monkeys | 6 | 2 | 50 | 4 | 50 | 50 | inconclusive |
| olive baboons | 9 | 4 | 100 | 5 | 40 | 67 | yes |
| guenons | 7 | 4 | 100 | 3 | 100 | 100 | yes |
| total | 57 | 29 | 86 | 28 | 46 | 67 | 60% of species |

[a]The expected number of philopatric individuals and dispersing individuals for each species is based on the sex of each individual and the overall pattern for sex-biased dispersal within that species.
[b]Any individual with an offset at or below the mean offset for the species (small offset) is assigned a local/philopatric attribution.
[c]Any individual with an offset above the mean offset for the species (large offset) is assigned a non-local/dispersing sex attribution.
[d]The dispersing sex was considered 'correctly identified' if a greater proportion of the dispersing sex had large offsets compared with the proportion of the philopatric sex with large offsets.

## 3.5. Enamel–environment offsets (bootstrapping)

We compared the enamel from each individual primate with the mean of the isotopic cluster(s) underlying the park area on which the primate skeletons were recovered. Enamel from Kanyawara and Sebitole individuals were compared with the Northern cluster $^{87}Sr/^{86}Sr$ mean ($0.7094 \pm 0.0010$), Kanyanchu individuals to the Southern cluster $^{87}Sr/^{86}Sr$ mean ($0.7169 \pm 0.0026$) and Ngogo individuals to the mean $^{87}Sr/^{86}Sr$ of the combined Ngogo and Southern clusters ($0.7189 \pm 0.005$), as the Ngogo park area falls on top of both of these clusters (figure 1), so it is necessary to assume that primates living there are equally likely to forage on either, and more likely, on both clusters during their lifetimes. Species with larger home ranges trended towards larger $^{87}Sr/^{86}Sr_{enamel-environment}$ offsets for the dispersing sex (figure 4b). Parametric bootstrapping methods showed significantly larger offsets in the expected dispersing sex ($p < 0.05$) in 98% of simulations for chimpanzees, 100% of simulations for olive baboons and 96% of simulations for guenons, but only 5% of simulation for red colobus monkeys and 39% for black and white colobus monkeys (figure 5b and table 3).

## 3.6. Enamel–environment offsets (categorical classification)

At the individual level, this method assigned non-local/dispersing and local/philopatric attributions that matched the expected attribution in 65% of individuals (68% of individuals from Ngogo and 62% of individuals from Kanyawara), including 79% of philopatric sex individuals and 50% of dispersing sex individuals (table 5). At the species level, all five species had a higher proportion of dispersing sex individuals fall above the species' mean $^{87}Sr/^{86}Sr_{enamel-environment}$ offset than individuals of the expected philopatric sex. Table 6 summarizes these results.

## 3.7. Enamel–environment offsets (using underlying geology to define the local environment)

To determine if defining the local environment based on statistical clusters is more reliable at identifying dispersing and philopatric primates than defining it based on underlying geology, we calculated $^{87}Sr/^{86}Sr_{enamel-environment}$ offsets comparing individuals from Kanywara and Sebitole with the mean $^{87}Sr/^{86}Sr$ of plants collected on the Toro Formation (0.7097), individuals from Ngogo with the mean $^{87}Sr/^{86}Sr$ of plants collected on the unidentified radiometric anomaly (0.7200), and the individual from Kanyanchu with the mean $^{87}Sr/^{86}Sr$ of plants collected on the Buganda Formation (0.7113). Parametric bootstrapping showed significantly larger offsets in the expected dispersing sex ($p < 0.05$) in 24% of simulations for chimpanzees, 55% of simulations for black and white colobus monkeys, 40% of simulations for red colobus monkeys, 81% of simulations for olive baboons and 97% of simulations for guenons. Categorically, this method assigned non-local/dispersing and local/philopatric attributions that matched the expected attribution in 47% of individuals, including 34% of philopatric sex individuals and 61% of dispersing sex individuals (table 7).

# 4. Discussion

Our goal in this study was to determine the degree to which strontium isotope ratios can differentiate between philopatric and dispersing individuals in primate communities, and what method and analytical approach most reliably identified sex-bias in dispersal patterns. In particular, we wanted to compare the efficacy of a standard approach taken in the archaeological literature with our novel offset values approach. Our approach is critically different for two primary reasons. First, while the standard method creates binary categories (local, non-local) based on environmental minima and maxima, our new method quantifies the amount of variation from a local mean to assess the likelihood of an individual being local or non-local based on the degree of offset relative to other members of the same species. Differences between individuals belonging to species with home range sizes or dispersal distances smaller than the previously designated local/non-local areas can still be identified, whereas they are overlooked using the standard approach. Second, we use hierarchical cluster analysis to generate geochemically unique 'local' areas rather than relying on geologic formations or other predetermined boundaries to group $^{87}Sr/^{86}Sr_{environment}$ samples. Use of these statistical clusters provides a higher degree of certainty whether or not a given $^{87}Sr/^{86}Sr_{enamel}$ ratio is indeed local to an area even when geologic or geographical areas overlap isotopically (such as Swartkrans and Sterkfontein caves in South Africa [9]), and/or in which sufficiently detailed geologic

**Table 5.** Comparison of expected and assigned 'local/philopatric' versus 'non-local/dispersing' attribution of Kibale primates. Expected attributions are based on biological sex and species dispersal patterns. Assigned attributions are determined by $^{87}Sr/^{86}Sr_{enamel-environment}$ offsets, categorized as 'small'[b] or 'large'[c] and using statistically identified clusters to define the local environment.

| species | no. of samples | expected no. of locals/philopatric sex[a] | % of expected locals assigned 'local' attribution[b] (%) | expected no. of non-locals/dispersing sex[a] | % of non-locals assigned 'non-local' attribution[c] (%) | % of total samples assigned to their expected attribution (local/non-local) (%) | dispersing sex correctly identified for this species?[d] |
|---|---|---|---|---|---|---|---|
| chimpanzees | 17 | 11 | 82 | 6 | 50 | 71 | yes |
| red colobus monkey | 18 | 8 | 75 | 10 | 30 | 50 | yes |
| black and white colobus monkeys | 6 | 2 | 50 | 4 | 75 | 67 | yes |
| olive baboons | 9 | 4 | 75 | 5 | 80 | 78 | yes |
| guenons | 7 | 4 | 100 | 3 | 33 | 71 | yes |
| total | 57 | 29 | 79 | 28 | 50 | 65 | 100% of species |

[a]The expected number of philopatric individuals and dispersing individuals for each species is based on the sex of each individual and the overall pattern for sex-biased dispersal within that species.
[b]Any individual with an offset at or below the mean offset for the species (small offset) is assigned a local/philopatric attribution.
[c]Any individual with an offset above the mean offset for the species (large offset) is assigned a non-local/dispersing sex attribution.
[d]The dispersing sex was considered 'correctly identified' if a greater proportion of the dispersing sex had large offsets compared with the proportion of the philopatric sex with large offsets.

**Table 6.** Summary table: were methods able to correctly identify the dispersing sex for each primate species?

| | standard archaeological method (local/non-local) | | offset methods | | | |
| | | | $^{87}Sr/^{86}Sr_{enamel-bone}$ offset | | $^{87}Sr/^{86}Sr_{enamel-environment}$ offset | |
| | >90% of individuals assigned expected attribution?[a] | categorical classification effective?[b] | bootstrapped comparison of means effective?[c] | categorical classification effective?[b] | bootstrapped comparison of means effective?[c] | categorical classification correctly effective?[b] |
| --- | --- | --- | --- | --- | --- | --- |
| chimpanzees | no | yes | no | yes | yes | yes |
| red colobus monkeys | no | yes | yes | yes | no | yes |
| black and white colobus monkeys | no | no | yes | no | no | yes |
| olive baboons | no | yes | yes | yes | yes | yes |
| guenons | no | no | yes | yes | yes | yes |

[a]The expected attribution (local or non-local) is based on the sex of the individual and the known sex-bias in dispersal patterns for that species.

[b]The categorical classification method was designated as 'effective' if a higher proportion of dispersing sex individuals fell outside the local isotopic minima and maxima (for the standard archaeological method) or were classified as large offset (for the offset methods) compared with the proportion of philopatric sex individuals.

[c]The bootstrap comparison of means was designated as 'effective' if over 90% of simulations returned a significant $p$-value ($p < 0.05$) when comparing offset values between males and females, with the dispersing sex having a significantly higher offset than the philopatric sex.

**Table 7.** Comparison of expected and assigned 'local/philopatric' versus 'non-local/dispersing' attribution of Kibale primates. Expected attributions are based on biological sex and species dispersal patterns. Assigned attributions are determined by $^{87}Sr/^{86}Sr_{enamel-environment}$ offsets categorized as 'small'[b] or 'large'[c] and using underlying geologic formations to define the local environment.

| species | no. of samples | expected no. of locals/philopatric sex[a] | % of expected locals assigned 'local' attribution[b] (%) | expected no. of non-locals/dispersing sex[a] | % of non-locals assigned 'non-local' attribution[c] (%) | % of total samples assigned to their expected attribution (local/non-local) (%) | dispersing sex correctly identified for this species?[d] |
|---|---|---|---|---|---|---|---|
| chimpanzees | 17 | 11 | 36 | 6 | 50 | 41 | no |
| red colobus monkeys | 18 | 8 | 50 | 10 | 70 | 61 | yes |
| black and white colobus monkeys | 6 | 2 | 50 | 4 | 50 | 50 | inconclusive |
| olive baboons | 9 | 4 | 0 | 5 | 60 | 33 | no |
| guenons | 7 | 4 | 25 | 3 | 67 | 43 | no |
| total | 57 | 29 | 34 | 28 | 61 | 47 | 20% of species |

[a]The expected number of philopatric individuals and dispersing individuals for each species is based on the sex of each individual and the overall pattern for sex-biased dispersal within that species.
[b]Any individual with an offset at or below the mean offset for the species (small offset) is assigned a local/philopatric attribution.
[c]Any individual with an offset above the mean offset for the species (large offset) is assigned a non-local/dispersing sex attribution.
[d]The dispersing sex was considered 'correctly identified' if a greater proportion of the dispersing sex had large offsets compared with the proportion of the philopatric sex with large offsets.

maps are not available (such as Kibale National Park), provided that such areas have spatially patterned isotopic heterogeneity.

At the individual level, using our known sample of primates from Kibale National Park, Uganda, the standard archaeological method assigned 36% of primates expected to be non-local/dispersers based on species and sex a matching attribution based on isotopic data. The $^{87}Sr/^{86}Sr_{enamel-bone}$ offset method assigned 46% of expected non-local/dispersing individuals a matching attribution, and the $^{87}Sr/^{86}Sr_{enamel-environment}$ offset method using statistical isotopic clusters assigned 50% of non-local/dispersing individuals a matching attribution. The $^{87}Sr/^{86}Sr_{enamel-environment}$ offset method using geologic formations to define the local environment assigned 34% of expected non-local/dispersing primates a matching attribution. At the species level, the standard archaeological method indicated the expected sex-bias in dispersal patterns in three of five species, but indicated the opposite of the expected sex-bias in two out of five species. The $^{87}Sr/^{86}Sr_{enamel-bone}$ offset method indicated the expected sex-bias in dispersal in three of five species using categorical classification based on large and small offsets, with inconclusive results in two of five species. The $^{87}Sr/^{86}Sr_{enamel-environment}$ offset method using statistical isotopic clusters indicated the expected sex-biases in dispersal in all five species tested using categorical classification based on large and small offsets, and in only one species when geologic formations were used instead of statistically identified clusters (tables 2, 4, 5 and 7).

While no method was perfect, the $^{87}Sr/^{86}Sr_{enamel-environment}$ offset method using statistical isotopic clusters and categorical classification of large and small offsets was the most accurate and effective approach for identifying sex-biases in dispersal examined in this study, particularly for species with large home ranges. Within the $^{87}Sr/^{86}Sr_{enamel-environment}$ offset method, the categorical classification approach was more reliable than the bootstrap approach. This may suggest that the issue with the comparison of means could be a product of small sample sizes. Proportionately calculating how many individuals of each sex have large offsets relative to the rest of the species is a more robust way to handle data of this sort when sample sizes are small.

Trends in the reliability of each method paint a more complex picture of when each method is most applicable. For example, for primates with large home ranges and dispersal distances (those meeting or exceeding the size of local isotopically homogeneous areas), comparing mean $^{87}Sr/^{86}Sr_{enamel-environment}$ offsets using statistical isotopic clusters (with large sample sizes) or proportions of each sex with large offset values from this proxy (for smaller sample sizes) was most reliable. For primates with small dispersal distances, comparing $^{87}Sr/^{86}Sr_{enamel-bone}$ offsets was also a reliable approach. However, bone is also the least reliable material to rely on for palaeontological reconstructions due to high susceptibility to diagenesis.

The $^{87}Sr/^{86}Sr_{enamel-environment}$ offsets method using the underlying geologic formations to define local environments had the lowest overall matching rate of all methods tested in this study, particularly for matching expected and assigned attributions for assumed philopatric sex individuals (47% matching, as compared with 60%, 65% and 67% in the standard archaeological approach, $^{87}Sr/^{86}Sr_{enamel-bone}$, and $^{87}Sr/^{86}Sr_{enamel-environment}$ using statistical isotopic clusters, respectively). However, the matching rate for assumed dispersing sex individuals is higher than that of the offset methods using the statistical clusters (61% as compared with 46% for $^{87}Sr/^{86}Sr_{enamel-bone}$ and 50% for $^{87}Sr/^{86}Sr_{enamel-environment}$ using statistical isotopic clusters) while the matching rate for assumed philopatric sex individuals is substantially lower (34% as compared with 86% for $^{87}Sr/^{86}Sr_{enamel-bone}$ and 79% for $^{87}Sr/^{86}Sr_{enamel-environment}$ using statistical isotopic clusters). This is most likely because the boundaries created by the geologic formations as shown on existing geologic maps doing a poor job separating areas of significant isotopic differences, as suggested by the lack of statistical differences between the $^{87}Sr/^{86}Sr$ of plants collected on the Toro and Buganda Formations (figure 2). Because of this, defining the 'local environment' by grouping isotopic data together by geologic formation boundaries does not reflect true local isoscapes, leading to a high mismatch rate for assumed-local individuals. The fact that geologic formation boundaries in Kibale do not reflect isotopic boundaries could be due to myriad factors, including insufficient detail available on geologic maps of the area, high isotopic heterogeneity, or heavy soil weathering; similar high variability in bioavailable strontium has been recorded elsewhere in East Africa as well [42].

There are many potential pitfalls when using data from modern environments to study mobility in the past, particularly in hominin contexts. In much of East Africa, home to some of the most detailed hominin fossil material, volcanic activity over the past 5 Myr has dramatically changed the geologic landscape. Changes in the hydrology of the East African Rift [83], including the formidable palaeo-Omo River, means many of these hominin fossil sites that were probably lacustrine or gallery forests today are dry and arid. Other studies have demonstrated the potentially dramatic shift in bioavailable strontium between riparian

and non-riparian environments [24,52]. The absence of such water systems in modern habitats can lead to differences in the isotopic ratios of modern and fossil material [84]. Robust, careful sampling procedures are, therefore, a necessary precursor to any reconstruction of past mobility [42]. Analytically, the authors recommend using the categorical classification of $^{87}Sr/^{86}Sr_{enamel-environment}$ offsets in fossil contexts because of its reasonably high reliability and lack of dependence on bone samples, which need to be treated with great care and special attention paid to diagenesis. In areas with ancient geologies or insufficiently detailed geologic maps, the authors recommend using statistical clustering of environmental $^{87}Sr/^{86}Sr$ to define the local environment in these calculations. Measuring and quantifying environmental isotopic clusters for a fossil context could be approached in numerous ways. The half-life of rubidium-87, which decays into strontium-87, is 48.8 Myr, so modern plant samples could be used as proxies for palaeo isoscapes [9] under conditions in which surface geology has been consistent since the time period of interest. There are other data proxies that could substitute for local environmental estimates as well. Fossils of small rodents and other micromammals can establish similar local environmental profiles to botanical samples [24,85], and botanical and microfaunal remains can be used in tandem to develop detailed isoscapes [42]. More work of this kind in hominin palaeoenvironments would greatly benefit the state of current research. Investigations into the potential of other proxies for local palaeoenvironments would also benefit future researchers. For example, while researchers have used carbon and oxygen isotopic records from palaeosols for decades for palaeoenvironmental reconstructions [86,87], the potential for strontium isotopes in palaeosols have not been thoroughly examined. Further methodological work would be needed before we could confidently use this method.

## 4.1. Potential sources of assignment error

The methods used here to test the reliability of different approaches to reconstructing sex-biases in philopatry rely on classifying primates as either 'local/philopatric' or 'non-local/disperser' based on the biological sex and species for each individual. There are numerous factors that could impact the results based on this assumption that warrant further investigation: variations in the age at which the sampled teeth mineralize, differences in home range size and variability in the fidelity of sex-biased dispersal rates.

### 4.1.1. Age at which sampled teeth mineralize

We sampled the outer enamel from first molars whenever present to ensure that mineralization of tooth enamel occurred prior to any dispersal and with minimal maternal effects [13]; however, for six individuals in the collection, first molars were not available. We were able to sample earlier forming teeth in two of these individuals (a first incisor in KFB 122, a male redtail monkey, and a first premolar in KFB 171, a male red colobus monkey) and second molars in the other four (KFB 57 and KFB 111, both female red colobus monkeys, and KFB 154 and KCP 9, both male chimpanzees) (electronic supplementary material, table S1). Second molars erupt before age eight in chimpanzees [88]. While variable, female chimpanzees typically do not disperse until after reaching sexual maturity around age thirteen [76–78]. Data on eruption times for colobus monkeys are sparser, but modelling based on brain size by Smith *et al.* [88] suggests that the final permanent tooth (the third molar) will erupt by 3.7 years. Dispersal in colobines is rare before the age of three [89,90], so it is unlikely (although not impossible) for dispersal to occur before the mineralization of the second permanent molar. While we cannot rule out that tooth selection caused these individuals' $^{87}Sr/^{86}Sr_{enamel}$ measurement to come from post-dispersal, it is reassuring that within our dataset, the offsets calculated for these individuals are not statistical outliers and only in one instance represent the minimum or maximum observed value (chimpanzee $^{87}Sr/^{86}Sr_{enamel-bone}$ range 0–0.0044, KFB 154 $^{87}Sr/^{86}Sr_{enamel-bone}$ offset = 0, KCP 9 $^{87}Sr/^{86}Sr_{enamel-bone}$ offset = 0.0005; chimpanzee $^{87}Sr/^{86}Sr_{enamel-environment}$ offset range = 0.0001–0.005, KFB 154 $^{87}Sr/^{86}Sr_{enamel-environment}$ offset = 0.0042, KCP 9 $^{87}Sr/^{86}Sr_{enamel-environment}$ offset = 0.0013; red colobus $^{87}Sr/^{86}Sr_{enamel-bone}$ range 0–0.0054, KFB 57 $^{87}Sr/^{86}Sr_{enamel-bone}$ offset = 0.0001, KFB 111 $^{87}Sr/^{86}Sr_{enamel-bone}$ offset = 0.0009; red colobus $^{87}Sr/^{86}Sr_{enamel-environment}$ offset range = 0.0001–0.0073, KFB 57 $^{87}Sr/^{86}Sr_{enamel-enviornment}$ offset = 0.0002, KFB 111 $^{87}Sr/^{86}Sr_{enamel-environment}$ offset = 0.005).

### 4.1.2. The impact of home range size

The observed success or failure of the tooth–environment and tooth–bone offset proxies is potentially also related to a combination of home range size and life-history variables (table 1). Chimpanzees

have highly variable home ranges based on their habitat [91,92]; in forests such as Kibale National Park, researchers measured home range sizes between 28–41 km$^2$, which is large relative to the size of the isotopic clusters [63,64]. Tissues formed while ranging over this large area incorporate a variety of strontium isotope ratios. We hypothesize that this averaging effect mutes the between-tissue difference *equally* for both philopatric and dispersing individuals, decreasing the reliability of tooth–bone offsets and rendering tooth–environment offsets a more reliable approach. Furthermore, male chimpanzees live in their natal home ranges for their entire lives; however, these ranges are large enough that they may inhabit different areas as a sub-adult with their mother and as an independent adult competing for a place in the group's hierarchy. These factors compound to further reduce isotopic differences between tooth and bone tissues within the philopatric sex for large-ranging, slow-growing primates, thus diminishing the expected differences between the sexes for this offset.

Species with relatively small home ranges compared with the isotopic clusters on the landscape (such as colobus monkeys and guenons) are likely to disperse a much shorter distance [93]. Comparisons between tooth enamel and the environment (defined as the mean isotopic ratio of the local cluster) might look the same for both dispersers and philopatric individuals in this scenario. However, these smaller home ranges also mean that each tissue incorporates a less variable strontium source during formation which increases the likelihood of detecting differences in between-tissues offsets in dispersing individuals compared with philopatric individuals. This can be true even in the event that dispersing individuals do not travel far enough to leave the 'local' isotopic cluster, as there is smaller-scale isotopic heterogeneity within each cluster (figure 1). In the present study, olive baboon home ranges approximate quite closely the spacing of isotopic variability on the landscape. In this 'Goldilocks' scenario, we see *both* tooth–bone offsets and tooth–environment offsets effectively predict sex-biases in their philopatry and dispersal patterns.

### 4.1.3. Variation in dispersal rates

Our assessments of method reliability depend on modelling that assumes all dispersal is locational, and that 100% of individuals conform to the expected dispersal patterns for their species. In reality, dispersal is a much more complex and non-binary. Locational dispersal, in which individuals leave their natal group's territory and enter into a new area, is the primary focus of this manuscript. However, individuals can also undergo social dispersal, in which they join different social groups, potentially within the same territory [94]. The methods presented here are not applicable to instance of exclusive social dispersal, as this behaviour does not necessarily correlate with changes in physical location. It is possible to undergo locational dispersal without undergoing social dispersal, such as when home ranges for an entire natal group shift [95], a nuance that these methods could not indicate.

The degree to which individuals within a given species follow the expected dispersal or philopatric pattern predicted by their sex varies substantially and is influenced by social and environmental factors, including status within dominance hierarchies, rates of affiliation and aggression, resource and mating competition, climate, and habitat fragmentation. We note that in our models, there was a bias towards misclassifying dispersing sex individuals; it in entirely possible (if not likely) that for some of these, the 'misclassification' is a true signal that they did not disperse at all.

These sources of error provide insight into why the categorical classification approach led to the most faithful reconstruction of species-level sex-biases in dispersal patterns, particularly when using $^{87}Sr/^{86}Sr_{enamel-environment}$ offsets and statistically identified clusters. Even when sex-biased patterns in dispersal are not followed by 100% of individuals within a primate group, it is reasonable to assume relatively more members of the dispersing sex will leave and more members of the philopatric sex will remain. Despite limitation at the individual level, when attributions are examined in aggregate for all individuals within a species, a greater proportion of dispersing sex individual had large offsets while a greater proportion of philopatric sex individuals had small offsets. While the models cannot predict individual-level attributions with complete confidence due to the overly simplified assumptions regarding the nature of dispersal, they can uncover sex-biases in dispersal patterns at the species level when enough individuals are included.

## 5. Conclusion

Strontium isotope ratios hold great potential for uncovering aspects of landscape use in fossil species, but methodological approaches are not one-size-fits-all. By considering the impacts of ecological

variables, such as range size and life history, this study demonstrates how different methodological approaches apply under different ecological and environmental conditions. For example, the $^{87}Sr/^{86}Sr_{enamel-environment}$ offset method described accurately indicates male philopatry in modern chimpanzees, despite large home range sizes, complex mobility patterns and social structures, and extended life histories. These complicating variables are relevant to early hominins as well, suggesting that this method will also be more reliable than those used in the past to reconstruct our own ancestors' dispersal patterns. Future work will further refine and validate these methods. For example, additional work investigating other philopatric patterns, including bisexual dispersal, conducting modern studies in more geochemically diverse areas, and validating environmental proxies available for fossil areas as appropriate substitutes for the botanical samples used here are all necessary future endeavours.

Ethics. All fieldwork, sample collection and sample import/export activities were approved by the Ugandan Wildlife Authority (EDO/35/01), the Ugandan National Committee on Science and Technology (NS 506), the US Department of Agriculture (PCIP-16-00229) and the US Fish and Wildlife Service (15US62228B/9)

Data accessibility. All data used in these analyses is available in the electronic supplementary material (tables S1 and S2). R code for cluster analysis and parametric bootstrapping is available in the electronic supplementary material file.

Authors' contributions. M.I.H. designed the study, performed field work and laboratory analyses, conducting statistical analyses and wrote the manuscript; S.V.N. performed field work and laboratory analyses and contributed to writing the manuscript; D.P.F. designed laboratory protocols, supervised laboratory analyses and contributed to writing the manuscript; all authors gave final approval for publication.

Competing interests. We declare we have no competing interests.

Funding. National Science Foundation Graduate Research Fellowship (grant no. DGE-0903444) (M.I.H.); Wenner-Gren Foundation (M.I.H.); the Leakey Foundation (M.I.H.); the University of New Mexico Departments of Anthropology and Graduate Studies (M.I.H.); and the University of Northern Colorado.

Acknowledgements. The authors would like to thank the Ugandan Wildlife Authority, the Ugandan National Committee on Science and Technology, the US Department of Agriculture and the US Fish and Wildlife Service for permits and permissions to conduct this research, Dr Emily Otali and the staff and field assistants at Kibale National Park, Lee Drake and Karissa Kilgore for manuscript feedback, and our reviewers for their thoughtful critiques and improvements.

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
