## [Peer Review File · Royal Society Open Science]

Review History

RSOS-200760.R0 (Original submission)

Review form: Reviewer 1

Is the manuscript scientifically sound in its present form?

Yes

Are the interpretations and conclusions justified by the results?

Yes

Is the language acceptable?

Yes

Do you have any ethical concerns with this paper?

No

Reports © 2021 The Reviewers; Decision Letters © 2021 The Reviewers and Editors; Responses © 2021 The Reviewers, Editors and Authors. Published by the Royal Society under the terms of the Creative Commons Attribution License <http://creativecommons.org/licenses/by/4.0/>, which permits unrestricted use, provided the original author and source are credited

Have you any concerns about statistical analyses in this paper?

No

Recommendation?

Accept with minor revision (please list in comments)

Comments to the Author(s)

Hamilton et al. present a new method for distinguishing local and non-local (philopatric and dispersing) individuals and species that relies on the offset (absolute value) between $87\text{Sr}/86\text{Sr}$ of tooth enamel and bone or tooth enamel and the local environment (using plants). The authors have incorporated a number of my previous suggested changes and additions, and these have definitely improved the readability and flow of the text. However, there are still some issues that need to be addressed before I think this is publishable. I outline these below.

First, I note that there are multiple mistakes in the text. In addition to the normal expected typos, there are sentences in the methods that are written more than once verbatim, and some table captions are incorrect (e.g., Table 1 does not include any isotope data)... This lack of attention to detail is concerning. I have done my best to repeatedly correct errors in this manuscript but the authors need to be responsible for the quality of their own work. At best, an editor might catch mistakes, but if things get published that are incorrect, that would be disappointing for everyone.

Second, I think that the offset method that the authors present definitely has promise. The language currently used in the introduction (lines 70-77) is nicely written, and it makes sense. I agree that using offsets to differentiate dispersing and philopatric individuals (or to identify which sex is dispersing for a species) could be a great approach. However, I think that HOW the authors present why the method is novel is rather weak. The authors focus mostly on emphasizing that their study is novel because they establish geochemical clusters on the landscape. For example, in the Conclusion paragraph, they write “the present study highlights that it is critical to establish meaningful geochemical clusters on the landscape thorough sampling and mapping, as geologic boundaries do not by default correspond to unique isotopic clusters” (lines 331-333). I don’t think that most researchers have previously assumed that geologic boundaries by default correspond to unique isotopic clusters. This is why empirical data are included from local animals or plants (to verify what local values are, typically on different geologies). This is nearly ubiquitous across strontium studies (and has been the standard practice for decades, as I pointed out in my previous review).

Instead, I think it’s the second point that the authors mention in the introduction (but don’t revisit later) that makes this method novel: i.e., that calculating offsets allows one to assess the likelihood that an individual is local or non-local based on how it compares to other members of the same species (or community) rather than relying on those maxima and minima designated by the local baselines.

Third, the Discussion is still lacking some important aspects. Mismatches between expected and observed numbers of philopatric vs. dispersing individuals could be due to sampling teeth that mineralized after dispersal, inaccuracies in the available geologic map, heterogeneity in the geologies themselves (see my note below regarding text on lines 91-95 below), and incorrect assumptions about the degree to which individuals truly are catholic to expectations of dispersing and not dispersing. I think all three of these points need to be touched on in the Discussion.

1. I would like to know how much tooth selection might affect offset results. The authors now explicitly mention that they tried to sample M1 for all individuals (and they have included which

tooth was sampled in the supplementary table), but they do not revisit how results might be affected for those individuals that had a different tooth sampled. In particular, I would think that M2 might have erupted post dispersal. I think that the authors need to: (1) Mention explicitly that there were six individuals that had a tooth other than M1 sampled in the Methods (after lines 116-117); (2) Be careful to only refer to tooth (not the earliest forming tooth and not the earliest forming molar) elsewhere in the text; and (3) Revisit if offset might be off for individuals that had a tooth other than M1 sampled (particularly pertinent for later-forming teeth like M2).

2. I still want to know how isotopic clusters for plants compare with the underlying geology (I have asked this previously). The authors repeatedly state that their method of developing isotopic clusters for environmental baselines from plants is far superior to working with preconceived notions of geology, but their justification for this is very vague and revolves around statements about geologies sometimes being indistinguishable (e.g., lines 91-95; lines 264-266). More importantly, the authors still do not compare their “isotopic clusters” with what would be expected based on underlying geology. I want to see what data look like for plants growing on the different geologic formations. I had also asked previously if the authors could please add the expected geologic boundaries for the different formations to Figure 1 (or perhaps make a complementary panel). I still think this would be valuable. Then the reader can see if plants that look unusual are at or near an expected geologic boundary (or not). The issues with poor geologic maps and old geologies very much need to be brought up in the Discussion.

3. I think the authors need to briefly discuss how their assumption of dispersal being nearly ubiquitous for the dispersing sex (and vice versa) might impact “accuracy” of strontium isotope predictions. In the Methods, the authors mention that they assume that 100% of individuals in dispersing sex disperse and 100% of individuals in philopatric sex stay put (without any justification for this decision). They note at least one known example at Kibale where this is not true (a female chimp), but I would think that this individual isn’t an isolated incident. This is important because the “True” number of philopatric and dispersing individuals is not actually known, and implying (or explicitly referring to) any expected number as “true” is misleading. The authors need to please add “expected” or “assumed” in front of all mentions of “correct” classification of individuals. For example, the text on lines 208-209 needs to be “83% of ASSUMED local/philopatric individuals and only 32% of ASSUMED non-local/dispersing individuals were correctly identified”. They should also make sure that they discuss how this assumption may or may not affect their results and interpretations of the data in the Discussion. Smaller points:

Keywords - It is my general understanding that words that are in the title aren’t also needed in the keywords. Please add the country where the study took place, and maybe include “plants” and/or names of the more interesting primate taxa. Also consider including key words related to statistical methods.

There are several instances of data interpretation in the Results (lines 218-220; 239-240; 245-250). To me, these would fit better in the Discussion.

Reference to a single case study (or even two case studies) cannot support a claim that something is common or typical. In particular, citing a single archaeology study and palaeoanthropological paper to support what the authors are calling “the standard archaeological method” is disappointing. If this is truly a standard method, surely the authors can cite more references to support this point.

The authors still have not told the reader what plant tissues were sampled. This is my third time requesting that this information please be provided. “Botanical sample” could be fruit, leaves,

flowers, stems, whole plants, etc. If the authors always collected leaves, say so. If different plant tissues were sampled, then please provide these details in the supplementary table.

Tables:

- The tables are hard to follow as presented. The formatting is quite awkward. In general, starting each table on a new page is reader friendly.
- Table 1 presents no isotope data (contrary to what the caption states)
- Full references belong in the reference cited section (not as table footnotes)
- In the caption for Table 2, "Falling outside of local environment minima or maxima = local/philopatric attribution" is incorrect.
- Table 4 – what is being compared (i.e. what does "proportion of significant ($p < 0.05$) bootstrap simulations" mean)?
- In tables 2, 5, and 6, what are the cutoffs for determining if something is conclusive or not? Please clarify in a footnote.
- **IMPORTANT:** As I mention above, the wording regarding number of dispersing versus local individuals in these tables is misleading. Bolding and underlining true in these table headers makes this even more misleading. As mentioned above, the "TRUE" # of locals and non-locals is unknown! "Assumed" or "expected" are the words the authors should be using. And the authors really can't be looking at ACCURACY of attribution given that the "true" information is assumed in the first place. They can assess and compare attributions.
- The organization of the tables is perplexing to me. Why does Table 3 come before Tables 4-6?
- How do Tables 3 and 7 differ? Presumably one is species level and the other individuals? More informative captions would help ("Summary of Method Efficacy" is not sufficient). Are there other places besides Line 250 where Table 7 should be cited?
- Table S2. How are entries organized in this table? I had initially thought species, locality, sex, but in some cases, the sexes are mixed up for each species at each locality.
- It would be helpful to provide summary offset data for each sex for each species in addition to species means. Please remember that numbers < 1 should have a 0 before the decimal place.

Figures and captions:

- Figure 1 - I have asked before about the geologic map for the region. Even if it is rather generalized, it would still be helpful (and provide reader transparency) to show the expected boundaries of the different units relative to the plant collection sites and primate habitats.
- For Figure 4, are all of the details regarding significance of bootstrapped offsets for each species needed both in this caption and in the main text? Please remind the reader what is being compared in these tests (e.g., sexes?).

Line by line things:

Lines 12-13: "The present study demonstrates that standard archeological methods used to determine 'local' and 'non-local' individuals are not reliable when applied to dispersal patterns in modern primate communities" needs context. Readers will not inherently know what the authors are talking about. Briefly define what the method is that the authors are considering standard (and presumably this should be method, and not plural methods?).

Lines 18-19: "relative to the isotopic clusters on the landscape" needs context. It will not be clear what this means to someone who is just starting to read the paper. Clusters in baseline bioavailable $87\text{Sr}/86\text{Sr}$ obtained from plants? Is this text needed?

Line 39 (and elsewhere): " $87\text{Sr}/86\text{Sr}$ ratio" is redundant. Either strontium isotope ratios, OR $87\text{Sr}/86\text{Sr}$.

Line 40: Tissues should be plural here

Lines 54-55: Consider if the sentence starting with “chimpanzees” should be moved up one sentence. This could help with the flow of the text here.

Line 57: “if” or “degree to which” would be better choices. “Whether” requires “or not”.

Lines 59-61: The authors looked at multiple species of primates. That’s still not clear at this point in the text (it is hinted at on line 69 but doesn’t become fully evident until much later in the text). Rather than say “in a modern primate habitat”. I would recommend “In sympatric primate species with varying home ranges, philopatry, and dispersal patterns” or “Within a modern primate community”. Please also mention the site name here or below on lines 69-70.

Line 62: Citing a single archaeological study and one palaeoanthropological study is not sufficient for supporting a statement that this is the most common method in archaeological strontium isotope literature!

Lines 74-76. Please clarify that “environment” is synonymous with local bioavailable $87\text{Sr}/86\text{Sr}$ or baseline $87\text{Sr}/86\text{Sr}$.

Lines 88-95 – I think adding text like what has been added here is helpful, but it’s also confusing as written. The authors refer to data from botanical samples here but then also refer to botanical data to address the issue of similar botanical data in the next paragraph... I’m also not sure that all of this information belongs here in the Methods. Some of it appears to be results, and justification for the authors’ argument that their new offset method is preferable to other methods. This is absolutely an appropriate place to discuss limitations of available geologic maps. the text could be reorganized slightly in this paragraph to make that point clearer up front rather than halfway down the paragraph. However, the text on lines 91-95 definitely does not seem like Methods to me. It also seems quite vague. Please revisit wording and be careful to focus on Methods (and not results) in the Methods. Also, note that “pattered” should be patterned on line 89.

Here, or perhaps in the Discussion, please explain how/why different geologic formations may be isotopically difficult to distinguish (in the context of what the geology is expected to be at Kibale). My understanding is that very old felsic geologies (e.g., Paleoproterozoic metamorphic rocks) have elevated $87\text{Sr}/86\text{Sr}$ (typically some of the highest measured) due to their age, and that they can be quite isotopically heterogeneous due to differential weathering of minerals over time as well as bioavailable Sr contributions from exogenous sources (like dust). I think this context would be very helpful for the reader, who is likely not familiar with any of this. Please add.

Lines 107+ - This is the meat of the study but the way it is worded makes it sound secondary (deleting also on line 107 would help a lot). Consider if this should be mentioned before the details of how the authors accounted for local variability in bioavailable $87\text{Sr}/86\text{Sr}$.

Line 114 – Site Table 1 here rather than Table S2.

Lines 114-119 – Site Table S2 after this text as it provides details about which element and tooth was sampled for each individual. As mentioned above, please make sure to add explicit mention of the six individuals who had a tooth other than M1 sampled.

Lines 120-132 – There’s redundancy in the text here (e.g., the lab name and location are mentioned twice). First finish the thought about plant sample collection (making sure to mention

what tissues were sampled). Then focus on sample prep and analysis. Some information regarding sample prep is still missing. How were bone and tooth powdered? Did the authors grind up entire teeth or did they remove enamel? Did they chemically pretreat samples in any way? Also note that on Line 122, "on" should be "in".

Line 143 – Consider if this should be a new subheading. It's data analysis (a separate thought from sample acquisition and analysis).

Lines 144-146 – Please justify this assumption.

Line 160 – The authors did sample the earliest forming molar for all individuals. Reword.

Line 194 – there's a typo "±"

Line 198 – "was" seems awkward here. Would be? Is?

Line 199 – Fauna should be capitalized

Lines 203-205 – Methods are laid out in the Methods section. No need to restate this here (but please do make sure that Methods are sufficiently detailed). Should "Standard Archaeological Method" be capitalized in the subheading?

Lines 208-209 – As mentioned above, I think that throughout the text the authors need to be clear that these are EXPECTED local/philopatric or non-local/dispersing individuals. This is only actually known for one female chimpanzee.

Lines 215-218 – Please remind the reader what the significance is referring to here. These species had significant what? Significantly different offset in $87\text{Sr}/86\text{Sr}$ between sexes? Expected dispersing and non-dispersing individuals? Does this language need to be included both in the text and the figure caption?

Lines 256 - 257 – Please clarify ONE standard approach (not the only standard approach). No need to state this again on Line 257 (just write "this approach").

Lines 258-260 - The wording here is confusing. minima and maxima are also isotopic ratios themselves. Please reword or cut.

Lines 260-270 – I think this information would fit much better in the introduction where the authors introduce the method rather than down here in the Discussion. There's some redundancy with what is said earlier, but also some novel information. Please move up and integrate with text in the intro and cut redundant wording.

Lines 262-264 – The authors cite one example here but refer to multiple previous studies. Please revisit wording. Also, as written, this is pretty vague and kind of misleading. the authors need to please make sure that they are clear that while this can happen, it's VERY context dependent. The geologies could also have entirely discrete $87\text{Sr}/86\text{Sr}$. It entirely depends on what the local geologies are. See my notes above about the complexities of very old, felsic geologies.

Lines 300-301 – This needs to be reworded. Underlying geology will ALWAYS be older than something deposited on top of it with rare exceptions (e.g., flood deposits). What's key is that the surface geology has been consistent over time (i.e. no major erosion or deposition since the organisms of interest expired).

Line 303 – is the idea of paleosols holding promise based on the authors’ own gut feelings or is there some existing study or publication that has suggested this that could be cited here?

Lines 308-321 – please cite Table 1 here, which provides home ranges for all of the species. I note that the authors state here that chimp home ranges can be up to 15 km² but Table 1 indicates chimp home ranges can be up to 27 km² (nearly two times bigger than 15 km²). Which is correct?

Lines 337-340 - The paper sort of fizzles in its final sentence. The authors are repeating themselves here; they already pointed out in the beginning of this conclusion paragraph that “the previous standard approach” doesn’t work in all contexts. I would recommend rewording (e.g., it is reasonable to expect early hominins to share many of these features, suggesting that these methods may be able to reconstruct our own ancestors’ dispersal habits“?), think bigger picture about what the next steps might be to help validate this method, or perhaps conjecture about the kinds of questions researchers may be able to answer using this new approach.

Review form: Reviewer 2 (David Watts)

Is the manuscript scientifically sound in its present form?

Yes

Are the interpretations and conclusions justified by the results?

Yes

Is the language acceptable?

Yes

Do you have any ethical concerns with this paper?

No

Have you any concerns about statistical analyses in this paper?

No

Recommendation?

Accept with minor revision (please list in comments)

Comments to the Author(s)

Please see attached file (Appendix A).

Decision letter (RSOS-200760.R0)

Dear Dr Hamilton,

The editors assigned to your paper ("Using Strontium Isotopes to Determine Philopatry and Dispersal in Primates") have now received comments from reviewers. We would like you to

revise your paper in accordance with the referee and Associate Editor suggestions which can be found below (not including confidential reports to the Editor). Please note this decision does not guarantee eventual acceptance.

Please submit a copy of your revised paper before 29-Jul-2020. Please note that the revision deadline will expire at 00.00am on this date. If we do not hear from you within this time then it will be assumed that the paper has been withdrawn. In exceptional circumstances, extensions may be possible if agreed with the Editorial Office in advance. We do not allow multiple rounds of revision so we urge you to make every effort to fully address all of the comments at this stage. If deemed necessary by the Editors, your manuscript will be sent back to one or more of the original reviewers for assessment. If the original reviewers are not available, we may invite new reviewers.

- Data accessibility

<http://datadryad.org/submit?journalID=RSOS&manu=RSOS-200760>

- Competing interests

- Authors' contributions

- Acknowledgements

- Funding statement

on behalf of Dr Alexander Ophir (Associate Editor) and Kevin Padian (Subject Editor)
openscience@royalsociety.org

Editor comments:

Both reviewers and the AE are supportive of the paper, and the reviewers each recommend some different modifications. The revision may not be so "major" and we hope you have no trouble answering their concerns in your next iteration. Thanks for submitting.

Associate Editor's comments (Dr Alexander Ophir):

Dear Dr. Hamilton,

Your manuscript has now been seen by two expert referees whose reports are at the end of this email.

As you will see, both reviewers had a positive impression of your paper, but both noted several points (most of which are minor, but some of which are substantive) that must be addressed before your paper can proceed. Fortunately, the reviewers have invested great effort into providing you constructive comments that will guide you as you improve your manuscript and address these concerns. Please take care to explicitly address each comment (some of which you

have seen in previous rounds of review). If you choose not to make a particular change, please provide a clear and justified reason for why. I also want to particularly highlight the point made by Reviewer 1 that although some typos, etc will be caught by the editorial staff, it is incumbent on you to take care to minimize these as a matter of professionalism.

Overall, I concur with their positive assessment of your manuscript and believe that this will ultimately be a very nice addition to the literature. I would like to congratulate you and your co-authors on this paper and I look forward to seeing your revised manuscript.

Alex Ophir
Associate Editor, RSOS

Associate Editor: 2
Comments to the Author:
(There are no comments.)

Comments to Author:

Reviewers' Comments to Author:
Reviewer: 1

Comments to the Author(s)

Hamilton et al. present a new method for distinguishing local and non-local (philopatric and dispersing) individuals and species that relies on the offset (absolute value) between $87\text{Sr}/86\text{Sr}$ of tooth enamel and bone or tooth enamel and the local environment (using plants). The authors have incorporated a number of my previous suggested changes and additions, and these have definitely improved the readability and flow of the text. However, there are still some issues that need to be addressed before I think this is publishable. I outline these below.

First, I note that there are multiple mistakes in the text. In addition to the normal expected typos, there are sentences in the methods that are written more than once verbatim, and some table captions are incorrect (e.g., Table 1 does not include any isotope data)... This lack of attention to detail is concerning. I have done my best to repeatedly correct errors in this manuscript but the authors need to be responsible for the quality of their own work. At best, an editor might catch mistakes, but if things get published that are incorrect, that would be disappointing for everyone.

Second, I think that the offset method that the authors present definitely has promise. The language currently used in the introduction (lines 70-77) is nicely written, and it makes sense. I agree that using offsets to differentiate dispersing and philopatric individuals (or to identify which sex is dispersing for a species) could be a great approach. However, I think that HOW the authors present why the method is novel is rather weak. The authors focus mostly on emphasizing that their study is novel because they establish geochemical clusters on the landscape. For example, in the Conclusion paragraph, they write "the present study highlights that it is critical to establish meaningful geochemical clusters on the landscape thorough sampling and mapping, as geologic boundaries do not by default correspond to unique isotopic clusters" (lines 331-333). I don't think that most researchers have previously assumed that geologic boundaries by default correspond to unique isotopic clusters. This is why empirical data are included from local animals or plants (to verify what local values are, typically on different geologies). This is nearly ubiquitous across strontium studies (and has been the standard practice for decades, as I pointed out in my previous review).

Instead, I think it's the second point that the authors mention in the introduction (but don't revisit later) that makes this method novel: i.e., that calculating offsets allows one to assess the

likelihood that an individual is local or non-local based on how it compares to other members of the same species (or community) rather than relying on those maxima and minima designated by the local baselines.

Third, the Discussion is still lacking some important aspects. Mismatches between expected and observed numbers of philopatric vs. dispersing individuals could be due to sampling teeth that mineralized after dispersal, inaccuracies in the available geologic map, heterogeneity in the geologies themselves (see my note below regarding text on lines 91-95 below), and incorrect assumptions about the degree to which individuals truly are catholic to expectations of dispersing and not dispersing. I think all three of these points need to be touched on in the Discussion.

1. I would like to know how much tooth selection might affect offset results. The authors now explicitly mention that they tried to sample M1 for all individuals (and they have included which tooth was sampled in the supplementary table), but they do not revisit how results might be affected for those individuals that had a different tooth sampled. In particular, I would think that M2 might have erupted post dispersal. I think that the authors need to: (1) Mention explicitly that there were six individuals that had a tooth other than M1 sampled in the Methods (after lines 116-117); (2) Be careful to only refer to tooth (not the earliest forming tooth and not the earliest forming molar) elsewhere in the text; and (3) Revisit if offset might be off for individuals that had a tooth other than M1 sampled (particularly pertinent for later-forming teeth like M2).

2. I still want to know how isotopic clusters for plants compare with the underlying geology (I have asked this previously). The authors repeatedly state that their method of developing isotopic clusters for environmental baselines from plants is far superior to working with preconceived notions of geology, but their justification for this is very vague and revolves around statements about geologies sometimes being indistinguishable (e.g., lines 91-95; lines 264-266). More importantly, the authors still do not compare their “isotopic clusters” with what would be expected based on underlying geology. I want to see what data look like for plants growing on the different geologic formations. I had also asked previously if the authors could please add the expected geologic boundaries for the different formations to Figure 1 (or perhaps make a complementary panel). I still think this would be valuable. Then the reader can see if plants that look unusual are at or near an expected geologic boundary (or not). The issues with poor geologic maps and old geologies very much need to be brought up in the Discussion.

3. I think the authors need to briefly discuss how their assumption of dispersal being nearly ubiquitous for the dispersing sex (and vice versa) might impact “accuracy” of strontium isotope predictions. In the Methods, the authors mention that they assume that 100% of individuals in dispersing sex disperse and 100% of individuals in philopatric sex stay put (without any justification for this decision). They note at least one known example at Kibale where this is not true (a female chimp), but I would think that this individual isn’t an isolated incident. This is important because the “True” number of philopatric and dispersing individuals is not actually known, and implying (or explicitly referring to) any expected number as “true” is misleading. The authors need to please add “expected” or “assumed” in front of all mentions of “correct” classification of individuals. For example, the text on lines 208-209 needs to be “83% of ASSUMED local/philopatric individuals and only 32% of ASSUMED non-local/dispersing individuals were correctly identified”. They should also make sure that they discuss how this assumption may or may not affect their results and interpretations of the data in the Discussion. Smaller points:

Keywords - It is my general understanding that words that are in the title aren’t also needed in the keywords. Please add the country where the study took place, and maybe include “plants”

and/or names of the more interesting primate taxa. Also consider including key words related to statistical methods.

There are several instances of data interpretation in the Results (lines 218-220; 239-240; 245-250). To me, these would fit better in the Discussion.

Reference to a single case study (or even two case studies) cannot support a claim that something is common or typical. In particular, citing a single archaeology study and palaeoanthropological paper to support what the authors are calling “the standard archaeological method” is disappointing. If this is truly a standard method, surely the authors can cite more references to support this point.

The authors still have not told the reader what plant tissues were sampled. This is my third time requesting that this information please be provided. “Botanical sample” could be fruit, leaves, flowers, stems, whole plants, etc. If the authors always collected leaves, say so. If different plant tissues were sampled, then please provide these details in the supplementary table.

Tables:

- The tables are hard to follow as presented. The formatting is quite awkward. In general, starting each table on a new page is reader friendly.
- Table 1 presents no isotope data (contrary to what the caption states)
- Full references belong in the reference cited section (not as table footnotes)
- In the caption for Table 2, “Falling outside of local environment minima or maxima = local/philopatric attribution” is incorrect.
- Table 4 – what is being compared (i.e. what does “proportion of significant ($p < 0.05$) bootstrap simulations” mean)?
- In tables 2, 5, and 6, what are the cutoffs for determining if something is conclusive or not? Please clarify in a footnote.
- IMPORTANT: As I mention above, the wording regarding number of dispersing versus local individuals in these tables is misleading. Bolding and underlining true in these table headers makes this even more misleading. As mentioned above, the “TRUE” # of locals and non-locals is unknown! “Assumed” or “expected” are the words the authors should be using. And the authors really can’t be looking at ACCURACY of attribution given that the “true” information is assumed in the first place. They can assess and compare attributions.
- The organization of the tables is perplexing to me. Why does Table 3 come before Tables 4-6?
- How do Tables 3 and 7 differ? Presumably one is species level and the other individuals? More informative captions would be helpful (“Summary of Method Efficacy” is not sufficient). Are there other places besides Line 250 where Table 7 should be cited?
- Table S2. How are entries organized in this table? I had initially thought species, locality, sex, but in some cases, the sexes are mixed up for each species at each locality.
- It would be helpful to provide summary offset data for each sex for each species in addition to species means. Please remember that numbers < 1 should have a 0 before the decimal place.

Figures and captions:

- Figure 1 - I have asked before about the geologic map for the region. Even if it is rather generalized, it would still be helpful (and provide reader transparency) to show the expected boundaries of the different units relative to the plant collection sites and primate habitats.
- For Figure 4, are all of the details regarding significance of bootstrapped offsets for each species needed both in this caption and in the main text? Please remind the reader what is being compared in these tests (e.g., sexes?).

Line by line things:

Lines 12-13: “The present study demonstrates that standard archeological methods used to determine ‘local’ and ‘non-local’ individuals are not reliable when applied to dispersal patterns in

modern primate communities” needs context. Readers will not inherently know what the authors are talking about. Briefly define what the method is that the authors are considering standard (and presumably this should be method, and not plural methods?).

Lines 18-19: “relative to the isotopic clusters on the landscape” needs context. It will not be clear what this means to someone who is just starting to read the paper. Clusters in baseline bioavailable $^{87}\text{Sr}/^{86}\text{Sr}$ obtained from plants? Is this text needed?

Line 39 (and elsewhere): “ $^{87}\text{Sr}/^{86}\text{Sr}$ ratio” is redundant. Either strontium isotope ratios, OR $^{87}\text{Sr}/^{86}\text{Sr}$.

Line 40: Tissues should be plural here

Lines 54-55: Consider if the sentence starting with “chimpanzees” should be moved up one sentence. This could help with the flow of the text here.

Line 57: “if” or “degree to which” would be better choices. “Whether” requires “or not”.

Lines 59-61: The authors looked at multiple species of primates. That’s still not clear at this point in the text (it is hinted at on line 69 but doesn’t become fully evident until much later in the text). Rather than say “in a modern primate habitat”. I would recommend “In sympatric primate species with varying home ranges, philopatry, and dispersal patterns” or “Within a modern primate community”. Please also mention the site name here or below on lines 69-70.

Line 62: Citing a single archaeological study and one palaeoanthropological study is not sufficient for supporting a statement that this is the most common method in archaeological strontium isotope literature!

Lines 74-76. Please clarify that “environment” is synonymous with local bioavailable $^{87}\text{Sr}/^{86}\text{Sr}$ or baseline $^{87}\text{Sr}/^{86}\text{Sr}$.

Lines 88-95 – I think adding text like what has been added here is helpful, but it’s also confusing as written. The authors refer to data from botanical samples here but then also refer to botanical data to address the issue of similar botanical data in the next paragraph... I’m also not sure that all of this information belongs here in the Methods. Some of it appears to be results, and justification for the authors’ argument that their new offset method is preferable to other methods. This is absolutely an appropriate place to discuss limitations of available geologic maps. the text could be reorganized slightly in this paragraph to make that point clearer up front rather than halfway down the paragraph. However, the text on lines 91-95 definitely does not seem like Methods to me. It also seems quite vague. Please revisit wording and be careful to focus on Methods (and not results) in the Methods. Also, note that “pattered” should be patterned on line 89.

Here, or perhaps in the Discussion, please explain how/why different geologic formations may be isotopically difficult to distinguish (in the context of what the geology is expected to be at Kibale). My understanding is that very old felsic geologies (e.g., Paleoproterozoic metamorphic rocks) have elevated $^{87}\text{Sr}/^{86}\text{Sr}$ (typically some of the highest measured) due to their age, and that they can be quite isotopically heterogeneous due to differential weathering of minerals over time as well as bioavailable Sr contributions from exogenous sources (like dust). I think this context would be very helpful for the reader, who is likely not familiar with any of this. Please add.

Lines 107+ - This is the meat of the study but the way it is worded makes it sound secondary (deleting also on line 107 would help a lot). Consider if this should be mentioned before the details of how the authors accounted for local variability in bioavailable $87\text{Sr}/86\text{Sr}$.

Line 114 - Site Table 1 here rather than Table S2.

Lines 114-119 - Site Table S2 after this text as it provides details about which element and tooth was sampled for each individual. As mentioned above, please make sure to add explicit mention of the six individuals who had a tooth other than M1 sampled.

Lines 120-132 - There's redundancy in the text here (e.g., the lab name and location are mentioned twice). First finish the thought about plant sample collection (making sure to mention what tissues were sampled). Then focus on sample prep and analysis. Some information regarding sample prep is still missing. How were bone and tooth powdered? Did the authors grind up entire teeth or did they remove enamel? Did they chemically pretreat samples in any way? Also note that on Line 122, "on" should be "in".

Line 143 - Consider if this should be a new subheading. It's data analysis (a separate thought from sample acquisition and analysis).

Lines 144-146 - Please justify this assumption.

Line 160 - The authors did sample the earliest forming molar for all individuals. Reword.

Line 194 - there's a typo "±"

Line 198 - "was" seems awkward here. Would be? Is?

Line 199 - Fauna should be capitalized

Lines 203-205 - Methods are laid out in the Methods section. No need to restate this here (but please do make sure that Methods are sufficiently detailed). Should "Standard Archaeological Method" be capitalized in the subheading?

Lines 208-209 - As mentioned above, I think that throughout the text the authors need to be clear that these are EXPECTED local/philopatric or non-local/dispersing individuals. This is only actually known for one female chimpanzee.

Lines 215-218 - Please remind the reader what the significance is referring to here. These species had significant what? Significantly different offset in $87\text{Sr}/86\text{Sr}$ between sexes? Expected dispersing and non-dispersing individuals? Does this language need to be included both in the text and the figure caption?

Lines 256 - 257 - Please clarify ONE standard approach (not the only standard approach). No need to state this again on Line 257 (just write "this approach").

Lines 258-260 - The wording here is confusing. minima and maxima are also isotopic ratios themselves. Please reword or cut.

Lines 260-270 - I think this information would fit much better in the introduction where the authors introduce the method rather than down here in the Discussion. There's some redundancy with what is said earlier, but also some novel information. Please move up and integrate with text in the intro and cut redundant wording.

Lines 262-264 – The authors cite one example here but refer to multiple previous studies. Please revisit wording. Also, as written, this is pretty vague and kind of misleading. The authors need to please make sure that they are clear that while this can happen, it's VERY context dependent. The geologies could also have entirely discrete $87\text{Sr}/86\text{Sr}$. It entirely depends on what the local geologies are. See my notes above about the complexities of very old, felsic geologies.

Lines 300-301 – This needs to be reworded. Underlying geology will ALWAYS be older than something deposited on top of it with rare exceptions (e.g., flood deposits). What's key is that the surface geology has been consistent over time (i.e. no major erosion or deposition since the organisms of interest expired).

Line 303 – is the idea of paleosols holding promise based on the authors' own gut feelings or is there some existing study or publication that has suggested this that could be cited here?

Lines 308-321 – please cite Table 1 here, which provides home ranges for all of the species. I note that the authors state here that chimp home ranges can be up to 15 km² but Table 1 indicates chimp home ranges can be up to 27 km² (nearly two times bigger than 15 km²). Which is correct?

Lines 337-340 - The paper sort of fizzles in its final sentence. The authors are repeating themselves here; they already pointed out in the beginning of this conclusion paragraph that “the previous standard approach” doesn't work in all contexts. I would recommend rewording (e.g., it is reasonable to expect early hominins to share many of these features, suggesting that these methods may be able to reconstruct our own ancestors' dispersal habits”), think bigger picture about what the next steps might be to help validate this method, or perhaps conjecture about the kinds of questions researchers may be able to answer using this new approach.

Reviewer: 2
Comments to the Author(s)
Please see attached file

Author's Response to Decision Letter for (RSOS-200760.R0)

See Appendix B.

RSOS-200760.R1 (Revision)

Review form: Reviewer 1

Is the manuscript scientifically sound in its present form?

No

Are the interpretations and conclusions justified by the results?

No

Is the language acceptable?

Yes

Do you have any ethical concerns with this paper?

No

Have you any concerns about statistical analyses in this paper?

No

Recommendation?

Major revision is needed (please make suggestions in comments)

Comments to the Author(s)

I do not appear to have access to a clean copy of the manuscript so I am going to limit referring to specific line items in my text below. I have marked up the submitted .pdf (Appendix C) and would ask both the editor and authors to please review my comments, questions, and highlighted points there.

The manuscript has improved. Specifically, the discussion of tooth selection and the addition of geologic boundaries and plant data from different geologies in Figure 1, both definitely help. The additional text about what else might affect results in the Discussion is also useful, although its organization is hard to follow (I revisit this point below). I also think that placing more emphasis on how the offsets help “assess like the likelihood that an individual is local or non-local based on how it compares to other members of the same species (or community) rather than relying on those maxima and minima designated by the local baselines” is good.

Despite these improvements, there are still several fundamental points that I think the authors need to address before this is publishable.

Most critically, the authors still haven't demonstrated that their “isotopic clusters” actually perform better than geologies would. The authors' continued decision to discount geology without actually investigating if using plants from different geologies to define “local environment” (and in turn calculate offsets) actually performs worse than their “statistical clusters” continues to be a critical shortcoming of the paper. Trying to argue that there is a fundamental flaw in using geology without actually showing it doesn't work is problematic. This is a very bold claim that could have major implications for other researchers. The authors need to actually incorporate geology into their paper and compare it with their new method if they want to make the claim that geology doesn't work. Specifically, I think the authors should both: (1) show how the “standard archaeological method” might differ if using geologies vs. isotopic clusters, and (2) calculate offsets for plants on different geologies and compare this to their offsets using “statistical clusters”

It is also still not possible to compare the “standard archaeological” method with the authors' new offset methods in the current version of the manuscript because:

1. The authors don't provide comparable datasets. Primates from Kanywara and Ngogo are compared to local min and maxima separately for the standard archaeological method, but as far as I can tell, data from all regions are combined in offset calculations, tables, and Figs 3 and 4 (i.e. we never see region-specific offsets). The authors need to please discuss how offset calculations and efficacy of distinguishing expected philopatric and dispersing individuals varies across the park. is it all good at all sites or are some sites at Kibale better than others, and if so why). I would expect this could be affected by isotopic heterogeneity (e.g., less useful in places that are more isotopically homogenous)... The authors present primate vs. plant data in Figure 2 but then

combine everyone in the offset calculations. I want to see how offsets might vary with location as well. Then one could more directly compare the different methods.

2. It is not clear what “environment” baseline the authors used for their standard archaeological method. They write “To test the efficacy of the standard archeological approach, we defined minimum and maximum “local” $87\text{Sr}/86\text{Sr}$ thresholds based on plant leaves collected from the Kanyawara and Ngogo study areas and compared the $87\text{Sr}/86\text{Sr}$ renamel of fauna collected from each area to this range” (lines 215-217 of the marked-up text). What are these groups? Do they correspond to either geology or isotopic clusters? How is this “standard”?

3. As I note below, the authors combine plant clusters for their offset calculations for Kanyanchu primates.

4. As mentioned above, the authors still don’t actually investigate geology in their paper. They need to go the necessary extra step of comparing primates to the geologies.

Other related major points:

1. The addition of geology to Figure 1 is helpful. Thank you. This map allowed me to note several things for the first time, which I will outline below. I would like to point out that many of the details in Figure 1 are so small that they are barely legible. This includes the axes and the datapoints themselves. I would also encourage the authors to make the two maps similar sizes or at least include a 3-km scale bar on the geologic map.

2. According to Figure 1, the text regarding the park’s geology on lines 114-117 of the marked-up text is incorrect. The authors write “Kanyawara and Sebitole are on the Toro Formation (an undifferentiated early Paleoproterozoic gneiss) while Kanyanchu and Ngogo are on the Buganda formation (a middle Paleoproterozoic quartzite). Ngogo also falls partially on an area designated an ‘unidentified radiometric anomaly’ by the Ugandan Society for Geology and Mines”. According to Figure 1, Kanyawara is on the Buganda Formation (albeit quite close to the boundary with the Toro Formation) and both Ngogo and Kanyanchu are on a radiometric anomaly...

The authors mention in the Methods that there are no geologic data at all where the geologic anomaly is (maybe I am incorrect but I actually feel like I can see geologic boundaries under the anomaly in Figure 1?). If geology is, indeed, unknown under the anomaly, then how can they write that the Ngogo and Kanyanchu are on the Buganda Formation? Moreover, if there are no data where the radiometric anomaly is, how did the Ugandan Society of Geology and Mines determine it exists (and what its approximate boundaries are)? How is the radiometric anomaly defined/ how was it previously discovered? And why don’t the authors revisit this previously described anomaly in the Discussion? For example, it would seem to me that the very high $87\text{Sr}/86\text{Sr}$ for plants in the “Ngogo” cluster geographically relate rather closely to the radiometric anomaly...

I also note that the authors mention there may be an influence of riparian habitat (non-local strontium from rivers?) in the Discussion. Is this relevant at Kibale? I don’t know the park. Is there a major river system? If so, it should be labeled in Figure 1. Have the authors considered if plants from riparian habitat differ from the surrounding soil? Could this explain some of the spatial heterogeneity they observe in strontium isotope ratios for plants?

3. The geology underlying the various “isotopic clusters” the authors define is also not nearly as cut and dry as the authors would lead the reader to believe. Specifically, the text at the beginning of the Results is quite misleading. It suggests that the isotopic clusters align with geology. But I can now see (thanks to this geologic map in Figure 1) that the authors’ isotopic clusters don’t match geologies like the authors state that they do (i.e. the northern cluster isn’t predominantly on the Toro Formation and the southern cluster isn’t predominantly on the Buganda Formation).

And Kanyanchu samples must also be on mixed geologies as Kanyanchu itself is on the radiometric anomaly. Isn't the whole point of creating these clusters to ignore underlying geology?

4. On a related note, the spatial distribution of the isotopically distinct clusters is not exactly straightforward. For example, there are relatively low $87\text{Sr}/86\text{Sr}$ that would be indicative of the "southern" cluster in northern parts of the park. It would seem to me that the spatial distribution of isotope data actually DOES conform somewhat to geologies. Couldn't this be readily explained if the boundaries of the different formations are perhaps mapped incorrectly?

5. Again on a related note, throughout the text, the authors state that they use local bioavailable $87\text{Sr}/86\text{Sr}$ defined by statistical clustering of botanical samples to assign "local environment", but apparently they combined both southern and Ngogo clusters to estimate "local" for Ngogo individuals? They do not broadcast this point (it is only mentioned once on lines 321-323 in marked up pdf.). Why did the authors do this? This makes very little sense to me given that a main message of the paper is that clusters should be used to define local.... Why didn't Ngogo individuals get compared to the Ngogo cluster? If the rationale is spatial heterogeneity in the clusters, then it would make just as much sense to combine the northern and southern clusters for individuals from Kanywara, but the authors didn't do that... What's the point in defining isotopically discrete clusters of plants if they are then ignored? And if the authors are combining clusters, then can this really be more accurate than geology? Maybe it is, but I have no way of knowing since they still haven't provided any of these comparisons in their paper. Surely all of this needs to be mulled over in the Discussion?

Smaller things:

1. Throughout the text, the authors refer to high and low offsets. Given that they are dealing with absolute values, I would think large and small might make more sense?

2. The language about "expected/anticipated" philopatric or dispersing individuals is still not clear in parts of the Results (e.g., lines 290-291 and 312-314 of the marked-up text).

3. Proofread! The authors state that they are very secure that their paper is now error free but a quick look through the paper shows that there are still issues. In addition to some grammar (numerous places that need commas still), the new text in the discussion has quite a few typos and the caption for Table 6 reads "Were methods were able to correctly identify the dispersing sex for each primate species?"

4. There are some analytical details missing in the methods. What concentrations of nitric acid were used, for example?

5. I still don't understand how the authors define "local isotopically homogenous areas"... Primates could move a lot further than 3 km and still be on the same geology (or isotopically discrete cluster). Is this based on where the primate groups live within the park? Is the idea simply that this is unlikely if we assume primates are equally likely to move in any direction, then there's a very good chance that they would wind up on an isotopically discrete landscape within 3 km? If the authors could please more clearly explain their rationale for 3 km, this would be greatly appreciated. Alternatively, perhaps what matters isn't 3 km per se but rather that there are primates with quite restricted home ranges <1 km) and those with much larger home ranges (chimps). There are also species with somewhat intermediate ranges (blue monkeys and olive baboons). I think it would be more informative if the authors considered their data in these terms rather than home ranges that are below or above a certain (perhaps arbitrary?) threshold of 3 km.

6. The new text in discussion under the heading “Potential Sources of Model Error” is useful but seems rather disjointed. For example, it is confusing that both dispersal from natal troupe and timing of tooth mineralization are discussed separately from “life history variables”... I also do not understand what the overview of the regional geology has to do with “model limitations”. And what do the authors mean when they refer to “models” and “modelling”? Are they referring to their different offset calculations?

7. The. Is the Discussion truly the most appropriate place for all of the details regarding Kibale’s geology and factors that can affect the degree to which bioavailable strontium reflects geology? These details provide context for why the authors are arguing that geologic boundaries, per se, may not be particularly useful for spatially partitioning a region. I would think at least some of this needs to be early in the paper so they authors can justify why they are proposing a novel method for understanding dispersal.

8. Table captions are still very short and vague. For example, that for Table 2 states: “Comparison of expected and assigned “local/philopatric” and “non-local/dispersing” attributions, based on $^{87}\text{Sr}/^{86}\text{Sr}$ enamel falling outside/within the local isotopic minima and maxima (“standard archeological method”)”. The reader is left to guess what local isotopic minima and maxima might be and how these were estimated. Also, expected and assigned attributions aren’t both based on strontium, but that’s how the caption reads. Both table and figure captions should provide enough details to be understandable on their own.

Review form: Reviewer 2 (David Watts)

Is the manuscript scientifically sound in its present form?

Yes

Are the interpretations and conclusions justified by the results?

Yes

Is the language acceptable?

Yes

Do you have any ethical concerns with this paper?

No

Have you any concerns about statistical analyses in this paper?

No

Recommendation?

Accept with minor revision (please list in comments)

Comments to the Author(s)

The authors have extensively re-written the manuscript. In the process, they have clarified their methods and their arguments and considerably improved the paper. They have adequately addressed my concerns and comments about the presentation, the behavioral ecology of their study species, and the literature on primate dispersal strategies. My sense is that they have also done a good job of addressing the concerns about the isotopic methods and I think the present version is acceptable for publication, but as before, I defer to Reviewer 1 to decide whether the isotopic material needs more work.

I found a few typographical errors, the occurrence of which is understandable given the difficulty of noticing all of them while so many tracked changes are visible. For the same reason, I might have missed others.

266-267: should be “isotopic values from the two main geological formations...were not significantly different from each other” (or “did not differ significantly”).

394: delete last “the”

Table 6: Fix “Were methods were able... (Why not just “Which methods correctly identified the dispersing sex for each primate species?” or “Methods that correctly identified the dispersing sex...”)

Decision letter (RSOS-200760.R1)

Dear Dr Hamilton,

The Editors assigned to your paper RSOS-200760.R1 "Using Strontium Isotopes to Determine Philopatry and Dispersal in Primates" have now received comments from reviewers and would like you to revise the paper in accordance with the reviewer comments and any comments from the Editors. Please note this decision does not guarantee eventual acceptance.

Please submit your revised manuscript and required files (see below) no later than 21 days from today's (ie 09-Sep-2020) date. Note: the ScholarOne system will 'lock' if submission of the revision is attempted 21 or more days after the deadline. If you do not think you will be able to meet this deadline please contact the editorial office immediately.

on behalf of Dr Alexander Ophir (Associate Editor) and Kevin Padian (Subject Editor)
openscience@royalsociety.org

Associate Editor Comments to Author (Dr Alexander Ophir):

Dear Dr. Hamilton,

I have received the reviews from the previously assigned reviewers and their comments can be found below. As you will see, although Reviewer 2 felt the manuscript is satisfactory, they have made it clear that they are unable to evaluate the material on isotopes and geology. Sadly, I am also in a similar boat, and so I must more heavily rely on the comments from Reviewer 1, who has been supportive of your manuscript in the past, but who also continues to raise several major critiques of this manuscript after reviewing it four times. I would therefore ask that you very carefully think about each of these points and do whatever is necessary to address them with careful justification and consideration. In particular, you should respond to the Reviewer's concerns that you have not adequately incorporated geologies into your analyses and a need to demonstrate that your methodology indeed provides better assessments than could be achieved by other methodologies. Direct comparisons are clearly preferable, but if that is not possible, then at least provide strong evidence for your conclusions.

I am in support of providing one more opportunity to quell the important critiques that Reviewer 1 has outlined in detail. RSOS aims to publish high-quality original research without the usual restrictions on scope, length, or impact. Reviewer 1 has raised issues that they perceive to be problems with the logic (or gaps therein) and poor justification for some of the claims you have made that - if true - would have major implications for your field. Therefore it is critical to demonstrate that these claims are of the highest quality and indeed justified, especially if the impact is likely to be far reaching. I wish you the best of luck as you revise your manuscript, and encourage you to pay close attention to the appended reviews and the marked up PDF that Reviewer 1 provided.

Alex Ophir
Associate Editor, RSOS

Reviewer comments to Author:

Reviewer: 1
Comments to the Author(s)

I do not appear to have access to a clean copy of the manuscript so I am going to limit referring to specific line items in my text below. I have marked up the submitted .pdf and would ask both the editor and authors to please review my comments, questions, and highlighted points there.

The manuscript has improved. Specifically, the discussion of tooth selection and the addition of geologic boundaries and plant data from different geologies in Figure 1, both definitely help. The additional text about what else might affect results in the Discussion is also useful, although its organization is hard to follow (I revisit this point below). I also think that placing more emphasis

on how the offsets help “assess like the likelihood that an individual is local or non-local based on how it compares to other members of the same species (or community) rather than relying on those maxima and minima designated by the local baselines” is good.

Despite these improvements, there are still several fundamental points that I think the authors need to address before this is publishable.

Most critically, the authors still haven't demonstrated that their “isotopic clusters” actually perform better than geologies would. The authors' continued decision to discount geology without actually investigating if using plants from different geologies to define “local environment” (and in turn calculate offsets) actually performs worse than their “statistical clusters” continues to be a critical shortcoming of the paper. Trying to argue that there is a fundamental flaw in using geology without actually showing it doesn't work is problematic. This is a very bold claim that could have major implications for other researchers. The authors need to actually incorporate geology into their paper and compare it with their new method if they want to make the claim that geology doesn't work. Specifically, I think the authors should both: (1) show how the “standard archaeological method” might differ if using geologies vs. isotopic clusters, and (2) calculate offsets for plants on different geologies and compare this to their offsets using “statistical clusters”

It is also still not possible to compare the “standard archaeological” method with the authors' new offset methods in the current version of the manuscript because:

1. The authors don't provide comparable datasets. Primates from Kanywara and Ngogo are compared to local min and maxima separately for the standard archaeological method, but as far as I can tell, data from all regions are combined in offset calculations, tables, and Figs 3 and 4 (i.e. we never see region-specific offsets). The authors need to please discuss how offset calculations and efficacy of distinguishing expected philopatric and dispersing individuals varies across the park. Is it all good at all sites or are some sites at Kibale better than others, and if so why? I would expect this could be affected by isotopic heterogeneity (e.g., less useful in places that are more isotopically homogenous)... The authors present primate vs. plant data in Figure 2 but then combine everyone in the offset calculations. I want to see how offsets might vary with location as well. Then one could more directly compare the different methods.
2. It is not clear what “environment” baseline the authors used for their standard archaeological method. They write “To test the efficacy of the standard archeological approach, we defined minimum and maximum “local” $87\text{Sr}/86\text{Sr}$ thresholds based on plant leaves collected from the Kanyawara and Ngogo study areas and compared the $87\text{Sr}/86\text{Sr}$ enamel of fauna collected from each area to this range” (lines 215-217 of the marked-up text). What are these groups? Do they correspond to either geology or isotopic clusters? How is this “standard”?
3. As I note below, the authors combine plant clusters for their offset calculations for Kanyanchu primates.
4. As mentioned above, the authors still don't actually investigate geology in their paper. They need to go the necessary extra step of comparing primates to the geologies.

Other related major points:

1. The addition of geology to Figure 1 is helpful. Thank you. This map allowed me to note several things for the first time, which I will outline below. I would like to point out that many of the details in Figure 1 are so small that they are barely legible. This includes the axes and the datapoints themselves. I would also encourage the authors to make the two maps similar sizes or at least include a 3-km scale bar on the geologic map.

2. According to Figure 1, the text regarding the park's geology on lines 114-117 of the marked-up text is incorrect. The authors write “Kanyawara and Sebitole are on the Toro Formation (an

undifferentiated early Paleoproterozoic gneiss) while Kanyanchu and Ngogo are on the Buganda formation (a middle Paleoproterozoic quartzite). Ngogo also falls partially on an area designated an 'unidentified radiometric anomaly' by the Ugandan Society for Geology and Mines". According to Figure 1, Kanyawara is on the Buganda Formation (albeit quite close to the boundary with the Toro Formation) and both Ngogo and Kanyanchu are on a radiometric anomaly...

The authors mention in the Methods that there are no geologic data at all where the geologic anomaly is (maybe I am incorrect but I actually feel like I can see geologic boundaries under the anomaly in Figure 1?). If geology is, indeed, unknown under the anomaly, then how can they write that the Ngogo and Kanyanchu are on the Buganda Formation? Moreover, if there are no data where the radiometric anomaly is, how did the Ugandan Society of Geology and Mines determine it exists (and what its approximate boundaries are)? How is the radiometric anomaly defined/ how was it previously discovered? And why don't the authors revisit this previously described anomaly in the Discussion? For example, it would seem to me that the very high $87\text{Sr}/86\text{Sr}$ for plants in the "Ngogo" cluster geographically relate rather closely to the radiometric anomaly...

I also note that the authors mention there may be an influence of riparian habitat (non-local strontium from rivers?) in the Discussion. Is this relevant at Kibale? I don't know the park. Is there a major river system? If so, it should be labeled in Figure 1. Have the authors considered if plants from riparian habitat differ from the surrounding soil? Could this explain some of the spatial heterogeneity they observe in strontium isotope ratios for plants?

3. The geology underlying the various "isotopic clusters" the authors define is also not nearly as cut and dry as the authors would lead the reader to believe. Specifically, the text at the beginning of the Results is quite misleading. It suggests that the isotopic clusters align with geology. But I can now see (thanks to this geologic map in Figure 1) that the authors' isotopic clusters don't match geologies like the authors state that they do (i.e. the northern cluster isn't predominantly on the Toro Formation and the southern cluster isn't predominantly on the Buganda Formation). And Kanyanchu samples must also be on mixed geologies as Kanyanchu itself is on the radiometric anomaly. Isn't the whole point of creating these clusters to ignore underlying geology?

4. On a related note, the spatial distribution of the isotopically distinct clusters is not exactly straightforward. For example, there are relatively low $87\text{Sr}/86\text{Sr}$ that would be indicative of the "southern" cluster in northern parts of the park. It would seem to me that the spatial distribution of isotope data actually DOES conform somewhat to geologies. Couldn't this be readily explained if the boundaries of the different formations are perhaps mapped incorrectly?

5. Again on a related note, throughout the text, the authors state that they use local bioavailable $87\text{Sr}/86\text{Sr}$ defined by statistical clustering of botanical samples to assign "local environment", but apparently they combined both southern and Ngogo clusters to estimate "local" for Ngogo individuals? They do not broadcast this point (it is only mentioned once on lines 321-323 in marked up pdf.). Why did the authors do this? This makes very little sense to me given that a main message of the paper is that clusters should be used to define local.... Why didn't Ngogo individuals get compared to the Ngogo cluster? If the rationale is spatial heterogeneity in the clusters, then it would make just as much sense to combine the northern and southern clusters for individuals from Kanywara, but the authors didn't do that... What's the point in defining isotopically discrete clusters of plants if they are then ignored? And if the authors are combining clusters, then can this really be more accurate than geology? Maybe it is, but I have no way of knowing since they still haven't provided any of these comparisons in their paper. Surely all of this needs to be mulled over in the Discussion?

Smaller things:

1. Throughout the text, the authors refer to high and low offsets. Given that they are dealing with absolute values, I would think large and small might make more sense?
2. The language about “expected/anticipated” philopatric or dispersing individuals is still not clear in parts of the Results (e.g., lines 290-291 and 312-314 of the marked-up text).
3. Proofread! The authors state that they are very secure that their paper is now error free but a quick look through the paper shows that there are still issues. In addition to some grammar (numerous places that need commas still), the new text in the discussion has quite a few typos and the caption for Table 6 reads “Were methods were able to correctly identify the dispersing sex for each primate species?”
4. There are some analytical details missing in the methods. What concentrations of nitric acid were used, for example?
5. I still don’t understand how the authors define “local isotopically homogenous areas”... Primates could move a lot further than 3 km and still be on the same geology (or isotopically discrete cluster). Is this based on where the primate groups live within the park? Is the idea simply that this is unlikely if we assume primates are equally likely to move in any direction, then there’s a very good chance that they would wind up on an isotopically discrete landscape within 3 km? If the authors could please more clearly explain their rationale for 3 km, this would be greatly appreciated. Alternatively, perhaps what matters isn’t 3 km per se but rather that there are primates with quite restricted home ranges (<1 km) and those with much larger home ranges (chimps). There are also species with somewhat intermediate ranges (blue monkeys and olive baboons). I think it would be more informative if the authors considered their data in these terms rather than home ranges that are below or above a certain (perhaps arbitrary?) threshold of 3 km.
6. The new text in discussion under the heading “Potential Sources of Model Error” is useful but seems rather disjointed. For example, it is confusing that both dispersal from natal troupe and timing of tooth mineralization are discussed separately from “life history variables”... I also do not understand what the overview of the regional geology has to do with “model limitations”. And what do the authors mean when they refer to “models” and “modelling”? Are they referring to their different offset calculations?
7. The. Is the Discussion truly the most appropriate place for all of the details regarding Kibale’s geology and factors that can affect the degree to which bioavailable strontium reflects geology? These details provide context for why the authors are arguing that geologic boundaries, per se, may not be particularly useful for spatially partitioning a region. I would think at least some of this needs to be early in the paper so they authors can justify why they are proposing a novel method for understanding dispersal.
8. Table captions are still very short and vague. For example, that for Table 2 states: “Comparison of expected and assigned “local/philopatric” and “non-local/dispersing” attributions, based on $^{87}\text{Sr}/^{86}\text{Sr}$ enamel falling outside/within the local isotopic minima and maxima (“standard archeological method”)”. The reader is left to guess what local isotopic minima and maxima might be and how these were estimated. Also, expected and assigned attributions aren’t both based on strontium, but that’s how the caption reads. Both table and figure captions should provide enough details to be understandable on their own.

Reviewer: 2

Comments to the Author(s)

The authors have extensively re-written the manuscript. In the process, they have clarified their methods and their arguments and considerably improved the paper. They have adequately addressed my concerns and comments about the presentation, the behavioral ecology of their study species, and the literature on primate dispersal strategies. My sense is that they have also done a good job of addressing the concerns about the isotopic methods and I think the present version is acceptable for publication, but as before, I defer to Reviewer 1 to decide whether the isotopic material needs more work.

I found a few typographical errors, the occurrence of which is understandable given the difficulty of noticing all of them while so many tracked changes are visible. For the same reason, I might have missed others.

266-267: should be “isotopic values from the two main geological formations...were not significantly different from each other” (or “did not differ significantly”).

394: delete last “the”

Table 6: Fix “Were methods were able... (Why not just “Which methods correctly identified the dispersing sex for each primate species?” or “Methods that correctly identified the dispersing sex...”)

===PREPARING YOUR MANUSCRIPT===

Your revised paper should include the changes requested by the referees and Editors of your manuscript. You should provide two versions of this manuscript and both versions must be provided in an editable format:
 one version identifying all the changes that have been made (for instance, in coloured highlight, in bold text, or tracked changes);
 a 'clean' version of the new manuscript that incorporates the changes made, but does not highlight them. This version will be used for typesetting if your manuscript is accepted.
 Please ensure that any equations included in the paper are editable text and not embedded images.

===PREPARING YOUR REVISION IN SCHOLARONE===

Author's Response to Decision Letter for (RSOS-200760.R1)

See Appendix D.

RSOS-200760.R2 (Revision)

Review form: Reviewer 3

Is the manuscript scientifically sound in its present form?

No

Are the interpretations and conclusions justified by the results?

No

Is the language acceptable?

Yes

Do you have any ethical concerns with this paper?

No

Have you any concerns about statistical analyses in this paper?

No

Recommendation?

Accept with minor revision (please list in comments)

Comments to the Author(s)

This is a worthy article which sets out in an important direction—to provide background data necessary for informed interpretation of strontium isotope ratios—especially as they pertain to the issue of primate (and hominid) residence patterns. I believe the article in its current form could be improved, however, with more careful attention to previous relevant research; the omissions make it seem that either the authors are unaware of critical issues long understood and written about, or deliberately ignored studies which might raise questions about their research methodology. I will simply state the problems as I see them, and leave it to the editors to determine if, in a revision, the issues are satisfactorily addressed. I don't feel the need to re-review.

1. The authors, in the abstract and throughout, make it sound as if comparing different skeletal tissues (eg, bone vs. teeth) is somehow 'novel'. On the contrary, the general concept was articulated by Jonathan Ericson as early as 1985 (Ericson, J. Strontium isotope characterization in the study of prehistoric human ecology; *J. Human. Ev.* 14:5 pp. 503-514,) and this was the first article to propose using strontium isotopes to address residence patterns. Moreover, the concept was articulated specifically for fossil hominids in an article they themselves cite (Sillen, et al. 1998 ref#22) Others have worked on the issue; the use of the word 'novel' feels inappropriate.

2. Lines 44-45: The authors state that “tooth enamel forms early in life and then is metabolically inert.” This is true, but a tremendous oversimplification; the difficulty is that different locations of enamel (between teeth, and within teeth) are calcifying at different moments in an individual’s growth and development, and these differences matter, particularly when analyzing first molars (see below)
3. I understand the logic of using cancellous bone as a proxy for a relatively recent signal, but somewhere it should be made clear that it is the LEAST useful material, when it exists at all, for looking at fossils.
4. I am perplexed that, in reviewing the studies pertaining to the fossil record (line 55), the authors omit Sillen and Balter (2018; Strontium isotopic aspects of Paranthropus robustus teeth: implications for habitat, residence, and growth; J. Human. Ev. 114: 118-130). This study is directly relevant, since these authors documented varying $87\text{Sr}/86\text{Sr}$ within teeth: most notably first molars. Obviously, we can’t expect the authors to go back and redo their research design, but they do need to articulate exactly from where they took their samples. The reason is that Sillen and Balter suggest that some first molar enamel, that calcifying before weaning, represents maternal residence. I suggest the authors write a few sentences acknowledging the issue and explaining how their sampling methodology avoids such tissue.
5. Line 129. I’m not sure I understand the logic of not looking at riparian leaf samples. I think it is that, way down on line 407, they say “...the local geology in the riparian zone soil plays a secondary role as a source of strontium in the plant and animal tissue.” It isn’t clear to me whether they are saying riparian zones don’t differ that much from the primary source, or that primates (and by extension hominids) don’t much exploit riparian resources. It needs clarification, because Sillen et al (1997) demonstrated for the Sterkfontein Valley that the difference between riparian and veld $87\text{Sr}/86\text{Sr}$ was, if anything more important than the geological map.
6. Line 377 sp. insufficiently
7. Lines 377-383. Instead of saying on line 377 that bone samples are ‘notoriously unreliable’, I would suggest using a phrase like, ‘need to be conducted with great care, and special attention to diagenesis’. Otherwise, the paragraph is inconsistent with citation 22 on line 384, which is based on analysis of small rodent fossil bones.
8. Lines 399-400. Based on the experience of the Sterkfontein Valley, I disagree that bedrock geology is a reasonable proxy for local water and vegetation. On the contrary, I would argue that bedrock geology is at best, a starting place for hypotheses, but there is no substitute for empirical isoscape mapping.
9. Lines 429. Again, the authors need to address the question of maternal alkaline earths in M1 enamel tissue, perhaps with a better description of their sampling methodology.

Review form: Reviewer 4

Is the manuscript scientifically sound in its present form?

Yes

Are the interpretations and conclusions justified by the results?

Yes

Is the language acceptable?

Yes

Do you have any ethical concerns with this paper?

No

Have you any concerns about statistical analyses in this paper?

No

Recommendation?

Major revision is needed (please make suggestions in comments)

Comments to the Author(s)

See my review in the attached document (Appendix E).

Decision letter (RSOS-200760.R2)

Dear Dr Hamilton

The Editors assigned to your paper RSOS-200760.R2 "Using Strontium Isotopes to Determine Philopatry and Dispersal in Primates" have now received comments from reviewers and would like you to revise the paper in accordance with the reviewer comments and any comments from the Editors. Please note this decision does not guarantee eventual acceptance.

Please submit your revised manuscript and required files (see below) no later than 21 days from today's (ie 23-Nov-2020) date. Note: the ScholarOne system will 'lock' if submission of the revision is attempted 21 or more days after the deadline. If you do not think you will be able to meet this deadline please contact the editorial office immediately.

Kind regards,

Andrew Dunn

on behalf of Prof Kevin Padian (Subject Editor)
openscience@royalsociety.org

Associate Editor Comments to Author:

We have sought new reviewers of this iteration of the paper, but it appears that further modification is still required before your manuscript may be accepted for publication. In general, we are not able to permit multiple rounds of revision, but on given the new reviewers, we are trying to be flexible to give you one final opportunity to amend your paper and get it over the line for publication. Please do your best to respond effectively and comprehensively to the comments received, ensuring a full point-by-point response as well as edited manuscript. These new reviewers will be invited to review your final version - if they are satisfied the paper is ready for acceptance, then we would be glad to accept the paper; however, if major concerns remain, I regret that we will not be able to accept the paper (you will have had ample opportunity at that point to make the paper acceptable to a number of referees). With this in mind, good luck and we look forward to receiving the revision soon.

Reviewer comments to Author:

Reviewer: 3

Comments to the Author(s)

This is a worthy article which sets out in an important direction – to provide background data necessary for informed interpretation of strontium isotope ratios – especially as they pertain to the issue of primate (and hominid) residence patterns. I believe the article in its current form could be improved, however, with more careful attention to previous relevant research; the omissions make it seem that either the authors are unaware of critical issues long understood and written about, or deliberately ignored studies which might raise questions about their research methodology. I will simply state the problems as I see them, and leave it to the editors to determine if, in a revision, the issues are satisfactorily addressed. I don't feel the need to re-review.

1. The authors, in the abstract and throughout, make it sound as if comparing different skeletal tissues (eg, bone vs. teeth) is somehow 'novel'. On the contrary, the general concept was articulated by Jonathan Ericson as early as 1985 (Ericson, J. Strontium isotope characterization in the study of prehistoric human ecology; *J. Human. Ev.* 14:5 pp. 503-514,) and this was the first article to propose using strontium isotopes to address residence patterns. Moreover, the concept was articulated specifically for fossil hominids in an article they themselves cite (Sillen, et al. 1998 ref#22) Others have worked on the issue; the use of the word 'novel' feels inappropriate.
2. Lines 44-45: The authors state that "tooth enamel forms early in life and then is metabolically inert." This is true, but a tremendous oversimplification; the difficulty is that different locations of enamel (between teeth, and within teeth) are calcifying at different moments in an individual's growth and development, and these differences matter, particularly when analyzing first molars (see below)
3. I understand the logic of using cancellous bone as a proxy for a relatively recent signal, but somewhere it should be made clear that it is the LEAST useful material, when it exists at all, for looking at fossils.
4. I am perplexed that, in reviewing the studies pertaining to the fossil record (line 55), the authors omit Sillen and Balter (2018; Strontium isotopic aspects of *Paranthropus robustus* teeth: implications for habitat, residence, and growth; *J. Human. Ev.* 114: 118-130). This study is directly relevant, since these authors documented varying $87\text{Sr}/86\text{Sr}$ within teeth: most notably first molars. Obviously, we can't expect the authors to go back and redo their research design, but they do need to articulate exactly from where they took their samples. The reason is that

Sillen and Balter suggest that some first molar enamel, that calcifying before weaning, represents maternal residence. I suggest the authors write a few sentences acknowledging the issue and explaining how their sampling methodology avoids such tissue.

5. Line 129. I'm not sure I understand the logic of not looking at riparian leaf samples. I think it is that, way down on line 407, they say "...the local geology in the riparian zone soil plays a secondary role as a source of strontium in the plant and animal tissue." It isn't clear to me whether they are saying riparian zones don't differ that much from the primary source, or that primates (and by extension hominids) don't much exploit riparian resources. It needs clarification, because Sillen et al (1997) demonstrated for the Sterkfontein Valley that the difference between riparian and veld $87\text{Sr}/86\text{Sr}$ was, if anything more important than the geological map.

6. Line 377 sp. insufficiently

7. Lines 377-383. Instead of saying on line 377 that bone samples are 'notoriously unreliable', I would suggest using a phrase like, 'need to be conducted with great care, and special attention to diagenesis'. Otherwise, the paragraph is inconsistent with citation 22 on line 384, which is based on analysis of small rodent fossil bones.

8. Lines 399-400. Based on the experience of the Sterkfontein Valley, I disagree that bedrock geology is a reasonable proxy for local water and vegetation. On the contrary, I would argue that bedrock geology is at best, a starting place for hypotheses, but there is no substitute for empirical isoscape mapping.

9. Lines 429. Again, the authors need to address the question of maternal alkaline earths in M1 enamel tissue, perhaps with a better description of their sampling methodology.

Reviewer: 4

Comments to the Author(s)

See my review in the attached document.

===PREPARING YOUR MANUSCRIPT===

If you have been asked to revise the written English in your submission as a condition of publication, you must do so, and you are expected to provide evidence that you have received language editing support. The journal would prefer that you use a professional language editing

service and provide a certificate of editing, but a signed letter from a colleague who is a native speaker of English is acceptable. Note the journal has arranged a number of discounts for authors using professional language editing services (<https://royalsociety.org/journals/authors/benefits/language-editing/>).

===PREPARING YOUR REVISION IN SCHOLARONE===

<https://royalsociety.org/journals/authors/author-guidelines/#supplementary-material> to include a suitable title and informative caption. An example of appropriate titling and captioning

may be found at https://figshare.com/articles/Table_S2_from_Is_there_a_trade-off_between_peak_performance_and_performance_breadth_across_temperatures_for_aerobic_sc_ope_in_teleost_fishes_/3843624.

Author's Response to Decision Letter for (RSOS-200760.R2)

See Appendix F.

RSOS-200760.R3 (Revision)

Review form: Reviewer 3

Is the manuscript scientifically sound in its present form?

Yes

Are the interpretations and conclusions justified by the results?

Yes

Is the language acceptable?

Yes

Do you have any ethical concerns with this paper?

No

Have you any concerns about statistical analyses in this paper?

No

Recommendation?

Accept with minor revision (please list in comments)

Comments to the Author(s)

All good, with one minor exception, on line 46: "Tooth enamel forms early in life and then is metabolically inert, although different enamel locations within the same tooth do calcify at different rates." The point is rather that they calcify at different times, not different rates.

Review form: Reviewer 4

Is the manuscript scientifically sound in its present form?

Yes

Are the interpretations and conclusions justified by the results?

No

Is the language acceptable?

Yes

Do you have any ethical concerns with this paper?

No

Have you any concerns about statistical analyses in this paper?

No

Recommendation?

Major revision is needed (please make suggestions in comments)

Comments to the Author(s)

RSOS-200760.R3

Second Review

Overall I'm happy with the changes the authors made to the manuscript. However, upon a second (and perhaps deeper) reading, there are still two major issues that stand out to me.

I would like the authors to clearly address how the statistical clusters of samples they generated differ from using just the range of $87\text{Sr}/86\text{Sr}$ within the bounds of each study area. Why is the "standard archaeological method" one that takes into account data from geology rather than $87\text{Sr}/86\text{Sr}$ from the study area? I comment on this below in several places, but I'm really not sure why, for example, anyone trying to identify local vs non local individuals in the Sebitole area would include plant $87\text{Sr}/86\text{Sr}$ collected from the southern reaches of the park to generate their local Sr signature. Right now this seems a bit like a straw man argument. What I'd like to see is a set of boxplots (maybe grouping it in with figure 2) that shows the distribution of plant $87\text{Sr}/86\text{Sr}$ for each study area (for example, all the Sr data from within the dotted lines in figure 1). I think that's going to align much better with the statistical clusters. Then the authors can really get into why the statistical clustering method is better than using "local" Sr ratios.

One issue that I failed to comment on in the first round of reviews is that this paper does not mention previous archaeological studies that have used similar approaches (bone-tooth pairs) to exploring mobility. The other reviewer commented on this, but I don't think the authors added a thorough enough discussion in their background section. It would be good to see the authors contextualize their study with a clear discussion of how the approaches used in this study compare to those used in the following:

Price, T. Douglas, Linda Manzanilla, and William D. Middleton. "Immigration and the ancient city of Teotihuacan in Mexico: a study using strontium isotope ratios in human bone and teeth." *Journal of Archaeological Science* 27, no. 10 (2000): 903-913.

Grupe, Gisela, T. Douglas Price, Peter Schröter, Frank Söllner, Clark M. Johnson, and Brian L. Beard. "Mobility of Bell Beaker people revealed by strontium isotope ratios of tooth and bone: a study of southern Bavarian skeletal remains." *Applied Geochemistry* 12, no. 4 (1997): 517-525.

Schweissing, Matthew Mike, and Gisela Grupe. "Stable strontium isotopes in human teeth and bone: a key to migration events of the late Roman period in Bavaria." *Journal of archaeological science* 30, no. 11 (2003): 1373-1383.

Or this one, which compares two different teeth:

Slater, Philip A., Kristin M. Hedman and Thomas E. Emerson. 2014. Immigrants at the Mississippian polity of Cahokia: strontium isotope evidence for population movement. *Journal of Archaeological Science* 44(0):117-127.

It might also be worth it to mention other studies of mobility that use paired tissues (one metabolically inert tissue compared to a metabolically active tissue):

Del Rio, Carlos Martínez, Pablo Sabat, Richard Anderson-Sprecher, and Sandra P. Gonzalez. "Dietary and isotopic specialization: the isotopic niche of three *Cinclodes* ovenbirds." *Oecologia* 161, no. 1 (2009): 149-159.

As well as mention some of the studies looking at dispersals in other taxa (like birds!)

Hobson, Keith A., Leonard I. Wassenaar, and Erin Bayne. "Using isotopic variance to detect long-distance dispersal and philopatry in birds: an example with ovenbirds and American redstarts." *The Condor* 106, no. 4 (2004): 732-743.

Also this recent paper may be of interest:

Crowley, Brooke Erin, and Laurie Rohde Godfrey. "Strontium isotopes support small home ranges for extinct lemurs." *Frontiers in Ecology and Evolution* 7 (2019): 490.

Line 99: Do the authors mean that the Sr will be consumed by weathering?

Line 196 and 197: use \pm instead of +/-

Line 219: Wouldn't the standard archaeological approach be better described as one that uses local $^{87}\text{Sr}/^{86}\text{Sr}$, rather than $^{87}\text{Sr}/^{86}\text{Sr}$ assigned to particular geologies? I'm thinking of the Slater et al. 2014 paper on Cahokia. They do not use a geologic map here, but just samples from sites from the American Bottom.

Line 220: The authors should explain why they chose to use the $^{87}\text{Sr}/^{86}\text{Sr}$ values of plants only from the Toro formation as their "local" signature. According to Figure 1, Kanyawara sits on both the Toro and Buganda formations, pretty equally. What about comparing the primates from Sebitole to the Toro formation, since that one is pretty homogenous? And for the Kanyawara primates, how about just using the $^{87}\text{Sr}/^{86}\text{Sr}$ values that are within the study area? I also don't think there is any information in the SI table that indicates clearly which animals came from which study area. Can the authors include a column in the table that indicates that?

Line 350: I'm not totally clear on why the authors used the mean value of all the plant samples collected from the Buganda formation to compare to the individual from Kanyanchu. The mean value is 0.7113, but clearly in Figure 1 all the plant samples from the Kanyanchu area are higher, closer to around 0.715, due to the proximity to that unidentified radiometric anomaly. I think it's a good exercise to compare the values from the geology, but I think it would also be good to compare the primates from each study area to the plants that were collected from each study area.

After reading through this a second time, I realize I'm not totally clear on what the standard archaeological approach means. In some places I take it to mean using the range of $^{87}\text{Sr}/^{86}\text{Sr}$ of any plant sample collected from the geologic substrate on which a site is located. But, other times

in the manuscript, like in the paragraph starting with Line 407 to line 411, it seems like the method I just described is compared to the “standard archaeological approach”.

Figure 1: This figure is much improved! A couple of suggestions to polish it. First, the big Africa map needs to be updated to include South Sudan. I also think it would look better if the authors just darkened the outline of Uganda in the big Africa map (rather than the rectangle), because the big Uganda map is just an image of Uganda. Then use two lines extending from the corners of the box marking the study area in the Uganda map to extend to two corners of the study area map. Also crop out that line above Uganda. A few things about the caption:

1. Specify that the boundary of the park is indicated by the SOLID black line.
2. The GREEN (not yellow) areas represent the Buganda formation
3. Specify in the caption what the dashed black lines mean

Figure 2: Remove all colors of the box and whisker plots. Because the X-axis labels are clearly marked, color is not necessary.

Figure 3: The pink shaded boxes in the two plots don't totally match in color. Also, with the box over the plots, it makes the coral circle markers (philopatric sex) appear red, so they don't match up with the legend (or are the markers in the plot actually a different color from those in the legend?). I also feel that the pink shaded areas indicating local range makes the figure look a little dated/like a draft. I think just a light grey box would look a lot cleaner.

Decision letter (RSOS-200760.R3)

Dear Dr Hamilton

On behalf of the Editors, we are pleased to inform you that your Manuscript RSOS-200760.R3 "Using Strontium Isotopes to Determine Philopatry and Dispersal in Primates" has been accepted for publication in Royal Society Open Science subject to minor revision in accordance with the referees' reports. Please find the referees' comments along with any feedback from the Editors below my signature.

Please submit your revised manuscript and required files (see below) no later than 7 days from today's (ie 07-Jan-2021) date. Note: the ScholarOne system will 'lock' if submission of the revision is attempted 7 or more days after the deadline. If you do not think you will be able to meet this deadline please contact the editorial office immediately.

Please note article processing charges apply to papers accepted for publication in Royal Society Open Science (<https://royalsocietypublishing.org/rsos/charges>). Charges will also apply to papers transferred to the journal from other Royal Society Publishing journals, as well as papers submitted as part of our collaboration with the Royal Society of Chemistry

(<https://royalsocietypublishing.org/rsos/chemistry>). Fee waivers are available but must be requested when you submit your revision (<https://royalsocietypublishing.org/rsos/waivers>).

on behalf of Prof Kevin Padian (Subject Editor)
openscience@royalsociety.org

Associate Editor Comments to Author:

A few changes/comments remain to be addressed, but given the efforts you have made to improve the paper thus far, it is hoped these will be relatively minor and - assuming you prepare a thorough final revision and rebuttal - it is unlikely further review will be necessary. While it seems there are divergent community views on the subject of the work, it would seem more valuable at this point to publish the revised version of the paper and allow the community to engage with the work in a public forum. Good luck with the revisions!

Reviewer comments to Author:

Reviewer: 3

Comments to the Author(s)

All good, with one minor exception, on line 46: "Tooth enamel forms early in life and then is metabolically inert, although different enamel locations within the same tooth do calcify at different rates." The point is rather that they calcify at different times, not different rates.

Reviewer: 4

Comments to the Author(s)

RSOS-200760.R3

Second Review

Overall I'm happy with the changes the authors made to the manuscript. However, upon a second (and perhaps deeper) reading, there are still two major issues that stand out to me.

I would like the authors to clearly address how the statistical clusters of samples they generated differ from using just the range of $87\text{Sr}/86\text{Sr}$ within the bounds of each study area. Why is the "standard archaeological method" one that takes into account data from geology rather than $87\text{Sr}/86\text{Sr}$ from the study area? I comment on this below in several places, but I'm really not sure why, for example, anyone trying to identify local vs non local individuals in the Sebitole area would include plant $87\text{Sr}/86\text{Sr}$ collected from the southern reaches of the park to generate their local Sr signature. Right now this seems a bit like a straw man argument. What I'd like to see is a set of boxplots (maybe grouping it in with figure 2) that shows the distribution of plant $87\text{Sr}/86\text{Sr}$ for each study area (for example, all the Sr data from within the dotted lines in figure 1). I think that's going to align much better with the statistical clusters. Then the authors can really get into why the statistical clustering method is better than using "local" Sr ratios.

One issue that I failed to comment on in the first round of reviews is that this paper does not mention previous archaeological studies that have used similar approaches (bone-tooth pairs) to exploring mobility. The other reviewer commented on this, but I don't think the authors added a thorough enough discussion in their background section. It would be good to see the authors contextualize their study with a clear discussion of how the approaches used in this study compare to those used in the following:

Price, T. Douglas, Linda Manzanilla, and William D. Middleton. "Immigration and the ancient city of Teotihuacan in Mexico: a study using strontium isotope ratios in human bone and teeth." *Journal of Archaeological Science* 27, no. 10 (2000): 903-913.

Grupe, Gisela, T. Douglas Price, Peter Schröter, Frank Söllner, Clark M. Johnson, and Brian L. Beard. "Mobility of Bell Beaker people revealed by strontium isotope ratios of tooth and bone: a study of southern Bavarian skeletal remains." *Applied Geochemistry* 12, no. 4 (1997): 517-525.

Schweissing, Matthew Mike, and Gisela Grupe. "Stable strontium isotopes in human teeth and bone: a key to migration events of the late Roman period in Bavaria." *Journal of archaeological science* 30, no. 11 (2003): 1373-1383.

Or this one, which compares two different teeth:

Slater, Philip A., Kristin M. Hedman and Thomas E. Emerson. 2014. Immigrants at the Mississippian polity of Cahokia: strontium isotope evidence for population movement. *Journal of Archaeological Science* 44(0):117-127.

It might also be worth it to mention other studies of mobility that use paired tissues (one metabolically inert tissue compared to a metabolically active tissue):

Del Rio, Carlos Martínez, Pablo Sabat, Richard Anderson-Sprecher, and Sandra P. Gonzalez. "Dietary and isotopic specialization: the isotopic niche of three *Cinclodes* ovenbirds." *Oecologia* 161, no. 1 (2009): 149-159.

As well as mention some of the studies looking at dispersals in other taxa (like birds!)

Hobson, Keith A., Leonard I. Wassenaar, and Erin Bayne. "Using isotopic variance to detect long-distance dispersal and philopatry in birds: an example with ovenbirds and American redstarts." *The Condor* 106, no. 4 (2004): 732-743.

Also this recent paper may be of interest:

Crowley, Brooke Erin, and Laurie Rohde Godfrey. "Strontium isotopes support small home ranges for extinct lemurs." *Frontiers in Ecology and Evolution* 7 (2019): 490.

Line 99: Do the authors mean that the Sr will be consumed by weathering?

Line 196 and 197: use \pm instead of +/-

Line 219: Wouldn't the standard archaeological approach be better described as one that uses local $^{87}\text{Sr}/^{86}\text{Sr}$, rather than $^{87}\text{Sr}/^{86}\text{Sr}$ assigned to particular geologies? I'm thinking of the Slater et al. 2014 paper on Cahokia. They do not use a geologic map here, but just samples from sites from the American Bottom.

Line 220: The authors should explain why they chose to use the $87\text{Sr}/86\text{Sr}$ values of plants only from the Toro formation as their “local” signature. According to Figure 1, Kanyawara sits on both the Toro and Buganda formations, pretty equally. What about comparing the primates from Sebitole to the Toro formation, since that one is pretty homogenous? And for the Kanyawara primates, how about just using the $87\text{Sr}/86\text{Sr}$ values that are within the study area? I also don’t think there is any information in the SI table that indicates clearly which animals came from which study area. Can the authors include a column in the table that indicates that?

Line 350: I’m not totally clear on why the authors used the mean value of all the plant samples collected from the Buganda formation to compare to the individual from Kanyanchu. The mean value is 0.7113, but clearly in Figure 1 all the plant samples from the Kanyanchu area are higher, closer to around 0.715, due to the proximity to that unidentified radiometric anomaly. I think it’s a good exercise to compare the values from the geology, but I think it would also be good to compare the primates from each study area to the plants that were collected from each study area.

After reading through this a second time, I realize I’m not totally clear on what the standard archaeological approach means. In some places I take it to mean using the range of $87\text{Sr}/86\text{Sr}$ of any plant sample collected from the geologic substrate on which a site is located. But, other times in the manuscript, like in the paragraph starting with Line 407 to line 411, it seems like the method I just described is compared to the “standard archaeological approach”.

Figure 1: This figure is much improved! A couple of suggestions to polish it. First, the big Africa map needs to be updated to include South Sudan. I also think it would look better if the authors just darkened the outline of Uganda in the big Africa map (rather than the rectangle), because the big Uganda map is just an image of Uganda. Then use two lines extending from the corners of the box marking the study area in the Uganda map to extend to two corners of the study area map. Also crop out that line above Uganda. A few things about the caption:

1. Specify that the boundary of the park is indicated by the SOLID black line.
2. The GREEN (not yellow) areas represent the Buganda formation
3. Specify in the caption what the dashed black lines mean

Figure 2: Remove all colors of the box and whisker plots. Because the X-axis labels are clearly marked, color is not necessary.

Figure 3: The pink shaded boxes in the two plots don’t totally match in color. Also, with the box over the plots, it makes the coral circle markers (philopatric sex) appear red, so they don’t match up with the legend (or are the markers in the plot actually a different color from those in the legend?). I also feel that the pink shaded areas indicating local range makes the figure look a little dated/like a draft. I think just a light grey box would look a lot cleaner.

===PREPARING YOUR MANUSCRIPT===

Your revised paper should include the changes requested by the referees and Editors of your manuscript. You should provide two versions of this manuscript and both versions must be provided in an editable format:
 one version identifying all the changes that have been made (for instance, in coloured highlight, in bold text, or tracked changes);
 a 'clean' version of the new manuscript that incorporates the changes made, but does not highlight them. This version will be used for typesetting.

===PREPARING YOUR REVISION IN SCHOLARONE===

-- Ensure that your data access statement meets the requirements at <https://royalsociety.org/journals/authors/author-guidelines/#data>. You should ensure that you cite the dataset in your reference list. If you have deposited data etc in the Dryad repository, please only include the 'For publication' link at this stage. You should remove the 'For review' link.

Author's Response to Decision Letter for (RSOS-200760.R3)

See Appendix G.

Decision letter (RSOS-200760.R4)

Dear Dr Hamilton,

It is a pleasure to accept your manuscript entitled "Using Strontium Isotopes to Determine Philopatry and Dispersal in Primates" in its current form for publication in Royal Society Open Science.

on behalf of Professor Kevin Padian (Subject Editor)
openscience@royalsociety.org

Appendix A

I did not look at the initial submission or the first revision of this manuscript, so I am commenting just on the current version and on the responses to comments from one previous reviewer. I realize this introduces the possibility of shifting the goalposts, as it were. Also, I am familiar with isotopic analysis, but not an expert, and I will trust the previous reviewer, who clearly is, and the authors' responses to that reviewer. So far as I am aware, previous reviews did not raise questions about the behavioral ecology of the primate species included in the analysis, so some of my comments will be the first time the authors have encountered such questions.

That said, I think the manuscript is quite interesting and the research has real value. Again, I am not an expert regarding isotopic methods, but the authors' "novel" method seems to hold considerable promise, although getting appropriate "environmental" signals from paleontological sites may be quite challenging. However, I have some substantive concerns and also will make some specific comments about the presentation of the material.

One substantive concern is the need to distinguish between locational dispersal (individuals move away from natal areas, or from areas to which they have already dispersed in space) and social dispersal (members of group living species emigrate from groups and immigrate into other groups). In a valuable 1996 paper, Lynn Isbell and Dirk Van Vuren made this important distinction clear (Isbell, L.A. & Van Vuren, D. Differential costs of locational and social dispersal and their consequences for female group-living primates. *Behaviour* 133:1-36). Individuals can undergo one, both, or neither of these processes. Male chimpanzees don't undergo either. Most female chimpanzees undergo both: they transfer between social groups and they move from one territory to that of another group. However, this is not universal, as the authors implicitly acknowledge when they state that researchers knew that one of the Kanyawara females had stayed in her natal community as an adult. A female gray-cheeked mangabey typically does not undergo social dispersal – females don't emigrate from their natal groups – but some females undergo locational dispersal when groups fission and establish separate home ranges, neither of which is identical to the original one (for a Kibale example, see K. Janmaat et al., 2009, *Intl J Primatol* 30: 443). Also, how exclusive home range or territory use could become an issue. Female guerezas apparently don't disperse from their natal groups – at least, this has not been documented in Kibale – but the home ranges of neighboring groups can overlap extensively, to the point where groups at Kanyawara can share virtually all of their home ranges with multiple other groups (T.R. Harris & C.A. Chapman 2007, *Primates* 48: 208-221); thus, a female could disperse socially, but not locationally, or at least locational dispersal would be very short. Their home ranges are small – although Harris & Chapman found their sizes can vary 5-fold – and overlap might not matter re. isotopic signatures (and male locational dispersal might occur on a scale that, on average, is sufficiently larger than the scale at which female home range occurs that you could still expect to pick up isotopic signals of dispersal reliably. But these issues may not be trivial. Dispersal is not a nice, clean categorical variable on an ecological time scale (even incorporating a third category for species in which individuals of both sexes disperse, at least conditionally – actually the case for most mammals, although sex biases in locational dispersal still often exist). I realize that this manuscript is not meant to be a review of primate dispersal strategies. However, I think you need to acknowledge this complexity. Moreover, you

might find it useful to think about the possible implications that variation in locational dispersal has for your over-arching research questions. For example, suppose females usually don't disperse *socially*, but home range stability is low and any individual female covers a wide habitat range over the course of her lifetime because her group shifts the areas they use? How much does home range overlap matter? To what extent could these complexities explain "mis-classifications" (a question that particularly occurred to me when I reached line 209 and saw that you correctly identified only 32% of "non-local" individuals)? I also worry about this when I see that you pooled data for redbills and blue monkeys and still had a sample of only 7. Yes, male natal social dispersal and female natal social philopatry seem to be universal in both, and redbills in Kibale are territorial...but...The bottom line is that these details of the behavioral ecology of the species in your sample need some acknowledgement.

Similar questions kept occurring to me when you referred to "correctly classified individuals", although that touches on another point. If I understand correctly, you established your criterion for correct classification by assuming that (a) dispersal = locational dispersal; (b) this is a categorical variable; (c) each species in your sample shows a universal sex bias and can be put either in the category "male (locational) dispersal/female (locational) philopatry" or the reverse. Thus, all male chimpanzees should show local isotopic signals (low offsets), for example, and any who don't are "mis-classified". For a start, it would greatly help readers if you state very clearly in the Methods your definition of "classification and your criterion for deciding whether someone was correctly classified. So, for example, in the par. starting on 221, you state that 67% of individuals were either members of the "dispersing sex" with high offsets members of the "philopatric sex" with low values. Is this just a way to say that we assumed that all females of species a, b disperse and all males of species d,e, and f disperse – locationally – that any male of species d with an offset below the mean is "mis-classified"? This seems to be the case, and if it is, you could make it much clearer. If it isn't, then perhaps you still need some clarification.

This brings up a more general substantive issue, one that the previous reviewer also raised in a somewhat different way: at multiple points in the manuscript, you have not really made it clear what the question is. Readers can infer this, but they shouldn't need to. For example:

Par. starting 88: "To address this..." What is "this"? You have not stated a question or explicitly framed an issue.

105: "bias" with regard to what? Again, you haven't clearly stated the issue.

Abstract:

15: "between faunal teeth and bone and/or local environment" – this would be much better/clearer as "by calculating offset values for comparison of teeth to bone or to local environments".

17: Bonobos are also "humans' closest living relatives".

19: Insert a comma after "landscape", and please check the rest of the manuscript closely for places where you should insert commas (e.g., put a comma after "plants" in 39).

Introduction:

27: should be “consequences for”

27: “The ancestral condition” for what?

28: This list (“aggression”, etc.) is not one of “abstract social parameters”. It is one of variables that we cannot directly infer from the fossil record.

30: “Pair-bonding” is uniquely human. Nor, for that matter, are “extended life histories”. You need a more nuance/clarity here. Exactly what are you comparing humans to?

33-34: “Employed more direct proxies” to do what?

34: “Vary by surface geology” should be “vary in association with variation in surface geology”.

36: Can they use the landscape *non-spatially*? Just “use the landscape”, or “space use”.

47-48: Just say “if the fossilized individuals died at or near the depositional sites”.

Par. starting 50: The “chimpanzee referential model” has come in for much criticism. I don’t agree with all the criticism, by any means, but one serious problem has been the paired assumptions that (a) recent hunter-gatherers were characterized by male dispersal and female philopatry, and (b) therefore this characterized the LCA. In fact, (a) is incorrect, at least with regard to ethnographic documentation of recent H-G residential patterns. This leaves the question of the LCA open. I think you need to acknowledge this.

Par, starting 62: This is very awkwardly written. (For one thing, here – a minimum...ratios” -- and elsewhere your subject and verb persons – singular vs. plural – don’t agree). This may seem like pedantry, but avoiding such awkwardness makes it more likely your audience will get your points.

Methods:

155: Should be “Once the samples cooled, we spiked them...”

136: Omit “down”

153-155: Readers can use your tables to figure out how many individuals were mis-classified and, importantly, how many males and females were mis-classified, but it would be nice to see summary statements in the text.

183-184: This gets back to a point I made above: “correctly classified” as what? As male or female?

Results:

197: You must mean “the greatest **straight line** distance” -- ? That is, the maximum diameter for a hypothetical circular “small” home range would be 3 km, and this would be the longest distance anyone could move in a straight line without leaving it.

214: What is “trended” supposed to mean?

225-226: As noted above: come of the “bias” toward “mis-identifying” individuals of the dispersing sex could be because they really did Not disperse locationally, at least not on a scale sufficient to affect isotopic signatures.

238-239: Fig. 4 shows HIGHER values for the philopatric sex -- ?? Are the labels in the legend wrong? (Or am I missing something?) Also, the figure doesn't contain any info about significance values.

264: A lesson I learned long ago: If you think it necessary to write “In other words”, you didn't say it right the first time.

267: “Isotopic ratios from botanical samples” – obviously you can't do this with fossil sites. As you mention below (in response to the previous reviewer's comments), micromammals and maybe paleosols possibly provide alternative ways to gain insights. But I think you need to make a clearer argument here, maybe by analogy: our results show that isotopic data derived from vegetation samples make acceptable proxies for saying “x” about sex differences in dispersal; we can't get such data from Miocene and Plio-Pleistocene fossil sites relevant to hominin evolution, but we could instead use data on “a” and “b” because...”.

273: “Likelihood of an individual being local...”: again, you are assuming (a) dispersal = locational dispersal, and (b) when a sex bias exists, it is absolute: 100% of males or 100% of females disperse locationally.

281: Here, despite your bootstrapping efforts, I am a bit puzzled. Dispersal is a categorical variable, and you assume the sex bias is universal: either 100% or 0% of males disperse, and 100% or 0% of females disperse. Nothing in between is possible. So, every female disperses, or none of them do. Every male disperses, or none of them do. The isotopic data get this right 50% of the time. I think I understand your underlying argument, but I also think you still need to do some work to make it completely clear.

302: You cite one reference; this does not make the method “commonly used”. Actually, my non-specialist understanding is that analysis of micromammals at Miocene & Plio-Pleistocene hominin sites is not common, in no small part because it is labor-intensive and not glamorous. In fact, it might be good if you modify this to highlight its usefulness (and you could cite Denne Reed's work) and put out a call to make it more common.

304: “but better methodological...” would be better as just “before we could confidently use these methods”.

306: “geochemicalLY”

309: Omit “also”

and Table 1: There are better references than Chapman & Wrangham (1993) for Kibale chimp home range sizes! Notably, for Kanyawara, see Wilson et al. 2012, *Anim Behav* 83: 277, and for Ngogo see Mitani et al. 2010, *Cur Biol* 20 (12) R507-508. Also, chimpanzee home

ranges can be $> 100 \text{ km}^2$ (Moore et al. 2012, *J Hum Evol* 112: 30-40). Some are 15 sq km or smaller, but that is really toward the low end of the range of variation.

312: Here and elsewhere, don't use italics. Make it clear from your prose what you think is most important.

315-316 (see also my comments above): I'm not sure this quite holds up logically. Small home ranges could mean that mean dispersal distance is too short for dispersers, as adults, to be in areas with isotopic signals different from their natal areas.

329: I don't think you've said anything about the potential of isotopic data to tell us about social behavior. Better to omit this, unless you can state explicitly what you think it could tell us.

337: Which approaches? You presented three, then concluded that one was most reliable, although ecological scale matters. This isn't a very strong concluding statement; it rather undermines some of what you said above.

Appendix B

Dear editors:

We are deeply appreciative of the time and investment the reviewers put in to improving this manuscript. We have included our responses (in bold) to each of the reviewer comments (*italicized*) below. The line references within this document refer to the clean manuscript copy included with this resubmission, although we include a Tracked Changes version of the manuscript as well for reference. We believe that this work is substantially improved by the included edits and thank you in advance for your consideration on its resubmission.

Reviewer 1 Comments:

Hamilton et al. present a new method for distinguishing local and non-local (philopatric and dispersing) individuals and species that relies on the offset (absolute value) between $87\text{Sr}/86\text{Sr}$ of tooth enamel and bone or tooth enamel and the local environment (using plants). The authors have incorporated a number of my previous suggested changes and additions, and these have definitely improved the readability and flow of the text. However, there are still some issues that need to be addressed before I think this is publishable. I outline these below.

First, I note that there are multiple mistakes in the text. In addition to the normal expected typos, there are sentences in the methods that are written more than once verbatim, and some table captions are incorrect (e.g., Table 1 does not include any isotope data).... This lack of attention to detail is concerning. I have done my best to repeatedly correct errors in this manuscript but the authors need to be responsible for the quality of their own work. At best, an editor might catch mistakes, but if things get published that are incorrect, that would be disappointing for everyone.

We agree that any published mistakes would be terribly disappointing, at that it is neither the job of the editor nor the reviewers to ensure the manuscripts are error-free. We have done multiple thorough read-throughs of the text and truly believe it is clean and free from typographical mistakes as we can be reasonably sure of.

Second, I think that the offset method that the authors present definitely has promise. The language currently used in the introduction (lines 70-77) is nicely written, and it makes sense. I agree that using offsets to differentiate dispersing and philopatric individuals (or to identify which sex is dispersing for a species) could be a great approach. However, I think that HOW the authors present why the method is novel is rather weak. The authors focus mostly on emphasizing that their study is novel because they establish geochemical clusters on the landscape. For example, in the Conclusion paragraph, they write “the present study highlights that it is critical to establish meaningful geochemical clusters on the landscape thorough sampling and mapping, as geologic boundaries do not by default correspond to unique isotopic clusters” (lines 331-333). I don’t think that most researchers have previously assumed that geologic boundaries by default correspond to unique isotopic clusters. This is why empirical data are included from local animals or plants (to verify what local values are, typically on different geologies). This is nearly ubiquitous across strontium studies (and has been the standard practice for decades, as I pointed out in my previous review).

Instead, I think it’s the second point that the authors mention in the introduction (but don’t revisit later) that makes this method novel: i.e., that calculating offsets allows one to assess the likelihood that an individual is local or non-local based on how it compares to other members of the same species (or community) rather than relying on those maxima and minima designated by the local baselines.

We agree that our most substantial contribution is the development of the offset method described here. We see how the previous structure and wording of the manuscript places a heavier emphasis on the use of statistical clustering rather than the offset method itself. While this is (to our knowledge) the first time cluster analysis has been used with strontium isotope ratios to form isoscapes in this way, that is more of a “means to an end” and not the most important contribution within this paper. We have significantly decreased the discussion of the clustering method in the Discussion to highlight the offset approach as the most central offering of this paper. We did revisit this second idea in the Conclusion in the previous manuscript version but see how it was overshadowed by the unnecessarily long discussion of the clustering method in the same paragraph. We hope that the manuscript is now more balanced towards the most important contributions (lines 271-286).

Third, the Discussion is still lacking some important aspects. Mismatches between expected and observed numbers of philopatric vs. dispersing individuals could be due to sampling teeth that mineralized after dispersal, inaccuracies in the available geologic map, heterogeneity in the geologies themselves (see my note below regarding text on lines 91-95 below), and incorrect assumptions about the degree to which individuals truly are catholic to expectations of dispersing and not dispersing. I think all three of these points need to be touched on in the Discussion.

We have restructured the discussion to more explicitly address each of these salient and important points one by one. The sections addressing each are indicated below each of the suggestions below.

1. I would like to know how much tooth selection might affect offset results. The authors now explicitly mention that they tried to sample M1 for all individuals (and they have included which tooth was sampled in the supplementary table), but they do not revisit how results might be affected for those individuals that had a different tooth sampled. In particular, I would think that M2 might have erupted post dispersal. I think that the authors need to: (1) Mention explicitly that there were six individuals that had a tooth other than M1 sampled in the Methods (after lines 116-117); (2) Be careful to only refer to tooth (not the earliest forming tooth and not the earliest forming molar) elsewhere in the text; and (3) Revisit if offset might be off for individuals that had a tooth other than M1 sampled (particularly pertinent for later-forming teeth like M2).

We have included a section addressing this on lines 328-342.

2. I still want to know how isotopic clusters for plants compare with the underlying geology (I have asked this previously). The authors repeatedly state that their method of developing isotopic clusters for environmental baselines from plants is far superior to working with preconceived notions of geology, but their justification for this is very vague and revolves around statements about geologies sometimes being indistinguishable (e.g., lines 91-95; lines 264-266). More importantly, the authors still do not compare their “isotopic clusters” with what would be expected based on underlying geology. I want to see what data look like for plants growing on the different geologic formations. I had also asked previously if the authors could please add the expected geologic boundaries for the different formations to Figure 1 (or perhaps make a complementary panel). I still think this would be valuable. Then the reader can see if plants that look unusual are at or near an expected geologic boundary (or not). The issues with poor geologic maps and old geologies very much need to be brought up in the Discussion.

We agree that the role of ancient geologies is a critical topic to address and have included a section on the implications of old geologies in the Discussion (lines 343-360). We have additionally discussed how these difficulties manifest in Kibale specifically on lines 361-370. We have added a geologic map for comparison to our clusters to Figure 1.

3. I think the authors need to briefly discuss how their assumption of dispersal being nearly ubiquitous for the dispersing sex (and vice versa) might impact “accuracy” of strontium isotope predictions. In the Methods, the authors mention that they assume that 100% of individuals in dispersing sex disperse and 100% of individuals in philopatric sex stay put (without any justification for this decision). They note at least one known example at Kibale where this is not true (a female chimp), but I would think that this individual isn’t an isolated incident. This is important because the “True” number of philopatric and dispersing individuals is not actually known, and implying (or explicitly referring to) any expected number as “true” is misleading. The authors need to please add “expected” or “assumed” in front of all mentions of “correct” classification of individuals. For example, the text on lines 208-209 needs to be “83% of ASSUMED local/philopatric individuals and only 32% of ASSUMED non-local/dispersing individuals were correctly identified”. They should also make sure that they discuss how this assumption may or may not affect their results and interpretations of the data in the Discussion.

We have included an explicit discussion of our model assumptions regarding the fidelity of dispersal patterns (and lack thereof) in the Discussion (lines 371-386). Furthermore, we have changed our language throughout the manuscript to indicate we are comparing assumed dispersal based on *expected* species-level sex-biases to the *assigned* dispersal attribute (local/non-local) from the isotopic data. We hope that this explanation is not only clearer, but more importantly that it is a more accurate reflection of the modeling we’ve done to test these different approaches.

Smaller points:

Keywords - It is my general understanding that words that are in the title aren’t also needed in the keywords. Please add the country where the study took place, and maybe include “plants” and/or names of the more interesting primate taxa. Also consider including key words related to statistical methods.

**We appreciate this insight into choosing proper keywords. We have updated them as follows:
Keywords: Uganda, chimpanzee, hominin evolution, cluster analysis, landscape use (Line 23)**

There are several instances of data interpretation in the Results (lines 218-220; 239-240; 245-250). To me, these would fit better in the Discussion.

The sentences referred to above have been removed from the Results section and their content integrated throughout the Discussion section.

Reference to a single case study (or even two case studies) cannot support a claim that something is common or typical. In particular, citing a single archaeology study and palaeoanthropological paper to support what the authors are calling “the standard archaeological method” is disappointing. If this is truly a standard method, surely the authors can cite more references to support this point.

We have included additional citations demonstrating the use of this method (line 65).

The authors still have not told the reader what plant tissues were sampled. This is my third time requesting that this information please be provided. “Botanical sample” could be fruit, leaves, flowers, stems, whole plants, etc. If the authors always collected leaves, say so. If different plant tissues were sampled, then please provide these details in the supplementary table.

We truly regret this this information was not forthcoming in the previous manuscript submissions; we certainly did not intend to be obtuse or difficult. All botanic samples are plant leaves. This information has been added throughout the text (ex: line 101) and in the title for Supplementary Table 1.

Tables:

- The tables are hard to follow as presented. The formatting is quite awkward. In general, starting each table on a new page is reader friendly.

We have edited the document to start each table on a new page. We have also edited the column names and bold formatting throughout the table to increase readability.

- Table 1 presents no isotope data (contrary to what the caption states)

The caption has been corrected, thank you for pointing out this error.

- Full references belong in the reference cited section (not as table footnotes)

We have moved these references into the proper section.

- In the caption for Table 2, “Falling outside of local environment minima or maxima = local/philopatric attribution” is incorrect.

Thank you. This has been corrected to read “Falling *inside* local environment minima or maxima = local/philopatric attribution.”

- Table 4 – what is being compared (i.e. what does “proportion of significant ($p < 0.05$) bootstrap simulations” mean)?

This title has been updated to be more clear: “Table 4 - Proportion of bootstrap simulations in which there was a significant difference ($p < 0.05$, Student’s t-test) in offset values ($^{87}\text{Sr}/^{86}\text{Sr}_{\text{enamel-bone}}$ and $^{87}\text{Sr}/^{86}\text{Sr}_{\text{enamel-environment}}$) between males and females”

- In tables 2, 5, and 6, what are the cutoffs for determining if something is conclusive or not? Please clarify in a footnote.

We have moved information from the table headers into the footnotes, with each column individually referenced and explained, in order to increase the clarity of the information being presented in these tables.

- **IMPORTANT:** As I mention above, the wording regarding number of dispersing versus local individuals in these tables is misleading. Bolding and underlining true in these table headers makes this even more misleading. As mentioned above, the “TRUE” # of locals and non-locals is unknown! “Assumed” or “expected” are the words the authors should be using. And the authors really can’t be looking at ACCURACY of attribution given that the “true” information is assumed in the first place. They can assess and compare attributions.

We appreciate the suggestions to make our wording more accurate and less misleading to the reader. We have updated the tables to use “expected” instead of “true” and updated the table headers to indicate a comparison of attributions (the expected, based on the sex and species, and the assigned attribution based on the isotopic data and the specific method being tested).

- The organization of the tables is perplexing to me. Why does Table 3 come before Tables 4-6?

- How do Tables 3 and 7 differ? Presumably one is species level and the other individuals? More

informative captions would be helpful (“Summary of Method Efficacy” is not sufficient). Are there other places besides Line 250 where Table 7 should be cited?

Upon reflection looking at our tables, we think the source of this confusion is that (previous) Table 3 is redundant to (previous) Tables 2, 5, and 6. (Previous) Table 7 serves as a more informative summary comparison of all methods than (previous) Table 3. We have updated (current) Table 6 (previous Table 7) to have a more informative title (“Summary Table: Were methods able to correctly identify the dispersing sex for each primate species?") and detailed footnotes for each column explaining the definition of ‘effective.’ We reference it in lieu of (previous) Table 3 throughout the text. We regret the confusion that this extra table caused and appreciate the dramatic increase in clarity that these changes provide.

- Table S2. How are entries organized in this table? I had initially thought species, locality, sex, but in some cases, the sexes are mixed up for each species at each locality.

The Table is now correctly organized by species, then location, then sex.

- It would be helpful to provide summary offset data for each sex for each species in addition to species means. Please remember that numbers <1 should have a 0 before the decimal place.

We have included the averages by sex under the species averages for each species and added the “0” in front of each decimal as needed.

Figures and captions:

- Figure 1 - I have asked before about the geologic map for the region. Even if it is rather generalized, it would still be helpful (and provide reader transparency) to show the expected boundaries of the different units relative to the plant collection sites and primate habitats.

We have included a version of the Figure 1 map showing the underlying geology of the area in relation to the plant samples collection sites and the park areas mentioned in the text. We have also included a boxplot showing the isotopic ratios of the plants collected on each geologic layer. We agree that this is very helpful context and compliments the increased discussion of the complex nature of the ancient felsic geology of the region coupled with the lack of detailed geologic maps.

- For Figure 4, are all of the details regarding significance of bootstrapped offsets for each species needed both in this caption and in the main text? Please remind the reader what is being compared in these tests (e.g., sexes?).

We have removed the data from the caption, as we agree it is redundant to the main text. We have revised the remaining caption to make clear that these are comparing offsets between sexes, by species.

Line by line things:

Lines 12-13: “The present study demonstrates that standard archeological methods used to determine ‘local’ and ‘non-local’ individuals are not reliable when applied to dispersal patterns in modern primate communities” needs context. Readers will not inherently know what the authors are talking about. Briefly define what the method is that the authors are considering standard (and presumably this should be method, and not plural methods?).

A very brief explanation has been added to provide context

Lines 18-19: “relative to the isotopic clusters on the landscape” needs context. It will not be clear what this means to someone who is just starting to read the paper. Clusters in baseline bioavailable $^{87}\text{Sr}/^{86}\text{Sr}$ obtained from plants? Is this text needed?

We agree that this clause (“relative to the isotopic clusters on the landscape”) can be omitted in the abstract for clarity.

Line 39 (and elsewhere): “ $^{87}\text{Sr}/^{86}\text{Sr}$ ratio” is redundant. Either strontium isotope ratios, OR $^{87}\text{Sr}/^{86}\text{Sr}$.

This has been changed to simply “ $^{87}\text{Sr}/^{86}\text{Sr}$ ” throughout the manuscript.

Line 40: Tissues should be plural here

We have corrected this.

Lines 54-55: Consider if the sentence starting with “chimpanzees” should be moved up one sentence. This could help with the flow of the text here.

We agree that this helps with the flow of this paragraph and have made the correction.

Line 57: “if” or “degree to which” would be better choices. “Whether” requires “or not”.

We have changed this to “if.”

Lines 59-61: The authors looked at multiple species of primates. That’s still not clear at this point in the text (it is hinted at on line 69 but doesn’t become fully evident until much later in the text). Rather than say “in a modern primate habitat”. I would recommend “In sympatric primate species with varying home ranges, philopatry, and dispersal patterns” or “Within a modern primate community”. Please also mention the site name here or below on lines 69-70.

We have changed “habitat” to “community,” and mention the name of the site (Kibale National Park, Uganda) on line 73.

Line 62: Citing a single archaeological study and one palaeoanthropological study is not sufficient for supporting a statement that this is the most common method in archaeological strontium isotope literature!

We had added additional citations to this statement.

Lines 74-76. Please clarify that “environment” is synonymous with local bioavailable $^{87}\text{Sr}/^{86}\text{Sr}$ or baseline $^{87}\text{Sr}/^{86}\text{Sr}$.

We have added this language to the text on line 78-80.

Lines 88-95 – I think adding text like what has been added here is helpful, but it’s also confusing as written. The authors refer to data from botanical samples here but then also refer to botanical data to address the issue of similar botanical data in the next paragraph... I’m also not sure that all of this information belongs here in the Methods. Some of it appears to be results, and justification for the authors’ argument that their new offset method is preferable to other methods. This is absolutely an

appropriate place to discuss limitations of available geologic maps. the text could be reorganized slightly in this paragraph to make that point clearer up front rather than halfway down the paragraph. However, the text on lines 91-95 definitely does not seem like Methods to me. It also seems quite vague. Please revisit wording and be careful to focus on Methods (and not results) in the Methods. Also, note that “pattered” should be patterned on line 89.

We have moved the discussion about the overlapping isotopic ratios on the two major geologic formations to the Results section (lines 214-216). We have also re-worded this section to better explain our cluster analysis approach to defining ‘local’ areas. We explain how this approach is helpful in both places without detailed geologic maps, and/or in areas with true isotopic overlap between neighboring geologic areas.

Here, or perhaps in the Discussion, please explain how/why different geologic formations may be isotopically difficult to distinguish (in the context of what the geology is expected to be at Kibale). My understanding is that very old felsic geologies (e.g., Paleoproterozoic metamorphic rocks) have elevated $87\text{Sr}/86\text{Sr}$ (typically some of the highest measured) due to their age, and that they can be quite isotopically heterogeneous due to differential weathering of minerals over time as well as bioavailable Sr contributions from exogenous sources (like dust). I think this context would be very helpful for the reader, who is likely not familiar with any of this. Please add.

We agree that this is helpful and important context. We have put a section about this topic in the Discussion, as suggested (lines 343-360).

Lines 107+ - This is the meat of the study but the way it is worded makes it sound secondary (deleting also on line 107 would help a lot). Consider if this should be mentioned before the details of how the authors accounted for local variability in bioavailable $87\text{Sr}/86\text{Sr}$.

We very much like the improved flow and emphasis that comes with reordering the faunal and botanical sample collection descriptions. We have made these changes.

Line 114 – Site Table 1 here rather than Table S2.

We have made this adjustment.

Lines 114-119 – Site Table S2 after this text as it provides details about which element and tooth was sampled for each individual. As mentioned above, please make sure to add explicit mention of the six individuals who had a tooth other than M1 sampled.

We have added text explicitly mentioning the 6 individuals with teeth other than M1 sampled (lines 97-98) in the Methods. We more fully discuss the implications of this in the Discussion section of the paper (lines 328-342), including a discussion of the eruption times of these teeth (from Smith et al 1994) and a recognition that this could impact the outcome of our models.

Lines 120-132 – There’s redundancy in the text here (e.g., the lab name and location are mentioned twice). First finish the thought about plant sample collection (making sure to mention what tissues were sampled). Then focus on sample prep and analysis. Some information regarding sample prep is still missing. How were bone and tooth powdered? Did the authors grind up entire teeth or did they remove enamel? Did they chemically pretreat samples in any way? Also note that on Line 122, “on” should be “in”.

The redundancies have been corrected. We have added in information about enamel and bone sample collection on lines 91-93. Samples were not chemically pre-treated beyond the procedures currently described in the Methods. “On” has been changed to “in” as suggested.

Line 143 – Consider if this should be a new subheading. It’s data analysis (a separate thought from sample acquisition and analysis).

We like the idea of subheadings within Methods and have added them to differentiate sample collection, laboratory analysis, and data analysis.

Lines 144-146 – Please justify this assumption.

We have added in a clear explanation for this assumption, which we recognize to be an idealized model of reality. We have provided an explanation of its limitations as well as a justification for its utility lines 150-168.

Line 160 – The authors did sample the earliest forming molar for all individuals. Reword.

This has been reworded to simply “tooth.”

Line 194 – there’s a typo “±-“

Thank you – this is corrected.

Line 198 – “was” seems awkward here. Would be? Is?

We agree that ‘would be’ flows more smoothly here.

Line 199 – Fauna should be capitalized

Thank you – corrected.

Lines 203-205 – Methods are laid out in the Methods section. No need to restate this here (but please do make sure that Methods are sufficiently detailed). Should “Standard Archaeological Method” be capitalized in the subheading?

We have removed this explanation from this section and integrated the relevant details throughout the Methods section.

Lines 208-209 – As mentioned above, I think that throughout the text the authors need to be clear that these are EXPECTED local/philopatric or non-local/dispersing individuals. This is only actually known for one female chimpanzee.

We have reworded the text throughout the Results section to match the language in Table 2 (expected vs assigned attribution).

Lines 215-218 –Please remind the reader what the significance is referring to here. These species had significant what? Significantly different offset in $87\text{Sr}/86\text{Sr}$ between sexes? Expected dispersing and non-dispersing individuals? Does this language need to be included both in the text and the figure caption?

We have reworded this to clarify that the comparison is between the offset values for each sex.

Lines 256 - 257 – Please clarify ONE standard approach (not the only standard approach). No need to state this again on Line 257 (just write “this approach”).

We have made these corrections and clarifications.

Lines 258-260 - The wording here is confusing. minima and maxima are also isotopic ratios themselves. Please reword or cut.

We have cut this section per your earlier recommendations.

Lines 260-270 – I think this information would fit much better in the introduction where the authors introduce the method rather than down here in the Discussion. There’s some redundancy with what is said earlier, but also some novel information. Please move up and integrate with text in the intro and cut redundant wording.

We have moved the necessary information from this section to lines 117-123, where the method is introduced. We agree that this greatly increases clarity in both areas and appreciate the suggestion.

Lines 262-264 – The authors cite one example here but refer to multiple previous studies. Please revisit wording. Also, as written, this is pretty vague and kind of misleading. the authors need to please make sure that they are clear that while this can happen, it’s VERY context dependent. The geologies could also have entirely discrete $87\text{Sr}/86\text{Sr}$. IT entirely depends on what the local geologies are. See my notes above about the complexities of very old, felsic geologies.

We have included this point in our response on line 117-123, with an emphasis that this is not an issue that we expect in ALL areas, but one that is very geologic context-dependent.

Lines 300-301 – This needs to be reworded. Underlying geology will ALWAYS be older than something deposited on top of it with rare exceptions (e.g., flood deposits). What’s key is that the surface geology has been consistent over time (i.e. no major erosion or deposition since the organisms of interest expired).

We have reworded this sentence, thank you for the suggested correction.

Line 303 – is the idea of paleosols holding promise based on the authors’ own gut feelings or is there some existing study or publication that has suggested this that could be cited here?

To our knowledge, strontium isotope records from paleosols have not been used in this way nor suggested, although clearly other isotope data from paleosols is prolific in the literature. The use of paleosols for other isotopic records is the root of our idea here, and we include citations for a few pieces of that literature. Its implementation in this context is our suggestion for future further investigation.

Lines 308-321 – please cite Table 1 here, which provides home ranges for all of the species. I note that the authors state here that chimp home ranges can be up to 15 km² but Table 1 indicates chimp home ranges can be up to 27 km² (nearly two times bigger than 15 km²). Which is correct?

We have included a citation for the table and clarified the chimpanzee home range size, including insight from our other reviewer that such home range sizes can vary quite widely based on the type

of habitat the chimpanzee community is in (open habitats vs closed forest habitats, such as in Kibale).

Lines 337-340 - The paper sort of fizzles in its final sentence. The authors are repeating themselves here; they already pointed out in the beginning of this conclusion paragraph that “the previous standard approach” doesn’t work in all contexts. I would recommend rewording (e.g., it is reasonable to expect early hominins to share many of these features, suggesting that these methods may be able to reconstruct our own ancestors’ dispersal habits”), think bigger picture about what the next steps might be to help validate this method, or perhaps conjecture about the kinds of questions researchers may be able to answer using this new approach.

We have removed the redundancy and added some bigger-picture suggestions to end the paper. Thank you for this suggestion.

Reviewer #2 Comments

I did not look at the initial submission or the first revision of this manuscript, so I am commenting just on the current version and on the responses to comments from one previous reviewer. I realize this introduces the possibility of shifting the goalposts, as it were. Also, I am familiar with isotopic analysis, but not an expert, and I will trust the previous reviewer, who clearly is, and the authors' responses to that reviewer. So far as I am aware, previous reviews did not raise questions about the behavioral ecology of the primate species included in the analysis, so some of my comments will be the first time the authors have encountered such questions.

We are extremely glad to have an expert in primate behavioral ecology provide their input to this paper. It is a critical component to this work, and the clarification and nuance you provide around the messy business of dispersal and philopatry is greatly appreciated. We hope that the following edits sufficiently address your concerns.

That said, I think the manuscript is quite interesting and the research has real value. Again, I am not an expert regarding isotopic methods, but the authors' "novel" method seems to hold considerable promise, although getting appropriate "environmental" signals from paleontological sites may be quite challenging. However, I have some substantive concerns and also will make some specific comments about the presentation of the material.

*One substantive concern is the need to distinguish between locational dispersal (individuals move away from natal areas, or from areas to which they have already dispersed in space) and social dispersal (members of group living species emigrate from groups and immigrate into other groups). In a valuable 1996 paper, Lynn Isbell and Dirk Van Vuren made this important distinction clear (Isbell, L.A. & Van Vuren, D. Differential costs of locational and social dispersal and their consequences for female group-living primates. *Behaviour* 133:1-36). Individuals can undergo one, both, or neither of these processes. Male chimpanzees don't undergo either. Most female chimpanzees undergo both: they transfer between social groups and they move from one territory to that of another group. However, this is not universal, as the authors implicitly acknowledge when they state that researchers knew that one of the Kanyawara females had stayed in her natal community as an adult. A female gray-cheeked mangabey typically does not undergo social dispersal – females don't emigrate from their natal groups – but some females undergo locational dispersal when groups fission and establish separate home ranges, neither of which is identical to the original one (for a Kibale example, see K. Janmaat et al., 2009, *Intl J Primatol* 30: 443). Also, how exclusive home range or territory use is could become an issue. Female guerezas apparently don't disperse from their natal groups – at least, this has not been documented in Kibale – but the home ranges of neighboring groups can overlap extensively, to the point where groups at Kanyawara can share virtually all of their home ranges with multiple other groups (T.R. Harris & C.A. Chapman 2007, *Primates* 48: 208-221); thus, a female could disperse socially, but not locationally, or at least locational dispersal would be very short. Their home ranges are small – although Harris & Chapman found their sizes can vary 5-fold – and overlap might not matter re. isotopic signatures (and male locational dispersal might occur on a scale that, on average, is sufficiently larger than the scale at which female home range occurs that you could still expect to pick up isotopic signals of dispersal reliably. But these issues may not be trivial. Dispersal is not a nice, clean categorical variable on an ecological time scale (even incorporating a third category for species in which individuals of both sexes disperse, at least conditionally – actually the case for most mammals, although sex biases in locational dispersal still often exist). I realize that this manuscript is not meant to be a review of primate dispersal strategies. However, I think you need to acknowledge this complexity. Moreover, you might find it useful to think about the possible implications that variation in locational dispersal has for your over-arching research questions. For example, suppose females usually don't disperse socially, but home range stability is low and any*

individual female covers a wide habitat range over the course of her lifetime because her group shifts the areas they use? How much does home range overlap matter? To what extent could these complexities explain “mis- classifications” (a question that particularly occurred to me when I reached line 209 and saw that you correctly identified only 32% of “non-local” individuals)? I also worry about this when I see that you pooled data for redtails and blue monkeys and still had a sample of only 7. Yes, male natal social dispersal and female natal social philopatry seem to be universal in both, and redtails in Kibale are territorial...but...The bottom line is that these details of the behavioral ecology of the species in your sample need some acknowledgement.

You are absolutely correct that we are using an overly simplistic version of dispersal to test the efficacy of these isotopic models, and that this deserves to both be recognized, described, and the limitations discussed. This is very helpful language and citations to discuss it, and we are very appreciative. We have included a brief acknowledgement of our model assumptions in our Methods section (lines 150-168) and a more detailed discussion of the ecological details and their implications for our results in the Discussion section (lines 371-386)

Similar questions kept occurring to me when you referred to “correctly classified individuals”, although that touches on another point. If I understand correctly, you established your criterion for correct classification by assuming that (a) dispersal = locational dispersal; (b) this is a categorical variable; (c) each species in your sample shows a universal sex bias and can be put either in the category “male (locational) dispersal/female (locational) philopatry” or the reverse. Thus, all male chimpanzees should show local isotopic signals (low offsets), for example, and any who don’t are “mis-classified”. For a start, it would greatly help readers if you state very clearly in the Methods your definition of “classification and your criterion for deciding whether someone was correctly classified. So, for example, in the par. starting on 221, you state that 67% of individuals were either members of the “dispersing sex” with high offsets members of the “philopatric sex” with low values. Is this just a way to say that we assumed that all females of species a, b disperse and all males of species d,e, and f disperse – locationally – that any male of species d with an offset below the mean is “mis-classified”? This seems to be the case, and if it is, you could make it much clearer. If it isn’t, then perhaps you still need some clarification.

We have changed the language here to avoid confusion around the word “correct.” Instead, we have reframed our results to indicate the proportion of individuals assigned a *matching* attribution (local or non-local) based on the isotopic data compared to the *expected* attribution from our simplified dispersal model (that all females chimpanzees and all female red colobus disperse, while all males remain philopatric, etc). References in the previous draft to “correct classifications” really meant that the isotopic data indicated the same attribution as was expected based on sex and species (and locational dispersal, as we discuss in the Discussion section). We appreciate your insight that the way this was previously written was not clear, and we hope that this version is much improved.

This brings up a more general substantive issue, one that the previous reviewer also raised in a somewhat different way: at multiple points in the manuscript, you have not really made it clear what the question is. Readers can infer this, but they shouldn’t need to. For example:

Par. starting 88: “To address this...” What is “this”? You have not stated a question or explicitly framed an issue.

We have reworded this section to more clearly indicate the problem with isotopic overlap between geologic formations (lines 117-123).

105: *“bias” with regard to what? Again, you haven’t clearly stated the issue.*

We have removed this particular phrase, as we agree it was unclear and did not add value to our argument.

Abstract:

15: *“between faunal teeth and bone and/or local environment” – this would be much better/clearer as “by calculating offset values for comparison of teeth to bone or to local environments”.*

We agree, and have made this correction. Thank you for the recommendation.

17: *Bonobos are also “humans’ closest living relatives”.*

We tried to reword to include this, but it did not flow well in an abstract. We’ve removed the clause as we do not want to be imprecise.

19: *Insert a comma after “landscape”, and please check the rest of the manuscript closely for places where you should insert commas (e.g., put a comma after “plants” in 39).*

We have closely edited the paper for proper punctuation and typos. Thank you for your careful eye and attention to detail here.

Introduction:

27: *should be “consequences for”*

Thank you – we have corrected this.

27: *“The ancestral condition” for what?*

We have reworded this sentence: “Ancestral behavior and socioecological conditions for the hominin clade are currently unresolved.”

28: *This list (“aggression”, etc.) is not one of “abstract social parameters”. It is one of variables that we cannot directly infer from the fossil record.*

Thank you – we have reworded this.

30: *“Pair-bonding” is uniquely human. Nor, for that matter, are “extended life histories”. You need a more nuance/clarity here. Exactly what are you comparing humans to?*

We have reworded this sentence to: “... the evolutionary trajectory of traits which differentiate humans from our fellow extant great apes...”

33-34: *“Employed more direct proxies” to do what?*

We have clarified “... to reconstruct behavior.”

34: *“Vary by surface geology” should be “vary in association with variation in surface geology”.*

We have reworded as suggested

36: *Can they use the landscape non-spatially? Just “use the landscape”, or “space use”.*

We have reworded as suggested.

47-48: *Just say “if the fossilized individuals died at or near the depositional sites”.*

We have reworded as suggested.

Par. starting 50: The “chimpanzee referential model” has come in for much criticism. I don’t agree with all the criticism, by any means, but one serious problem has been the paired assumptions that (a) recent hunter-gatherers were characterized by male dispersal and female philopatry, and (b) therefore this characterized the LCA. In fact, (a) is incorrect, at least with regard to ethnographic documentation of recent H-G residential patterns. This leaves the question of the LCA open. I think you need to acknowledge this.

We agree that this should not be presented as consensus, and have included reference to and citations of some of these critiques (lines 55-56)

Par, starting 62: This is very awkwardly written. (For one thing, here – a minimum...ratios” -- and elsewhere your subject and verb persons – singular vs. plural – don’t agree). This may seem like pedantry, but avoiding such awkwardness makes it more likely your audience will get your points.

We have changed the wording around in this paragraph to increase clarity.

Methods:

155: *Should be “Once the samples cooled, we spiked them...”* 136: *Omit “down”*

Corrected as suggested

153-155: Readers can use your tables to figure out how many individuals were mis-classified and, importantly, how many males and females were mis-classified, but it would be nice to see summary statements in the text.

We include this in the Results section.

183-184: *This gets back to a point I made above: “correctly classified” as what? As male or female?*

We have removed references to “correctly classifying” throughout the paper and reworded this for clarity: “We calculated the proportion of individuals with an assigned category (local/non-local) that matched their expected category based on sex and species dispersal patterns.”

Results:

197: You must mean “the greatest **straight line** distance” -- ? That is, the maximum diameter for a hypothetical circular “small” home range would be 3 km, and this would be the longest distance anyone could move in a straight line without leaving it.

Yes – we have added this for clarity

214: What is “trended” supposed to mean?

We have changed this to simply “had”

225-226: As noted above: come of the “bias” toward “mis-identifying” individuals of the dispersing sex could be because they really did Not disperse locationally, at least not on a scale sufficient to affect isotopic signatures.

We agree, and have included a discussion of this important point in the Discussion section (lines 371-386)

238-239: Fig. 4 shows HIGHER values for the philopatric sex -- ?? Are the labels in the legend wrong? (Or am I missing something?) Also, the figure doesn’t contain any info about significance values.

The graph was generated with the colored reverse – thank you for catching this error. We have corrected the graph and added *s to indicate significant differences between the sexes in > 90% of bootstrapped simulations.

264: A lesson I learned long ago: If you think it necessary to write “In other words”, you didn’t say it right the first time.

A lesson worth keeping! We have changed this section substantially based on suggestions from the other Reviewer, included removing this wording.

267: “Isotopic ratios from botanical samples” – obviously you can’t do this with fossil sites. As you mention below (in response to the previous reviewer’s comments), micromammals and maybe paleosols possibly provide alternative ways to gain insights. But I think you need to make a clearer argument here, maybe by analogy: our results show that isotopic data derived from vegetation samples make acceptable proxies for saying “x” about sex differences in dispersal; we can’t get such data from Miocene and Plio-Pleistocene fossil sites relevant to hominin evolution, but we could instead use data on “a” and “b” because...”.

We have substantially revised this section based on recommendations from the other Reviewer but have incorporated the spirit of this suggestion into Lines 312-325.

273: “Likelihood of an individual being local...”: again, you are assuming (a) dispersal = locational dispersal, and (b) when a sex bias exists, it is absolute: 100% of males or 100% of females disperse locationally.

Yes – we have included a discussion of this assumption and the biases implied with it in the Discussion section (lines 371-386). Thank you for highlighting this important set of assumptions in these models.

281: *Here, despite your bootstrapping efforts, I am a bit puzzled. Dispersal is a categorical variable, and you assume the sex bias is universal: either 100% or 0% of males disperse, and 100% or 0% of females disperse. Nothing in between is possible. So, every female disperses, or none of them do. Every male disperses, or none of them do. The isotopic data get this right 50% of the time. I think I understand your underlying argument, but I also think you still need to do some work to make it completely clear.*

We have re-worded these conclusions. The percentages indicate the proportion of individuals expected to be non-local/dispersers (under our idealized dispersal model) which were assigned that same, matching attribution (“non-local/disperser”) by the isotopic data under each model (the standard archeological approach, the tooth-to-bone offset, and the tooth-to-environment offset). We hope that this makes the conclusions more clear and also more honest (as you point out, the assignments cannot be “correct” unless we know for sure which individuals did or did not disperse during life, which we cannot know for everyone).

302: *You cite one reference; this does not make the method “commonly used”. Actually, my non-specialist understanding is that analysis of micromammals at Miocene & Plio-Pleistocene hominin sites is not common, in no small part because it is labor-intensive and not glamorous. In fact, it might be good if you modify this to highlight its usefulness (and you could cite Denne Reed’s work) and put out a call to make it more common.*

We agree that this is an excellent opportunity to call for more work in this area. We have cited Dene Reed’s contributions here and added a sentence calling for more of this type of data. Thank you for this suggestion!

304: *“but better methodological...” would be better as just “before we could confidently use these methods”.*

Reworded as suggested

306: *“geochemicalLY”*

Corrected

309: *Omit “also”*

Corrected

310 and Table 1: *There are better references than Chapman & Wrangham (1993) for Kibale chimp home range sizes! Notably, for Kanyawara, see Wilson et al. 2012, Anim Behav 83: 277, and for Ngogo see Mitani et al. 2010, Cur Biol 20 (12) R507-508. Also, chimpanzee home ranges can be > 100 km² (Moore et al. 2012, J Hum Evol 112: 30-40). Some are 15 sq km or smaller, but that is really toward the low end of the range of variation.*

Thank you for these reference suggestions. We have updated our home range estimates based on the Kanyawara and Ngogo papers both in-text and in Table 1 and refer to the Moore (2012) paper as well as Moore et al 2018, Am Jour Prim 80(8) to illustrate the wide range of possible home ranges in different habitats.

312: *Here and elsewhere, don't use italics. Make it clear from your prose what you think is most important.*

Italics have been removed throughout.

315-316 (*see also my comments above*): *I'm not sure this quite holds up logically. Small home ranges could mean that mean dispersal distance is too short for dispersers, as adults, to be in areas with isotopic signals different from their natal areas.*

We agree – this is exactly why we predict the tooth-environment offset metric was not effective for small home range species. We have reworded this section to try and make this point more clear.

329: *I don't think you've said anything about the potential of isotopic data to tell us about social behavior. Better to omit this, unless you can state explicitly what you think it could tell us.*

Omitted as suggested

337: *Which approaches? You presented three, then concluded that one was most reliable, although ecological scale matters. This isn't a very strong concluding statement; it rather undermines some of what you said above.*

We have re-written the concluding paragraph.

We are deeply appreciative of our reviewers, both of whom clearly spent quite a bit of time and energy carefully reviewing this text. The manuscript is much stronger, clearer, and more accessible to the reader thanks to their efforts. Thank you for your consideration on this resubmission.

Yours Sincerely,

Marian Hamilton

Appendix C**ROYAL SOCIETY
OPEN SCIENCE****Using Strontium Isotopes to Determine Philopatry and
Dispersal in Primates**

Journal:	Royal Society Open Science
Manuscript ID	RSOS-200760.R1
Article Type:	Research
Date Submitted by the Author:	09-Aug-2020
Complete List of Authors:	Hamilton, Marian; University of Northern Colorado, Anthropology Fernandez, Diego; University of Utah Nelson, Sherry; University of New Mexico, Anthropology
Subject:	behaviour < BIOLOGY, ecology < BIOLOGY, evolution < BIOLOGY
Keywords:	Philopatry, Strontium isotopes, Hominin evolution, Dispersal, Chimpanzee
Subject Category:	Organismal and Evolutionary Biology

Author-supplied statements

Relevant information will appear here if provided.

Ethics

Does your article include research that required ethical approval or permits?:

Yes

Statement (if applicable):

All fieldwork, sample collection, and sample import/export activities were approved by The Ugandan Wildlife Authority (EDO/35/01), the Ugandan National Committee on Science and Technology (NS 506), the US Department of Agriculture (PCIP-16-00229), the US Fish and Wildlife Service (15US62228B/9)

Data

It is a condition of publication that data, code and materials supporting your paper are made publicly available. Does your paper present new data?:

Yes

Statement (if applicable):

The new data presented in this article is available in Tables 1 and 2 of the ESM file.

Conflict of interest

I/We declare we have no competing interests

Statement (if applicable):

CUST_STATE_CONFLICT :No data available.

Authors' contributions

This paper has multiple authors and our individual contributions were as below

Statement (if applicable):

MIH designed the study, performed field work and laboratory analyses, conducting statistical analyses, and wrote the manuscript; SVN performed field work and laboratory analyses and contributed to writing the manuscript; DPF designed laboratory protocols, supervised laboratory analyses, and contributed to writing the manuscript; all authors gave final approval for publication.

Dear editors:

We are deeply appreciative of the time and investment the reviewers put in to improving this
manuscript. We have included our responses (in bold) to each of the reviewer comments (*italicized*)
below. The line references within this document refer to the clean manuscript copy included with
this resubmission, although we include a Tracked Changes version of the manuscript as well for
reference. We believe that this work is substantially improved by the included edits and thank you
in advance for your consideration on its resubmission.

----

**Reviewer 1 Comments:**

*Hamilton et al. present a new method for distinguishing local and non-local (philopatric and dispersing)*
*individuals and species that relies on the offset (absolute value) between $87\text{Sr}/86\text{Sr}$ of tooth enamel and*
*bone or tooth enamel and the local environment (using plants). The authors have incorporated a number*
*of my previous suggested changes and additions, and these have definitely improved the readability and*
*flow of the text. However, there are still some issues that need to be addressed before I think this is*
*publishable. I outline these below.*

*First, I note that there are multiple mistakes in the text. In addition to the normal expected typos, there*
*are sentences in the methods that are written more than once verbatim, and some table captions are*
*incorrect (e.g., Table 1 does not include any isotope data).... This lack of attention to detail is*
*concerning. I have done my best to repeatedly correct errors in this manuscript but the authors need to be*
*responsible for the quality of their own work. At best, an editor might catch mistakes, but if things get*
*published that are incorrect, that would be disappointing for everyone.*

**We agree that any published mistakes would be terribly disappointing, at that it is neither the job**
**of the editor nor the reviewers to ensure the manuscripts are error-free. We have done multiple**
**thorough read-throughs of the text and truly believe it is clean and free from typographical**
**mistakes as we can be reasonably sure of.**

*Second, I think that the offset method that the authors present definitely has promise. The language*
*currently used in the introduction (lines 70-77) is nicely written, and it makes sense. I agree that using*
*offsets to differentiate dispersing and philopatric individuals (or to identify which sex is dispersing for a*
*species) could be a great approach. However, I think that HOW the authors present why the method is*
*novel is rather weak. The authors focus mostly on emphasizing that their study is novel because they*
*establish geochemical clusters on the landscape. For example, in the Conclusion paragraph, they write*
*“the present study highlights that it is critical to establish meaningful geochemical clusters on the*
*landscape thorough sampling and mapping, as geologic boundaries do not by default correspond to*
*unique isotopic clusters” (lines 331-333). I don’t think that most researchers have previously assumed*
*that geologic boundaries by default correspond to unique isotopic clusters. This is why empirical data are*
*included from local animals or plants (to verify what local values are, typically on different geologies).*
*This is nearly ubiquitous across strontium studies (and has been the standard practice for decades, as I*
*pointed out in my previous review).*

*Instead, I think it’s the second point that the authors mention in the introduction (but don’t revisit later)*
*that makes this method novel: i.e., that calculating offsets allows one to assess the likelihood that an*
*individual is local or non-local based on how it compares to other members of the same species (or*
*community) rather than relying on those maxima and minima designated by the local baselines.*

We agree that our most substantial contribution is the development of the offset method described here. We see how the previous structure and wording of the manuscript places a heavier emphasis on the use of statistical clustering rather than the offset method itself. While this is (to our knowledge) the first time cluster analysis has been used with strontium isotope ratios to form isoscapes in this way, that is more of a “means to an end” and not the most important contribution within this paper. We have significantly decreased the discussion of the clustering method in the Discussion to highlight the offset approach as the most central offering of this paper. We did revisit this second idea in the Conclusion in the previous manuscript version but see how it was overshadowed by the unnecessarily long discussion of the clustering method in the same paragraph. We hope that the manuscript is now more balanced towards the most important contributions (lines 271-286).

Third, the Discussion is still lacking some important aspects. Mismatches between expected and observed numbers of philopatric vs. dispersing individuals could be due to sampling teeth that mineralized after dispersal, inaccuracies in the available geologic map, heterogeneity in the geologies themselves (see my note below regarding text on lines 91-95 below), and incorrect assumptions about the degree to which individuals truly are catholic to expectations of dispersing and not dispersing. I think all three of these points need to be touched on in the Discussion.

We have restructured the discussion to more explicit address each of these salient and important points one by one. The sections addressing each are indicated below each of the suggestions below.

1. I would like to know how much tooth selection might affect offset results. The authors now explicitly mention that they tried to sample M1 for all individuals (and they have included which tooth was sampled in the supplementary table), but they do not revisit how results might be affected for those individuals that had a different tooth sampled. In particular, I would think that M2 might have erupted post dispersal. I think that the authors need to: (1) Mention explicitly that there were six individuals that had a tooth other than M1 sampled in the Methods (after lines 116-117); (2) Be careful to only refer to tooth (not the earliest forming tooth and not the earliest forming molar) elsewhere in the text; and (3) Revisit if offset might be off for individuals that had a tooth other than M1 sampled (particularly pertinent for later-forming teeth like M2).

We have included a section addressing this on lines 328-342.

2. I still want to know how isotopic clusters for plants compare with the underlying geology (I have asked this previously). The authors repeatedly state that their method of developing isotopic clusters for environmental baselines from plants is far superior to working with preconceived notions of geology, but their justification for this is very vague and revolves around statements about geologies sometimes being indistinguishable (e.g., lines 91-95; lines 264-266). More importantly, the authors still do not compare their “isotopic clusters” with what would be expected based on underlying geology. I want to see what data look like for plants growing on the different geologic formations. I had also asked previously if the authors could please add the expected geologic boundaries for the different formations to Figure 1 (or perhaps make a complementary panel). I still think this would be valuable. Then the reader can see if plants that look unusual are at or near an expected geologic boundary (or not). The issues with poor geologic maps and old geologies very much need to be brought up in the Discussion.

We agree that the role of ancient geologies is a critical topic to address and have included a section on the implications of old geologies in the Discussion (lines 343-360). We have additionally discussion how these difficulties manifest in Kibale specifically on lines 361-370. We have added a geologic map for comparison to our clusters to Figure 1.

But the authors still don't explore how their statistical plant clusters might compare with geology... Figure 1 suggests to me that geology is still the most likely primary driver of spatial variability in $^{87}\text{Sr}/^{86}\text{Sr}$.

Why might there be spatial clusters in plant $^{87}\text{Sr}/^{86}\text{Sr}$ if it's not geology, particularly those very high ratios?

The authors mention both dust and riparian sources in their discussion. Is there a reason why dust might have spatially variable influence at Kibale? Are there rivers that should be on the map?

3. I think the authors need to briefly discuss how their assumption of dispersal being nearly ubiquitous for
the dispersing sex (and vice versa) might impact “accuracy” of strontium isotope predictions. In the
Methods, the authors mention that they assume that 100% of individuals in dispersing sex disperse and
100% of individuals in philopatric sex stay put (without any justification for this decision). They note at
least one known example at Kibale where this is not true (a female chimp), but I would think that this
individual isn’t an isolated incident. This is important because the “True” number of philopatric and
dispersing individuals is not actually known, and implying (or explicitly referring to) any expected
number as “true” is misleading. The authors need to please add “expected” or “assumed” in front of all
mentions of “correct” classification of individuals. For example, the text on lines 208-209 needs to be
“83% of ASSUMED local/philopatric individuals and only 32% of ASSUMED non-local/dispersing
individuals were correctly identified”. They should also make sure that they discuss how this assumption
may or may not affect their results and interpretations of the data in the Discussion.

**We have included an explicit discussion of our model assumptions regarding the fidelity of**
**dispersal patterns (and lack thereof) in the Discussion (lines 371-386). Furthermore, we have**
**changed our language throughout the manuscript to indicate we are comparing assumed dispersal**
**based on expected species-level sex-biases to the assigned dispersal attribute (local/non-local) from**
**the isotopic data. We hope that this explanation is not only clearer, but more importantly that it is a**
**more accurate reflection of the modeling we’ve done to test these different approaches.**

Smaller points:

*Keywords - It is my general understanding that words that are in the title aren’t also needed in the*
*keywords. Please add the country where the study took place, and maybe include “plants” and/or names*
*of the more interesting primate taxa. Also consider including key words related to statistical methods.*

**We appreciate this insight into choosing proper keywords. We have updated them as follows:**
**Keywords: Uganda, chimpanzee, hominin evolution, cluster analysis, landscape use (Line 23)**

*There are several instances of data interpretation in the Results (lines 218-220; 239-240; 245-250). To*
*me, these would fit better in the Discussion.*

**The sentences referred to above have been removed from the Results section and their content**
**integrated throughout the Discussion section.**

*Reference to a single case study (or even two case studies) cannot support a claim that something is*
*common or typical. In particular, citing a single archaeology study and palaeoanthropological paper to*
*support what the authors are calling “the standard archaeological method” is disappointing. If this is*
*truly a standard method, surely the authors can cite more references to support this point.*

**We have included additional citations demonstrating the use of this method (line 65).**

*The authors still have not told the reader what plant tissues were sampled. This is my third time*
*requesting that this information please be provided. “Botanical sample” could be fruit, leaves, flowers,*
*stems, whole plants, etc. If the authors always collected leaves, say so. If different plant tissues were*
*sampled, then please provide these details in the supplementary table.*

**We truly regret this this information was not forthcoming in the previous manuscript submissions;**
**we certainly did not intend to be obtuse or difficult. All botanic samples are plant leaves. This**
**information has been added throughout the text (ex: line 101) and in the title for Supplementary**
**Table 1.**

Tables:

- *The tables are hard to follow as presented. The formatting is quite awkward. In general, starting each table on a new page is reader friendly.*

We have edited the document to start each table on a new page. We have also edited the column names and bold formatting throughout the table to increase readability.

- *Table 1 presents no isotope data (contrary to what the caption states)*

The caption has been corrected, thank you for pointing out this error.

- *Full references belong in the reference cited section (not as table footnotes)*

We have moved these references into the proper section.

- *In the caption for Table 2, “Falling outside of local environment minima or maxima = local/philopatric attribution” is incorrect.*

Thank you. This has been corrected to read “Falling *inside* local environment minima or maxima = local/philopatric attribution.”

- *Table 4 – what is being compared (i.e. what does “proportion of significant ($p < 0.05$) bootstrap simulations” mean)?*

This title has been updated to be more clear: “Table 4 - Proportion of bootstrap simulations in which there was a significant difference ($p < 0.05$, Student’s t-test) in offset values ($^{87}\text{Sr}/^{86}\text{Sr}_{\text{enamel-bone}}$ and $^{87}\text{Sr}/^{86}\text{Sr}_{\text{enamel-environment}}$) between males and females”

- *In tables 2, 5, and 6, what are the cutoffs for determining if something is conclusive or not? Please clarify in a footnote.*

We have moved information from the table headers into the footnotes, with each column individually referenced and explained, in order to increase the clarity of the information being presented in these tables.

- *IMPORTANT: As I mention above, the wording regarding number of dispersing versus local individuals in these tables is misleading. Bolding and underlining true in these table headers makes this even more misleading. As mentioned above, the “TRUE” # of locals and non-locals is unknown! “Assumed” or “expected” are the words the authors should be using. And the authors really can’t be looking at ACCURACY of attribution given that the “true” information is assumed in the first place. They can assess and compare attributions.*

We appreciate the suggestions to make our wording more accurate and less misleading to the reader. We have updated the tables to use “expected” instead of “true” and updated the table headers to indicate a comparison of attributions (the expected, based on the sex and species, and the assigned attribution based on the isotopic data and the specific method being tested).

- *The organization of the tables is perplexing to me. Why does Table 3 come before Tables 4-6?*

- *How do Tables 3 and 7 differ? Presumably one is species level and the other individuals? More*

informative captions would be helpful (“Summary of Method Efficacy” is not sufficient). Are there other places besides Line 250 where Table 7 should be cited?

Upon reflection looking at our tables, we think the source of this confusion is that (previous) Table 3 is redundant to (previous) Tables 2, 5, and 6. (Previous) Table 7 serves as a more informative summary comparison of all methods than (previous) Table 3. We have updated (current) Table 6 (previous Table 7) to have a more informative title (“Summary Table: Were methods able to correctly identify the dispersing sex for each primate species?) and detailed footnotes for each column explaining the definition of ‘effective.’ We reference it in lieu of (previous) Table 3 throughout the text. We regret the confusion that this extra table caused and appreciate the dramatic increase in clarity that these changes provide.

- Table S2. How are entries organized in this table? I had initially thought species, locality, sex, but in some cases, the sexes are mixed up for each species at each locality.

The Table is now correctly organized by species, then location, then sex.

- It would be helpful to provide summary offset data for each sex for each species in addition to species means. Please remember that numbers <1 should have a 0 before the decimal place.

We have included the averages by sex under the species averages for each species and added the “0” in front of each decimal as needed.

Figures and captions:

- Figure 1 - I have asked before about the geologic map for the region. Even if it is rather generalized, it would still be helpful (and provide reader transparency) to show the expected boundaries of the different units relative to the plant collection sites and primate habitats.

We have included a version of the Figure 1 map showing the underlying geology of the area in relation to the plant samples collection sites and the park areas mentioned in the text. We have also included a boxplot showing the isotopic ratios of the plants collected on each geologic layer. We agree that this is very helpful context and compliments the increased discussion of the complex nature of the ancient felsic geology of the region coupled with the lack of detailed geologic maps.

Yes it is helpful context but it still isn't sufficient incorporation of geology.

- For Figure 4, are all of the details regarding significance of bootstrapped offsets for each species needed both in this caption and in the main text? Please remind the reader what is being compared in these tests (e.g., sexes?).

We have removed the data from the caption, as we agree it is redundant to the main text. We have revised the remaining caption to make clear that these are comparing offsets between sexes, by species.

Line by line things:

Lines 12-13: “The present study demonstrates that standard archeological methods used to determine ‘local’ and ‘non-local’ individuals are not reliable when applied to dispersal patterns in modern primate communities” needs context. Readers will not inherently know what the authors are talking about. Briefly define what the method is that the authors are considering standard (and presumably this should be method, and not plural methods?).

**A very brief explanation has been added to provide context**

*Lines 18-19: “relative to the isotopic clusters on the landscape” needs context. It will not be clear what*
*this means to someone who is just starting to read the paper. Clusters in baseline bioavailable $^{87}\text{Sr}/^{86}\text{Sr}$*
*obtained from plants? Is this text needed?*

**We agree that this clause (“relative to the isotopic clusters on the landscape”) can be omitted in the**
**abstract for clarity.**

*Line 39 (and elsewhere): “ $^{87}\text{Sr}/^{86}\text{Sr}$ ratio” is redundant. Either strontium isotope ratios, OR $^{87}\text{Sr}/^{86}\text{Sr}$.*

**This has been changed to simply “ $^{87}\text{Sr}/^{86}\text{Sr}$ ” throughout the manuscript.**

*Line 40: Tissues should be plural here*

**We have corrected this.**

*Lines 54-55: Consider if the sentence starting with “chimpanzees” should be moved up one sentence.*
*This could help with the flow of the text here.*

**We agree that this helps with the flow of this paragraph and have made the correction.**

*Line 57: “if” or “degree to which” would be better choices. “Whether” requires “or not”.*

**We have changed this to “if.”**

*Lines 59-61: The authors looked at multiple species of primates. That’s still not clear at this point in the*
*text (it is hinted at on line 69 but doesn’t become fully evident until much later in the text). Rather than*
*say “in a modern primate habitat”. I would recommend “In sympatric primate species with varying home*
*ranges, philopatry, and dispersal patterns” or “Within a modern primate community”. Please also*
*mention the site name here or below on lines 69-70.*

**We have changed “habitat” to “community,” and mention the name of the site (Kibale National**
**Park, Uganda) on line 73.**

*Line 62: Citing a single archaeological study and one palaeoanthropological study is not sufficient for*
*supporting a statement that this is the most common method in archaeological strontium isotope*
*literature!*

**We had added additional citations to this statement.**

*Lines 74-76. Please clarify that “environment” is synonymous with local bioavailable $^{87}\text{Sr}/^{86}\text{Sr}$ or*
*baseline $^{87}\text{Sr}/^{86}\text{Sr}$.*

**We have added this language to the text on line 78-80.**

*Lines 88-95 – I think adding text like what has been added here is helpful, but it’s also confusing as*
*written. The authors refer to data from botanical samples here but then also refer to botanical data to*
*address the issue of similar botanical data in the next paragraph... I’m also not sure that all of this*
*information belongs here in the Methods. Some of it appears to be results, and justification for the*
*authors’ argument that their new offset method is preferable to other methods. This is absolutely an*

appropriate place to discuss limitations of available geologic maps. the text could be reorganized slightly
in this paragraph to make that point clearer up front rather than halfway down the paragraph. However,
the text on lines 91-95 definitely does not seem like Methods to me. It also seems quite vague. Please
revisit wording and be careful to focus on Methods (and not results) in the Methods. Also, note that
“pattered” should be patterned on line 89.

**We have moved the discussion about the overlapping isotopic ratios on the two major geologic**
**formations to the Results section (lines 214-216). We have also re-worded this section to better**
**explain our cluster analysis approach to defining ‘local’ areas. We explain how this approach is**
**helpful in both places without detailed geologic maps, and/or in areas with true isotopic overlap**
**between neighboring geologic areas.**

*Here, or perhaps in the Discussion, please explain how/why different geologic formations may be*
*isotopically difficult to distinguish (in the context of what the geology is expected to be at Kibale). My*
*understanding is that very old felsic geologies (e.g., Paleoproterozoic metamorphic rocks) have elevated*
*$^{87}\text{Sr}/^{86}\text{Sr}$ (typically some of the highest measured) due to their age, and that they can be quite*
*isotopically heterogeneous due to differential weathering of minerals over time as well as bioavailable Sr*
*contributions from exogenous sources (like dust). I think this context would be very helpful for the reader,*
*who is likely not familiar with any of this. Please add.*

**We agree that this is helpful and important context. We have put a section about this topic in the**
**Discussion, as suggested (lines 343-360).**

*Lines 107+ - This is the meat of the study but the way it is worded makes it sound secondary (deleting*
*also on line 107 would help a lot). Consider if this should be mentioned before the details of how the*
*authors accounted for local variability in bioavailable $^{87}\text{Sr}/^{86}\text{Sr}$.*

**We very much like the improved flow and emphasis that comes with reordering the faunal and**
**botanical sample collection descriptions. We have made these changes.**

*Line 114 – Site Table 1 here rather than Table S2.*

**We have made this adjustment.**

*Lines 114-119 – Site Table S2 after this text as it provides details about which element and tooth was*
*sampled for each individual. As mentioned above, please make sure to add explicit mention of the six*
*individuals who had a tooth other than M1 sampled.*

**We have added text explicitly mentioning the 6 individuals with teeth other than M1 sampled (lines**
**97-98) in the Methods. We more fully discuss the implications of this in the Discussion section of the**
**paper (lines 328-342), including a discussion of the eruption times of these teeth (from Smith et al**
**1994) and a recognition that this could impact the outcome of our models.**

*Lines 120-132 – There’s redundancy in the text here (e.g., the lab name and location are mentioned*
*twice). First finish the thought about plant sample collection (making sure to mention what tissues were*
*sampled). Then focus on sample prep and analysis. Some information regarding sample prep is still*
*missing. How were bone and tooth powdered? Did the authors grind up entire teeth or did they remove*
*enamel? Did they chemically pretreat samples in any way? Also note that on Line 122, “on” should be*
*“in”.*

*Interesting. Is this typical? I thought samples typically go through a series of chemical*
*pretreatment steps for isotopic research. Perhaps that's only if they are ancient/fossil?*

**The redundancies have been corrected. We have added in information about enamel and bone**
**sample collection on lines 91-93. Samples were not chemically pre-treated beyond the procedures**
**currently described in the Methods. “On” has been changed to “in” as suggested.**

*Line 143 – Consider if this should be a new subheading. It's data analysis (a separate thought from*
*sample acquisition and analysis).*

**We like the idea of subheadings within Methods and have added them to differentiate sample**
**collection, laboratory analysis, and data analysis.**

*Lines 144-146 – Please justify this assumption.*

**We have added in a clear explanation for this assumption, which we recognize to be an idealized**
**model of reality. We have provided an explanation of its limitations as well as a justification for its**
**utility lines 150-168.**

*Line 160 – The authors did sample the earliest forming molar for all individuals. Reword.*

**This has been reworded to simply “tooth.”**

*Line 194 – there's a typo “±-“*

**Thank you – this is corrected.**

*Line 198 – “was” seems awkward here. Would be? Is?*

**We agree that ‘would be’ flows more smoothly here.**

*Line 199 – Fauna should be capitalized*

**Thank you – corrected.**

*Lines 203-205 – Methods are laid out in the Methods section. No need to restate this here (but please do*
*make sure that Methods are sufficiently detailed). Should “Standard Archaeological Method” be*
*capitalized in the subheading?*

**We have removed this explanation from this section and integrated the relevant details throughout**
**the Methods section.**

*Lines 208-209 – As mentioned above, I think that throughout the text the authors need to be clear that*
*these are EXPECTED local/philopatric or non-local/dispersing individuals. This is only actually known*
*for one female chimpanzee.*

**We have reworded the text throughout the Results section to match the language in Table 2**
**(expected vs assigned attribution).**

*Lines 215-218 – Please remind the reader what the significance is referring to here. These species had*
*significant what? Significantly different offset in 87Sr/86Sr between sexes? Expected dispersing and non-*
*dispersing individuals? Does this language need to be included both in the text and the figure caption?*

We have reworded this to clarify that the comparison is between the offset values for each sex.

Lines 256 - 257 – Please clarify ONE standard approach (not the only standard approach). No need to state this again on Line 257 (just write “this approach”).

We have made these corrections and clarifications.

Lines 258-260 - The wording here is confusing. minima and maxima are also isotopic ratios themselves. Please reword or cut.

We have cut this section per your earlier recommendations.

Lines 260-270 – I think this information would fit much better in the introduction where the authors introduce the method rather than down here in the Discussion. There’s some redundancy with what is said earlier, but also some novel information. Please move up and integrate with text in the intro and cut redundant wording.

We have moved the necessary information from this section to lines 117-123, where the method is introduced. We agree that this greatly increases clarity in both areas and appreciate the suggestion.

Lines 262-264 – The authors cite one example here but refer to multiple previous studies. Please revisit wording. Also, as written, this is pretty vague and kind of misleading. the authors need to please make sure that they are clear that while this can happen, it’s VERY context dependent. The geologies could also have entirely discrete $^{87}\text{Sr}/^{86}\text{Sr}$. IT entirely depends on what the local geologies are. See my notes above about the complexities of very old, felsic geologies.

We have included this point in our response on line 117-123, with an emphasis that this is not an issue that we expect in ALL areas, but one that is very geologic context-dependent.

Lines 300-301 – This needs to be reworded. Underlying geology will ALWAYS be older than something deposited on top of it with rare exceptions (e.g., flood deposits). What’s key is that the surface geology has been consistent over time (i.e. no major erosion or deposition since the organisms of interest expired).

We have reworded this sentence, thank you for the suggested correction.

Line 303 – is the idea of paleosols holding promise based on the authors’ own gut feelings or is there some existing study or publication that has suggested this that could be cited here?

To our knowledge, strontium isotope records from paleosols have not been used in this way nor suggested, although clearly other isotope data from paleosols is prolific in the literature. The use of paleosols for other isotopic records is the root of our idea here, and we include citations for a few pieces of that literature. Its implementation in this context is our suggestion for future further investigation.

Lines 308-321 – please cite Table 1 here, which provides home ranges for all of the species. I note that the authors state here that chimp home ranges can be up to 15 km² but Table 1 indicates chimp home ranges can be up to 27 km² (nearly two times bigger than 15 km²). Which is correct?

We have included a citation for the table and clarified the chimpanzee home range size, including insight from our other reviewer that such home range sizes can vary quite widely based on the type

**of habitat the chimpanzee community is in (open habitats vs closed forest habitats, such as in**
**Kibale).**

*Lines 337-340 - The paper sort of fizzles in its final sentence. The authors are repeating themselves here;*
*they already pointed out in the beginning of this conclusion paragraph that “the previous standard*
*approach” doesn’t work in all contexts. I would recommend rewording (e.g., it is reasonable to expect*
*early hominins to share many of these features, suggesting that these methods may be able to reconstruct*
*our own ancestors’ dispersal habits”?), think bigger picture about what the next steps might be to help*
*validate this method, or perhaps conjecture about the kinds of questions researchers may be able to*
*answer using this new approach.*

**We have removed the redundancy and added some bigger-picture suggestions to end the paper.**
**Thank you for this suggestion.**

Reviewer #2 Comments

I did not look at the initial submission or the first revision of this manuscript, so I am commenting just on the current version and on the responses to comments from one previous reviewer. I realize this introduces the possibility of shifting the goalposts, as it were. Also, I am familiar with isotopic analysis, but not an expert, and I will trust the previous reviewer, who clearly is, and the authors' responses to that reviewer. So far as I am aware, previous reviews did not raise questions about the behavioral ecology of the primate species included in the analysis, so some of my comments will be the first time the authors have encountered such questions.

We are extremely glad to have an expert in primate behavioral ecology provide their input to this paper. It is a critical component to this work, and the clarification and nuance you provide around the messy business of dispersal and philopatry is greatly appreciated. We hope that the following edits sufficiently address your concerns.

That said, I think the manuscript is quite interesting and the research has real value. Again, I am not an expert regarding isotopic methods, but the authors' "novel" method seems to hold considerable promise, although getting appropriate "environmental" signals from paleontological sites may be quite challenging. However, I have some substantive concerns and also will make some specific comments about the presentation of the material.

*One substantive concern is the need to distinguish between locational dispersal (individuals move away from natal areas, or from areas to which they have already dispersed in space) and social dispersal (members of group living species emigrate from groups and immigrate into other groups). In a valuable 1996 paper, Lynn Isbell and Dirk Van Vuren made this important distinction clear (Isbell, L.A. & Van Vuren, D. Differential costs of locational and social dispersal and their consequences for female group-living primates. *Behaviour* 133:1-36). Individuals can undergo one, both, or neither of these processes. Male chimpanzees don't undergo either. Most female chimpanzees undergo both: they transfer between social groups and they move from one territory to that of another group. However, this is not universal, as the authors implicitly acknowledge when they state that researchers knew that one of the Kanyawara females had stayed in her natal community as an adult. A female gray-cheeked mangabey typically does not undergo social dispersal – females don't emigrate from their natal groups – but some females undergo locational dispersal when groups fission and establish separate home ranges, neither of which is identical to the original one (for a Kibale example, see K. Janmaat et al., 2009, *Intl J Primatol* 30: 443). Also, how exclusive home range or territory use is could become an issue. Female guerezas apparently don't disperse from their natal groups – at least, this has not been documented in Kibale – but the home ranges of neighboring groups can overlap extensively, to the point where groups at Kanyawara can share virtually all of their home ranges with multiple other groups (T.R. Harris & C.A. Chapman 2007, *Primates* 48: 208-221); thus, a female could disperse socially, but not locationally, or at least locational dispersal would be very short. Their home ranges are small – although Harris & Chapman found their sizes can vary 5-fold – and overlap might not matter re. isotopic signatures (and male locational dispersal might occur on a scale that, on average, is sufficiently larger than the scale at which female home range occurs that you could still expect to pick up isotopic signals of dispersal reliably. But these issues may not be trivial. Dispersal is not a nice, clean categorical variable on an ecological time scale (even incorporating a third category for species in which individuals of both sexes disperse, at least conditionally – actually the case for most mammals, although sex biases in locational dispersal still often exist). I realize that this manuscript is not meant to be a review of primate dispersal strategies. However, I think you need to acknowledge this complexity. Moreover, you might find it useful to think about the possible implications that variation in locational dispersal has for your over-arching research questions. For example, suppose females usually don't disperse socially, but home range stability is low and any*

individual female covers a wide habitat range over the course of her lifetime because her group shifts the
areas they use? How much does home range overlap matter? **To what extent could these complexities**
**explain “mis- classifications” (a question that particularly occurred to me when I reached line 209 and**
**saw that you correctly identified only 32% of “non-local” individuals)? I also worry about this when I**
**see that you pooled data for redtails and blue monkeys and still had a sample of only 7.** Yes, male natal
social dispersal and female natal social philopatry seem to be universal in both, and redtails in Kibale
are territorial...but...The bottom line is that these details of the behavioral ecology of the species in your
sample need some acknowledgement.

**You are absolutely correct that we are using an overly simplistic version of dispersal to test the**
**efficacy of these isotopic models, and that this deserves to both be recognized, described, and the**
**limitations discussed. This is very helpful language and citations to discuss it, and we are very**
**appreciative. We have included a brief acknowledgement of our model assumptions in our Methods**
**section (lines 150-168) and a more detailed discussion of the ecological details and their implications**
**for our results in the Discussion section (lines 371-386)**

*Similar questions kept occurring to me when you referred to “correctly classified individuals”, although*
*that touches on another point. If I understand correctly, you established your criterion for correct*
*classification by assuming that (a) dispersal = locational dispersal; (b) this is a categorical variable; (c)*
*each species in your sample shows a universal sex bias and can be put either in the category “male*
*(locational) dispersal/female (locational) philopatry” or the reverse. Thus, all male chimpanzees should*
*show local isotopic signals (low offsets), for example, and any who don’t are “mis-classified”. For a*
*start, it would greatly help readers if you state very clearly in the Methods your definition of*
*“classification and your criterion for deciding whether someone was correctly classified. So, for example,*
*in the par. starting on 221, you state that 67% of individuals were either members of the “dispersing sex”*
*with high offsets members of the “philopatric sex” with low values. Is this just a way to say that we*
*assumed that all females of species a, b disperse and all males of species d,e, and f disperse – locationally*
*– that any male of species d with an offset below the mean is “mis-classified”? This seems to the case,*
*and if it is, you could make it much clearer. If it isn’t, then perhaps you still need some clarification.*

**We have changed the language here to avoid confusion around the word “correct.” Instead, we**
**have reframed our results to indicate the proportion of individuals assigned a *matching* attribution**
**(local or non-local) based on the isotopic data compared to the *expected* attribution from our**
**simplified dispersal model (that all females chimpanzees and all female red colobus disperse, while**
**all males remain philopatric, etc). References in the previous draft to “correct classifications” really**
**meant that the isotopic data indicated the same attribution as was expected based on sex and**
**species (and locational dispersal, as we discuss in the Discussion section). We appreciate your**
**insight that the way this was previously written was not clear, and we hope that this version is much**
**improved.**

*This brings up a more general substantive issue, one that the previous reviewer also raised in a somewhat*
*different way: at multiple points in the manuscript, you have not really made it clear what the question is.*
*Readers can infer this, but they shouldn’t need to. For example:*

*Par. starting 88: “To address this...” What is “this”? You have not stated a question or explicitly framed*
*an issue.*

**We have reworded this section to more clearly indicate the problem with isotopic overlap between**
**geologic formations (lines 117-123).**

105: “bias” with regard to what? Again, you haven’t clearly stated the issue.

**We have removed this particular phrase, as we agree it was unclear and did not add value to our**
**argument.**

*Abstract:*

15: “between faunal teeth and bone and/or local environment” – this would be much better/clearer as
“by calculating offset values for comparison of teeth to bone or to local environments”.

**We agree, and have made this correction. Thank you for the recommendation.**

17: Bonobos are also “humans’ closest living relatives”.

**We tried to reword to include this, but it did not flow well in an abstract. We’ve removed the clause**
**as we do not want to be imprecise.**

19: Insert a comma after “landscape”, and please check the rest of the manuscript closely for places
where you should insert commas (e.g., put a comma after “plants” in 39).

**We have closely edited the paper for proper punctuation and typos. Thank you for your careful eye**
**and attention to detail here.**

*Introduction:*

27: should be “consequences for”

**Thank you – we have corrected this.**

27: “The ancestral condition” for what?

**We have reworded this sentence: “Ancestral behavior and socioecological conditions for the**
**hominin clade are currently unresolved.”**

28: This list (“aggression”, etc.) is not one of “abstract social parameters”. It is one of variables that we
cannot directly infer from the fossil record.

**Thank you – we have reworded this.**

30: “Pair-bonding” is uniquely human. Nor, for that matter, are “extended life histories”. You need a
more nuance/clarity here. Exactly what are you comparing humans to?

**We have reworded this sentence to: “... the evolutionary trajectory of traits which differentiate**
**humans from our fellow extant great apes...”**

33-34: “Employed more direct proxies” to do what?

**We have clarified “... to reconstruct behavior.”**

34: “Vary by surface geology” should be “vary in association with variation in surface geology”.

**We have reworded as suggested**

36: Can they use the landscape non-spatially? Just “use the landscape”, or “space use”.

**We have reworded as suggested.**

47-48: Just say “if the fossilized individuals died at or near the depositional sites”.

**We have reworded as suggested.**

*Par. starting 50: The “chimpanzee referential model” has come in for much criticism. I don’t agree with*
*all the criticism, by any means, but one serious problem has been the paired assumptions that (a) recent*
*hunter-gatherers were characterized by male dispersal and female philopatry, and (b) therefore this*
*characterized the LCA. In fact, (a) is incorrect, at least with regard to ethnographic documentation of*
*recent H-G residential patterns. This leaves the question of the LCA open. I think you need to*
*acknowledge this.*

**We agree that this should not be presented as consensus, and have included reference to and**
**citations of some of these critiques (lines 55-56)**

*Par, starting 62: This is very awkwardly written. (For one thing, here – a minimum...ratios” -- and*
*elsewhere your subject and verb persons – singular vs. plural – don’t agree). This may seem like*
*pedantry, but avoiding such awkwardness makes it more likely your audience will get your points.*

**We have changed the wording around in this paragraph to increase clarity.**

*Methods:*

155: Should be “Once the samples cooled, we spiked them...” 136: Omit “down”

**Corrected as suggested**

*153-155: Readers can use your tables to figure out how many individuals were mis-classified and,*
*importantly, how many males and females were mis-classified, but it would be nice to see summary*
*statements in the text.*

**We include this in the Results section.**

183-184: This gets back to a point I made above: “correctly classified” as what? As male or female?

**We have removed references to “correctly classifying” throughout the paper and reworded this for**
**clarity: “We calculated the proportion of individuals with an assigned category (local/non-local)**
**that matched their expected category based on sex and species dispersal patterns.”**

*Results:*

197: You must mean “the greatest **straight line** distance” -- ? That is, the maximum diameter for a
hypothetical circular “small” home range would be 3 km, and this would be the longest distance anyone
could move in a straight line without leaving it.

**Yes – we have added this for clarity**

214: What is “trended” supposed to mean?

**We have changed this to simply “had”**

225-226: *As noted above: come of the “bias” toward “mis-identifying” individuals of the dispersing sex*
*could be because they really did Not disperse locationally, at least not on a scale sufficient to affect*
*isotopic signatures.*

**We agree, and have included a discussion of this important point in the Discussion section (lines**
**371-386)**

238-239: Fig. 4 shows HIGHER values for the philopatric sex -- ?? Are the labels in the legend wrong?
(Or am I missing something?) Also, the figure doesn’t contain any info about significance values.

**The graph was generated with the colored reverse – thank you for catching this error. We have**
**corrected the graph and added *s to indicate significant differences between the sexes in > 90% of**
**bootstrapped simulations.**

264: *A lesson I learned long ago: If you think it necessary to write “In other words”, you didn’t say it* I love this!
*right the first time.*

**A lesson worth keeping! We have changed this section substantially based on suggestions from the**
**other Reviewer, included removing this wording.**

267: *“Isotopic ratios from botanical samples” – obviously you can’t do this with fossil sites. As you*
*mention below (in response to the previous reviewer’s comments), micromammals and maybe paleosols*
*possibly provide alternative ways to gain insights. But I think you need to make a clearer argument here,*
*maybe by analogy: our results show that isotopic data derived from vegetation samples make acceptable*
*proxies for saying “x” about sex differences in dispersal; we can’t get such data from Miocene and Plio-*
*Pleistocene fossil sites relevant to hominin evolution, but we could instead use data on “a” and “b”*
*because...”.*

**We have substantially revised this section based on recommendations from the other Reviewer but**
**have incorporated the spirit of this suggestion into Lines 312-325.**

273: “Likelihood of an individual being local...”: again, you are assuming (a) dispersal = locational
dispersal, and (b) when a sex bias exists, it is absolute: 100% of males or 100% of females disperse
locationally.

**Yes – we have included a discussion of this assumption and the biases implied with it in the**
**Discussion section (lines 371-386). Thank you for highlighting this important set of assumptions in**
**these models.**

281: Here, despite your bootstrapping efforts, I am a bit puzzled. Dispersal is a categorical variable, and you assume the sex bias is universal: either 100% or 0% of males disperse, and 100% or 0% of females disperse. Nothing in between is possible. So, every female disperses, or none of them do. Every male disperses, or none of them do. The isotopic data get this right 50% of the time. I think I understand your underlying argument, but I also think you still need to do some work to make it completely clear.

We have re-worded these conclusions. The percentages indicate the proportion of individuals expected to be non-local/dispersers (under our idealized dispersal model) which were assigned that same, matching attribution (“non-local/disperser”) by the isotopic data under each model (the standard archeological approach, the tooth-to-bone offset, and the tooth-to-environment offset). We hope that this makes the conclusions more clear and also more honest (as you point out, the assignments cannot be “correct” unless we know for sure which individuals did or did not disperse during life, which we cannot know for everyone).

302: You cite one reference; this does not make the method “commonly used”. Actually, my non-specialist understanding is that analysis of micromammals at Miocene & Plio-Pleistocene hominin sites is not common, in no small part because it is labor-intensive and not glamorous. In fact, it might be good if you modify this to highlight its usefulness (and you could cite Denne Reed’s work) and put out a call to make it more common. I guess I don’t understand what the authors mean when they say this isn’t common. Using data from micromammals is the primary way that researchers have established what “local” should be in hominin research.

We agree that this is an excellent opportunity to call for more work in this area. We have cited Dene Reed’s contributions here and added a sentence calling for more of this type of data. Thank you for this suggestion!

304: “but better methodological...” would be better as just “before we could confidently use these methods”.

Reworded as suggested

306: “geochemicalLY”

Corrected

309: Omit “also”

Corrected

310 and Table 1: There are better references than Chapman & Wrangham (1993) for Kibale chimp home range sizes! Notably, for Kanyawara, see Wilson et al. 2012, *Anim Behav* 83: 277, and for Ngogo see Mitani et al. 2010, *Cur Biol* 20 (12) R507-508. Also, chimpanzee home ranges can be > 100 km² (Moore et al. 2012, *J Hum Evol* 112: 30-40). Some are 15 sq km or smaller, but that is really toward the low end of the range of variation.

Thank you for these reference suggestions. We have updated our home range estimates based on the Kanyawara and Ngogo papers both in-text and in Table 1 and refer to the Moore (2012) paper as well as Moore et al 2018, *Am Jour Prim* 80(8) to illustrate the wide range of possible home ranges in different habitats.

312: *Here and elsewhere, don't use italics. Make it clear from your prose what you think is most*
*important.*

**Italics have been removed throughout.**

315-316 (see also my comments above): *I'm not sure this quite holds up logically. Small home ranges*
*could mean that mean dispersal distance is too short for dispersers, as adults, to be in areas with isotopic*
*signals different from their natal areas.*

**We agree – this is exactly why we predict the tooth-environment offset metric was not effective for**
**small home range species. We have reworded this section to try and make this point more clear.**

329: *I don't think you've said anything about the potential of isotopic data to tell us about social*
*behavior. Better to omit this, unless you can state explicitly what you think it could tell us.*

**Omitted as suggested**

337: *Which approaches? You presented three, then concluded that one was most reliable, although*
*ecological scale matters. This isn't a very strong concluding statement; it rather undermines some of*
*what you said above.*

**We have re-written the concluding paragraph.**

**We are deeply appreciative of our reviewers, both of whom clearly spent quite a bit of time**
**and energy carefully reviewing this text. The manuscript is much stronger, clearer, and more**
**accessible to the reader thanks to their efforts. Thank you for your consideration on this**
**resubmission.**

**Yours Sincerely,**

44

**Marian Hamilton**

Using Strontium Isotopes to Determine Philopatry and Dispersal in Primates: A Case Study from Kibale National Park

Marian I Hamilton^{1,3}, Diego P Fernandez², Sherry V Nelson³

¹ University of Northern Colorado, Department of Anthropology

² University of Utah, Department of Geology and Geochemistry

³ University of New Mexico, Department of Anthropology

Abstract: Strontium isotope ratios ($^{87}\text{Sr}/^{86}\text{Sr}$) allow researchers to track changes in mobility throughout an animal's life ~~and~~. ~~They~~ could theoretically be used to reconstruct sex-biases in philopatry and dispersal patterns in primates. ~~Dispersal patterns are~~, a life history variable that correlates with numerous aspects of behavior and socio-ecology that are elusive in the fossil record. The present study demonstrates that ~~the~~ standard archeological methods used to ~~determine~~ ~~differentiate between~~ 'local' and 'non-local' individuals, ~~which involves comparing faunal isotopic ratios to environmental isotopic minima and maxima~~, ~~are~~ not reliable when applied to dispersal patterns ~~in modern primate communities~~. This study instead introduces a novel methodological approach: ~~using calculated offset values~~ ~~calculating offset values to compare $^{87}\text{Sr}/^{86}\text{Sr}$ of teeth to that of bone or local environments~~ ~~between faunal teeth and bone and/or the local environment~~. We demonstrate this new method's effectiveness using data from five species of primates, ~~including chimpanzees~~, from Kibale National Park, Uganda, ~~including humans~~ ~~closest living relative~~, chimpanzees. Tooth-to-bone offsets reliably indicate sex-biases in dispersal for primates ~~whose with small home ranges are small relative to the isotopic clusters on the landscape~~ while tooth-to-environment offset comparisons ~~are reliable~~ for primates with larger home ranges. Tooth-to-environment offsets yield the most reliable predictions of species' sex-biases in dispersal.

Also yield? Then this sentence follows from the previous one.

Keywords: ~~Uganda, chimpanzee~~ ~~Philopatry, strontium isotopes~~, hominin evolution, ~~dispersal~~ ~~cluster analysis, landscape use~~

Introduction and Background

Patterns of mobility and landscape use, such as sex-biased philopatry and dispersal, have direct consequences ~~on~~ ~~for~~ primate social behavior. ~~Ancestral~~ ~~The~~ ~~ehavior~~ and ~~socioecological~~ ~~ancestral~~ conditions for the hominin clade ~~are~~ currently unresolved¹⁻³. ~~A reconstruction of~~ dispersal patterns could provide a proxy for ~~estimating~~ ~~reconstructing~~ ~~behavioral~~ ~~more abstract social parameters~~ ~~variables that are~~ ~~not directly observable in the fossil record~~, including patterns of aggression and affiliation⁴, social bonding, coalition formation⁵, and the evolutionary trajectory of ~~uniquely human traits~~ ~~traits which~~

just one example of a modern primate community in Africa. I wouldn't say this necessarily is true across the board.

It's not that it isn't necessarily reliable so much as it might be compromised by geologic age or other aspects of the local environment/ the primates themselves.

~~differentiate humans from our fellow extant great apes~~ (e.g., ~~extended~~ life histories, pair-bonding,
 cooperative hunting, multi-family social structures, female-female bonding). Previous work relies heavily
 on modern primate behavioral models and assumptions based on patterns of sexual dimorphism^{6,7}.
 Recently, researchers have employed more direct proxies ~~to reconstruct behavior~~, including hormonal
 measurements⁸ ~~and~~ ~~and~~ geochemical data, ~~includingsuch as~~ strontium isotope ratios⁹. Strontium isotope
 ratios ~~vary-differ by-across varying~~ surface geologies⁹ and have the potential to act as a mobility tracer,
 providing empirical insights into the way extinct species ~~spatially~~-utilized their landscapes¹⁰. In geologic
 formations, the ratio of a heavier isotope of strontium (⁸⁷Sr) to ~~the-a~~ lighter isotope (⁸⁶Sr) varies in
 accordance with bedrock age and type. The strontium isotope ratio (⁸⁷Sr/⁸⁶Sr) of underlying bedrock is
 incorporated into soil and plants; and then into calcium-bearing tissues of animals eating those plants with
 minimal fractionation ~~effects~~^{10,11,12}. Therefore, the ⁸⁷Sr/⁸⁶Sr ~~ratio~~-within faunal tissues, including bones
 and tooth enamel, reflects the isotopic fingerprint of the area where the animal lived during that tissue's
 formation. Different tissues capture different life stages. Tooth enamel forms early in life and then is
 metabolically inert, capturing a juvenile (pre-dispersal) isotopic fingerprint. ~~Enamel is also robust~~
 ~~against diagenetic effects due to its high density and low porosity~~. Bone turns over throughout life and
 therefore reflects the time period closer to the end of an individual's life (often the adult, post-dispersal
 period). In fossils, bone is more susceptible to diagenetic alterations¹³. Botanical samples, soil, and
 contemporaneous micromammal remains from the area of a fossil's deposition can also approximate an
 'adult' isotopic signal, ~~if the fossilized individuals died at or near the depositional site with the assumption~~
 ~~that the fossil is deposited near where that individual ended its life~~. Comparisons of juvenile (e.g., ~~tooth~~
 enamel) to adult (e.g., ~~local environment~~) isotopic signatures can therefore trace ~~faunal~~ movement
 across geologically heterogeneous landscapes. I think the authors mean "an individual's" here?

Researchers have used strontium isotope ratios to track mobility patterns across a wide array of
 environments and species¹⁴⁻²⁰. ~~However, research to date, although with~~ ~~has~~ limited applications in the
 ~~hominin fossil record~~^{9,21,22}; and none ~~to-date~~ ~~in~~ modern primate communities. ~~Chimpanzees are~~
 ~~frequently used as models for the last common ancestor (LCA) of chimpanzees and humans~~²³⁻²⁶, ~~although~~
 ~~this model is not without criticisms~~²⁷⁻²⁹. In ~~this~~ male-philopatric ~~chimpanzees~~ ~~species~~, dispersal patterns
 have downstream impacts ~~including~~ male coalition formation and violence, decreased female
 gregariousness ~~in comparison to males~~, and ~~complex~~ social bond formation. ~~Chimpanzees are frequently~~
 ~~used as models for the last common ancestor (LCA) of chimpanzees and humans~~. Evidence that hominin
 ancestors also followed a male-philopatric pattern would be ~~a keystone~~ ~~important~~ support for
 ~~their continuing to use~~ ~~continued used~~ ~~chimpanzees as models for the LCA and ancestral hominin~~
 ~~behavior~~ ~~LCA models~~. Assessing ~~whether-if~~ strontium isotope ratios can accurately indicate such mobility
 patterns, especially ~~under conditions such~~ as large home ranges and complex social structures, is a critical
Are large home ranges and complex social structures really "conditions"? How about "especially when home ranges are large or social structures are complex"?

The key isn't that they died locally' it's that they lived locally. An animal could have had a death march to the place it died.

What does "is a critical methodological determination" mean? Do the authors mean that determining if this works is critical if we want to use this method?

**methodological determination.** To this end, the purposes of this investigation are to (1) assess the
 reliability of strontium isotope ratios for predicting dispersal patterns in a modern primate **habitat**
 **community** and (2) identify the most accurate methods to reconstruct these dispersal patterns.
 The most common method in archeological strontium isotope literature^{9,13,25,20,22,30-35} to
 differentiate local (presumably philopatric) and non-local (dispersing) individuals is to **first, (1) use**
 **botanical, soil, or micromammal samples** to define a minimum and maximum "local" strontium isotope
 ratios **for a pre-defined local areas using botanical, soil, or micromammal samples collected from discrete**
 **pre-defined zones, typically corresponding to underlying bedrock deposits such as discrete geologic**
 **formations or community boundaries,** and **second, (2) compare** this to an individual's **earliest forming**
 **available tooth enamel isotopic ratio ($^{87}\text{Sr}/^{86}\text{Sr}_{\text{enamel}}$), to this range.** For this standard archeological
 approach, individuals with $^{87}\text{Sr}/^{86}\text{Sr}_{\text{enamel}}$ falling outside of the local **isotopic area minimum and maximum**
 are considered non-local **while and** individuals with $^{87}\text{Sr}/^{86}\text{Sr}_{\text{enamel}}$ falling within the local **cluster range** are
 considered local. In this paper, we first assess the reliability of this method to **categorize** philopatric -vs-
 dispersing primates by applying it to a modern **skeletal collection from Kibale National Park, Uganda** in
 which the **expectations for** dispersing and philopatric sex are known for each species. Next, we determine
 the reliability of a novel approach **which compares offsets between these adult and juvenile isotopic**
 proxies to differentiate between dispersing and philopatric individuals, rather than using direct isotopic
 ratios for binary categorization. We compare the efficacy of two such offset proxies: tooth enamel-to-
 bone ($^{87}\text{Sr}/^{86}\text{Sr}_{\text{enamel-bone}}$) and tooth enamel-to-local environment ($^{87}\text{Sr}/^{86}\text{Sr}_{\text{enamel-environment}}$), **with The local**
 **environment is the local environment being defined equivalent to the local bioavailable $^{87}\text{Sr}/^{86}\text{Sr}$ and is**
 **defined by statistical clustering of botanical isotopic values rather than by predetermined bedrock map**
 **boundaries.** We expect all offsets will be greater in the dispersing sex than the philopatric sex.

Methods

**Sample collection:** This study samples **plant and faunal faunal and botanical** strontium isotope
 ratios in Kibale National Park, a rainforest in southwestern Uganda. **We collected bone and tooth enamel**
 **samples from 57 primates housed in existing opportunistically-gathered skeletal collections at the**
 **Makerere Biological Research Station in Kibale National Park, Uganda, with assistance from the Kibale**
 **Chimpanzee Project and the Ngogo Chimpanzee Project in summer 2016. These samples included four**
 **female-philopatric species -- black and white colobus monkeys (*Colobus guereza*), olive baboons (*Papio*
 ***anubis*), and two species of guenons, blue monkeys (*Cercopithecus mitis*) and redtail monkeys (*C.*
 ***ascanius*); and two male-philopatric species -- chimpanzees (*Pan troglodytes schweinfurthii*) and red**
 **colobus monkeys (*Procolobus badius*) (Table 1). Teeth and bones were first cleaned of surface dirt and**
 **debris using a Dremel drill. We then removed approximately 15 mg of enamel or bone using a diamond******

We assume that the...

Alternatively the authors could say "we define local environment as XX"

impregnated Dremel drill bit. Areas of cancellous bone (such as the ends of long bones, ribs, or areas on
 the skull) which have a more rapid turnover rate than areas of cortical bone were preferentially sampled
 when available to increase the likelihood of a post-dispersal isotopic signature in bones. First molars,
 which erupt around the time of weaning, were sampled when available to ensure pre-dispersal isotopic
 signatures in teeth. Six individuals did not have first molars available to sample and instead first
 premolars, first incisors, or second molars were sampled (see Discussion). Only fully mature adult
 individuals were included in the sample. Individuals were assigned provenience data based on collection
 location (Supplementary Table 1).
 Is "data" needed here? Doesn't provenience stand alone?

Samples-Botanical samples include GPS-referenced botanical-plant leaves specimens collected
 opportunistically along existing trails in approximately one-kilometer grids during the summers of 2014,
 2015, and 2016. These samples came from four study areas of the Kibale National Park: Kanyawara,
 Sebitole, Kanyanchu, and Ngogo (Figure 1, Supplementary Table 2). These areas are underlain by
 multiple geologic layers²⁴. Kanyawara and Sebitole are on the Toro Formation (an undifferentiated early
 Paleoproterozoic gneiss) while Kanyanchu and Ngogo are on the Buganda formation (a middle
 Paleoproterozoic quartzite). Ngogo also falls partially on an area designated an 'unidentified radiometric
 anomaly' by the Ugandan Society for Geology and Mines^{36,24}. Available geologic maps for this area lack
 high-resolution detail, making it difficult to assess with confidence which geologic area plant samples are
 from. For the 'unidentified radiometric anomaly' area, no reliable geologic data was available at all. In
 addition to these knowledge gaps, it is often difficult to predict bioavailable ⁸⁷Sr/⁸⁶Sr from bedrock
 geology in areas comprising ancient, heavily weathered tropical soils (such as those in Kibale National
 Park) as much of the geologic parent material has long since weathered away. These ancient felsic
 geologies can be highly isotopically heterogeneous due to differential weathering and the influence of
 atmospheric strontium sources. Thus, neighboring geologic formations are likely to have a high degree of
 isotopic overlap, making them difficult to distinguish geochemically even when they are well defined
 geographically (see Discussion).

and despite extensive and pattered isotopic variability across Kibale National Park, Given these
 limitations there were not statistical differences in the isotopic ratios from botanical samples collected on
 the two major geologic formations, using boundaries of geologic formations as default definitions for
 local areas would inevitably lead to (the Toro and the Buganda) (Wileox-Mann-Whitney U-test, U=108,
 p>0.1)-faunal ⁸⁷Sr/⁸⁶Sr_{enamel} values equally likely to be from multiple areas. This makes it impossible to
 draw further conclusions. A lack of detailed maps notwithstanding, different geologic formations can
 have overlapping strontium isotope profiles; de facto defining them as unique zones for the purposes of
 identifying local and non-local individuals is therefore not as accurate as grouping together areas which
 may differ geologically, but are isotopically difficult to distinguish due to this overlap.

I'm a little confused. In figure 1 it looks like there are geologic boundaries under the polygon showing the anomaly. Is there no geology exposed so nobody knows?

How is this additional? Isn't this exactly why it is hard to figure out the geology?

The authors need to cite references here. This is not common knowledge.

~~To As an alternative~~ address this, we used botanic samples to statistically define isotopic clusters
 ~~directly rather than automatically assigning each sample to a geologic group based on its collection~~
 ~~location~~ we used hierarchical cluster analysis to statistically identify isotopic clusters, which we then used
 to ~~define possible local areas~~ **(Supplementary Table 21)**. This approach is beneficial because it can be
 ~~applied in areas lacking detailed geologic maps and it provides an alternative approach for defining~~
 ~~geochemically meaningful local areas when neighboring geologies do in fact have overlapping isotopic~~
 ~~profiles.~~

To define isotopic clusters on the landscape, we included all non-riparian ~~plant leaf~~ samples ($N =$
 64) in a hierarchical cluster analysis using an average linkage distance and then used a dendrogram^{37,27} to
 assign each plant sample to a cluster group. We compared the isotopic profiles of the resulting clusters
 using Kruskal-Wallis rank sum tests and qualitatively assessed their geographical boundaries, including
 their relationship ~~to park study areas~~, based on maps generated using the ‘ggmap’ package in R Studio
 (version 0.99.902) **(Figure 1)**. ~~By keeping botanic isotopic ratios initially independent of group~~
 ~~designations and then statistically grouping those ratios to form geochemically meaningful clusters, we~~
 ~~both avoid potential bias and strengthen the predictive power of our dispersal models.~~

*We also collected bone and tooth enamel samples from 57 primates housed in existing*
 *opportunistically gathered skeletal collections at the Makerere Biological Research Station in Kibale*
 *National Park, Uganda with assistance from the Kibale Chimpanzee Project and the Ngogo Chimpanzee*
 *Project in summer 2016. These included four female philopatric species — black and white colobus*
 *monkeys (Colobus guereza), olive baboons (Papio anubis), and two species of guenons, blue monkeys*
 *(Cercopithecus mitis) and redtail monkeys (C. ascanius); and two male philopatric species —*
 *chimpanzees (Pan troglodytes schweinfurthii) and red colobus monkeys (Procolobus badius)*
 *(Supplementary Table 2). Areas of cancellous bone (such as the ends of long bones, ribs, or areas on the*
 *skull) which have a more rapid turnover rate than areas of cortical bone were preferentially sampled*
 *when available to increase the likelihood of a post-dispersal isotopic signature in bones; first molars,*
 *which erupt around the time of weaning were sampled when possible to ensure pre-dispersal isotopic*
 *signatures in teeth. Only fully mature adult individuals were included in the sample. Individuals were*
 *assigned provenience data based on collection location.*

*Laboratory analysis:* We performed all sample preparation and $^{87}\text{Sr}/^{86}\text{Sr}$ analysis in a clean lab at
 the University of Utah, Salt Lake City, USA in the Department of Geology and Geophysics following
 standard protocols. Immediately following collection in Kibale National Park, we dried all plant leaf
 samples in a food dehydrator between 50-60 degrees Celsius, then manually homogenized the sample
 for storage in an airtight bag. We performed all additional sample preparation and $^{87}\text{Sr}/^{86}\text{Sr}$ analysis in a
 clean lab following standard protocols at the University of Utah, Salt Lake City, UT, USA in the

It is not clear what dataset the authors are referring to here. Must make it clear that they are using PLANTS (or perhaps other sedentary materials like soils or relatively immobile organisms like rodents).

Department of Geology and Geophysics. All subsequent sample preparation and analysis was completed
 in a clean lab at the University of Utah, Salt Lake City, USA in the Department of Geology and
 Geophysics. We prepared 50 mg of each plant sample and about 5 mg of powdered tooth or bone for
 digestion. We placed plant samples in PTFE digestion micro-vessels using with 2 ml of clean nitric acid.
 For plants, a microwave digester (Ethos EZ microwave digestion system, Milestone, Inc., Shelton, CT,
 USA) digested the dried samples at 200°-degrees-Celsius for 20 minutes to oxidize the sample matrix and
 obtain a solution with Sr and other trace elements. Powdered tooth and bone samples were digested in 2
 178 mL of concentrated nitric acid (HNO₃) at room temperature under a laminar flow bench for 20 minutes to
 179 break down the sample matrix and transfer the elements of interest into a solution. Tooth and bone
 samples were digested in two milliliters of acetic acid under a laminar flow hood. This “cold” digestion
 decreased the dissolution of any soil or heavy contaminants that may have been mixed in with the enamel
 or bone powder (D. Fernandez, personal communication).

Once samples cooled, we spiked all sample digests with an internal indium standard and
 used an Agilent 7500ce quadrupole ICPMS to measure the concentration of strontium in each sample.
 Using this measurement, we prepared a 1 ml one- or three-ml solution of each sample containing 30
 186 ppb 200 ng strontium-Sr and purified it using the PrepFAST purification system (ESI, Omaha,
 Nebraska). We then dried the down-purified samples on a PTFE-covered hot plate at 250 degrees
 Celsius under a laminar flow hood for approximately two hours, concentrating all present solids. We
 rehydrated the solids using two 2 milliliters-mls of 2.4% clean nitric acid. Finally, a Neptune Plus multi-
 collector ICP-MS (Thermo Finnigan, Bremen, Germany) took the strontium isotope measurements. Each
 sample was followed by a blank. Every three samples, we ran one standard-certified- reference material
 sample (NIST SRM 987, National Institute of Standards and Technology, Gaithersburg, MD, USA). Our
 mean and standard deviation for reference standards was 0.710287 +/- 0.000011.

*Data analysis:* We first assessed the reliability of the standard archeological approach to
 classifying local and non-local individuals. To do this To compare the efficacy of the standard
 archeological approach to our offset methods (tooth enamel-to-bone offsets and tooth enamel-to-
 environment offsets), we assigned each primate in our sample a “local/philopatric” or “non-
 local/disperser” attribution based on sex and species-level sex biases in dispersal patterns. we
 assumed This assignment necessitates an idealized expectation- that all adult primates of the dispersing sex
 for each species were in fact individuals who locationally dispersed (non-locals), and all those of the
 philopatric sex were individuals who did not disperse (locals). The authors recognize that this is an overly
 simplified version of a dispersal model; in reality, rates of dispersal from natal groups can vary widely
 and are dependent on resource availability, social factors, habitat fragmentation, and other factors^{38,39}. For
 example, in male-philopatric chimpanzees, male dispersals are virtually unheard of due to lethal

What was concentration of acid?

This is confusing as written. Do the authors mean solids that didn't digest, or do they mean they precipitated solids out of solution? They were left with a film of precipitated solids?

intergroup aggression among males⁴⁰, but female dispersal rates from natal groups vary from 50% to
 nearly 100%⁴¹⁻⁴⁴. In fact, we ~~We made one exception to this rule within our own dataset~~ for sample KCP
 6, an adult female chimpanzee from the Kanyawara community known by researchers to have been born
 in that community and failed to ever disperse (M. Muller, personal communication). Even in situations
 where dispersal by members of a species' dispersing sex is not ubiquitous, it is reasonable that a greater
 proportion of members of this sex will disperse compared to members of the species' philopatric sex⁴⁵⁻⁴⁸.
 We can therefore use the frequency of matches ^{spell out words.} vs mis-matches between the ecologically expected
 attribution (based on species-level sex-biases) and the assigned attribution (from the isotopic model) to
 determine the degree of reliability for a given modeling approach, even knowing that some expected
 attributions may be inaccurate for some individuals.

To test the efficacy of the standard archeological approach, we defined minimum and maximum
 "local" $^{87}\text{Sr}/^{86}\text{Sr}$ thresholds based on plant leaves collected from the Kanyawara and Ngogo study areas
 and compared the $^{87}\text{Sr}/^{86}\text{Sr}_{\text{enamel}}$ of fauna collected from each area to this range. We compared the
 strontium isotope ratios of each individual's earliest forming available tooth ($^{87}\text{Sr}/^{86}\text{Sr}_{\text{enamel}}$) to the
 minimum and maximum environmental values of the park area from which the individual was collected.
 Individuals with $^{87}\text{Sr}/^{86}\text{Sr}_{\text{enamel}}$ falling outside of the local area were ~~classified assigned as~~ "non-local/
 disperser" attributions while individual with $^{87}\text{Sr}/^{86}\text{Sr}_{\text{enamel}}$ falling within the local area ~~were were assigned~~
 ~~a "classified as local/philopatric" attribution.~~ We then calculated the number of ~~classification errors~~
 ~~mismatches between the expected and assigned attributions, based on known sex and species dispersal~~
 ~~patterns.~~ We assessed efficacy at the individual level (how many individuals ~~were placed in the correct~~
 ~~category had assigned attributions which matched their expected attributions~~) and at the species level (if
 ~~the overall patterns within each species correctly indicated which sex dispersed and which was~~
 ~~philopatric a greater proportion of dispersing sex individuals were assigned "non-local/disperser"~~
 ~~attributions based on the isotopic data compared to the proportion of philopatric individuals).~~

~~We then~~Next, we tested the efficacy of our new offset method. To determine offsets between
 juvenile and adult strontium signals for each individual, we first calculated the difference between the
 strontium isotope ratio of an individual's ~~earliest forming molar tooth enamel~~ and that of ~~their bones~~
 ($^{87}\text{Sr}/^{86}\text{Sr}_{\text{enamel-bone}}$). ~~Second~~Then, we calculated the offset between an individual's ~~earliest forming~~
 ~~molar tooth enamel~~ and the mean of the local environment, as defined by the average $^{87}\text{Sr}/^{86}\text{Sr}$ ratio of the
 plants within the isotopic cluster on which the remains were recovered ($^{87}\text{Sr}/^{86}\text{Sr}_{\text{enamel-environment}}$). As with
 the standard archeological method, we assessed efficacy at the individual and species level for each
 offset.

For both offset methods, we transformed each offset into its absolute value for analysis because
 we are interested in the *magnitude* of the difference between the juvenile and adult signatures, rather than

exception to what? Exception to dispersal.
 Could also reword to say, We know at least
 one female in our dataset did not disperse.

The authors haven't yet mentioned that they are
 comparing "assigned" and "expected" dispersal using
 their offsets. I think this needs to come before they
 can refer back to doing this.

its bone

Or could make this a simpler sentence and say calculated the
 isotopic difference between an individual's tooth enamel and bone
 ($^{87}\text{Sr}/^{86}\text{Sr}_{\text{enamel-bone}}$).

the directionality. ~~The direction of dispersal determines; whether or not the juvenile isotopic ratio was is~~
~~higher or lower than the adult isotopic ratio-ratio, leading to either a positive or negative offset value. This~~
~~information would vary based on the direction of dispersal, and was not relevant to this particular study.~~

None of this seems necessary.

~~Because of~~Due to small sample sizes, a common problem in paleontological studies, we could not
 use standard frequentist statistics to investigate differences in offset values. Instead, we used parametric
 bootstrapping, which uses the mean and standard deviation of offset values collected for each sex in each
 primate group to draw normal distributions. We randomly sampled 100 data points from ~~these~~
~~distributions~~ and compared the offsets between the sexes for each species using Student's t-tests. We
 repeated this simulation 10,000 times and determined the percentage of times that the p-value was
 significant ($p < 0.05$). Parametric bootstrapping guards against under-estimations of true population
 variance and spurious fine structures that nonparametric methods can propagate. By iterating the
 bootstrap 10,000 times, we could go beyond a frequentist interpretation to assess the probability of
 retrieving a significant p-value. This method was considered effective if such a probability was greater
 than 90%.

but this is not a paleontological study. Please clarify "which is also a common problem in paleontological studies".

From each distributions, or across all distributions?

Finally, we categorically classified each individual as either 'high offset' or 'low offset' for both
 $^{87}\text{Sr}/^{86}\text{Sr}_{\text{enamel-bone}}$ and $^{87}\text{Sr}/^{86}\text{Sr}_{\text{enamel-environment}}$. High-offset individuals were those with offsets above the
 species' mean offset value; low-offset individuals were those with offsets below the species' mean. In an
 idealized data set, high-offset individuals would be non-local/dispersers, while low-offset individuals
 would be local/philopatric. We ~~assessed-calculated~~ the proportion of individuals ~~correctly-with an~~
~~assigned categorized attribution (local/non-local) that matched their expected attribution based on sex and~~
~~species dispersal patterns. Finally, we and~~ compared the proportions of high and low offset individuals
 between sexes to determine if a higher proportion of high offsets came from the dispersing sex. ~~This -~~
~~This allowed~~ us to assess if the method could ~~be used to~~ accurately predict species-level sex-biases in
 dispersal patterns.

Given that the authors are working with absolute values (right?) would it make more sense to think in terms of "large" and "small"?

Really it's offsets larger than the mean and smaller than the mean. Above and below suggests directionality, which gets confusing since the authors are working with absolute values.

Results

Just say leaf samples

Environmental: ~~Botanical (plant leaf) samples show extensive and patterned isotopic variability~~
~~across Kibale National Park (Figure 1). However, there were not statistical differences in the isotopic~~
~~ratios from botanical samples collected on the two major geologic formations (the Toro and the Buganda)~~
~~(Wilcoxon Mann-Whitney U-test, $U = 157$, $p > 0.1$).~~ Hierarchical cluster analysis identified three
 isotopically unique clusters in Kibale National Park ~~which roughly, but not completely, conformed to the~~
~~three geologic zones indicated on the bedrock map of the park: a "Northern" cluster predominantly on the~~
~~geologic Toro Formation and including the Kanyawara and Sebitole park areas (mean $^{87}\text{Sr}/^{86}\text{Sr} = 0.7094 \pm$~~
~~0.001), a "Southern" cluster predominantly on the Buganda Formation and including the Kanyanchu park~~

Way more concise to say "samples collected on the Toro and Buganda Formations did not differ isotopically" (present stats result). Also need to add that plants from the anomaly had considerably higher $^{87}\text{Sr}/^{86}\text{Sr}$ than either mapped geology.

The text here is very misleading. It suggests that the clustering aligns with geology, which the authors have gone to great lengths to suggest is not the case.

As far as I can tell looking at Figure 1, which finally shows geologies, the northern cluster is underlain by both the Toro and Buganda Formation.

Kanyanchu samples must also be on mixed geologies as Kanyanchu itself is on the radiometric anomaly.

I don't know where the spatial divisions are drawn between the plant clusters but there are a whole bunch of "southern cluster" plants (i.e. green) even north of Ngogo....

The text here is very misleading. It suggests that the clustering aligns with geology, which the authors have gone to great lengths to suggest is not the case.

As far as I can tell looking at Figure 1, which finally shows geologies, the northern cluster is underlain by both the Toro and Buganda Formation.

Kanyanchu samples must also be on mixed geologies as Kanyanchu itself is on the radiometric anomaly.

I don't know where the spatial divisions are drawn between the plant clusters but there are a whole bunch of "southern cluster" plants (i.e. green) even north of Ngogo.... In fact, it looks like there is a cluster of green between Kanyawara and Sebitole.... Is this still considered southern?

area as well as a portion of Ngogo (mean $^{87}\text{Sr}/^{86}\text{Sr} = 0.7169 \pm 0.0026$), and the Ngogo cluster comprising an extremely geographically constrained cluster area on the top an unclassified "unidentified radiometric anomaly" underlying the remainder of the Ngogo study site, which we refer to as the "Ngogo" cluster (mean $^{87}\text{Sr}/^{86}\text{Sr} = 0.7273 \pm 0.004$) (Figure 1, Kruskal Wallance rank sum test, chi-squared = 51.712, df = 2, $p < 0.001$). The greatest straight-line distance that an individual could travel and remain on a completely isotopically homogenous area was/would be approximately 3 km. This value defines 'large' and 'small' home ranges relative to Kibale's isotopic variation. Fauna who could reasonably range and disperse distances greater than or equal to 3 km have 'large home ranges' in this context (here, chimpanzees and olive baboons) while those reasonably ranging < 3 km have 'small home ranges' (all colobus monkeys and guenons) (Table 1).

Local vs Non-local (Standard Archeological Method): Following protocol for the common archeological method, we defined a minimum and maximum "local" value based on plants collected from the Kanyawara and Ngogo study areas and compared the $^{87}\text{Sr}/^{86}\text{Sr}_{\text{enamel}}$ of fauna collected from each area. We correctly identified 60% of individual Kibale primates were assigned their expected attribution (as local/philopatric or non-local/dispersing) using this method (Table 2, Figure 2) based on known sex and sex biases in species dispersal patterns. There is a strong bias towards correctly matching expected/assigned attributions for identifying locals/philopatric individuals and misidentifying mismatched expected/assigned attributions for non-locals/dispersers: 83% of local/philopatric individuals and only 32% of non-local/dispersing individuals were correctly identified assigned their expected attribution. At the species level, three out of five species (female-philopatric chimpanzees, female-philopatric red colobus monkeys, and male-philopatric olive baboons) had a higher proportion of individuals of the dispersing sex fall outside of the local minima and maxima, and therefore which would lead researchers to infer the correct sex bias in dispersal patterns if this were an unknown assemblage. (Table 3).

Enamel-Bone Offsets (bootstrapping): Nearly all species trended had towards higher $^{87}\text{Sr}/^{86}\text{Sr}_{\text{enamel-bone}}$ offset for the dispersing sex (Figure 3a). Olive baboons and guenons had significant p -values by different $^{87}\text{Sr}/^{86}\text{Sr}_{\text{enamel-bone}}$ offsets between the sexes ($p < 0.05$) in 100% of bootstrapped simulations, black and white colobus monkeys in 92% of simulations, and red colobus monkeys in 99% of simulations. Chimpanzees, however, returned significant p -values had significantly different offsets between the sexes in only 5% of simulations (Figure 4a, Table 3). These data suggest that $^{87}\text{Sr}/^{86}\text{Sr}_{\text{enamel-bone}}$ offsets can discriminate between philopatric and dispersing sexes for many primate species, but not for species with relatively large home ranges, such as chimpanzees.

Enamel-Bone Offsets (categorical classification): 67% of individual primates were correctly classified as local/philopatric or non-local/dispersing based on whether they fell into the individuals with

I'm still confused about how this value was calculated. I see a whole bunch of red in the middle of the park and a whole bunch of green down in the southern end. An individual could readily walk >3.5 km or even 10 km and still be within the same isotopic cluster.

I'm sorry but as written, this still reads that we know the dispersal status of each individual, which is not the case. Can't have expected at the end of the sentence. Have to say "expected local/philopatric individuals" and "expected non-local/dispersing individuals".

It would be more meaningful if the authors could please clarify HOW things are significantly different. Specifically, is offset for the dispersing sex larger? Hopefully it's not the other way around.

high offsets ~~category~~ (greater than the species' mean offset, ~~indicating dispersal~~) were assigned a "non-local/disperser" attribution; individuals with ~~or~~ low-offsets ~~category~~ (a lower offset than the species' mean, ~~indicating philopatry~~) were assigned a "local/philopatric" attribution. Sixty-seven percent of individual primates were assigned an attribution which matched their expected attribution based on sex and species. As with the standard archeological method, there was a bias in favor of ~~correctly~~ ~~identifying~~ ~~matching~~ expected/assigned attributions for local/philopatric individuals (86% ~~correct~~ ~~matching~~) and mis-~~identifying~~ ~~matched~~ expected/assigned attributions for non-local/dispersing individuals (only 46% ~~correct~~ ~~matching~~) (Table 45). This method was nonetheless more accurate than the standard archeological method. At the species level, red colobus monkeys, olive baboons, and guenons all had a greater proportion of ~~dispersing-sex individuals~~ ~~the dispersing-sex~~ with high offsets compared to the proportion of ~~philopatric-sex individuals~~ ~~the philopatric-sex~~. ~~C~~, while chimpanzees and black and white colobus monkeys had approximately equal proportions of each sex fall above the group mean. (Table 3).

Enamel-Environment Offsets (bootstrapping): Enamel from Kanyawara and Sebitole individuals were compared to the Northern cluster $^{87}\text{Sr}/^{86}\text{Sr}$ mean (0.7094 ± 0.0010), Kanyanchu individuals to the Southern cluster $^{87}\text{Sr}/^{86}\text{Sr}$ mean (0.7169 ± 0.0026), and Ngogo individuals to the mean $^{87}\text{Sr}/^{86}\text{Sr}$ of the combined Ngogo and Southern clusters (0.7189 ± 0.005) based on the cluster(s) underlying each park area. Species with larger home ranges trended towards higher $^{87}\text{Sr}/^{86}\text{Sr}_{\text{enamel-environment}}$ offsets for the dispersing sex (Figure 3b). Parametric bootstrapping methods showed significant differences between the sexes ($p < 0.05$) in 98% of simulations for chimpanzees, 100% of simulations for olive baboons, and 96% of simulations for guenons, but only 5% of simulation for red colobus monkeys and 39% for black and white colobus monkeys (Figure 4b, Table 34). These results indicate that $^{87}\text{Sr}/^{86}\text{Sr}_{\text{enamel-environment}}$ offsets may be more applicable to species with large home ranges relative to the isotopic variation on the landscape.

Enamel-Environment Offsets (categorical classification): At the individual level, this method ~~correctly~~ ~~classified~~ ~~assigned~~ non-local/dispersing and local/philopatric attributions that matched the expected attribution in 65% of individuals, including 79% of local/philopatric-sex individuals and 50% of non-local/dispersing-sex individuals (Table 56). At the species level, 100% of species had a higher proportion of ~~dispersing-sex the individuals~~ ~~dispersing sex~~ than the philopatric sex fall above the taxon's species' mean $^{87}\text{Sr}/^{86}\text{Sr}_{\text{enamel-environment}}$ offset than individuals of the expected philopatric sex (Table 3). This was overall the most reliable method tested. This may suggest that the issue with the comparison of means may be a product of small sample size, and proportionately calculating how many individuals of each sex have high offsets relative to the rest of the species is a more robust way to handle data of this sort when sample sizes are low. Conservatively, sex differences in $^{87}\text{Sr}/^{86}\text{Sr}_{\text{enamel-environment}}$ offsets can be considered reliable for species with large home ranges relative to landscape isotopic variation. Table 67

Again, language is still an issue. I think here the authors might mean for the local/philopatric SEX (they still don't know know if individuals were local/philopatric).

I'm confused. The authors defined clusters of plants but then didn't use these clusters for local comparisons? Why didn't Ngogo individuals get compared to the Ngogo cluster? What's the point in creating a cluster if it is then ignored?

This language is clearer. Thank you.

Would be more straightforward to say "all 5" since percents are used to represent individuals in the previous sentences.

summarizes these results.

Discussion and Conclusion

**This study's goal** was to determine the degree to which strontium isotope ratios can differentiate
 between philopatric and dispersing individuals in primate communities, and what method and analytical
 approach most reliably identified sex-bias in dispersal patterns. In particular, we wanted to compare the
 efficacy of **the a** standard approach taken in the archeological literature to our novel offset values
 approach. Our approach is critically different **from the standard archeological approach** for two primary
 reasons. **First**:

**1.** ~~— Rather than relying on previously mapped boundaries (geologic or geographic) to define~~
 ~~“local” isotopic minima and maxima, we use the isotopic ratios themselves to generate geochemically~~
 ~~meaningful clusters. This approach yields two benefits: first, it allows for this method to be applied in~~
 ~~areas without detailed geologic maps available. Second, it greatly diminishes isotopic overlap between~~
 ~~local areas. Previous studies which use underlying geology as de facto boundaries for discrete “local”~~
 ~~areas run into the problem that isotopically, these different geologic areas overlap⁹. In other words, an~~
 ~~⁸⁷Sr/⁸⁶Sr_{enamel} ratio that falls within the predetermined local range based on the geologic formation on~~
 ~~which it was found could simultaneously overlap isotopically with a neighboring geologic formation as~~
 ~~well. By using statistical clusters of isotopic ratios from botanical samples to draw “local” vs “non-local”~~
 ~~boundaries rather than assuming isotopic boundaries follow geologic boundaries allows researchers to~~
 ~~draw more geochemically meaningful lines across the landscape and assess with a higher degree of~~
 ~~certainty whether a given ⁸⁷Sr/⁸⁶Sr_{enamel} ratio is indeed local to an area.~~

**2.** ~~w~~While the standard method creates binary categories (“local,” “non-local”) based on
 environmental minima and maxima, our new method quantifies the amount of variation from a local mean
 to assess the likelihood of an individual being local or non-local based on the degree of offset relative to
 other members of the same species. ~~This means that D~~differences **between** individuals belonging to **among?**
 species with home range sizes or dispersal distances smaller than the previously designated local/non-
 local areas can still be identified, whereas they are overlooked using the standard approach. **Second, we**
 **use hierarchical cluster analysis to generate geochemically unique “local” areas rather than relying on**
 **geologic formations or other predetermined boundaries to group ⁸⁷Sr/⁸⁶Sr_{environment} samples. Use of these**
 **statistical clusters provides a higher degree of certainty whether a given ⁸⁷Sr/⁸⁶Sr_{enamel} ratio is indeed local**
 **to an area even when geologic or geographic areas overlap isotopically (such as Swartkrans and**
 **Sterkfontein caves in South Africa⁹), and/or in which sufficiently detailed geologic maps are not available**
 **(such as Kibale National Park).**

At the individual level, using **our known sample** of primates from Kibale National Park, Uganda,

What does this mean?

Do the authors mean they are able to assume which sex disperses since these are extant species (i.e. species with known behaviors)?

A small thing but worth mentioning - studies are inanimate things. They don't do things. Researchers do things.

Better wording would be “Our goal in this study was to determine...”

But the authors combined statistical clusters for Ngogo individuals, which I find perplexing. Why did they do that? How does that help improve certainty that a given ratio is indeed local?

I STILL want to know how results would differ if the authors used the mean values for plants on different geologies rather than these lumped clusters to calculate their offsets. How can the authors claim their cluster method is better without ever having done that comparison?

used?

awkward. Do the authors mean whether or not? that?

This of course also requires that there is geographic isotopic heterogeneity. That important point doesn't come across in the present message.

the standard archeological method accurately categorized 35% of dispersing sex primates expected to be
 non-local/dispersers based on species and sex were assigned as such a matching attribution based on
 isotopic data. The $^{87}\text{Sr}/^{86}\text{Sr}_{\text{enamel-bone}}$ offset method correctly classified assigned 46% of expected non-
 local/dispersing individuals a matching attribution, and the $^{87}\text{Sr}/^{86}\text{Sr}_{\text{enamel-environment}}$ offset method correctly
 assigned identified 50% of non-local/dispersing individuals individuals a matching attribution. (Tables 2,
 5, 6). At the species level, the standard archeological method incorrectly indicated inferred the expected the
 sex bias in dispersal patterns in three of five species and incorrectly indicated the opposite of the expected
 attributed the sex-bias in two out of five species. T; the $^{87}\text{Sr}/^{86}\text{Sr}_{\text{enamel-bone}}$ offset method correctly but? Seems like the authors should point out this is counter (and doesn't do th
 attributed indicated the expected sex-bias in dispersal in three of five species using categorical
 classification based on high and low offsets, with inconclusive results in two of five species. T; the
 $^{87}\text{Sr}/^{86}\text{Sr}_{\text{enamel-environment}}$ offset method accurately indicated the expected sex-biases in dispersal in all five
 species tested using categorical classification based on high and low offsets (Tables 2, 4, 5)(Table 3).

While no method was perfect, the $^{87}\text{Sr}/^{86}\text{Sr}_{\text{enamel-environment}}$ offset method using categorical
 classification of high and low offsets was the most accurate and effective approach examined herein this
 study, particularly for species with large home ranges. Within the $^{87}\text{Sr}/^{86}\text{Sr}_{\text{enamel-environment}}$ offset method, the
 categorical classification approach was more reliable than the bootstrap approach. This may suggest that
 the issue with the comparison of means could be a product of small sample size. Proportionately
 calculating how many individuals of each sex have high offsets relative to the rest of the species is a more
 robust way to handle data of this sort when sample sizes are low. small?

Trends in the reliability of each method paint a more complex picture of the eircumstances in
 which when each method may be the most dependable is most applicable. For example, for primates with
 large home ranges and dispersal distances (those meeting or exceeding the size of local isotopically
 homogenous areas), comparing mean $^{87}\text{Sr}/^{86}\text{Sr}_{\text{enamel-environment}}$ offsets (with large sample sizes) or
 proportions of each sex with high offset values from this proxy (for smaller sample sizes) was most
 reliable. For primates with dispersal distances smaller than a landscape's isotopic clusters with bone
 isotopic ratios available, comparing $^{87}\text{Sr}/^{86}\text{Sr}_{\text{enamel-bone}}$ offsets was also a reliable approach.

For fossil contexts, the authors recommend using the categorical classification of $^{87}\text{Sr}/^{86}\text{Sr}_{\text{enamel-}}$
 environment offsets because of its reasonably high reliability and lack of reliance on bone samples, which are
 notoriously unreliable after fossilization. Measuring and quantifying environmental isotopic clusters for a
 fossil context could be approached in numerous ways. The half-life of rubidium-87, which decays into
 strontium-87, is 48.8 million years, so modern environmental samples could be used as proxies for paleo
 isoscapes⁹ under conditions in which the age of the underlying geology is older than the fossils of
 interest surface geology has been consistent since the time period of interest. There are other data proxies
 that could substitute for local environmental estimates as well. FFossils of small rodents and other

This wording is clear. Thank you.

Commented [DP1]: Identified seems not needed after assigned

Remind the reader most accurate and effective approach for... identifying dispersing individuals? Distinguishing sex biases in dispersal patterns?

delete

What about comparing median values?

Sentence seems unnecessarily complicated. Of course this can only be done for animals that have bone data... The point is that this also works for primates with relatively small dispersal distances.

I think here the authors specifically mean modern plants?

micromammals, are also commonly used²² to establish similar local environmental profiles to botanical samples^{22,50} but have been underutilized in the past, and could be exceedingly useful in applying these methods to fossil contexts. More work of this kind in hominin paleoenvironments would greatly benefit the state of current research. Strontium isotope ratios of finally, while researchers have used carbon and oxygen isotopic records from paleosols for decades for paleoenvironmental reconstructions^{51,52}, strontium isotopes in paleosols have been less thoroughly examined. This approach from fossil deposits also could hold promise, but further methodological work would be needed before its implementation we could confidently use this method. Additional work investigating other philopatric patterns, including bisexual dispersal, and conducting modern studies in more geochemical diverse areas would further solidify and refine the method recommendations presented here.

Potential Sources of Model Error

Age of sampled tooth mineralization: We sampled first molars whenever present to ensure that mineralization of tooth enamel occurred prior to any dispersal; however, for six individuals in the collection, first molars were not available. We were able to sample earlier forming teeth in two of these individuals (a first incisor in KFB 122, a male redbellied monkey, and a first premolar in KFB 171, a male red colobus monkey) and second molars in the other four (KFB 57 and KFB 111, both female red colobus monkeys, and KFB 154 and KCP 9, both male chimpanzees) (**Supplementary Table 1**). Second molars erupt before age eight in chimpanzees⁵³. While variable, female chimpanzees typically do not disperse until after reaching sexual maturity around age thirteen⁴¹⁻⁴³. Data on eruption times for colobus monkeys is sparser, but modeling based on brain size by Smith (1994) suggests that the final permanent tooth (the third molar) will erupt by 3.7 years. Dispersal in colobines is rare before the age of three^{54,55}, so it is unlikely (although not impossible) for dispersal to occur before the mineralization of the second permanent molar. While we cannot rule out that tooth selection caused these individuals' $^{87}\text{Sr}/^{86}\text{Sr}_{\text{enamel}}$ measurement to come from post-dispersal, it is reassuring that within our data set, the offsets calculated for these individuals are neither outliers nor even the most extreme values; all fall within the range observed for individuals with available first molars.

Strontium sources and the role of local geology: Although for many areas the local bedrock is the main source of bioavailable soil strontium, important contributions from atmospheric or hydrological origin exist in some ecosystems⁵⁶. In general, a number of processes contribute to the strontium contained in the biosphere: for example, weathering of bedrock from groundwater⁵⁷, plant root uptake from soil^{58,59} and bedrock⁶⁰, precipitation containing marine aerosols⁶¹, dust transport^{10,62}, and ecosystem recycling⁶⁰. For instance, when allochthonous sediments (e.g., glacial or eolian deposits) are the main soil parent

I am not sure why the authors say they have been underutilized in the past. Is this not the primary way that researchers establish local $^{87}\text{Sr}/^{86}\text{Sr}$ for fossils?

There is always the issue that rodents might not be local (see Copeland et al. 2010)

Copeland, S.R., Sponheimer, M., Lee-Thorp J.A., de Ruiter, D.J., le Roux, P.J., Grimes, V., Codron, D., Berger, L.R. & Richards, M.P. (2010) Using strontium isotopes to study site accumulation processes. *Journal of Taphonomy*, 8, 115-127.

If possible, it would be nice to have an opening/introduction paragraph to this section.

This wording is confusing. The authors mean "age at which sampled teeth mineralized"

Data are

Could the authors show a supplementary figure that demonstrates this? Or at least provide the ranges? Don't make the reader dig dig dig for this.

material, the $^{87}\text{Sr}/^{86}\text{Sr}$ can be very different from the bedrock⁶³. The local $^{87}\text{Sr}/^{86}\text{Sr}$ is best known from
 bioavailable Sr contained in plants or small-range mammal teeth⁵⁸, although when these data do not exist,
 the bedrock geology is a reasonable proxy for the local water and vegetation^{56,64}.

For old oxisols in tropical savannas or forests like those in East Africa, the parent material
 (bedrock, till, alluvium, etc) would be almost completely consumed by weathering. Then, bioavailable
 strontium may be primarily derived from atmospheric sources⁶⁵. In this way, the local geology may play a
 subordinate role to the strontium found in the soil. In addition, bioavailable strontium in soil within the
 riparian zone typically reflects the isotopic ratio of the strontium dissolved in the river water, which in
 turn is related to the weathering of distal upstream rocks at higher elevation⁶⁶. The local geology in the
 riparian zone soil plays again a secondary role as a source of the strontium in the plant and animal tissue.
 Bioavailable $^{87}\text{Sr}/^{86}\text{Sr}$ signature is thus best determined in these cases from plant and small-range animal
 tissue.

Kibale National Park is underlain by a geology that lends itself to high isotopic variability. First,
 the geologic formations here are quite old, dating to the early and middle Paleoproterozoic era
 (approximately 2 – 2.5 billion year ago). The Toro formation comprises undifferentiated acid and basic
 gneisses while the neighboring Buganda formation comprises primarily quartzites³⁶. These felsic
 geologies, in addition to being very ancient, are high in their Rb/Sr ratios⁶⁴. Taken together, these factors
 predict highly radiogenic strontium isotope ratios. Second, we can expect a high degree of isotopic
 heterogeneity even within a single geologic formation due to differential mineral weathering. We see
 these expectations illustrated most strikingly in samples collected near the Ngogo park area, where we
 measured the most radiogenic strontium isotope ratios in the park and observed the highest variability;
 samples collected within less than a kilometer of each other ranged from 0.7129 to 0.7339 (see **Figure 1**).

*Variation in dispersal rates:* Our assessments of method reliability depend on modeling that
 assumes all dispersal is locational, and that 100% of individuals conform to the expected dispersal
 patterns for their species. In reality, dispersal is a much more complex and non-binary. Locational
 dispersal, in which individuals leave their natal group's territory and enter into a new area, is the primary
 focus of this manuscript. However, individuals can also undergo social dispersal, in which they join
 different social groups, potentially within the same territory⁶⁷. The methods presented here are not
 applicable to instance of exclusive social dispersal, as this behavior does not necessarily correlate with
 changes in physical location. It is possible to undergo locational dispersal without undergoing social
 dispersal, such as when home ranges for an entire natal group shift⁶⁸, a nuance that these methods could
 not indicate.

The degree to which individuals within a given species follow the expected dispersal or
 philopatric pattern predicted by their sex varies substantially and is influenced by social and

I don't see how any of these details about geology lead to model error. This all seems like model justification (i.e. explanation as to why 'isotopic clusters' are better than relying on geologic boundaries).

Also, why are the authors referring to their method as a model?

The authors STILL need to show that geology doesn't work. They put a lot of faith in their claim that it doesn't and that "environment clusters" are better but they still don't justify this.

Is there a river at Kibale? If so, this seems important. Is there a riparian effect?

This background on geology is super useful and important but it belongs earlier. It sets the stage for why the authors are interested in relying on something other than bedrock geologic boundaries to distinguish different local regions.

I think its quite interesting that the authors found rather low ratios in the northern cluster.

The authors suggested that there is a radiometric anomaly here earlier in their paper and in Figure 1. I had thought this was previously identified by others... Why isn't that mentioned here? If this is something just discovered by the authors, then why do they show it as a geology in Figure 1? I'm confused.

Are the authors saying it is ALSO a possibility that can't be identified using Sr isotopes? I don't follow.

environmental factors, including status within dominance hierarchies, rates of affiliation and aggression,
 resource and mating competition, climate, and habitat fragmentation. We note that in our models, there
 was a bias towards mis-classifying dispersing sex individuals; it is entirely possible (if not likely) that for
 some of these, the “mis-classification” is a true signal that they did not disperse at all.

These sources of error provide insight into why the categorical classification approach led to the
 most faithful reconstruction of species-level sex-biases in dispersal patterns, particularly when using
 $^{87}\text{Sr}/^{86}\text{Sr}_{\text{enamel-environment}}$ offsets. Even when sex-biased patterns in dispersal are not followed by 100% of
 individuals within a primate group, it is reasonable to assume relatively more members of the dispersing
 sex will leave and more members of the philopatric sex will remain. Despite modeling limitation at the
 individual level, when attributions are examined in aggregate for all individuals within a species, a greater
 proportion of dispersing-sex individual had high offsets while a greater proportion of philopatric sex
 individuals had low offsets. While the models cannot predict individual-level attributions with complete
 confidence due to the overly simplified assumptions regarding the nature of dispersal, they can uncover
 sex biases in dispersal patterns at the species level when enough individuals are included.

The impact of life history variables: The observed success or failure of the tooth-environment and
 tooth-bone offset proxies is also potentially also related to a combination of home range size and life
 history variables (Table 1). Chimpanzees have highly variable home ranges based on their habitat^{69,70}; in
 forests such as Kibale National Park, researchers measured home range sizes between home ranges up to
 28-41±5 km², which is large relative to the size of the isotopic clusters^{71,72}. ~~clusters in Kibale~~-Tissues
 formed while ranging over this large area incorporate a variety of strontium isotope ratios. We
 hypothesize that this averaging effect mutes the between-tissue difference equally for both philopatric and
 dispersing individuals, decreasing the reliability of tooth-bone offsets and renderings tooth-environment
 offsets a more reliable approach. However, for species with relatively small home ranges compared to
 the isotopic clusters on the landscape (such as colobus monkeys and guenons); each tissue incorporates
 only a small, specific strontium source during formation. are likely to disperse a much shorter distance.
 Comparisons between tooth enamel and the environment (defined as the mean isotopic ratio of the local
 cluster) might look the same for both dispersers and philopatric individuals in this scenario. However,
 these smaller home ranges also mean that each tissue incorporates a less variable strontium source during
 formation which increases the likelihood of detecting differences in ~~The differences between tissues~~
 ~~formed at different time intervals, such as bone and enamel, are thus accentuated for~~ offsets in dispersing
 individuals compared to philopatric individuals. This can be true even in the event that dispersing
 individuals do not travel far enough to leave the “local” isotopic cluster, as there is smaller-scale isotopic
 heterogeneity within each cluster (see Figure 1). In the present study, olive baboon home ranges
 approximate quite closely the spacing of isotopic variability on the landscape. In this “Goldilocks”

Agreed. This logic makes sense.

What did the authors model? I don't know what they are referring to when they refer to models....

Aren't tooth eruption and dispersal life history variables?

Don't the authors mean including Kibale, or specifically at Kibale?

But the authors combined clusters!

I agree this idea about olive baboon home ranges makes sense, but why is this separate from the discussion of dispersal above?

It seems to me like this section could be reorganized and it would flow better. And also the authors need to remove the geology overview as that's not a model limitation... It provides context for why the authors wanted to do this study...

scenario, we see *both* tooth-bone offsets and tooth-environment offsets effectively predict sex-biases in
 their philopatry and dispersal patterns.

Additionally, chimpanzees have a slower life history than monkeys, meaning their enamel takes a
 longer time to form⁵³²⁸. This can further exacerbate this averaging effect. ~~While male~~ Male chimpanzees
 ~~do~~ live in their natal home ranges for their entire lives; however, these ranges are large enough that ~~they~~
 they may inhabit different ~~included~~ areas as a sub-adult with their mother and as an independent adult
 competing for a place in the group's hierarchy. These factors compound to further reduce isotopic
 differences between tooth and bone tissues within the philopatric sex for large-ranging, slow-growing
 primates, thus diminishing the expected differences between the sexes for this offset.

While strontium isotope ratios hold great potential for uncovering aspects of ~~social behavior and~~
 landscape use ~~for in~~ fossil species, but methodological approaches are not one-size-fits-all. By
 considering the impacts of ecological variables, such as range size and life history, this study
 demonstrates how different methodological approaches apply under different ecological and
 environmental conditions. ~~the previous standard approach is not reliably effective for all primates. The~~
 ~~present study highlights that it is critical to establish meaningful geochemical clusters on the landscape~~
 ~~thorough sampling and mapping, as geologic boundaries do not by default correspond to unique isotopic~~
 ~~clusters. Furthermore, the size of these isotopic clusters and the distance between isotopically~~
 ~~homogenous areas must be regarded in relationship to the ranging behavior of the species of interest to~~
 ~~ensure the most accurate reconstruction of dispersal patterns. For example, the ⁸⁷Sr/⁸⁶Sr_{enamel-environment}~~
 ~~offset methods described here increase the accuracy of predicting sex-biases in dispersal beyond~~
 ~~previously existing approaches to similar data. They accurately demonstrate indicates~~ male philopatry in
 modern chimpanzees, despite large home range sizes, ~~and~~ complex mobility patterns ~~and social~~
 ~~structures, and extended life histories due to underlying social factors; it is reasonable to expect.~~ ~~These~~
 ~~complicating variables are relevant to early hominins to share many of these features as well,~~ suggesting
 that these this methods will also be more reliable than those used in the past to reconstruct our own
 ancestors' dispersal habits patterns. Future work will further refine and validate these methods. For
 example, additional work investigating other philopatric patterns, including bisexual dispersal, conducting
 modern studies in more geochemically diverse areas, and validating environmental proxies available for
 fossil areas as appropriate substitutes for the botanical samples used here are all necessary future
 endeavors.

 **Data Accessibility:** All data used in these analyses is available in the Supplemental Material (Tables 1
 and 2).

Ok... shouldn't this be part of the discussion on tooth eruption and mineralization (which is currently a separate section above)?

I agree. This is definitely one of this paper's contributions.

What method? The authors have only mentioned "strontium isotopes" above.

What methods are the authors referring to here??

Now which methods? Ones used in the past? New ones? Ones presented in this paper?

**Ethics Statement:** All fieldwork, sample collection, and sample import/export activities were approved
by The Ugandan Wildlife Authority (EDO/35/01), the Ugandan National Committee on Science and
Technology (NS 506), the US Department of Agriculture (PCIP-16-00229), the US Fish and Wildlife
Service (15US62228B/9)

**Competing Interests:** The authors declare no competing interests.

**Authors' Contributions:** MIH designed the study, performed field work and laboratory analyses,
conducting statistical analyses, and wrote the manuscript; SVN performed field work and laboratory
analyses and contributed to writing the manuscript; DPF designed laboratory protocols, supervised
laboratory analyses, and contributed to writing the manuscript; all authors gave final approval for
publication.

**Acknowledgements:** The authors would like to thank the Ugandan Wildlife Authority, the Ugandan
National Committee on Science and Technology, the US Department of Agriculture, and the US Fish and
Wildlife Service for permits and permissions to conduct this research, Dr. Emily Oтали and the, staff and
field assistants at Kibale National Park, Lee Drake and Karissa Kilgore for manuscript feedback, and our
reviewers for their thoughtful feedbackcritiques and improvements.

562
**Funding:** National Science Foundation Graduate Research Fellowship (DGE-0903444) (MIH), Wenner-
Gren Foundation (MIH), the Leakey Foundation (MIH), and the University of New Mexico Departments
of Anthropology and Graduate Studies (MIH).

38 569 **References**

- 1. Wilkins JF, & Marlowe FW. (2006). Sex-biased migration in humans: what should we expect
from genetic data? Bioessays 28:290–300.
- 571
2. Hill, K. R., Walker, R. S., Božičević, M., Eder, J., Headland, T., Hewlett, B., ... & Wood, B.
(2011). Co-residence patterns in hunter-gatherer societies show unique human social structure.
Science, 331(6022), 1286-1289.
3. Murdock, G. P. (1981). *Atlas of world cultures*. University of Pittsburgh Press.
- 4. Goodall, J., Bandora, A., Bergmann, E., Busse, C., Matama, H., Mpongo, E., Pierce, A. & Riss,
D. (1979). Intercommunity inter- actions in the chimpanzee population of the Gombe National

Park. In: *The Great Apes* (Ed. by D. Hamburg & E. McCown), pp. 13–54. Menlo Park:
Benjamin/Cummings.
- 5. Watts, D. P., & Mitani, J. C. (2001). Boundary patrols and intergroup encounters in wild
chimpanzees. *Behaviour*, 138(3), 299-327.
- 6. Lovejoy, C. O. (2009). Reexamining human origins in light of *Ardipithecus ramidus*. *Science*,
326(5949), 74-74e8.
- 7. Gordon, A. D., Green, D. J., & Richmond, B. G. (2008). Strong postcranial size dimorphism in
Australopithecus afarensis: results from two new resampling methods for multivariate data sets
with missing data. *American Journal of Physical Anthropology*, 135(3), 311-328.
- 8. Nelson, E., & Shultz, S. (2010). Finger length ratios (2D: 4D) in anthropoids implicate reduced
prenatal androgens in social bonding. *American Journal of Physical Anthropology*, 141(3), 395-
405.
- 9. Copeland SR, Sponheimer M, de Ruiter DJ, Lee-Thorp JA, Codron D, le Roux PJ,
Grimes V, & Richards MP. (2011). Strontium isotope evidence for landscape use by early
hominins. *Nature* 474: 76-78.
- 10. Beard BL, & Johnson CM. (2000). Strontium isotope composition of skeletal material
can determine the birth place and geographic mobility of humans and animals. *Journal of*
Forensic Science 45(5): 1049-1061
- 11. Capo RC, Stewart BW, & Chadwich OA. (1998). Strontium isotopes as tracers of
ecosystem processes: theory and method. *Geoderma* 82: 197-225.
- 12. Bentley RA. (2006). Strontium isotopes from the earth to the archeological skeleton: a review.
Journal of Archeological Method and Theory 13: 135-187.
- 13. Price TD, Burton JH, & Bentley RA. (2002). The characterization of biologically available
strontium isotopic ratios for investigation of prehistoric migration. *Archeometry* 44: 117-135.
- 14. Britton, K., Grimes, V., Dau, J., & Richards, M. P. (2009). Reconstructing faunal migrations
using intra-tooth sampling and strontium and oxygen isotope analyses: a case study of modern
caribou (*Rangifer tarandus granti*). *Journal of Archaeological Science*, 36(5), 1163-1172.
- 15. Vogel JC, Eglinton B, & Auret JM. (1990). Isotope fingerprints in elephant bone and ivory.
Nature 346:747–749
- 16. Radloff, F. G. T., Mucina, L., Bond, W. J., & Le Roux, P. J. (2010). Strontium isotope analyses
of large herbivore habitat use in the Cape Fynbos region of South Africa. *Oecologia*, 164(2), 567-
578.
- 17. Kennedy BP, Folt CL, Blum JD, & Chamberlain CP. (1997). Natural isotope markers in salmon.
Nature 387:766–767

18. Hoppe, K. A., Koch, P. L., Carlson, R. W., & Webb, S. D. (1999). Tracking mammoths
and mastodons: reconstruction of migratory behavior using strontium isotope ratios.
Geology, 27(5), 439-442.
- 19. Sjögren, K. G., & Price, T. D. (2013). A complex Neolithic economy: isotope evidence
for the circulation of cattle and sheep in the TRB of western Sweden. Journal of
Archaeological Science, 40(1), 690-704.
- 20. Feranec RS, Hadly EA, & Payton A. (2007). Determining landscape use of Holocene
mammals using strontium isotopes. Oecologia 153: 943-950.
- 21. Richards, M., Harvati, K., Grimes, V., Smith, C., Smith, T., Hublin, J. J., ... & Panagopoulou, E.
(2008). Strontium isotope evidence of Neanderthal mobility at the site of Lakonis, Greece using
laser-ablation PIMMS. Journal of Archaeological Science, 35(5), 1251-1256.
- 22. Sillen A, Hall G, Richardson S, & Armstron R. (1998). 87Sr/86Sr ratios in modern and fossil
food- webs of the Sterkfontein Valley: Implications for early hominid habitat preference.
Geochimica et Cosmochimica Acta 62: 2463-2478.
- 23. Muller MN, Wrangham RW, & Pilbeam DR (Editors). (2017). Chimpanzees and Human
Evolution. Cambridge, MA: Harvard University Press.
- 24. Wrangham, R., & Pilbeam, D. (2002). African apes as time machines. In *All apes great and small*
(pp. 5-17). Springer, Boston, MA.
- 25. McGrew, W. C. (2010). In search of the last common ancestor: new findings on wild
chimpanzees. *Philosophical transactions of the Royal Society B: Biological sciences*, 365(1556),
3267-3276.
- 26. Moore, J. (1996). 20-Savanna chimpanzees, referential models and the last common ancestor.
Great ape societies, 275.
- 27. Sayers, K., Lovejoy, C. O., Emery, N. J., Clayton, N. S., Hunt, K. D., Laland, K. N., ... & Strier,
K. B. (2008). The chimpanzee has no clothes: a critical examination of Pan troglodytes in models
of human evolution. *Current Anthropology*, 49(1), 87-114.
- 28. Almécija, S. (2016). Pitfalls reconstructing the last common ancestor of chimpanzees and
humans. *Proceedings of the National Academy of Sciences*, 113(8), E943-E944.
- 29. Hill, K. R., Walker, R. S., Božičević, M., Eder, J., Headland, T., Hewlett, B., ... & Wood, B.
(2011). Co-residence patterns in hunter-gatherer societies show unique human social structure.
science, 331(6022), 1286-1289.
- 30. Wright, L. E. (2005). Identifying immigrants to Tikal, Guatemala: defining local variability in
strontium isotope ratios of human tooth enamel. *Journal of Archaeological Science*, 32(4), 555-
566.

31. Bastos, M. Q., Souza, S. M., Santos, R. V., Lima, B. A., Santos, R. V., & Rodrigues-Carvalho, C. (2011). Human mobility on the Brazilian coast: an analysis of strontium isotopes in archaeological human remains from Forte Marechal Luz sambaqui. *Anais da Academia Brasileira de Ciências*, 83(2), 731-743.
- 32. Sjögren, K. G., Price, T. D., & Ahlström, T. (2009). Megaliths and mobility in south-western Sweden. Investigating relationships between a local society and its neighbours using strontium isotopes. *Journal of anthropological archaeology*, 28(1), 85-101.
- 33. Bentley, R. A., Price, T. D., & Stephan, E. (2004). Determining the 'local' \$^{87}\text{Sr}/^{86}\text{Sr}\$ range for archaeological skeletons: a case study from Neolithic Europe. *Journal of Archaeological Science*, 31(4), 365-375.
- 34. English, N. B., Betancourt, J. L., Dean, J. S., & Quade, J. (2001). Strontium isotopes reveal distant sources of architectural timber in Chaco Canyon, New Mexico. *Proceedings of the National Academy of Sciences*, 98(21), 11891-11896.
- 35. Hodell DA, Quinn RL, Brenner M, & Kamenov G. (2004). Spatial variation of strontium isotopes in the Maya region: A tool for tracking ancient human migration. *Journal of Archeological Science* 31: 585-601.
- 36. Westerhof, A. B., Härmä, P., Isabirye, E., Katto, E., Koistinen, T., Kuosmanen, E., ... & Mänttari, I. (2014). Geology and geodynamic development of Uganda with explanation of the 1: 1,000,000 scale geological map.
- 37. Manning, C. D., & Schütze, H. (1999). *Foundations of statistical natural language processing* (Vol. 999). Cambridge: MIT press.
- 38. Stumpf, R. M., Emery Thompson, M., Muller, M. N., & Wrangham, R. W. (2009). The context of female dispersal in Kanyawara chimpanzees. *Behaviour*, 146, 629-656.
- 39. McCarthy, M. S., Lester, J. D., Langergraber, K. E., Stanford, C. B., & Vigilant, L. (2018). Genetic analysis suggests dispersal among chimpanzees in a fragmented forest landscape in Uganda. *American Journal of Primatology*, 80(9), e22902.
- 40. Watts, D. P., Muller, M., Amsler, S. J., Mbabazi, G., & Mitani, J. C. (2006). Lethal intergroup aggression by chimpanzees in Kibale National Park, Uganda. *American Journal of Primatology: Official Journal of the American Society of Primatologists*, 68(2), 161-180.
- 41. Boesch, C., & Boesch-Achermann, H. (2000). *The chimpanzees of the Taï Forest*. New York: Oxford University Press.
- 42. Nishida, T. (Ed.). (1990). *The chimpanzees of the Mahale Mountains: Sexual and life history strategies*. Tokyo: University of Tokyo Press.
- 43. Reynolds V. (2005). *Chimpanzees of the Budongo forest*. New York: Oxford University Press.

44. Wroblewski, E. E., Murray, C. M., Keele, B. F., Schumacher-Stankey, J. C., Hahn, B. H., &
Pusey, A. E. (2009). Male dominance rank and reproductive success in chimpanzees, *Pan*
troglodytes schweinfurthii. *Animal behaviour*, 77(4), 873-885.
- 45. Ekernas, L. S., & Cords, M. (2007). Social and environmental factors influencing natal dispersal
in blue monkeys, *Cercopithecus mitis stuhlmanni*. *Animal Behaviour*, 73(6), 1009-1020.
- 46. McCarthy, M. S., Lester, J. D., Langergraber, K. E., Stanford, C. B., & Vigilant, L. (2018).
Genetic analysis suggests dispersal among chimpanzees in a fragmented forest landscape in
Uganda. *American Journal of Primatology*, 80(9), e22902.
- 47. Fischer, J., Higham, J. P., Alberts, S. C., Barrett, L., Beehner, J. C., Bergman, T. J., ... & da Silva,
M. J. F. (2019). The Natural History of Model Organisms: Insights into the evolution of social
systems and species from baboon studies. *Elife*, 8, e50989.
- 48. Akinyi, M. Y., Gesquiere, L. R., Franz, M., Onyango, P. O., Altmann, J., & Alberts, S. C. (2017).
Hormonal correlates of natal dispersal and rank attainment in wild male baboons. *Hormones and*
behavior, 94, 153-161.
- 49. Bowman, J., Jaeger, J. A., & Fahrig, L. (2002). Dispersal distance of mammals is proportional to
home range size. *Ecology*, 83(7), 2049-2055.
- 50. Reed, D.N., 2007. Serengeti micromammals and their implications for Olduvai
Paleoenvironments. In: Bobe, R., Alemseged, Z., Behrensmeyer, A.K. (Eds.), *Hominin*
Environments in the East African Pliocene: An Assessment of the Faunal Evidence. Springer,
Dordrecht, pp. 21-255.
- 51. Wynn, J. G. (2000). Paleosols, stable carbon isotopes, and paleoenvironmental interpretation of
Kanapoi, Northern Kenya. *Journal of Human Evolution*, 39(4), 411-432.
- 52. Levin, N. E., Brown, F. H., Behrensmeyer, A. K., Bobe, R., & Cerling, T. E. (2011). Paleosol
carbonates from the Omo Group: Isotopic records of local and regional environmental change in
East Africa. *Palaeogeography, Palaeoclimatology, Palaeoecology*, 307(1-4), 75-89.
- 53. Smith, B.H., Crummett, T. L., & Brandt, K. L. (1994). Ages of eruption of primate teeth: a
compendium for aging individuals and comparing life histories. *American Journal of Physical*
Anthropology, 37(S19), 177-231.
- 54. Teichroeb JA, Wikberg EC, & Sicotte P. (2009). Female dispersal patterns in six groups of ursine
colobus (*Colobus vellerosus*): infanticide avoidance is important. *Behaviour* 146: 551–582.
- 55. Wikberg, E. C., Sicotte, P., Campos, F. A., & Ting, N. (2012). Between-group variation in female
dispersal, kin composition of groups, and proximity patterns in a black-and-white colobus
monkey (*Colobus vellerosus*). *PLOS one*, 7(11), e48740.

56. Bataille, C. P., Crowley, B. E., Wooller, M. J., & Bowen, G. J. (2020). Advances in global
bioavailable strontium isoscapes. *Palaeogeography, Palaeoclimatology, Palaeoecology*, 109849.
- 57. Katz, B. G., & Bullen, T. D. (1996). The combined use of 87Sr/86Sr and carbon and water
isotopes to study the hydrochemical interaction between groundwater and lakewater in mantled
karst. *Geochimica et Cosmochimica Acta*, 60(24), 5075–
5087. [https://doi.org/10.1016/s0016-7037\(96\)00296-7](https://doi.org/10.1016/s0016-7037(96)00296-7)
- 58. Maurer, A. F., Galer, S. J., Knipper, C., Beierlein, L., Nunn, E. V., Peters, D., ... & Schöne, B. R.
(2012). Bioavailable 87Sr/86Sr in different environmental samples—Effects of anthropogenic
contamination and implications for isoscapes in past migration studies. *Science of the Total*
Environment, 433, 216-229.
- 59. Flockhart, D. T., Kyser, T. K., Chipley, D., Miller, N. G., & Norris, D. R. (2015). Experimental
evidence shows no fractionation of strontium isotopes (87Sr/86Sr) among soil, plants, and
herbivores: implications for tracking wildlife and forensic science. *Isotopes in environmental and*
health studies, 51(3), 372-381.
- 60. Perakis, S. S., & Pett-Ridge, J. C. (2019). Nitrogen-fixing red alder trees tap rock-derived
nutrients. *Proceedings of the National Academy of Sciences*, 116(11), 5009-5014.
- 61. Miller, E. K., Blum, J. D., & Friedland, A. J. (1993). Determination of soil exchangeable-cation
loss and weathering rates using Sr isotopes. *Nature*, 362(6419), 438-441.
- 62. Miller, O. L., Solomon, D. K., Fernandez, D. P., Cerling, T. E., & Bowling, D. R. (2014).
Evaluating the use of strontium isotopes in tree rings to record the isotopic signal of dust
deposited on the Wasatch Mountains. *Applied geochemistry*, 50, 53-65.
- 63. Widga, C., Walker, J. D., & Boehm, A. (2017). Variability in Bioavailable 87 Sr/86 Sr in the
North American Midcontinent. *Open Quaternary*, 3(1).
- 64. Bataille, C. P., & Bowen, G. J. (2012). Mapping 87Sr/86Sr variations in bedrock and water for
large scale provenance studies. *Chemical Geology*, 304, 39-52.
- 65. Chadwick, O. A., Derry, L. A., Bern, C. R., & Vitousek, P. M. (2009). Changing sources of
strontium to soils and ecosystems across the Hawaiian Islands. *Chemical Geology*, 267(1-2), 64-
76.
- 66. Hamilton, M., Nelson, S. V., Fernandez, D. P., & Hunt, K. D. (2019). Detecting riparian habitat
preferences in “savanna” chimpanzees and associated Fauna with strontium isotope ratios:
Implications for reconstructing habitat use by the chimpanzee-human last common ancestor.
American Journal of Physical Anthropology, 170(4), 551-564.
- 67. Van Vuren, D., & Isbell, L. A. (1996). Differential costs of locational and social dispersal and
their consequences for female group-living primates. *Behaviour*, 133(1-2), 1-36.

68. Janmaat, K. R., Olupot, W., Chancellor, R. L., Arlet, M. E., & Waser, P. M. (2009). Long-term
site fidelity and individual home range shifts in *Lophocebus albigena*. *International Journal of*
Primateology, 30(3), 443-466.
- 69. Moore, J., Black, J., Hernandez-Aguilar, R. A., Idani, G. I., Piel, A., & Stewart, F. (2017).
Chimpanzee vertebrate consumption: savanna and forest chimpanzees compared. *Journal of*
Human Evolution, 112, 30-40.
- 70. Moore, J. F., Mulindahabi, F., Gatorano, G., Niyigaba, P., Ndikubwimana, I., Cipolletta, C., &
Masozera, M. K. (2018). Shifting through the forest: home range, movement patterns, and diet of
the eastern chimpanzee (*Pan troglodytes schweinfurthii*) in Nyungwe National Park, Rwanda.
American journal of primatology, 80(8), e22897.
- 71. Wilson, M. L., Kahlenberg, S. M., Wells, M., & Wrangham, R. W. (2012). Ecological and social
factors affect the occurrence and outcomes of intergroup encounters in chimpanzees. *Animal*
Behaviour, 83(1), 277-291.
- 72. Mitani, J. C., Watts, D. P., & Amstler, S. J. (2010). Lethal intergroup aggression leads to territorial
expansion in wild chimpanzees. *Current biology, 20(12), R507-R508.*
- 73. Chapman, C. A., & Pavelka, M. S. (2005). Group size in folivorous primates: ecological
constraints and the possible influence of social factors. *Primates, 46(1), 1-9.*
- 74. Rowell, T. E. (1966). Forest living baboons in Uganda. *Journal of Zoology, 149(3), 344-364.*
- 75. Struhsaker, T. T., & Leland, L. (1979). Socioecology of five sympatric monkey species in the
Kibale Forest, Uganda. *Advances in the Study of Behavior, 9, 159-228.*
- 76. Butynski, T. M. (1990). Comparative Ecology of Blue Monkeys (*Cercopithecus mitis*) in
High-and Low-Density Subpopulations. *Ecological Monographs, 60(1), 1-26.*
- Wilkins JF, Marlowe FW. (2006). Sex-biased migration in humans: what should we expect from genetic
data? *Bioessays 28:290-300.*
- 1. Hill, K. R., Walker, R. S., Božičević, M., Eder, J., Headland, T., Hewlett, B., ... & Wood, B.
(2011). Co-residence patterns in hunter-gatherer societies show unique human social structure.
*Science, 331(6022), 1286-1289.*
- 2. Murdock, G. P. (1981). *Atlas of world cultures*. University of Pittsburgh Press.
- 3. Goodall, J., Bandora, A., Bergmann, E., Busse, C., Matama, H., Mpongo, E., Pierce, A. & Riss,
D. (1979). Intercommunity inter-actions in the chimpanzee population of the Gombe National
Park. In: *The Great Apes* (Ed. by D. Hamburg & E. McCown), pp. 13-54. Menlo Park:
Benjamin/Cummings.

- 4. Watts, D. P., & Mitani, J. C. (2001). Boundary patrols and intergroup encounters in wild
chimpanzees. *Behaviour*, 138(3), 299-327.
- 5. Lovejoy, C. O. (2009). Reexamining human origins in light of *Ardipithecus ramidus*. *Science*,
326(5949), 74-74e8.
- 6. Gordon, A. D., Green, D. J., & Richmond, B. G. (2008). Strong posterianal size dimorphism in
*Australopithecus afarensis*: results from two new resampling methods for multivariate data sets
with missing data. *American Journal of Physical Anthropology*, 135(3), 311-328.
- 7. Nelson, E., & Shultz, S. (2010). Finger length ratios (2D: 4D) in anthropoids implicate reduced
prenatal androgens in social bonding. *American Journal of Physical Anthropology*, 141(3), 395-
405.
- 8. Copeland SR, Sponheimer M, de Ruiter DJ, Lee-Thorp JA, Codron D, le Roux PJ,
Grimes V, and Richards MP. (2011). Strontium isotope evidence for landscape use by
early hominins. *Nature* 474: 76-78.
- 9. Beard BL, Johnson CM. (2000). Strontium isotope composition of skeletal material can
determine the birth place and geographic mobility of humans and animals. *Journal of*
*Forensic Science* 45(5): 1049-1061
- 10. Capo RC, Stewart BW, Chadwich OA. (1998). Strontium isotopes as tracers of
ecosystem processes: theory and method. *Geoderma* 82: 197-225.
- 11. Bentley RA. (2006). Strontium isotopes from the earth to the archeological skeleton: a review.
*Journal of Archeological Method and Theory* 13: 135-187.
- 12. Price TD, Burton JH, Bentley RA. (2002). The characterization of biologically available
strontium isotopic ratios for investigation of prehistoric migration. *Archeometry* 44: 117-135.
- 13. Britton, K., Grimes, V., Dau, J., & Richards, M. P. (2009). Reconstructing faunal migrations
using intra-tooth sampling and strontium and oxygen isotope analyses: a case study of modern
caribou (*Rangifer tarandus granti*). *Journal of Archaeological Science*, 36(5), 1163-1172.
- 14. Vogel JC, Eglinton B, Auret JM (1990) Isotope fingerprints in elephant bone and ivory. *Nature*
346:747-749
- 15. Radloff, F. G. T., Mucina, L., Bond, W. J., & Le Roux, P. J. (2010). Strontium isotope analyses
of large herbivore habitat use in the Cape Fynbos region of South Africa. *Oecologia*, 164(2), 567-
578.
- 16. Kennedy BP, Folt CL, Blum JD, Chamberlain CP (1997) Natural isotope markers in salmon.
*Nature* 387:766-767

17. Hoppe, K. A., Koch, P. L., Carlson, R. W., & Webb, S. D. (1999). Tracking mammoths
and mastodons: reconstruction of migratory behavior using strontium isotope ratios.
*Geology*, 27(5), 439-442.
- 18. Sjögren, K. G., & Price, T. D. (2013). A complex Neolithic economy: isotope evidence
for the circulation of cattle and sheep in the TRB of western Sweden. *Journal of*
*Archaeological Science*, 40(1), 690-704.
- 19. Feranec RS, Hadly EA, Payton A. (2007). Determining landscape use of Holocene
mammals using strontium isotopes. *Oecologia* 153: 943-950.
- 20. Richards, M., Harvati, K., Grimes, V., Smith, C., Smith, T., Hublin, J. J., ... & Panagopoulou, E.
(2008). Strontium isotope evidence of Neanderthal mobility at the site of Lakonis, Greece using
laser-ablation PIMMS. *Journal of Archaeological Science*, 35(5), 1251-1256.
- 21. Sillen A, Hall G, Richardson S, Armstrong R. (1998). 87Sr/86Sr ratios in modern and fossil food-
webs of the Sterkfontein Valley: Implications for early hominid habitat preference. *Geochimica et*
*Cosmochimica Acta* 62: 2463-2478.
- 22. Muller MN, Wrangham RW, Pilbeam DR (Editors). (2017). *Chimpanzees and Human*
*Evolution*. Cambridge, MA: Harvard University Press.
- 23. Westerhof, A. B., Härmä, P., Isabirye, E., Katto, E., Koistinen, T., Kuosmanen, E., ... & Mänttari,
I. (2014). Geology and geodynamic development of Uganda with explanation of the 1: 1,000,000
scale geological map.
- 24. Hodell DA, Quinn RL, Brenner M, Kamenov G. (2004). Spatial variation of strontium isotopes in
the Maya region: A tool for tracking ancient human migration. *Journal of Archeological Science*
31: 585-601.
- 25. Bowman, J., Jaeger, J. A., & Fahrig, L. (2002). Dispersal distance of mammals is proportional to
home range size. *Ecology*, 83(7), 2049-2055.
- 26. Manning, C. D., & Schütze, H. (1999). *Foundations of statistical natural language processing*
(Vol. 999). Cambridge: MIT press.
- 27. Smith, BH., Crummett, T. L., & Brandt, K. L. (1994). Ages of eruption of primate teeth: a
compendium for aging individuals and comparing life histories. *American Journal of Physical*
*Anthropology*, 37(S19), 177-231.

**Figure Captions**

**Figure 1:** (A) Geologic map of Kibale National Park with locations and strontium isotope ratios of
collected plant leaf samples. The boundaries of the park are outlined in black; yellow shaded areas
represent the Buganda Formation (middle Paleoproterozoic quartzite); brown shaded areas represent the

846 Toro Formation (early Paleoproterozoic gneiss); blue shaded areas represent an ‘unidentified radiometric
anomaly’²⁴. Map reproduced with the permission of the OneGeology. All rights reserved. (B) Boxplot of
strontium isotope ratios from plant samples by the geologic formation on which they were collected. (C)
Plant ⁸⁷Sr/⁸⁶Sr in Kibale National Park by isotopic cluster, as identified through hierarchical cluster
analysis; (DB) Strontium isotope ratios of plant samples by isotopic clusters – “Northern” (red): 0.7078 -
0.712, mean = 0.7094 +/- 0.0010; “Southern” (green): 0.7129 - 0.7219, mean = 0.7169 +/- 0.0026;
“Ngogo” cluster (blue) 0.7234 - 0.7339, mean = 0.7273 +/- 0.0043. Clusters were generated using a
dendrogram with average linkage distance to group similar data points together and mapped using the
‘ggmap’ package in R Studio (version 0.99.902).
**Figure 2:** ⁸⁷Sr/⁸⁶Sr of tooth enamel: (A) Ngogo fauna; (B) Kanyawara fauna. Colored bands show
minimum and maximum values of local plants (0.7129 - 0.7339 for Ngogo, 0.7062 - 0.7201 for
Kanyawara). Other study areas (Kanyanchu, Sebitole) not included due to small sample sizes.
**Figure 3:** Offsets between tooth and bone strontium isotope ratios (A) and tooth and local environment
strontium isotope ratios (B) for individuals of each species of Kibale primate by sex. Offset values are
shown as absolute values in order to compare the magnitude of each offset.
**Figure 4:** Boxplots illustrating bootstrapped cComparison of ⁸⁷Sr/⁸⁶Sr offsets between-in (A)-enamel and
bone (A), and (B)-enamel and environment (B) between the expected dispersing and philopatric sex for
strontium isotope ratios within individuals of each species of Kibale primate-by sex. Bars show median
offset for each sex; diamonds show mean offset. * indicates a significant difference (p<0.05) between the
sexes in greater than 90% of bootstrap simulations. Significant results (p < 0.05) in (A) 92% (black and
white colobus), 5% (chimpanzees), 100% (olive baboons), 99% (red colobus monkeys), and 100%
(guenons) of 10,000 bootstrapped simulations of difference in mean for enamel-bone offsets, and (B) 39%
(black and white colobus), 98% (chimpanzees), 100% (olive baboons), 5% (red colobus monkeys), and
96% (guenons) of 10,000 bootstrapped simulations of difference in mean for enamel-environment offsets.

Table 1 – Kibale Primate Ecological Parameters and Sample Sizes				
Species (Scientific Name)	Species (Common Name)	Home Range	Dispersing Sex	Sample Size
Colobus guereza	Black and White colobus monkey	0.12 – 0.28 km ² ⁽⁷³⁾	Males	6
Papio anubis	Olive baboon	~5 km ² ⁽⁷⁴⁾	Males	9
Cercopithecus ascanius	Redtail monkey (guenon)	0.28 – 0.68 km ² ⁽⁷⁵⁾	Males	5
Cercopithecus mitis	Blue monkey (guenon)	0.5 – 3.3 km ² ⁽⁷⁶⁾	Males	2
Pan troglodytes schweinfurthii	Common chimpanzee	28-41 km ² ^(71,72)	Females	17
Procolobus badius	Red colobus	~ 0.65 km ² ⁽⁷⁵⁾	Females	18

Are these home ranges specifically for primates at Kibale? The authors had previously had much larger ranges for chimpanzees (all smaller than 28 km²). The table caption is too vague to interpret this.

Still not 100% clear. Expected is based on the literature while assigned is based on strontium.

Table 2 – Comparison of expected and assigned “local/philopatric” and “non-local/dispersing” attributions, based on $^{87}\text{Sr}/^{86}\text{Sr}_{\text{enamel}}$ falling outside/within the local isotopic minima and maxima (“standard archeological method”)
 And the local maxima and minima are based on... geology?

Species	# of samples	Expected # of locals / philopatric sex ¹	% of expected locals assigned ‘local’ attribution ²	Expected # of non-locals / dispersing sex ¹	% of non-locals assigned ‘non-local’ attribution ³	% of total samples assigned to their expected attribution (local / non-local)	Dispersing sex correctly identified for this species? ⁴
Chimpanzees	17	11	82%	6	50%	71%	Yes
Red colobus monkey	18	8	100%	10	20%	56%	Yes
Black and white colobus monkeys	6	2	0%	4	50%	33%	No
Olive baboons	9	4	100%	5	40%	67%	Yes
Guenons	7	4	75%	3	0%	43%	No
TOTAL	57	29	83%	28	32%	60%	60% of species

¹ The expected number of philopatric individuals and dispersing individuals for each species is based on the sex of each individual and the overall pattern for sex-biased dispersal within that species

² Any individual falling within the local environment isotopic minima/maxima was assigned a local/philopatric attribution

³ Any individual falling outside of local environment isotopic minima/maxima was assigned a non-local/dispersing attribution

⁴ The dispersing sex was considered “correctly identified” if a greater proportion of the dispersing sex fell outside of the local isotopic minima and maxima compared to the proportion of the philopatric sex falling outside the local isotopic minima and maxima

Table 3 - Proportion of bootstrap simulations in which there was a significant difference ($p < 0.05$, Student's t-test) in offset values ($^{87}\text{Sr}/^{86}\text{Sr}_{\text{enamel-bone}}$ and $^{87}\text{Sr}/^{86}\text{Sr}_{\text{enamel-environment}}$) between males and females

Species	$^{87}\text{Sr}/^{86}\text{Sr}_{\text{enamel-bone}}$	$^{87}\text{Sr}/^{86}\text{Sr}_{\text{enamel-environment}}$
Chimpanzees	5%	98%
Red colobus monkey	99%	100%
Black and white colobus monkeys	92%	39%
Olive baboons	100%	100%
Guenons	100%	96%

"significant difference" is only partially informative. Was the mean for the dispersing sex always higher?

Table 4 - Comparison of expected and assigned “local/philopatric” vs “non-local/dispersing” attribution to individuals based on categorical classification (high/low) of $^{87}\text{Sr}/^{86}\text{Sr}_{\text{enamel-bone}}$ offsets

Species	# of samples	Expected # of locals / philopatric sex ¹	% of expected locals assigned ‘local’ attribution ²	Expected # of non-locals / dispersing sex ¹	% of non-locals assigned ‘non-local’ attribution ³	% of total samples assigned to their expected attribution (local / non-local)	Dispersing sex correctly identified for this species? ⁴
Chimpanzees	17	11	72%	6	33%	59%	Inconclusive
Red colobus monkey	18	8	88%	10	40%	61%	Yes
Black and white colobus monkeys	6	2	50%	4	50%	50%	Inconclusive
Olive baboons	9	4	100%	5	40%	67%	Yes
Guenons	7	4	100%	3	100%	100%	Yes
TOTAL	57	29	86%	28	46%	67%	60% of species

¹ The expected number of philopatric individuals and dispersing individuals for each species is based on the sex of each individual and the overall pattern for sex-biased dispersal within that species

² Any individual with an offset at or below the mean offset for the species (“low offset”) is assigned a local/philopatric attribution

³ Any individual with an offset above the mean offset for the species (“high offset”) is assigned a non-local/dispersing sex attribution

⁴ The dispersing sex was considered “correctly identified” if a greater proportion of the dispersing sex had high offsets compared to the proportion of the philopatric sex with high offsets

Table 5 - Comparison of expected and assigned “local/philopatric” vs “non-local/dispersing” attribution to individuals based on categorical classification (high/low) of $^{87}\text{Sr}/^{86}\text{Sr}_{\text{enamel-environment}}$ offsets

Species	# of samples	Expected # of locals / philopatric sex ¹	% of expected locals assigned ‘local’ attribution ²	Expected # of non-locals / dispersing sex ¹	% of non-locals assigned ‘non-local’ attribution ³	% of total samples assigned to their expected attribution (local / non-local)	Dispersing sex correctly identified for this species? ⁴
Chimpanzees	17	11	82%	6	50%	71%	Yes
Red colobus monkey	18	8	75%	10	30%	50%	Yes
Black and white colobus monkeys	6	2	50%	4	75%	67%	Yes
Olive baboons	9	4	75%	5	80%	78%	Yes
Guenons	7	4	100%	3	33%	71%	Yes
TOTAL	57	29	79%	28	50%	65%	100% of species

¹ The expected number of philopatric individuals and dispersing individuals for each species is based on the sex of each individual and the overall pattern for sex-biased dispersal within that species

² Any individual with an offset at or below the mean offset for the species (“low offset”) is assigned a local/philopatric attribution

³ Any individual with an offset above the mean offset for the species (“high offset”) is assigned a non-local/dispersing sex attribution

⁴ The dispersing sex was considered “correctly identified” if a greater proportion of the dispersing sex had high offsets compared to the proportion of the philopatric sex with high offsets

Table 6 – Summary Table: Were methods were able to correctly identify the dispersing sex for each primate species?

	Standard Archeological Method (Local / Non-local)		Offset Methods			
			$^{87}\text{Sr}/^{86}\text{Sr}_{\text{enamel-bone}}$ Offset		$^{87}\text{Sr}/^{86}\text{Sr}_{\text{enamel-environment}}$ Offset	
	> 90% of individuals assigned expected attribution ¹ ?	Categorical classification effective? ²	Bootstrapped comparison of means effective? ³	Categorical classification effective? ²	Bootstrapped comparison of means effective? ³	Categorical classification correctly effective? ²
Chimpanzees	No	Yes	No	Yes	Yes	Yes
Red colobus monkeys	No	Yes	Yes	Yes	No	Yes
Black and white colobus monkeys	No	No	Yes	No	No	Yes
Olive baboons	No	Yes	Yes	Yes	Yes	Yes
Guenons	No	No	Yes	Yes	Yes	Yes

¹ The expected attribution (local or non-local) is based on the sex of the individual and the known sex-bias in dispersal patterns for that species.

² The categorical classification method was designated as ‘effective’ if a higher proportion of dispersing-sex individuals fell outside the local isotopic minima and maxima (for the standard archeological method) or were classified as high-offset (for the offset methods) compared to the proportion of philopatric-sex individuals.

³ The bootstrap comparison of means was designated as ‘effective’ if over 90% of simulations returned a significant p-value ($p < 0.05$) when comparing offset values between males and females, with the dispersing sex having a significantly higher offset than the philopatric sex.

Table 1 – Kibale Primate Ecological Parameters and Sample Sizes and $^{87}\text{Sr}/^{86}\text{Sr}$ ratios

Species (Scientific Name)	Species (Common Name)	Home Range	Dispersing Sex	Sample Size
Pan troglodytes	Common chimpanzee	5-27 km ² (¹)	Females	17
Colobus guereza	Black and White colobus monkey	0.12 – 0.28 km ² (⁷³ ₂)	Males	6
Papio anubis	Olive baboon	~5 km ² (⁷⁴ ₃)	Males	9
Procolobus badius	Red colobus	~0.65 km ² (⁴)	Females	18
Cercopithecus ascanius	Redtail monkey (guenon)	0.28 – 0.68 km ² (⁷⁵ ₄)	Males	5
Cercopithecus mitis	Blue monkey (guenon)	0.5 – 3.3 km ² (⁷⁶ ₅)	Males	2
(1) Chapman, C. A., & Wrangham, R. W. (1993). Range use of the forest chimpanzees of Kibale: implications for the understanding of chimpanzee social organization. American Journal of Primatology, 31(4), 263-273. (2) Chapman, C. A., & Pavelka, M. S. (2005). Group size in folivorous primates: ecological constraints and the possible influence of social factors. Primates, 46(1), 1-9. (3) Rowell, T. E. (1966). Forest living baboons in Uganda. Journal of Zoology, 149(3), 344-364. (4) Struhsaker, T. T., & Leland, L. (1979). Socioecology of five sympatric monkey species in the Kibale Forest, Uganda. Advances in the Study of Behavior, 9, 159-228. (5) Butynski, T. M. (1990). Comparative Ecology of Blue Monkeys (Cercopithecus mitis) in High and Low-Density Subpopulations. Ecological Monographs, 60(1), 1-26.				
Pan troglodytes schweinfurthii	Common chimpanzee	28-41 km ² (⁷¹ , ⁷²)	Females	17
Procolobus badius	Red colobus	~ 0.65 km ² (⁷⁵)	Females	18

So... 5-27 was totally wrong previously and the actual home ranges are WAY larger? I'm guessing this is because they are now citing Kibale-specific data.

Table 2 — Comparison of expected and assigned “local/philopatric” and “non-local/dispersing” attributions, based on $^{87}\text{Sr}/^{86}\text{Sr}_{\text{enamel}}$. Accuracy of attribution of local/philopatric vs non-local/dispersing status to individuals falling within/outside/within the local isotopic minima and maxima (“common standard archeological method”)—accuracy assessment)
 — Falling outside of local environment minima or maxima = non-local/dispersing attribution
 — Falling outside of local environment minima or maxima = local/philopatric attribution
 — Dispersing sex was correctly identified if a greater proportion of individuals of that sex fell within the local minima and maxima compared to individuals of the philopatric sex

Species	# of samples	True Expected # of locals / philopatric sex ¹	% of expected locals correctly identified as signed ‘local’ attribution ²	True Expected # of non-locals / dispersing sex ¹	% of non-locals assigned ‘non-local’ attribution ³ correctly identified	% of total samples correctly categorized assigned to their expected attribution (local / non-local)	Dispersing sex correctly identified for this species? ⁴
Chimpanzees	17	11	82%	6	50%	71%	Yes
Red colobus monkey	18	8	100%	10	20%	56%	Yes
Black and white colobus monkeys	6	2	0%	4	50%	33%	No
Olive baboons	9	4	100%	5	40%	67%	Yes
Guenons	7	4	75%	3	0%	43%	No
TOTAL	57	29	83%	28	32%	60%	60% of species

¹ The expected number of philopatric individuals and dispersing individuals for each species is based on the sex of each individual and the overall pattern for sex-biased dispersal within that species

² Any individual falling within the local environment isotopic minima/maxima was assigned a local/philopatric attribution

³ Any individual falling outside of local environment isotopic minima/maxima was assigned a non-local/dispersing attribution

⁴ The dispersing sex was considered “correctly identified” if a greater proportion of the dispersing sex fell outside of the local isotopic minima and maxima compared to the proportion of the philopatric sex falling outside the local isotopic minima and maxima See Table 3 for details on assessing accurate reconstruction of species-level sex-bias in dispersal patterns

Table 3—Comparison of species-level sex-bias reconstruction efficacy. For the standard archeological method, percentages indicate the proportion of individuals of each sex that fall outside of the “local” isotopic boundaries. For the offset methods ($^{87}\text{Sr}/^{86}\text{Sr}_{\text{enamel-bone}}$ and $^{87}\text{Sr}/^{86}\text{Sr}_{\text{enamel-environment}}$), percentages indicate the proportion of each sex that falls into the “high offset” category (falling above the mean offset for that species).

Taxa		Chimpanzee (N=17)	Red colobus (N=18)	Black and White Colobus (N=6)	Olive Baboon (N=9)	Guenon (N=7)
Proportions of individuals falling outside the “local” isotopic boundaries						
Standard archeological method	Philopatric sex	18%	0%	100%	0%	25%
	Dispersing sex	50%	25%	50%	40%	0%
Proportions of individuals falling into the “high offset” category						
$^{87}\text{Sr}/^{86}\text{Sr}_{\text{enamel-bone}}$ Offset	Philopatric sex	27%	10%	50%	0%	0%
	Dispersing sex	33%	40%	50%	40%	100%
$^{87}\text{Sr}/^{86}\text{Sr}_{\text{enamel-environment}}$ Offset	Philopatric sex	18%	25%	40%	25%	0%
	Dispersing sex	50%	30%	100%	80%	33%

Table 34 - Proportion of significant ($p < 0.05$) bootstrap simulations in which there was a significant difference ($p < 0.05$, Student’s t-test) in offset values ($^{87}\text{Sr}/^{86}\text{Sr}_{\text{enamel-bone}}$ and $^{87}\text{Sr}/^{86}\text{Sr}_{\text{enamel-environment}}$) between males and females for $^{87}\text{Sr}/^{86}\text{Sr}_{\text{enamel-bone}}$ and $^{87}\text{Sr}/^{86}\text{Sr}_{\text{enamel-environment}}$ offset methods

Species	$^{87}\text{Sr}/^{86}\text{Sr}_{\text{enamel-bone}}$	$^{87}\text{Sr}/^{86}\text{Sr}_{\text{enamel-environment}}$
Chimpanzees	5%	98%
Red colobus monkey	99%	100%
Black and white colobus monkeys	92%	39%
Olive baboons	100%	100%
Guenons	100%	96%

Table 45 - Accuracy Comparison of expected and attribution-assigned “of local/philopatric” vs “non-local/dispersing” status attribution to individuals using based on categorical classification (high/low) of $^{87}\text{Sr}/^{86}\text{Sr}$ enamel-bone offsets

— High offset (above species mean) = non-local/dispersing attribution
 — Low offset (below species mean) = local/philopatric attribution
 - Dispersing sex was correctly identified if a greater proportion of individuals of that sex fell into the high offset category compared to individuals of the philopatric sex

Species	# of samples	Expected # of locals / philopatric sex¹	% of expected locals assigned ‘local’ attribution²	Expected # of non-locals / dispersing sex¹	% of non-locals assigned ‘non-local’ attribution³	% of total samples assigned to their expected attribution (local / non-local)	Dispersing sex correctly identified for this species?⁴
Species	# of samples	True # of locals / philopatric sex	% locals correctly identified	True # of non-locals / dispersing sex	% non-locals correctly identified	% of total samples correctly categorized	Dispersing sex correctly identified?⁴
Chimpanzees	17	11	72%	6	33%	59%	Inconclusive
Red colobus monkey	18	8	88%	10	40%	61%	Yes
Black and white colobus monkeys	6	2	50%	4	50%	50%	Inconclusive
Olive baboons	9	4	100%	5	40%	67%	Yes
Guenons	7	4	100%	3	100%	100%	Yes
TOTAL	57	29	86%	28	46%	67%	60% of species

¹ The expected number of philopatric individuals and dispersing individuals for each species is based on the sex of each individual and the overall pattern for sex-biased dispersal within that species

² Any individual with an offset at or below the mean offset for the species (“low offset”) is assigned a local/philopatric attribution

³ Any individual with an offset above the mean offset for the species (“high offset”) is assigned a non-local/dispersing sex attribution

⁴ The dispersing sex was considered “correctly identified” if a greater proportion of the dispersing sex had high offsets compared to the proportion of the philopatric sex with high offsets

[†] See Table 3 for details on assessing accurate reconstruction of species-level sex-bias in dispersal patterns

|

Table 5 - Comparison of expected and assigned “local/philopatric” vs “non-local/dispersing” attribution to individuals based on categorical classification (high/low) of $^{87}\text{Sr}/^{86}\text{Sr}$ enamel-environment offsets

Table 6 – Accuracy of attribution of local/philopatric vs non-local/dispersing status to individuals using $^{87}\text{Sr}/^{86}\text{Sr}$ enamel-environment offsets

High offset (above species mean) = non-local/dispersing attribution

Low offset (below species mean) = local/philopatric attribution

Dispersing sex was correctly identified if a greater proportion of individuals of that sex fell into the high offset category compared to individuals of the philopatric sex

Species	# of samples	Expected # of locals / philopatric sex ¹	% of expected locals assigned ‘local’ attribution ²	Expected # of non-locals / dispersing sex ¹	% of non-locals assigned ‘non-local’ attribution ³	% of total samples assigned to their expected attribution (local / non-local) ⁴	Dispersing sex correctly identified for this species? ⁴
Chimpanzees	17	11	82%	6	50%	71%	Yes
Red colobus monkey	18	8	75%	10	30%	50%	Yes
Black and white colobus monkeys	6	2	50%	4	75%	67%	Yes
Olive baboons	9	4	75%	5	80%	78%	Yes
Guenons	7	4	100%	3	33%	71%	Yes
TOTAL	57	29	79%	28	50%	65%	100% of species

¹ The expected number of philopatric individuals and dispersing individuals for each species is based on the sex of each individual and the overall pattern for sex-biased dispersal within that species

² Any individual with an offset at or below the mean offset for the species (“low offset”) is assigned a local/philopatric attribution

³ Any individual with an offset above the mean offset for the species (“high offset”) is assigned a non-local/dispersing sex attribution

⁴ The dispersing sex was considered “correctly identified” if a greater proportion of the dispersing sex had high offsets compared to the proportion of the philopatric sex with high offsets

[†] See Table 3 for details on assessing accurate reconstruction of species-level sex-bias in dispersal patterns

Table 67 – Summary of Method Efficacy Table: Were methods able to correctly identify the dispersing sex for each primate species?

	Standard Archeological Method (Local / Non-local)		Offset Methods			
			$^{87}\text{Sr}/^{86}\text{Sr}_{\text{enamel-bone}}$ Offset		$^{87}\text{Sr}/^{86}\text{Sr}_{\text{enamel-environment}}$ Offset	
	> 90% of individuals correctly classified as assigned expected attribution ¹ ?	Categorical classification effective? ²⁺	Bootstrapped comparison of means effective? ^{3,2}	Categorical classification effective? ²⁺	Bootstrapped comparison of means effective? ^{3,2}	Categorical classification correctly effective? ²⁺
Chimpanzees	No	Yes	No	Yes	Yes	Yes
Red colobus monkeys	No	Yes	Yes	Yes	No	Yes
Black and white colobus monkeys	No	No	Yes	No	No	Yes
Olive baboons	No	Yes	Yes	Yes	Yes	Yes
Guenons	No	No	Yes	Yes	Yes	Yes

Summary table of methodological effectiveness:

¹ The expected attribution (local or non-local) is based on the sex of the individual and the known sex-bias in dispersal patterns for that species.

²⁺ The categorical classification method was designated as ‘effective’ if a higher proportion of dispersing-sex individuals fell outside the local isotopic minima and maxima (for the standard archeological method) or were classified as high-offset (for the offset methods) or fell outside the local minima and maxima (for the standard archeological method) compared to the proportion of philopatric-sex individuals, thereby correctly identifying the dispersing sex as such.

^{3,2} The bootstrap comparison of means was designated as ‘effective’ if over 90% of simulations returned a significant p-value ($p < 0.05$) when comparing offset values between males and females, with the dispersing sex having a significantly higher offset than the philopatric sex.

Figure 1 - Botanical sample $^{87}\text{Sr}/^{86}\text{Sr}$ relative to underlying geologic formations (A, B) and statistically identified isotopic clusters (C, D) in Kibale National Park

(A) Geologic map of Kibale National Park with locations and strontium isotope ratios of collected plant leaf samples. The boundaries of the park are outlined in black; yellow shaded areas represent the Buganda Formation (middle Paleoproterozoic quartzite); brown shaded areas represent the Toro Formation (early Paleoproterozoic gneiss); blue shaded areas represent an 'unidentified radiometric anomaly'²⁴. Map reproduced with the permission of the OneGeology. All rights reserved. (B) Boxplot of strontium isotope ratios from plant samples by the geologic formation on which they were collected. (C) Plant $^{87}\text{Sr}/^{86}\text{Sr}$ in Kibale National Park by isotopic cluster, as identified through hierarchical cluster analysis; (D) Strontium isotope ratios of plant samples by isotopic clusters - "Northern" (red): 0.7078 - 0.712, mean = 0.7094 +/- 0.0010; "Southern" (green): 0.7129 - 0.7219, mean = 0.7169 +/- 0.0026; "Ngogo" cluster (blue) 0.7234 - 0.7339, mean = 0.7273 +/- 0.0043. Clusters were generated using a dendrogram with average linkage distance to group similar data points together and mapped using the 'ggmap' package in R Studio (version 0.99.902).

1083x812mm (72 x 72 DPI)

Figure 2 - Strontium Isotope Ratios in Ngogo (A) and Kanyawara (B) Primates

wording. Not just "fauna". Primates. Please use complete sentences in figure captions.

$^{87}\text{Sr}/^{86}\text{Sr}$ of tooth enamel: (A) Ngogo fauna; (B) Kanyawara fauna. Colored bands show minimum and maximum values of local plants (0.7129 - 0.7339 for Ngogo, 0.7062 - 0.7201 for Kanyawara). Other study areas (Kanyanchu, Sebitole) not included due to small sample sizes.

1083x812mm (72 x 72 DPI)

Figure 3 - Offsets between tooth-bone (A) and tooth-local environment (B) strontium isotope ratios: Kibale National Park primates

Would love to see offsets for primates from the different localities (to compare to the "standard" method in Fig. 2).

Offsets between tooth and bone strontium isotope ratios (A) and tooth and local environment strontium isotope ratios (B) for individuals of each species of primate by sex. Offset values are shown as absolute values in order to compare the magnitude of each offset.

1083x812mm (72 x 72 DPI)

Figure 4: Bootstrapped simulations comparing $^{87}\text{Sr}/^{86}\text{Sr}_{\text{enamel-bone}}$ (A) and $^{87}\text{Sr}/^{86}\text{Sr}_{\text{enamel-environment}}$ (B) strontium isotope ratio offsets between sexes in Kibale National Park primates

Boxplots illustrating bootstrapped comparison of $^{87}\text{Sr}/^{86}\text{Sr}$ offsets in enamel and bone (A), and enamel and environment (B) between the expected dispersing and philopatric sex for each species of primate. Bars show median offset for each sex; diamonds show mean offset. * indicates a significant difference ($p < 0.05$) between the sexes in greater than 90% of bootstrap simulations.

1083x812mm (72 x 72 DPI)

Appendix D

Dear editors,

We are very grateful for the extensive and detail-oriented review provided for this version of our manuscript. We have addressed each point raised by the reviewer in the letter below and in the Tracked Changes copy of our manuscript. The reviewer's comments appear in italics while our responses are in bold. The line numbers referenced below are in relation to the Clean Copy version of the manuscript uploaded as part of our resubmission. We thank both you and the reviewer for their time and hope that the present version of our manuscript addresses all concerns.

Response to Reviewer 1:

I do not appear to have access to a clean copy of the manuscript so I am going to limit referring to specific line items in my text below. I have marked up the submitted .pdf and would ask both the editor and authors to please review my comments, questions, and highlighted points there.

We are sorry to hear that the reviewer did not get access to our clean copy and we appreciate them working around with the tracked-changes version. We have done our best to follow their comments within the PDF document. Smaller adjustments to wording and phrasing are deeply appreciated and changes have been made accordingly, although we have not listed all minor recommendations from the marked-up PDF individually within this response letter.

In addition to the comments included in the letter below, in the marked-up PDF the reviewer asks why there might be spatially patterned isotopic values in plants if not for geology. We have offered numerous potential explanations for the isotopic heterogeneity seen in Kibale in the Discussion section, many with great thanks to the suggestions from the reviewer in previous rounds. While this is an excellent question and one that we are happy to provide hypotheses around, addressing it empirically is beyond the scope of this paper. Our objective here was to show that, even in area without detailed geologic maps available or in areas with neighboring geologic areas that isotopically overlap, it is possible to quantify spatially patterned isotopic variation in ways that are meaningful to address questions of animal mobility. A detailed investigation into the geology of the Kibale National Park area would be an important contribution to the literature but is not the goal of the present manuscript.

The manuscript has improved. Specifically, the discussion of tooth selection and the addition of geologic boundaries and plant data from different geologies in Figure 1, both definitely help. The additional text about what else might affect results in the Discussion is also useful, although its organization is hard to follow (I revisit this point below). I also think that placing more emphasis on how the offsets help “assess like the likelihood that an individual is local or non-local based on how it compares to other members of the same species (or community) rather than relying on those maxima and minima designated by the local baselines” is good.

Despite these improvements, there are still several fundamental points that I think the authors need to address before this is publishable.

Most critically, the authors still haven't demonstrated that their “isotopic clusters” actually perform better than geologies would. The authors' continued decision to discount geology without actually

investigating if using plants from different geologies to define “local environment” (and in turn calculate offsets) actually performs worse than their “statistical clusters” continues to be a critical shortcoming of the paper. Trying to argue that there is a fundamental flaw in using geology without actually showing it doesn’t work is problematic. This is a very bold claim that could have major implications for other researchers. The authors need to actually incorporate geology into their paper and compare it with their new method if they want to make the claim that geology doesn’t work. Specifically, I think the authors should both: (1) show how the “standard archaeological method” might differ if using geologies vs. isotopic clusters, and (2) calculate offsets for plants on different geologies and compare this to their offsets using “statistical clusters”

We have performed the direct comparisons requested by the reviewer and explained these additional Methods on lines 182-186 and 194-198. For the standard archeological method, there were no differences in the outcome of the analyses using geologic vs statistical cluster boundaries, with the exception of one female chimpanzee who was “correctly” categorized as a non-local/disperser when using the geologic boundaries and “incorrectly” categorized when using the clusters. This is explained on lines 253-257. For the tooth-to-environment offset, using geologic boundaries performed markedly worse than the statistically identified clusters. We have included a new sub-section in the Results section detailing the outcome of our tooth-environment offset comparisons when using the geologic formations instead of the statistical clusters found on lines 296-308 with follow-up discussion on lines 357-374 in the Discussion.

It is also still not possible to compare the “standard archaeological” method with the authors’ new offset methods in the current version of the manuscript because:

1. The authors don’t provide comparable datasets. Primates from Kanywara and Ngogo are compared to local min and maxima separately for the standard archaeological method, but as far as I can tell, data from all regions are combined in offset calculations, tables, and Figs 3 and 4 (i.e. we never see region-specific offsets). The authors need to please discuss how offset calculations and efficacy of distinguishing expected philopatric and dispersing individuals varies across the park. is it all good at all sites or are some sites at Kibale better than others, and if so why). I would expect this could be affected by isotopic heterogeneity (e.g., less useful in places that are more isotopically homogenous)... . The authors present primate vs. plant data in Figure 2 but then combine everyone in the offset calculations. I want to see how offsets might vary with location as well. Then one could more directly compare the different methods.

We do not combine all regions together to calculate the offsets; every offset calculated is by definition region-specific. The offset calculation for each primate is calculated as the isotopic difference between its tooth enamel and the mean of the isotopic cluster on which it lived, which varies from park location to park location (explained on lines 192-194 and 276-283; the environmental mean used for the calculation for each primate is listed in Supplemental Table 1). We have added more to lines 192-194 to ensure that this point is clear. Within the text of the manuscript, we examine the proportion of matching assigned/expected attributions for Kibale primates as a whole for all methods (lines 244-246, 266-268, 291-292). Because the standard archeological method requires direct comparisons between tooth enamel and environmental data, it is necessary to show them on separate plots (as in Figure 2) whereas the offset calculations integrate the different baseline environmental isotopic conditions into the calculation itself, making it possible to illustrate the data together (Figure 3). These differences in illustration do not represent a difference in combining versus separating regions differently between the methods.

We do agree that a breakdown of matching/non-matching attributes by site is a useful metric to share and have included this for the standard archeological method (line 246) and each offset method (lines 268 and 291). The differences between park regions is not particularly pronounced for the offset methods but is more substantial for the standard archeological method. This is likely due to the use of the isotopic clusters in the offset method analyses, as it does a more faithful job reflecting the isotopic variability relevant to that particular region rather than the geologic groupings used in the standard archeological approach (in our accounting, most likely due to inaccurate or insufficiently detailed maps of the area).

2. *It is not clear what “environment” baseline the authors used for their standard archaeological method. They write “To test the efficacy of the standard archeological approach, we defined minimum and maximum “local” $^{87}\text{Sr}/^{86}\text{Sr}$ thresholds based on plant leaves collected from the Kanyawara and Ngogo study areas and compared the $^{87}\text{Sr}/^{86}\text{Sr}$ of fauna collected from each area to this range” (lines 215-217 of the marked-up text). What are these groups? Do they correspond to either geology or isotopic clusters? How is this “standard”?*

The minimum and maximum values for each park area are based on the plant samples collected from the geologic formation on which that park area sits. We have reworded to make this more clear in the main text (lines 172-174, 239-244) and in the caption for Figure 2.

3. *As I note below, the authors combine plant clusters for their offset calculations for Kanyanchu primates.*

This is not accurate. As explained on lines 279-280, “Kanyanchu individuals [are compared] to the Southern cluster $^{87}\text{Sr}/^{86}\text{Sr}$ mean (0.7169 ± 0.0026).” We do combine the values of the Southern and Ngogo cluster for comparison to the Ngogo primates and assume this is what the reviewer is referencing. As illustrated in Figure 1C and explained on lines 220-222 of the previous submission, this is because the Ngogo park area is located on top of both clusters. It sits on the boundary between the statistically identified “Ngogo cluster” (in blue, Figure 1) and the statistically identified “Southern” cluster (in green, Figure 1). Because it is positioned on this boundary line, it would be arbitrary to assume that Ngogo primates live exclusively on one or the other. We therefore combine the data from the clusters in order to calculate offsets for the Ngogo primates. If other park areas also fell along cluster boundaries, we would have followed similar protocols there; however, other park areas fall exclusively within single isotopic clusters. We have added an additional sentence explaining this on lines 280-283 and for transparency and accuracy.

4. As mentioned above, the authors still don’t actually investigate geology in their paper. They need to go the necessary extra step of comparing primates to the geologies.

We have added in direct comparisons to the analysis using only the geology, as referenced earlier in this letter.

Other related major points:

1. *The addition of geology to Figure 1 is helpful. Thank you. This map allowed me to note several things for the first time, which I will outline below. I would like to point out that many of the details in Figure 1 are so small that they are barely legible. This includes the axes and the datapoints themselves. I would also encourage the authors to make the two maps similar sizes or at least include a 3-km scale bar on*

the geologic map.

2. According to Figure 1, the text regarding the park's geology on lines 114-117 of the marked-up text is incorrect. The authors write "Kanyawara and Sebitole are on the Toro Formation (an undifferentiated early Paleoproterozoic gneiss) while Kanyanchu and Ngogo are on the Buganda formation (a middle Paleoproterozoic quartzite). Ngogo also falls partially on an area designated an 'unidentified radiometric anomaly' by the Ugandan Society for Geology and Mines". According to Figure 1, Kanyawara is on the Buganda Formation (albeit quite close to the boundary with the Toro Formation) and both Ngogo and Kanyanchu are on a radiometric anomaly...

The authors mention in the Methods that there are no geologic data at all where the geologic anomaly is (maybe I am incorrect but I actually feel like I can see geologic boundaries under the anomaly in Figure 1?). If geology is, indeed, unknown under the anomaly, then how can they write that the Ngogo and Kanyanchu are on the Buganda Formation? Moreover, if there are no data where the radiometric anomaly is, how did the Ugandan Society of Geology and Mines determine it exists (and what its approximate boundaries are)? How is the radiometric anomaly defined/ how was it previously discovered? And why don't the authors revisit this previously described anomaly in the Discussion? For example, it would seem to me that the very high $87\text{Sr}/86\text{Sr}$ for plants in the "Ngogo" cluster geographically relate rather closely to the radiometric anomaly...

We thank the reviewer for finding an error in the boundaries drawn around the radiometric anomaly in Figure 1A. We have corrected these boundaries in the revised figures here and adjusted text to match. We have also increased the font size and included a 3-km scale on both maps in the figure.

We do not have any insights into how the Ugandan Society of Geology and Mines identified the radiometric anomaly, nor any specific information on what they meant when they designated it as such (see Westerhof et al 2014 reference for all available information on this area). These are excellent questions that the reviewer raises, and ones that we would love to know more about; however, that information is not, to the best of our knowledge, available at this time. Based on some of the comments on the marked-up PDF, we want to be abundantly clear that we did not identify this radiometric anomaly ourselves, and do not claim to have done so anywhere in the manuscript. This area is marked and identified on the map in the Westerhof et al 2014 reference. We do agree that the very high ratios identified in our statistical clustering (the "Ngogo cluster") fall on the area designated by the USGM as the radiometric anomaly, and that this is probably not a coincidence. However, our very radiometric values fall into a significantly smaller areas than that designated as the anomaly by the USGM. Identifying the source of these very radiometric values would be a fascinating and important extension to this work but falls outside of the scope of this manuscript. The lack of available information about the radiometric anomaly, its geology, its definition, etc are, however, a clear illustration of why our statistical clustering method is preferable in places without abundant or detailed geologic maps.

The geologic boundaries that the Reviewer suspects they may see in Figure 1 are darker colorations caused by the trees in the Kibale forest as seen on Google Maps, not a geologic feature. We tried to make the geologic features evident with the brown, yellow, and blue colorations; we have made the area identified as the anomaly on the USGM map more opaque in our new version in order to avoid any confusion.

We have edited the text for clarity around the geologic context of the park areas (lines 105-111).

I also note that the authors mention there may be an influence of riparian habitat (non-local strontium from rivers?) in the Discussion. Is this relevant at Kibale? I don't know the park. Is there a major river system? If so, it should be labeled in Figure 1. Have the authors considered if plants from riparian habitat differ from the surrounding soil? Could this explain some of the spatial heterogeneity they observe in strontium isotope ratios for plants?

This is an excellent question. We have published on the impact of non-local strontium on the isotopic profiles of riparian habitats in other parks in Uganda (Hamilton et al 2019, AJPA 170(4), 551-564) and performed similar tests to those outlined in the referenced paper in Kibale. There were no significant riparian effects measured along Kibale rivers. We suspect that this lack of riparian signal in the strontium isotope ratios of plants growing along Kibale's rivers is due to the abundance of water in the Kibale system as a whole (in contrast with the Toro-Semliki Wildlife Refuge, for example, which is a gallery forest surrounded by grassland). We have labeled the largest river in the park, the Dura River, on the maps in Figure 1 as suggested.

3. The geology underlying the various "isotopic clusters" the authors define is also not nearly as cut and dry as the authors would lead the reader to believe. Specifically, the text at the beginning of the Results is quite misleading. It suggests that the isotopic clusters align with geology. But I can now see (thanks to this geologic map in Figure 1) that the authors' isotopic clusters don't match geologies like the authors state that they do (i.e. the northern cluster isn't predominantly on the Toro Formation and the southern cluster isn't predominantly on the Buganda Formation). And Kanyanchu samples must also be on mixed geologies as Kanyanchu itself is on the radiometric anomaly. Isn't the whole point of creating these clusters to ignore underlying geology?

The statistical clustering is a tool that we designed because the geologies of the area are not well defined (as in the case of the radiometric anomaly) and because the available geologic boundaries are not able to be differentiated isotopically (see Figure 1B). We agree with the reviewer that the alignment is not at all cut and dry and have amended our wording on lines 234-238 to more clearly reflect that.

4. On a related note, the spatial distribution of the isotopically distinct clusters is not exactly straightforward. For example, there are relatively low $87\text{Sr}/86\text{Sr}$ that would be indicative of the "southern" cluster in northern parts of the park. It would seem to me that the spatial distribution of isotope data actually DOES conform somewhat to geologies. Couldn't this be readily explained if the boundaries of the different formations are perhaps mapped incorrectly?

We agree that while there is spatial patterning to the isotopic values, there absolutely is variability! However, we disagree that it could be explained through a simple error in geologic boundary mapping. For example, there are relatively low values (~ 0.710) located in the far north of the park at Sebitole, which is unquestionably on the Toro formation. Similar values are clustered south of Kanyawara, which is entirely on the Buganda formation. Similarly, intermediary values (~ 0.715) are located in the far south of the park (Toro Formation), northwest from Kanyanchu (Buganda Formation) and around Ngogo (radiometric anomaly). The observation that the reviewer is making is exactly what led us to devise our statistical clustering tool; there is absolutely patterned isotopic variation in the park, which should make tracking animals' movements (our ultimate goal!) absolutely possible in this setting. However, the existing geologic formation boundaries were clearly not reflecting where the isotopic variation was. Whether this is because of the heterogenous nature of

ancient geologies, inaccurately drawn geologic maps, or any combination of causes, the statistical clustering tool allows us to nevertheless quantify, categorize, and use the existing isotopic variation to answer our ecological question.

5. Again on a related note, throughout the text, the authors state that they use local bioavailable $^{87}\text{Sr}/^{86}\text{Sr}$ defined by statistical clustering of botanical samples to assign "local environment", but apparently they combined both southern and Ngogo clusters to estimate "local" for Ngogo individuals? They do not broadcast this point (it is only mentioned once on lines 321-323 in marked up pdf.). Why did the authors do this? This makes very little sense to me given that a main message of the paper is that clusters should be used to define local.... Why didn't Ngogo individuals get compared to the Ngogo cluster? If the rationale is spatial heterogeneity in the clusters, then it would make just as much sense to combine the northern and southern clusters for individuals from Kanywara, but the authors didn't do that... What's the point in defining isotopically discrete clusters of plants if they are then ignored? And if the authors are combining clusters, then can this really be more accurate than geology? Maybe it is, but I have no way of knowing since they still haven't provided any of these comparisons in their paper. Surely all of this needs to be mulled over in the Discussion?

We agree with the Reviewer that this decision could be articulated more clearly and have done so on lines 231-233 and 381-383. We combine the Southern and Ngogo cluster values for the Ngogo primates not because of heterogeneity within the clusters, but because the Ngogo park area sits directly on top of both of these two clusters (explained on lines 381-383, and in response to this point earlier in the letter). If other park areas also sat on boundary lines between clusters, we would combine those as well, but this is not the case. The purpose of identifying these isotopic clusters is to determine where significantly different isotopic zones fall within the park; while it would absolutely be ideal for each park area of interest to fall cleanly on one cluster or another, unfortunately that is not the case here.

Smaller things:

1. Throughout the text, the authors refer to high and low offsets. Given that they are dealing with absolute values, I would think large and small might make more sense?

We agree with this suggestion and have changed the wording throughout accordingly.

2. The language about "expected/anticipated" philopatric or dispersing individuals is still not clear in parts of the Results (e.g., lines 290-291 and 312-314 of the marked-up text).

We have added in additional clarifications around this in the mentioned lines as well as elsewhere throughout the paper.

3. Proofread! The authors state that they are very secure that their paper is now error free but a quick look through the paper shows that there are still issues. In addition to some grammar (numerous places that need commas still), the new text in the discussion has quite a few typos and the caption for Table 6 reads "Were methods were able to correctly identify the dispersing sex for each primate species?"

Thank you - we have removed the extra "were" in the title of Table 6. We have followed APA guidelines as far as comma placement and grammar throughout this article and have employed the services of a technical writer to edit for grammatical issues. Any disagreements over punctuation

placement is most likely a difference of style preference or differences in convention adherence. We are happy to follow guidelines other than APA if it is recommended.

4. There are some analytical details missing in the methods. What concentrations of nitric acid were used, for example?

We have added this information into the Methods section as requested.

5. I still don't understand how the authors define "local isotopically homogenous areas"... Primates could move a lot further than 3 km and still be on the same geology (or isotopically discrete cluster). Is this based on where the primate groups live within the park? Is the idea simply that this is unlikely if we assume primates are equally likely to move in any direction, then there's a very good chance that they would wind up on an isotopically discrete landscape within 3 km? If the authors could please more clearly explain their rationale for 3 km, this would be greatly appreciated. Alternatively, perhaps what matters isn't 3 km per se but rather that there are primates with quite restricted home ranges (<1 km) and those with much larger home ranges (chimps). There are also species with somewhat intermediate ranges (blue monkeys and olive baboons). I think it would be more informative if the authors considered their data in these terms rather than home ranges that are below or above a certain (perhaps arbitrary?) threshold of 3 km.

We agree that this idea can be conveyed more clearly without discussing a specific threshold. We have removed this section and discuss home ranges elsewhere as appropriate.

6. The new text in discussion under the heading "Potential Sources of Model Error" is useful but seems rather disjointed. For example, it is confusing that both dispersal from natal troupe and timing of tooth mineralization are discussed separately from "life history variables"... I also do not understand what the overview of the regional geology has to do with "model limitations". And what do the authors mean when they refer to "models" and "modelling"? Are they referring to their different offset calculations?

We have renamed the section "Potential Sources of Assignment Error" to increase clarity. We also reorganized this section as suggested in the marked PDF, included a brief introduction to the section, and moved the extended discussion about Kibale's geology to its own area of the Discussion section. We have also changed the word "model" and "modeling" to increase clarity. As requested in the marked up PDF, we also note the offset values for individuals with M2's sampled along with species ranges.

7. The. Is the Discussion truly the most appropriate place for all of the details regarding Kibale's geology and factors that can affect the degree to which bioavailable strontium reflects geology? These details provide context for why the authors are arguing that geologic boundaries, per se, may not be particularly useful for spatially partitioning a region. I would think at least some of this needs to be early in the paper so they authors can justify why they are proposing a novel method for understanding dispersal.

We believe that the details of this are best suited for the Discussion but agree that it is important justification for the statistical clustering used in this paper. We previously discuss this on lines 139-146. A few additional sentences have been added for increased clarity and weight.

8. Table captions are still very short and vague. For example, that for Table 2 states: "Comparison of expected and assigned "local/philopatric" and "non-local/dispersing" attributions, based on

87Sr/86Srenamel falling outside/within the local isotopic minima and maxima (“standard archeological method”). The reader is left to guess what local isotopic minima and maxima might be and how these were estimated. Also, expected and assigned attributions aren’t both based on strontium, but that’s how the caption reads. Both table and figure captions should provide enough details to be understandable on their own.

We have added more detail to the table headings.

We thank you and the reviewer again for your time and attention. We deeply appreciate the opportunity to improve the manuscript once again and look forward to hearing back from you.

Sincerely,

Marian Hamilton

Appendix E

Hamilton et al. Using Strontium Isotopes to Determine Philopatry and Dispersal in Primates

In this manuscript, the authors use differences in $^{87}\text{Sr}/^{86}\text{Sr}$ of primate teeth formed before dispersal and compare those to environmental Sr ratios (where the animals died) and to Sr ratios recovered from bone, which presents an average of $^{87}\text{Sr}/^{86}\text{Sr}$ over the last few years of life. The goal here is to assess whether Sr ratios are useful indicators of dispersal patterns. I think this is a useful study that would be of interest to many researchers, and the bioavailable Sr data presented here is also a nice contribution. Overall, I think this manuscript could benefit from deeper discussions of why bioavailable $^{87}\text{Sr}/^{86}\text{Sr}$ may not correspond to local geology, and below I outline several areas for improvement.

Line 45: maybe define dispersal earlier, for those non-primatologists out there.

Line 47-48: not just fossils! I would also add that this is compared to enamel, and a couple words on why (e.g. porosity).

Line 72: cite previous studies outlining this approach (I'm thinking Price et al. 2002 on using ratios within $\pm 2\sigma$ of the mean)

Line 93: delete "impregnated", sub in "-tipped"

Line 96: citation for higher turnover in cancellous vs cortical bone? Does this differ with element?

Line 97-100: I think it would be nice to include an SI table with the ages at which these different teeth start to form crowns/mineralize/erupt in the different species. Also include age at weaning in that chart. Update—I see you now get to this in the discussion, but I still think it's worth including a table with all the known eruption/mineralization data that's out there. That'd be a useful resource for other researchers.

Figure 1: I think this needs to be broken up into several different figures: the maps on one, and the boxplots on the other. I also think the map figure needs an inset to show where in Uganda this is, and also, where Uganda is within the African continent. In my experience many people are unfamiliar with African geography. Right now these figures look very much like a first draft. The fonts in the boxplot y axes are way too small to read, I would make those bigger and consistently use serif or sans serif fonts for the axis labels (I think Arial is probably best, but ultimately that's up to the authors, as long as they're consistent). Also, the latitude/longitude markers are really small and not always visible in the maps. I would also get rid of the white box around the scale bar, and change this to a round number (maybe 5 km?). In Figure 1A the Unidentified radiometric anomaly looks like a lake, so I would change the color! I also wonder if Figure 1C is necessary? The authors include the botanical sample data in both maps, so the only difference is the interpolated landscape, which, I'm not sure how helpful that is. I'd also consider changing the color ramp for the plant $^{87}\text{Sr}/^{86}\text{Sr}$. Red indicates low $^{87}\text{Sr}/^{86}\text{Sr}$...but we usually

think of red as indicating a higher ratio! I'd consider using the color codes in the recent paper on spatial variation in $^{87}\text{Sr}/^{86}\text{Sr}$ in Kenya and Tanzania:

Janzen, A., Bataille, C., Copeland, S.R., Quinn, R.L., Ambrose, S.H., Reed, D., Hamilton, M., Grimes, V., Richards, M.P., le Roux, P. and Roberts, P., 2020. Spatial variation in bioavailable strontium isotope ratios ($^{87}\text{Sr}/^{86}\text{Sr}$) in Kenya and northern Tanzania: Implications for ecology, paleoanthropology, and archaeology. *Palaeogeography, Palaeoclimatology, Palaeoecology*, p.109957.

I would also get rid of the really dark border around the color ramp.

Even though the Janzen et al. 2020 study doesn't cover Uganda, I would cite it as a good example of how bioavailable $^{87}\text{Sr}/^{86}\text{Sr}$ doesn't always agree with underlying geology, due to aeolian or fluvial inputs, for example. Do Janzen et al. have samples from similar geologies as in this study? What were the $^{87}\text{Sr}/^{86}\text{Sr}$ of those? I think reviewing that and setting up some expectations for the author's own data would be useful here (or flesh this out more in the paragraphs starting around line 392 by referencing some of the results from the Janzen et al. paper).

I think it would also be a good idea to discuss some of the geological research that's been done in the area.

Link, K., Koehn, D., Barth, M.G., Tiberindwa, J.V., Barifajjo, E., Aanyu, K. and Foley, S.F., 2010. Continuous cratonic crust between the Congo and Tanzania blocks in western Uganda. *International Journal of Earth Sciences*, 99(7), pp.1559-1573.

Bell, Keith, and J. L. Powell. "Strontium isotopic studies of alkalic rocks: the potassium-rich lavas of the Birunga and Toro—Ankole Regions, East and Central Equatorial Africa." *Journal of Petrology* 10, no. 3 (1969): 536-572.

What's also missing from the map figure is some other clear landmark. I'm having trouble comparing the maps in the manuscript to maps in other papers. Maybe the authors could zoom out a bit in the map? Or this problem might be solved with the addition of an inset showing exactly where this is in Uganda.

Also, is the Dura River indicated by a dotted line? I'd change this to maybe a blue line that's more easily identifiable as a river. This could be easily digitized in ArcGIS from a map.

More comments on the figure. For the boxplots, could the authors indicate in the x axis what the northern southern and Ngogo clusters are? That's missing from the figure. Also, it would be useful to either outline or mark those clusters in a clear way on the map.

The authors mentioned that Ngogo park sits over two clusters. Is it possible to show the outlines of each park on the figure instead of just a point? I think that'll help the readers quite a bit. Right now we have no idea of the extent of those parks.

I also wonder if it's possible to employ an approach like in the Janzen et al. study (which is really explained in the paper by Bataille et al. (2018) for western Europe), in which other geological/atmospheric variables are used to create a predictive Sr map.

Also, maybe the authors could symbolize the clusters by marker shape in the map?

Were plant samples cleaned before analysis? How much dust/dirt could have been on them? I would just mention this to assuage any concerns readers might have about this.

Line 151: what's the expected value of SRM 987? And how many total runs of the standard during the span of analysis?

Figure 2: there are two different captions for this? The lower one is better! I would shift the font sizes here, the y axis Sr ratios are really hard to see. The "Dispersal category" legend heading font is too big.

Figure 3: also has two captions, the lower one is the more descriptive of the two. Here the y axis values are easier to read. I would make the font consistent, right now it's a mix of serif and sans serif fonts. I'd stick with Arial or something like that.

Line 235-238: The clusters should be indicated on the map more clearly, I suggest using different marker shapes for points that fall into each cluster, whereas the color of the marker can still reflect $^{87}\text{Sr}/^{86}\text{Sr}$ of the sample. I'd also reword to "The southern cluster includes *parts* of the Buganda and Toro formations...".

Line 375-377: too many "reliable"s here! Switch up wording.

Line 381: "modern plant sampleS" samples should be plural

In the paragraph starting with line 375, I think it's also worth discussing some of the factors that Janzen et al. (2020) bring up about the usefulness of modern Sr isoscapes in exploring mobility in the past. They also provide a good example of using plants and microfaunal data (both modern and archaeological) together to create an isoscape. I'd also look at van der Lubbe's study on shifts in $^{87}\text{Sr}/^{86}\text{Sr}$ of Lake Turkana waters over the Holocene.

van der Lubbe, HJL, J Krause-Nehring, A Junginger, Y Garcin, JCA Joordens, GR Davies, C Beck, Craig S Feibel, TC Johnson and Hubert B Vonhof
2017 Gradual or abrupt? Changes in water source of Lake Turkana (Kenya) during the African Humid Period inferred from Sr isotope ratios. *Quaternary Science Reviews* 174:1-12.

Line 387: this line on Sr in paleosols is sort of out of left field, so I think a line on why that would be useful would benefit this section. Also, what do we know about how faithfully they record bioavailable Sr ratios, which is what we're truly interested in here?

Line 442: Starting here and through to the end of the paragraph is an awful lot of data. Maybe this could be put in a table instead (though if the authors decide to keep it in the text I think that's probably fine).

I strongly argue that all of the discussion about geology (currently in the Discussion section) should be moved up to a background section. We need to know why Sr is a useful element to track mobility in the first place!!! This of course depends on the geological variability of the area. Lay this out early on and I think the paper will read a lot more clearly, plus set up some nice expectations for the data. It really doesn't make any sense to relegate this important information to after the results are presented.

Other recommendations: I'm not well-versed in the cluster analysis or parametric bootstrapping. However, I'd like to see the R code used for these included in the SI, along with the results of these methods. This would be useful for researchers seeking to employ similar methods in their own study areas.

Appendix F

Dear editors –

We would like to express our genuine gratitude towards Reviewers 3 and 4 for their careful eye to details and thoughtful suggestions. We believe the present manuscript is stronger for their input, and we hope that we have satisfactorily addressed each of their concerns. We have addressed each point raised by the reviewers in the letter below and in the Tracked Changes copy of our manuscript. The reviewer's comments appear in italics while our responses are in bold. The line numbers referenced below are in relation to the Clean Copy version of the manuscript uploaded as part of our resubmission.

We would also like to thank you for the extra time and work that it takes to secure additional reviewers and for giving us this opportunity to continue to refine our manuscript. We hope that you and your families are all keeping healthy and doing well.

Thank you again for your time and consideration of this resubmission.

Best,

Drs. Marian Hamilton, Sherry Nelson, and Diego Fernandez

Reviewer comments to Author:

Reviewer: 3

Comments to the Author(s)

This is a worthy article which sets out in an important direction—to provide background data necessary for informed interpretation of strontium isotope ratios—especially as they pertain to the issue of primate (and hominid) residence patterns. I believe the article in its current form could be improved, however, with more careful attention to previous relevant research; the omissions make it seem that either the authors are unaware of critical issues long understood and written about, or deliberately ignored studies which might raise questions about their research methodology. I will simply state the problems as I see them, and leave it to the editors to determine if, in a revision, the issues are satisfactorily addressed. I don't feel the need to re-review.

1. The authors, in the abstract and throughout, make it sound as if comparing different skeletal tissues (eg, bone vs. teeth) is somehow 'novel'. On the contrary, the general concept was articulated by Jonathan Ericson as early as 1985 (Ericson, J. Strontium isotope characterization in the study of prehistoric human ecology; J. Human. Ev. 14:5 pp. 503-514,) and this was the first article to propose using strontium isotopes to address residence patterns. Moreover, the concept was articulated specifically for fossil hominids in an article they themselves cite (Sillen, et al. 1998 ref#22) Others have worked on the issue; the use of the word 'novel' feels inappropriate.

We certainly do not mean to imply anywhere in this manuscript that comparing bones and teeth is our novel contribution – rather, it is our analytical process (offset calculations) that we present as novel. We have re-worded our abstract and included the seminal Ericson 1985 reference in the introduction – thank you for the suggestion.

2. Lines 44-45: The authors state that “tooth enamel forms early in life and then is metabolically inert.” This is true, but a tremendous oversimplification; the difficulty is that different locations of

enamel (between teeth, and within teeth) are calcifying at different moments in an individual's growth and development, and these differences matter, particularly when analyzing first molars (see below)

We have rephrased this to capture this important nuance and included a citation for the Sillen and Balter (2018) paper, referenced in the comment below. Thank you for pushing for this important clarification.

3. I understand the logic of using cancellous bone as a proxy for a relatively recent signal, but somewhere it should be made clear that it is the LEAST useful material, when it exists at all, for looking at fossils.

We agree. We refer to the risk of diagenesis within fossilized bone on Line 50-52 and have reworded the sentence there to emphasize this point. We then introduce environmental samples as an alternative for obtaining an 'adult' isotopic ratio. We have added a reiteration of this point into our Discussion on line 404-406 as well.

4. I am perplexed that, in reviewing the studies pertaining to the fossil record (line 55), the authors omit Sillen and Balter (2018; Strontium isotopic aspects of Paranthropus robustus teeth: implications for habitat, residence, and growth; J. Human. Ev. 114: 118-130). This study is directly relevant, since these authors documented varying $87\text{Sr}/86\text{Sr}$ within teeth: most notably first molars. Obviously, we can't expect the authors to go back and redo their research design, but they do need to articulate exactly from where they took their samples. The reason is that Sillen and Balter suggest that some first molar enamel, that calcifying before weaning, represents maternal residence. I suggest the authors write a few sentences acknowledging the issue and explaining how their sampling methodology avoids such tissue.

Thank you for pointing out this omission. We have added the reference in on Line 56 as well as earlier, in clarification of the mineralizing timing of tooth enamel (line 46). We also addressed this in our sampling methods on lines 171-179.

5. Line 129. I'm not sure I understand the logic of not looking at riparian leaf samples. I think it is that, way down on line 407, they say "...the local geology in the riparian zone soil plays a secondary role as a source of strontium in the plant and animal tissue." It isn't clear to me whether they are saying riparian zones don't differ that much from the primary source, or that primates (and by extension hominids) don't much exploit riparian resources. It needs clarification, because Sillen et al (1997) demonstrated for the Sterkfontein Valley that the difference between riparian and veld $87\text{Sr}/86\text{Sr}$ was, if anything more important than the geological map.

This is an excellent question, particularly given the stark contrast between riparian and non-riparian areas observed in South Africa. We have published on the impact of non-local strontium on the isotopic profiles of riparian habitats in other parks in Uganda (Hamilton et al 2019, AJPA 170(4), 551-564) and performed similar tests to those outlined in the referenced paper in Kibale. There were no significant riparian effects measured along Kibale rivers. We suspect that this lack of riparian signal in the strontium isotope ratios of plants growing along Kibale's rivers is due to the abundance of water in the Kibale system as a whole (in contrast with the Toro-Semliki Wildlife Refuge, for example, which is a gallery forest surrounded by grassland). We have explicitly mentioned the lack of riparian-driven variation in Kibale on lines 228-229.

6. Line 377 sp. Insufficiently

Thank you – corrected.

7. Lines 377-383. *Instead of saying on line 377 that bone samples are 'notoriously unreliable', I would suggest using a phrase like, 'need to be conducted with great care, and special attention to diagenesis'. Otherwise, the paragraph is inconsistent with citation 22 on line 384, which is based on analysis of small rodent fossil bones.*

Thank you – corrected.

8. Lines 399-400. *Based on the experience of the Sterkfontein Valley, I disagree that bedrock geology is a reasonable proxy for local water and vegetation. On the contrary, I would argue that bedrock geology is at best, a starting place for hypotheses, but there is no substitute for empirical isoscape mapping.*

We agree that there is no substitute or better practice than empirical mapping. We think it is worth stating this in no uncertain terms, and have done so throughout where appropriate (ex: line 97, line 433)

9. Lines 429. *Again, the authors need to address the question of maternal alkaline earths in M1 enamel tissue, perhaps with a better description of their sampling methodology.*

Thank you – we have added in reference to our earlier clarification of possible material effects and sampling procedures.

Reviewer comments to Author:

Reviewer: 3

Hamilton et al. Using Strontium Isotopes to Determine Philopatry and Dispersal in Primates

In this manuscript, the authors use differences in $^{87}\text{Sr}/^{86}\text{Sr}$ of primate teeth formed before dispersal and compare those to environmental Sr ratios (where the animals died) and to Sr ratios recovered from bone, which presents an average of $^{87}\text{Sr}/^{86}\text{Sr}$ over the last few years of life. The goal here is to assess whether Sr ratios are useful indicators of dispersal patterns. I think this is a useful study that would be of interest to many researchers, and the bioavailable Sr data presented here is also a nice contribution. Overall, I think this manuscript could benefit from deeper discussions of why bioavailable $^{87}\text{Sr}/^{86}\text{Sr}$ may not correspond to local geology, and below I outline several areas for improvement.

Line 45: maybe define dispersal earlier, for those non-primatologists out there.

We have added in definitions of philopatry and dispersal at the beginning of the manuscript.

Line 47-48: not just fossils! I would also add that this is compared to enamel, and a couple words on why (e.g. porosity).

Thank you – we have clarified this and added in the porosity explanation as suggested.

Line 72: cite previous studies outlining this approach (I'm thinking Price et al. 2002 on using ratios within $\pm 2\sigma$ of the mean)

We have added these citations as suggested.

Line 93: delete “impregnated”, sub in “-tipped”

Changed as suggested.

Line 96: citation for higher turnover in cancellous vs cortical bone? Does this differ with element?

We have added in two citations for this information (line 138). This is overall bone turnover rate from the osteological medical literature; we were not able to find information suggesting variation by element.

Line 97-100: I think it would be nice to include an SI table with the ages at which these different teeth start to form crowns/mineralize/erupt in the different species. Also include age at weaning in that chart. Update—I see you now get to this in the discussion, but I still think it's worth including a table with all the known eruption/mineralization data that's out there. That'd be a useful resource for other researchers.

We agree that these are important data to compile, and that such a reference would be beneficial to many (the authors included!). However, for many of the species that we include in this study, there are not detailed studies on the specific age of M1 eruption, although plenty of foundational work documenting that M1 mineralization and eruptions occurs long before sexual maturity and any possible dispersal event related to it (for example: Swindler, D. R. (2002). *Primate dentition: an introduction to the teeth of non-human primates* (Vol. 32). Cambridge University Press; Smith BH (1989) Dental development as a measure of life history in primates. *Evolution* 43:683–688; Smith, BH., Crummett, T. L., & Brandt, K. L. (1994). Ages of eruption of primate teeth: a compendium for aging individuals and comparing life histories. *American Journal of Physical Anthropology*,

37(S19), 177-231). The existence of such fine-tuned data for some species, such as chimpanzees, is thanks to meticulous data collection under primarily captive conditions that simply has not been conducted for species such as colobus monkeys or guenons. Because of the numerous missing specific age data points for tooth mineralization and eruption, this does not translate well into a table format. We have included more information on line 139-140 regarding tooth eruption, and a reference to the portion of the Discussion where this is explored in greater detail on line 150. We hope that this will help direct an interested reader immediately to the relevant section of the manuscript.

Even though the Janzen et al. 2020 study doesn't cover Uganda, I would cite it as a good example of how bioavailable $^{87}\text{Sr}/^{86}\text{Sr}$ doesn't always agree with underlying geology, due to aeolian or fluvial inputs, for example. Do Janzen et al. have samples from similar geologies as in this study? What were the $^{87}\text{Sr}/^{86}\text{Sr}$ of those? I think reviewing that and setting up some expectations for the author's own data would be useful here (or flesh this out more in the paragraphs starting around line 392 by referencing some of the results from the Janzen et al. paper).

This is a wonderful suggestion, and we have incorporated ideas from Janzen et al 2020 into the paragraph to set up more clear expectation for high variability within our data (this paragraph previously in the Discussion, now in the Introduction, per your other suggestion – lines 86-124). While there are a few data points from similarly aged rocks in the Janzen et al dataset, there are not many (N=4 Precambrian samples, broadly), and varied dramatically in their specific ages and lithologies, making a direct comparison with our data difficult.

I think it would also be a good idea to discuss some of the geological research that's been done in the area.

*Link, K., Koehn, D., Barth, M.G., Tiberindwa, J.V., Barifaijo, E., Aanyu, K. and Foley, S.F., 2010. Continuous cratonic crust between the Congo and Tanzania blocks in western Uganda. *International Journal of Earth Sciences*, 99(7), pp.1559-1573.*

*Bell, Keith, and J. L. Powell. "Strontium isotopic studies of alkalic rocks: the potassium-rich lavas of the Birunga and Toro—Ankole Regions, East and Central Equatorial Africa." *Journal of Petrology* 10, no. 3 (1969): 536-572.*

Thank you for these reference suggestions – we have built up the discussion of the regional geology on lines 107-111.

I also wonder if it's possible to employ an approach like in the Janzen et al. study (which is really explained in the paper by Bataille et al. (2018) for western Europe), in which other geological/atmospheric variables are used to create a predictive Sr map.

We agree that this would be an interesting exercise and one that would be a good comparison to our empirical isoscape, but it falls outside of the goals and objectives of the current manuscript. We would love to see future researchers build on our work to do this and compare the predictive results to ours.

Were plant samples cleaned before analysis? How much dust/dirt could have been on them? I would just mention this to assuage any concerns readers might have about this.

We cleaned plants of any dirt or debris prior to drying and homogenizing. We have included a sentence about this on line 181-182.

Line 151: what's the expected value of SRM 987? And how many total runs of the standard during the span of analysis?

We have included these data on lines 196 and 198.

Line 375-377: too many "reliable"s here! Switch up wording.

Reworded – thank you!

Line 381: "modern plant sampleS" samples should be plural

Corrected.

In the paragraph starting with line 375, I think it's also worth discussing some of the factors that Janzen et al. (2020) bring up about the usefulness of modern Sr isoscapes in exploring mobility in the past. They also provide a good example of using plants and microfaunal data (both modern and archaeological) together to create an isoscape. I'd also look at van der Lubbe's study on shifts in $87\text{Sr}/86\text{Sr}$ of Lake Turkana waters over the Holocene.

*van der Lubbe, HJL, J Krause-Nehring, A Junginger, Y Garcin, JCA Joordens, GR Davies, C Beck, Craig S Feibel, TC Johnson and Hubert B Vonhof 2017 Gradual or abrupt? Changes in water source of Lake Turkana (Kenya) during the African Humid Period inferred from Sr isotope ratios. *Quaternary Science Reviews* 174:1-12.*

Thank you – we have added in a discussion of these difficulties to the start of this paragraph (lines 425-433) including the suggested citations (van der Lubbe et al 2017, Janzen et al 2020)

Line 387: this line on Sr in paleosols is sort of out of left field, so I think a line on why that would be useful would benefit this section. Also, what do we know about how faithfully they record bioavailable Sr ratios, which is what we're truly interested in here?

To the best of our knowledge, the fidelity with which paleosols retain bioavailable Sr information has never been investigated – we include it here as a potentially promising avenue for future research. We have added to our thoughts in these sentences to make them feel more integrated into the present manuscript.

Line 442: Starting here and through to the end of the paragraph is an awful lot of data. Maybe this could be put in a table instead (though if the authors decide to keep it in the text I think that's probably fine).

These data are all available in table format in Supplementary Table 1. Previous reviewers suggested that we explicitly pull the relevant information out in the text for more rapid accessibility to anyone trying to verify our statement.

I strongly argue that all of the discussion about geology (currently in the Discussion section) should be moved up to a background section. We need to know why Sr is a useful element to track mobility in the first place!!! This of course depends on the geological variability of the area. Lay this out early on and I think the paper will read a lot more clearly, plus set up some nice expectations for the data. It really doesn't make any sense to relegate this important information to after the results are presented.

We agree that this reformatting greatly helps the readability of the paper! We have moved this information from the Discussion to the Background section at the beginning (lines 86-124) and made edits throughout to smooth the transitioned text.

(We have put all the comments on figures together below with our remarks – thank you for taking such time and care to evaluate them, both aesthetically and for clarity of information):

Figure 1: I think this needs to be broken up into several different figures: the maps on one, and the boxplots on the other. I also think the map figure needs an inset to show where in Uganda this is, and also, where Uganda is within the African continent. In my experience many people are unfamiliar with African geography. Right now these figures look very much like a first draft.

We agree and have split Figure 1 into (new) Figures 1 and 2. We have included orienting maps to the location of Uganda and the location of Kibale within Uganda.

The fonts in the boxplot y axes are way too small to read, I would make those bigger and consistently use serif or sans serif fonts for the axis labels (I think Arial is probably best, but ultimately that's up to the authors, as long as they're consistent). Also, the latitude/longitude markers are really small and not always visible in the maps. I would also get rid of the white box around the scale bar, and change this to a round number (maybe 5 km?). In Figure 1A the Unidentified radiometric anomaly looks like a lake, so I would change the color! I also wonder if Figure 1C is necessary? The authors include the botanical sample data in both maps, so the only difference is the interpolated landscape, which, I'm not sure how helpful that is. I'd also consider changing the color ramp for the plant $87\text{Sr}/86\text{Sr}$. Red indicates low $87\text{Sr}/86\text{Sr}$...but we usually think of red as indicating a higher ratio! I'd consider using the color codes in the recent paper on spatial variation in $87\text{Sr}/86\text{Sr}$ in Kenya and Tanzania:

*Janzen, A., Bataille, C., Copeland, S.R., Quinn, R.L., Ambrose, S.H., Reed, D., Hamilton, M., Grimes, V., Richards, M.P., le Roux, P. and Roberts, P., 2020. Spatial variation in bioavailable strontium isotope ratios ($87\text{Sr}/86\text{Sr}$) in Kenya and northern Tanzania: Implications for ecology, paleoanthropology, and archaeology. *Palaeogeography, Palaeoclimatology, Palaeoecology*, p.109957.*

I would also get rid of the really dark border around the color ramp.

We have made the following changes per your suggestions:

- **Fonts have all been enlarged and changed to Arial, included the lat/long markers**
- **White box around the scale bar is removed and scale changed to 5km**
- **Unidentified radiometric anomaly is now dark orange instead of blue**
- **We have changed the color ramp as suggested, following the general model in Janzen et al 2020**
- **We have removed old Figure 1C and included the information all on one map (new Figure 1)**
- **We have removed the borders around the color ramp**

What's also missing from the map figure is some other clear landmark. I'm having trouble comparing the maps in the manuscript to maps in other papers. Maybe the authors could zoom out a bit in the map? Or this problem might be solved with the addition of an inset showing exactly where this is in Uganda. Also, is the Dura River indicated by a dotted line? I'd change this to maybe a blue line that's more easily identifiable as a river. This could be easily digitized in ArcGIS from a map.

We have made the Dura River into a blue solid line as suggested. We believe that the orienting map features on the new Figure 1 should help orient the reader more clearly as to the location of the park.

More comments on the figure. For the boxplots, could the authors indicate in the x axis what the northern southern and Ngogo clusters are? That's missing from the figure. Also, it would be useful to either outline or mark those clusters in a clear way on the map.

We have included the samples grouped with each cluster as unique shapes (squares, circles, or triangle) on the new Figure 1. We have included the cluster labels on the boxplot in new Figure 2.

The authors mentioned that Ngogo park sits over two clusters. Is it possible to show the outlines of each park on the figure instead of just a point? I think that'll help the readers quite a bit. Right now we have no idea of the extent of those parks.

We have placed dotted circles around the approximate boundaries of each park area. It is worth noting that these park area boundaries are not firm or topographic; rather, they reflect the (mildly) flexible territories of the chimpanzee communities studies at each park location. The areas indicated on the map are accurate, but should not be regarded as precise, fixed boundaries.

Also, maybe the authors could symbolize the clusters by marker shape in the map?

This is a wonderful idea! We have done so in Figure 1.

Figure 2: there are two different captions for this? The lower one is better! I would shift the font sizes here, the y axis Sr ratios are really hard to see. The "Dispersal category" legend heading font is too big.

We have increased the font sizes and decreased the legend heading font size. I am unsure about the reference to two different captions? Our captions are included in the main article text file, at the bottom of the document per submission requirements.

Figure 3: also has two captions, the lower one is the more descriptive of the two. Here the y axis values are easier to read. I would make the font consistent, right now it's a mix of serif and sans serif fonts. I'd stick with Arial or something like that.

We have made the fonts consistently Arial throughout, as suggested.

Line 235-238: The clusters should be indicated on the map more clearly, I suggest using different marker shapes for points that fall into each cluster, whereas the color of the marker can still reflect $^{87}\text{Sr}/^{86}\text{Sr}$ of the sample. I'd also reword to "The southern cluster includes parts of the Buganda and Toro formations..."

Adjusted as suggested.

Other recommendations: I'm not well-versed in the cluster analysis or parametric bootstrapping. However, I'd like to see the R code used for these included in the SI, along with the results of these methods. This would be useful for researchers seeking to employ similar methods in their own study areas.

We are more than happy to provide R code for these analyses and have done so in our Supplementary Materials file. Thank you for this suggestion. The dendrogram output for the cluster analysis has been included as Supplementary Figure 1. The results from the parametric bootstrapping are described in the manuscript (lines 306-311 and 324-336).

Appendix G

Dear editors –

We thank you again for your patience and diligence in working with our manuscript. We deeply appreciate the help and comments of the reviewers and hope that these most recent incorporations and additional requested comparisons complete the manuscript to your standards. We are very appreciative of the opportunity to publish these data in an open, accessible format for future researchers, and add our proposed approach to strontium isotope analysis into the scientific discourse.

Below please find our responses to the most recent reviewer comments in bold, with reviewer comments in *italics*. The referenced line numbers correspond to the attached Clean Copy manuscript file.

Thank you again for the work that you do, and happy New Year.

Sincerely,

Marian Hamilton, Sherry Nelson, and Diego Fernandez

Reviewer comments to Author:

Reviewer: 3

Comments to the Author(s)

All good, with one minor exception, on line 46: "Tooth enamel forms early in life and then is metabolically inert, although different enamel locations within the same tooth do calcify at different rates." The point is rather that they calcify at different times, not different rates.

Thank you – we have corrected this wording.

Reviewer: 4

Comments to the Author(s)

RSOS-200760.R3

Second Review

Overall I'm happy with the changes the authors made to the manuscript. However, upon a second (and perhaps deeper) reading, there are still two major issues that stand out to me.

I would like the authors to clearly address how the statistical clusters of samples they generated differ from using just the range of $^{87}\text{Sr}/^{86}\text{Sr}$ within the bounds of each study area. Why is the "standard archaeological method" one that takes into account data from geology rather than

87Sr/86Sr from the study area? I comment on this below in several places, but I'm really not sure why, for example, anyone trying to identify local vs non local individuals in the Sebitole area would include plant 87Sr/86Sr collected from the southern reaches of the park to generate their local Sr signature. Right now this seems a bit like a straw man argument. What I'd like to see is a set of boxplots (maybe grouping it in with figure 2) that shows the distribution of plant 87Sr/86Sr for each study area (for example, all the Sr data from within the dotted lines in figure 1). I think that's going to align much better with the statistical clusters. Then the authors can really get into why the statistical clustering method is better than using "local" Sr ratios.

The standard archeological method requires a pre-determined definition of "local" strontium isotope ratios; this can be based on geologic formation, site associations, community boundaries, and potentially numerous other ways as well (as we explain on lines 69-74). Site-specific definitions are more common and more appropriate for large-scale migration and mobility studies, such as those examining human mobility at Teotihuacan and Cahokia, or those looking at large-scale animal migration (such as birds or mammoths). Studies working within a more limited geographic area more commonly use geology (ex: Sillen et al 1998, Copeland et al 2009), which is why we focused on that definition here as a comparison to our newly proposed method. We included a reference to this on lines 173-175 and included site association local definitions within the discussion of the statistical clustering method's usefulness on the following lines.

It is critical to note here that the way in which "local" is defined not the salient detail in our discussion of this approach. Rather, the important aspect of the standard approach (which our method presents an alternative to) is in the categorization of individuals as 'local' or 'non-local' in comparison to the local environmental signature (no matter how it is originally defined).

The ineffectiveness of the standard archeological approach in Kibale does not change regardless of whether 'local' is defined based on geology, as presented in the manuscript, or site association. However, because site association is a method found often in the literature, we agree that it is fair and in the spirit of thoroughness to include it as a point of comparison to our offset method. We have done so on lines 237-241 and 324-330. We also have included the requested boxplot in Figure 2 showing the breakdown of isotopic ratios by nearest site association, using those associations listed in Supplemental Table 2. We include a discussion of this plot on lines 287-294.

Using site association as opposed to geologic formation association does not change the effectiveness of the standard archeological method in identifying dispersing/philopatric individuals within our dataset.

One issue that I failed to comment on in the first round of reviews is that this paper does not mention previous archaeological studies that have used similar approaches (bone-tooth pairs) to exploring mobility. The other reviewer commented on this, but I don't think the authors added a thorough enough discussion in their background section. It would be good to see the authors contextualize their study with a clear discussion of how the approaches used in this study compare to those used in the following:

Price, T. Douglas, Linda Manzanilla, and William D. Middleton. "Immigration and the ancient city of Teotihuacan in Mexico: a study using strontium isotope ratios in human bone and teeth." *Journal of Archaeological Science* 27, no. 10 (2000): 903-913.

Grupe, Gisela, T. Douglas Price, Peter Schröter, Frank Söllner, Clark M. Johnson, and Brian L. Beard. "Mobility of Bell Beaker people revealed by strontium isotope ratios of tooth and bone: a study of southern Bavarian skeletal remains." *Applied Geochemistry* 12, no. 4 (1997): 517-525.

Schweissing, Matthew Mike, and Gisela Grupe. "Stable strontium isotopes in human teeth and bone: a key to migration events of the late Roman period in Bavaria." *Journal of archaeological science* 30, no. 11 (2003): 1373-1383.

Or this one, which compares two different teeth:

Slater, Philip A., Kristin M. Hedman and Thomas E. Emerson. 2014. Immigrants at the Mississippian polity of Cahokia: strontium isotope evidence for population movement. *Journal of Archaeological Science* 44(0):117-127.

It might also be worth it to mention other studies of mobility that use paired tissues (one metabolically inert tissue compared to a metabolically active tissue):

Del Rio, Carlos Martínez, Pablo Sabat, Richard Anderson-Sprecher, and Sandra P. Gonzalez. "Dietary and isotopic specialization: the isotopic niche of three *Cinclodes* ovenbirds." *Oecologia* 161, no. 1 (2009): 149-159.

As well as mention some of the studies looking at dispersals in other taxa (like birds!)

Hobson, Keith A., Leonard I. Wassenaar, and Erin Bayne. "Using isotopic variance to detect long-distance dispersal and philopatry in birds: an example with ovenbirds and American redstarts." *The Condor* 106, no. 4 (2004): 732-743.

Also this recent paper may be of interest:

Crowley, Brooke Erin, and Laurie Rohde Godfrey. "Strontium isotopes support small home ranges for extinct lemurs." *Frontiers in Ecology and Evolution* 7 (2019): 490.

Thank you for these reference recommendations. We have included a sentence situating our work within the context of these foundational studies; as we have stressed previously, in no way do we want to imply that the novel contribution from this work is simply assaying two different tissues! We have included context on lines 84-89.

Line 99: Do the authors mean that the Sr will be consumed by weathering?

The original strontium isotope ratio from the consumed parent material would be lost due to weathering. We have restructured the sentence to clarify meaning here.

Line 196 and 197: use \pm instead of +/-

Corrected, thank you.

Line 219: Wouldn't the standard archaeological approach be better described as one that uses local $^{87}\text{Sr}/^{86}\text{Sr}$, rather than $^{87}\text{Sr}/^{86}\text{Sr}$ assigned to particular geologies? I'm thinking of the Slater et al. 2014 paper on Cahokia. They do not use a geologic map here, but just samples from sites from the American Bottom.

As mentioned above, there are many ways used in the literature to define "local" when using the standard archeological method. We reference them in our introduction to it on lines 69-75. We have added in a discussion and comparison of using site association to define "local" on lines 237-241 and 324-330 in order to be as thorough in our comparison as possible.

Line 220: The authors should explain why they chose to use the $^{87}\text{Sr}/^{86}\text{Sr}$ values of plants only from the Toro formation as their "local" signature. According to Figure 1, Kanyawara sits on both the Toro and Buganda formations, pretty equally. What about comparing the primates from Sebitole to the Toro formation, since that one is pretty homogenous? And for the Kanyawara primates, how about just using the $^{87}\text{Sr}/^{86}\text{Sr}$ values that are within the study area? I also don't think there is any information in the SI table that indicates clearly which animals came from which study area. Can the authors include a column in the table that indicates that?

We have only one primate from Sebitole included in this data set, which is why Sebitole is not used as a test site within the paper. Our explanation for using the geology instead of the park area is included above. When park area association is use instead, the results from the standard archeology method remain unchanged. We have included this analysis and results on lines 237-241 and 324-330.

This information on what park area each animal was collected from is currently included in SI Table 1, under the "Location" column.

Line 350: I'm not totally clear on why the authors used the mean value of all the plant samples collected from the Buganda formation to compare to the individual from Kanyanchu. The mean value is 0.7113, but clearly in Figure 1 all the plant samples from the Kanyanchu area are higher, closer to around 0.715, due to the proximity to that unidentified radiometric anomaly. I think it's a good exercise to compare the values from the geology, but I think it would also be good to compare the primates from each study area to the plants that were collected from each study area.

We agree that because defining a local area based on site association is a common method found in the literature, this is indeed a reasonable exercise to include. We have done so on lines 237-241 and 324-330.

After reading through this a second time, I realize I'm not totally clear on what the standard

archaeological approach means. In some places I take it to mean using the range of $^{87}\text{Sr}/^{86}\text{Sr}$ of any plant sample collected from the geologic substrate on which a site is located. But, other times in the manuscript, like in the paragraph starting with Line 407 to line 411, it seems like the method I just described is compared to the “standard archaeological approach”.

We explain the standard archeological method on lines 69-75 (underlined emphasis added for the purposes of this comment): “The most common method in archeological strontium isotope literature to differentiate local (presumably philopatric) and non-local (dispersing) individuals is to first, use botanical, soil, or micromammal samples to define minimum and maximum strontium isotope ratios for a pre-defined local site, such as discrete geologic formations, site associations, or community boundaries, and second, compare an individual’s enamel $^{87}\text{Sr}/^{86}\text{Sr}$ to this range. For this standard archeological approach, individuals with $^{87}\text{Sr}/^{86}\text{Sr}_{\text{enamel}}$ falling outside of the local isotopic minimum and maximum are considered non-local and individuals with $^{87}\text{Sr}/^{86}\text{Sr}_{\text{enamel}}$ falling within the local range are considered local.” The important aspect is not which boundary is selected to define the local area (geology, local site association, community boundaries, etc) but rather the subsequent comparison and categorization of an individual as belonging to that local area or not. It is this binary categorization that fails to indicate sex biases in dispersal in Kibale, and what this manuscript provides as an alternative approach to.

Figure 1: This figure is much improved! A couple of suggestions to polish it. First, the big Africa map needs to be updated to include South Sudan. I also think it would look better if the authors just darkened the outline of Uganda in the big Africa map (rather than the rectangle), because the big Uganda map is just an image of Uganda. Then use two lines extending from the corners of the box marking the study area in the Uganda map to extend to two corners of the study area map. Also crop out that line above Uganda. A few things about the caption:

- 1. Specify that the boundary of the park is indicated by the SOLID black line.*
- 2. The GREEN (not yellow) areas represent the Buganda formation*
- 3. Specify in the caption what the dashed black lines mean*

Thank you – we have made all recommended changes. We agree that this reviewer’s suggestions have dramatically improved this figure and are very grateful for them.

Figure 2: Remove all colors of the box and whisker plots. Because the X-axis labels are clearly marked, color is not necessary.

Corrected as suggested.

Figure 3: The pink shaded boxes in the two plots don’t totally match in color. Also, with the box over the plots, it makes the coral circle markers (philopatric sex) appear red, so they don’t match up with the legend (or are the markers in the plot actually a different color from those in the legend?). I also feel that the pink shaded areas indicating local range makes the figure look a little dated/like a draft. I think just a light grey box would look a lot cleaner.

Corrected as suggested.